# Unified Mechanism-Specific Amplification by Subsampling and Group Privacy Amplification

**Jan Schuchardt[1], Mihail Stoian[2],\* Arthur Kosmala[1],\* Stephan Günnemann[1]**
{j.schuchardt, a.kosmala, s.guennemann}@tum.de, mihail.stoian@utn.de
[1]Dept. of Computer Science & Munich Data Science Institute, Technical University of Munich
[2]Dept. of Engineering, University of Technology Nuremberg

## Abstract

Amplification by subsampling is one of the main primitives in machine learning with differential privacy (DP): Training a model on random batches instead of complete datasets results in stronger privacy. This is traditionally formalized via *mechanism-agnostic* subsampling guarantees that express the privacy parameters of a subsampled mechanism as a function of the original mechanism's privacy parameters. We propose the first general framework for deriving *mechanism-specific* guarantees, which leverage additional information beyond these parameters to more tightly characterize the subsampled mechanism's privacy. Such guarantees are of particular importance for *privacy accounting*, i.e., tracking privacy over multiple iterations. Overall, our framework based on conditional optimal transport lets us derive existing and novel guarantees for approximate DP, accounting with Rényi DP, and accounting with dominating pairs in a unified, principled manner. As an application, we analyze how subsampling affects the privacy of groups of multiple users. Our tight mechanism-specific bounds outperform tight mechanism-agnostic bounds and classic group privacy results.

## 1 Introduction

Composability and amplification by subsampling are two key properties of Differential Privacy (DP) [1–3] that make it possible to provide provable privacy guarantees for machine learning algorithms that iteratively learn from batches of data.

Composability means that privacy gracefully deteriorates when iteratively applying a differentially private mechanism to a dataset [2]. Kairouz et al. [4], Murtagh and Vadhan [5] ultimately derived tight composition theorems that optimally characterize the privacy parameters $(\varepsilon', \delta')$ of a composed mechanism as a function of the component mechanisms' parameters $(\varepsilon, \delta)$. However, later work demonstrated that these guarantees are only tight in a *mechanism-agnostic* sense [6]. Stronger *mechanism-specific* composition guarantees can often be obtained by using additional information beyond the fact that the mechanisms satisfies some notion of DP. Methods for tracking privacy over multiple iterations (e.g. [6–12]) are refererred to as *privacy accountants*.

Amplification by subsampling means that privacy can be strenghtened by applying a differentially private mechanism to randomly sampled batches of a dataset [13, 14]. Similar to Kairouz et al. [4], Balle et al. [15] ultimately proposed a framework for deriving tight subsampling theorems that optimally characterize the privacy parameters $(\varepsilon', \delta')$ of a subsampled mechanism as a function of the underlying *base mechanism*'s parameters $(\varepsilon, \delta)$.

However, their framework for deriving subsampling theorems has two key limitations. Firstly, as we shall demonstrate, the resultant guarantees are generally only tight in a mechanism-agnostic

---

\*Equal contribution

38th Conference on Neural Information Processing Systems (NeurIPS 2024).

sense: Given $(\varepsilon, \delta)$, one can construct *some* worst-case $(\varepsilon, \delta)$-DP mechanism that is $(\varepsilon', \delta')$-DP under subsampling. The specific mechanism at hand may however be significantly more private. Secondly, their framework does not explain how to derive subsampling guarantees for privacy accountants. In fact, there is so far no unified framework for deriving subsampling guarantees for privacy accountants.

To address these two limitations, we propose a novel framework for *unified mechanism-specific amplification by subsampling* analysis, using conditional optimal transport. Our proposed framework (1) lets us derive mechanism-specific subsampling guarantees, which can be stronger than mechanism-agnostic guarantees, (2) lets us recover mechanism-agnostic guarantees via a pessimistic upper bound – essentially subsuming the approach from [15] – and (3) lets us derive guarantees for approximate differential privacy [2], moments accounting [6] (i.e., Rényi differential privacy [7]), and accounting with dominating pairs [10] (e.g., numerical accounting [16, 8, 17–21]) in a unified, principled manner.

As a practical application, we consider the problem of tightly analyzing group privacy under subsampling (see Fig. 1). Assume a scenario where $K$ individuals may independently decide to contribute to a dataset. Further assume that this dataset is processed by randomly sampling a batch and applying an $(\varepsilon, \delta)$-DP *base mechanism*. A special case of this setting is discussed in a recent technical note [22], which assumes that the $K$ individuals collaboratively agree to either contribute or not contribute all their data and that the base mechanism is Gaussian. In the general setting we consider, the best known method for providing privacy guarantees for the entire group is to (1) use existing bounds to show that the subsampled mechanism guarantees $(\varepsilon', \delta')$-DP for individuals and (2) use the group privacy property[23] to show that this implies $(K \cdot \varepsilon', K \cdot e^{K \cdot \varepsilon'} \cdot \delta')$-DP for the group.

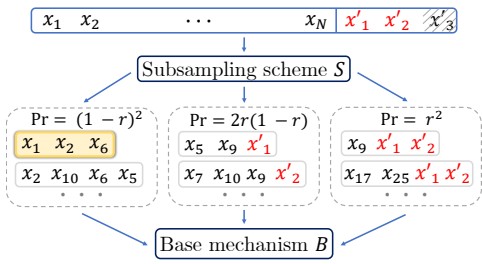

Figure 1: Group members $x'_1, x'_2$ contribute to a dataset, while group member $x'_3$ does not. For small subsampling rates $r$, it is unlikely to access a single $(\mathrm{Pr} = 2r(1-r))$ or even both $(\mathrm{Pr} = r^2)$ inserted elements when applying a base mechanism $B$ to a subsampled batch (e.g., the yellow one). This further obfuscates which data was contributed by members of group $\{x'_1, x'_2, x'_3\}$.

Our proposed framework lets us derive stronger, tight *group privacy amplification* guarantees. By analyzing group privacy and subsampling jointly, these guarantees accurately capture that it is unlikely for a large fraction of the group's data to simultaneously appear in a batch. Our framework further lets us derive dominating pairs [10], which enables tight tracking of group privacy under composition. Overall, our main contributions are that we

- demonstrate for the first time that there is a qualitative difference between mechanism-specific and mechanism-agnostic tightness in privacy amplification,
- propose a general framework for deriving mechanism-specific amplification by subsampling guarantees, subsuming prior work on mechanism-agnostic amplification,
- develop the first unified method for deriving subsampling guarantees for privacy accounting,
- and derive the first tight subsampling guarantees for general group privacy.

As part of our group privacy analysis, we also significantly generalize recent results derived for subsampled matrix mechanisms [24], which is of independent interest. Experimental evaluation demonstrates that our tight mechanism-specific guarantees outperform both tight mechanism-agnostic bounds and post-hoc use of the group privacy property.

## 2 Background and preliminaries

**Problem setting.** We consider the same setting as Balle et al. [15], but with an additional privacy accounting and group privacy perspective. We assume a space of datasets $\mathbb{X}$, a space of batches $\mathbb{Y}$ that can be constructed from these datasets, and an output space $\mathbb{Z}$. For the sake of exposition, we assume that $\mathbb{X}$ is the powerset $\mathcal{P}(\mathbb{A})$ of some finite discrete set $\mathbb{A}$, that $\mathbb{Y} = \{y \subseteq x \mid x \in \mathbb{X}\}$, and that $\mathbb{Z} = \mathbb{R}^D$. We further consider a random *subsampling scheme* $S : \mathbb{X} \to \mathbb{Y}$ that generates batches from datasets, and a random *base mechanism* $B : \mathbb{Y} \to \mathbb{R}^D$ that maps batches to outputs. We write $s_x(y)$ for the pmf of $S(X)$ and $b_y(z)$ for the pdf of $B(y)$. In Appendix D, we introduce a more general setting that admits arbitrary spaces, for which we derive all our results. Our goal is to

provide formal privacy guarantees for the *subsampled* mechanism $M = B \circ S$, which takes a dataset, subsamples a batch from it, and then applies the random base mechanism to this batch.

**Dataset and batch relations.** Specifically, we want to prove privacy under *neighboring relations* $\simeq_{\mathbb{X}} \subseteq \mathbb{X}^2$ between datasets. For example, the *insertion/removal* relation $x \simeq_{\pm, \mathbb{X}} x'$ implies that $x' = x \cup \{a\}$ or $x' = x \setminus \{a\}$ for some $a$. The *substitution* relation $x \simeq_{\Delta, \mathbb{X}} x'$ implies that $x' = x \setminus \{a\} \cup \{a'\}$ for some $a, a'$. In addition, we assume that $\mathbb{Y}$ is equipped with a batch neighboring relation $\simeq_{\mathbb{Y}} \subseteq \mathbb{Y}^2$. The batch relation $\simeq_{\mathbb{Y}}$ can be distinct from the dataset relation $\simeq_{\mathbb{X}}$.

**Subsampling schemes.** While we consider arbitrary subsampling schemes $S : \mathbb{X} \to \mathbb{Y}$, we shall later apply our framework to three particularly common ones: Subsampling *without replacement* and *with replacement* sample sets and multisets of fixed size $|y| = q$ uniformly at random. *Poisson* subsampling includes each element of $x$ with independent probability ("rate") $r \in [0, 1]$.

**Approximate differential privacy.** A mechanism $M : \mathbb{X} \to \mathbb{R}^D$ is privacy-preserving under symmetric neighboring relation $\simeq_{\mathbb{X}}$ when the distributions of $M(x), M(x')$ with densities $m_x, m'_x$ are almost indistinguishable for all $x \simeq_{\mathbb{X}} x'$. Approximate differential privacy (ADP) [2] quantifies this via hockey stick divergences [25]:

**Definition 2.1.** For $\varepsilon \geq 0$, a mechanism $M : \mathbb{X} \to \mathbb{R}^D$ is $(\varepsilon, \delta)$-DP under symmetric relation $\simeq_{\mathbb{X}}$ iff $\forall x \simeq_{\mathbb{X}} x' : H_{e^\varepsilon}(m_x || m_{x'}) \leq \delta$ with $H_\alpha(m_x || m_{x'}) = \int_{\mathbb{R}^D} \max\{m_x(z)/m_{x'}(z) - \alpha, 0\} \cdot m_{x'}(z) \, dz$.

Note that this divergence-based definition of approximate differential privacy is equivalent to requiring for all datasets $x \simeq_{\mathbb{X}} x'$ and all events $S \subseteq \mathbb{R}^D$ that $\Pr[M(x) \in S] \leq e^\varepsilon \cdot \Pr[M(x') \in S] + \delta$.

**Rényi differential privacy.** Privacy accounting was first popularized by methods that used moment bounds instead of $(\varepsilon, \delta)$ pairs to better track privacy under composition [6, 26, 27]. These methods were later developed into a moments-based notion of privacy – Rényi differential privacy (RDP) [7]:

**Definition 2.2.** For $\alpha \geq 1$, a mechanism $M : \mathbb{X} \to \mathbb{R}^D$ is $(\alpha, \rho)$-RDP under symmetric relation $\simeq_{\mathbb{X}}$ iff $\forall x \simeq_{\mathbb{X}} x' : \log(\Lambda_\alpha(m_x || m_{x'})) / (\alpha - 1) \leq \rho$ with $\Lambda_\alpha(m_x || m_{x'}) = \int_{\mathbb{R}^D} m_x(z)^\alpha \cdot m_{x'}(z)^{1-\alpha} dz$.

Note that $\Lambda_\alpha$ *is not the Rényi divergence, but its $\alpha$th moment, i.e., a scaled and exponentiated Rényi divergence.* We use this notation to eliminate exponential terms that arise in amplification for RDP (see [28–30]). If a mechanism is $(\alpha, \rho)$-RDP, its $T$-fold self-composition is $(\alpha, T \cdot \rho)$-RDP. These RDP parameters can then be converted to ADP parameters $(\varepsilon, \delta)$, albeit in a lossy manner [7, 31, 32].

**Dominating pairs.** Later work proposed other numerical [16, 8, 17–21] or analytical accounting techniques [9, 33, 28, 34, 35]. Zhu et al. [10] developed a unified view on these works by introducing the notion of *dominating pairs*:

**Definition 2.3.** A pair of distributions $(P, Q)$ with densities $(p, q)$ is a dominating pair for mechanism $M$ under neighboring relation $\simeq_{\mathbb{X}}$, if $H_\alpha(m_x || m_{x'}) \leq H_\alpha(p || q)$ for all $x \simeq_{\mathbb{X}} x'$ and all $\alpha \geq 0$.

Given a dominating pair $(P, Q)$, one can use various numerical or analytic techniques, such as convolution of privacy loss distributions [16] or central limit theorems of tradeoff functions [9], to track the privacy of mechanism $M$ under composition (see Fig. 2 in [10]).

**Group privacy.** The *group privacy property* is the graceful decay of privacy when considering multiple user's data. This is normally formalized via the notion of induced distance (see [7, 15, 23]):

**Definition 2.4.** The distance $d_{\mathbb{X}}(x, x')$ induced by relation $\simeq_{\mathbb{X}}$ is the length of the shortest sequence $(x_1, \ldots, x_{K-1}) \in \mathbb{X}^{K-1}$ such that $x \simeq_{\mathbb{X}} x_1, \forall k : x_k \simeq_{\mathbb{X}} x_{k+1}$, and $x_{K-1} \simeq_{\mathbb{X}} x'$.

*Example* 2.5. Let $\simeq_{\mathbb{X}}$ be the insertion/removal relation $\simeq_{\pm}$. Then the induced distance $d_{\mathbb{X}}$ between $x = \{1, 2\}$ and $x' = \{2, 3\}$ is 2, because $\{1, 2\} \simeq_{\pm} \{1, 2, 3\} \simeq_{\pm} \{2, 3\}$.

**Proposition 2.6** (Vadhan [23]). *If mechanism $M : \mathbb{X} \to \mathbb{R}^D$ is $(\varepsilon, \delta)$-DP under relation $\simeq_{\mathbb{X}}$, then it is $(K \cdot \varepsilon, K \cdot e^{K \cdot \varepsilon} \cdot \delta)$-DP under group relation $\{(x, x') \in \mathbb{X}^2 \mid d_{\mathbb{X}}(x, x') = K\}$.*

That is, for sufficiently small $\varepsilon$ the ADP parameters deteriorate appproximately linearly with induced distance. As baselines for our experiments, we use even tighter bounds (see Appendix C.1.4).

## 3 Unified mechanism-specific amplification by subsampling

Our goal is to develop a general procedure for (tightly) deriving ADP parameters $(\varepsilon', \delta')$, RDP parameters $(\alpha', \rho')$, or dominating pairs $(P', Q')$ for the subsampled mechanism $M = B \circ S$ with

subsampling scheme $S$ and base mechanism $B$. Based on Definitions 2.1 to 2.3, this requires that we evaluate or bound the hockey stick divergence $H_\alpha$ or the scaled and exponentiated Rényi divergence $\Lambda_\alpha$ between the distributions of $M(x)$ and $M(x')$ with $x \simeq_\mathbb{X} x'$. To discuss both simultaneously, let us write $\Psi_\alpha$ for either $H_\alpha$ or $\Lambda_\alpha$ and assume that $\alpha \geq 0$ or $\alpha \geq 1$, respectively.

There are two challenges: Firstly, the distribution of $M(x)$ is high-dimensional mixture distribution with one component per possible batch $y \in \mathbb{Y}$ and weights given by batch probabilitites $s_x(y)$, i.e., $m_x(z) = \sum_{y \in \mathbb{Y}} b_y(z) \cdot s_x(y)$. Secondly, there may be no simple analytic formula for the component densities. For instance, evaluating the density of perturbed model gradients in noisy SGD [36] requires backpropagation. Both challenges make it hard to evaluate or bound $\Psi_\alpha(m_x || m_{x'})$.

A useful property that will help us in addressing the first problem of having a large number of mixture components is the joint convexity of $\Psi_\alpha$ in the space of probability density functions [37, 38]:

**Lemma 3.1.** *Consider arbitrary densities* $f_1^{(1)}, f_2^{(1)}, f_1^{(2)}, f_2^{(2)} : \mathbb{R}^D \to \mathbb{R}_+$ *and weight* $w \in [0,1]$. *Then,* $\Psi_\alpha(w f_1^{(1)} + (1-w) f_2^{(1)} || w f_1^{(2)} + (1-w) f_2^{(2)}) \leq w \Psi_\alpha(f_1^{(1)} || f_1^{(2)}) + (1-w) \Psi_\alpha(f_2^{(1)} || f_2^{(2)})$.

In our case, the $f$s can be base mechanism densities $b_y$ with different $y \in \mathbb{Y}$, and the weights can be given by the subsampling distribution. We could thus upper-bound the mixture divergence $\Psi_\alpha(m_x || m_{x'})$ in terms of component divergences – if the mixtures had identical weights. This is generally not the case, since subsampling mass functions $s_x(\cdot)$ and $s_{x'}(\cdot)$ depend on datasets $x \neq x'$.

To still leverage the joint convexity of $\Psi_\alpha(m_x || m_{x'})$, we want to rewrite $m_x$ and $m_{x'}$ as mixtures with identical weights. This is exactly what is offered by *couplings* between probability mass functions. Intuitively, a coupling between two pmfs $p_1, p_2 : \mathbb{Y} \to [0,1]$ specifies for every $y_1, y_2 \in \mathbb{Y}$ how much probability should be transported from $y_1$ to $y_2$ to transform $p_1$ into $p_2$. Couplings can be formalized and generalized to multiple pmfs as follows (for a more thorough introduction, see [39]):

**Definition 3.2.** A coupling between $N$ pmfs $p_1, \ldots, p_N : \mathbb{Y} \to [0,1]$ on batch space $\mathbb{Y}$ is a joint pmf $\gamma : \mathbb{Y}^N \to [0,1]$ where the $n$th marginal is $p_n$, i.e., $p_n(y_n^*) = \sum_{\boldsymbol{y} \in \mathbb{Y}^N} \mathbb{1}[y_n = y_n^*] \cdot \gamma(\boldsymbol{y})$.

Given a valid coupling $\gamma$ between subsampling mass functions $s_x$ and $s_{x'}$, we can use marginalization to rewrite $m_x(z)$ as $\sum_{\boldsymbol{y} \in \mathbb{Y}^2} b_{y_1}(z) \gamma(\boldsymbol{y})$. Similarly, we can rewrite $m_{x'}(z)$ as $\sum_{\boldsymbol{y} \in \mathbb{Y}^2} b_{y_2}(z) \gamma(\boldsymbol{y})$. Now, both $m_x$ and $m_{x'}$ are mixtures with identical weights and one component per pair of batches in $\mathbb{Y}^2$. We can thus recursively apply Lemma 3.1 to show (full proof in Appendix E):

**Theorem 3.3.** *Consider a subsampled mechanism* $M = B \circ S$, *and an arbitrary coupling* $\gamma$ *between subsampling mass functions* $s_x(\cdot)$ *and* $s_{x'}(\cdot)$. *Then,*

$$\Psi_\alpha(m_x || m_{x'}) \leq \sum_{\boldsymbol{y} \in \mathbb{Y}^2} c_\alpha(y^{(1)}, y^{(2)}) \cdot \gamma(y^{(1)}, y^{(2)}) \tag{1}$$

*with cost function* $c_\alpha : \mathbb{Y} \times \mathbb{Y} \to \mathbb{R}_+$ *defined by* $c_\alpha(y^{(1)}, y^{(2)}) = \Psi_\alpha(b_{y^{(1)}} || b_{y^{(2)}})$.

We write $y^{(1)}$ instead of $y_1$ to simplify later notations. While every coupling $\gamma$ yields a valid upper bound, the guarantees can be tightened by finding a coupling $\gamma^*$ that minimizes the r.h.s. of Eq. (1). We thus have an *optimal transport problem*, where the cost $c_\alpha$ of transporting probability from batch $y^{(1)}$ to batch $y^{(2)}$ depends on the divergence $\Psi_\alpha$ of base mechanism densities $b_{y^{(1)}}$ and $b_{y^{(2)}}$.

Unfortunately, experimental evaluation in Appendix B.1.4 shows that the resultant guarantees can be much weaker than those from prior work – even with an optimal coupling $\gamma^*$. This is because Theorem 3.3 results from recursively applying the joint convexity property (Lemma 3.1) to $\Psi_\alpha(m_x || m_{x'})$. Each recursive step splits each mixture into two smaller mixtures and further upper-bounds the divergence that is achieved by *our specific subsampled mechanism $M$ on our specific pair of datasets* $x, x'$. Upon fully decomposing the overall divergence into divergences between single-mixture components, this sequence of bounds is in fact larger than even the divergence $\Psi_\alpha(\tilde{m}_{\tilde{x}} || \tilde{m}_{\tilde{x}'})$ achieved by *a worst-case subsampled mechanism $\tilde{M}$ on a worst-case pair of datasets* $\tilde{x}, \tilde{x}'$. To overcome this limitation, we propose to limit the recursion depth in order to obtain a tighter upper bound that matches *our specific subsampled mechanism $M$ on a worst-case pair of datasets* $\hat{x}, \hat{x}'$.

Limiting the recursion depth to which Lemma 3.1 is applied means upper-bounding $\Psi_\alpha(m_x || m_{x'})$ in terms of mixture divergences that have not been fully decomposed into their individual components. Specifically, we propose to do so by defining an optimal transport problem between *multiple* subsampling mass functions conditioned on different events (proof in Appendix E):

**Theorem 3.4.** *Consider a subsampled mechanism $M = B \circ S$. Further consider two disjoint partitionings $\bigcup_{i=1}^{I} A_i = \mathbb{Y}$ and $\bigcup_{j=1}^{J} E_j = \mathbb{Y}$ of batch space $\mathbb{Y}$ such that all $A_i$ and $E_j$ have non-zero probability under the distribution of $S_x$ and $S_{x'}$, respectively. Let $\gamma$ be an arbitrary coupling between conditional mass functions $s_x(\cdot \mid A_1), \ldots, s_x(\cdot \mid A_I), s_{x'}(\cdot \mid E_1), \ldots, s_{x'}(\cdot \mid E_J)$. Then,*

$$\Psi_\alpha(m_x || m_{x'}) \leq \sum_{\boldsymbol{y} \in \mathbb{Y}^{I+J}} c_\alpha(\boldsymbol{y}^{(1)}, \boldsymbol{y}^{(2)}) \cdot \gamma((\boldsymbol{y}^{(1)}, \boldsymbol{y}^{(2)})),$$

*with cost function $c_\alpha : \mathbb{Y}^I \times \mathbb{Y}^J \to \mathbb{R}_+$ defined by*

$$c_\alpha(\boldsymbol{y}^{(1)}, \boldsymbol{y}^{(2)}) = \Psi_\alpha \left( \sum_{i=1}^{I} b_{y_i^{(1)}} \cdot \Pr[S(x) \in A_i] || \sum_{j=1}^{J} b_{y_j^{(2)}} \cdot \Pr[S(x') \in E_j] \right). \tag{2}$$

In other words: We now have an optimal transport problem between $I + J$ probability mass functions coupled by $\gamma$. The transport cost $c_\alpha$ is a divergence between two mixtures. The components of the first mixture are base mechanisms densities given batches $y_i^{(1)} \in \mathbb{Y}$ from batch tuple $\boldsymbol{y}^{(1)} \in \mathbb{Y}^I$. The weights of the first mixture are probabilities of events $A_i$. The second mixture is defined analogously.

Note that we can recover Theorem 3.3 by conditioning on a single event, i.e., $A_1 = \mathbb{Y}, E_1 = \mathbb{Y}$. As we shall demonstrate, a more fine-grained partitioning lets us obtain tighter bounds that match the divergence $\Psi_\alpha(m_{\tilde{x}} || m_{\tilde{x}'})$ of our specific subsampled mechanism $M$ on a worst-case pair of datasets $\hat{x}, \hat{x}'$. In the extreme case of defining event per possible batch, Theorem 3.4 holds with equality.

Overall, Theorem 3.4 reduces the broad problem of bounding mixture divergences to the canonical problem of optimal transport between conditional distributions. But before we can apply it, we need to address two open problems: Evaluating the cost function and designing optimal couplings.

**Cost function bound.** Not having an analytic expression for every base mechanism density $b_y(z)$ may make it intractable to evaluate cost function $c_\alpha$. We thus propose to bound it via an approach that is inherent to differential privacy: Considering worst-case inputs (proof in Appendix E).

**Proposition 3.5.** *Consider $\boldsymbol{y}^{(1)} \in \mathbb{Y}^I, \boldsymbol{y}^{(2)} \in \mathbb{Y}^J$, and cost function $c$ defined in Eq. (2). Let $d_\mathbb{Y}$ be the distance induced by $\simeq_\mathbb{Y}$ (see Definition 2.4). Then, $c_\alpha(\boldsymbol{y}^{(1)}, \boldsymbol{y}^{(2)}) \leq \hat{c}_\alpha(\boldsymbol{y}^{(1)}, \boldsymbol{y}^{(2)})$, with*

$$\hat{c}_\alpha(\boldsymbol{y}^{(1)}, \boldsymbol{y}^{(2)}) = \max_{\hat{\boldsymbol{y}}^{(1)}, \hat{\boldsymbol{y}}^{(2)}} c_\alpha(\hat{\boldsymbol{y}}^{(1)}, \hat{\boldsymbol{y}}^{(2)}) \tag{3}$$

*subject to $\forall k, l \in \{1, 2\}, \forall t, u : d_\mathbb{Y}(\hat{y}_t^{(k)}, \hat{y}_u^{(l)}) \leq d_\mathbb{Y}(y_t^{(k)}, y_u^{(l)})$ and $\hat{\boldsymbol{y}}^{(1)} \in \mathbb{Y}^I, \hat{\boldsymbol{y}}^{(2)} \in \mathbb{Y}^J$.*

Put differently: We construct two new mixtures with components that are adversarially chosen to maximize divergence while retaining the pairwise distances between batches in $\boldsymbol{y}^{(1)}$ and $\boldsymbol{y}^{(2)}$. Note that, for the special case of $\Psi_\alpha = H_\alpha$ and $I = J = 1$, this bound corresponds to the "group privacy profile" in [15]. As we shall demonstrate in the next sections, this bound $\hat{c}$ can often be evaluated using high-level information about the base mechanism $B : \mathbb{Y} \to \mathbb{R}^D$, such as global sensitivity.

**Sufficient optimality condition.** While every coupling $\gamma$ yields a valid upper bound in Theorem 3.4, this bound can be tightened by designing an optimal coupling $\gamma^*$. To inform this design, we generalize the notion of distance-compatible couplings from [15] to an arbitrary number of distributions:

**Definition 3.6.** A coupling $\gamma$ between mass functions $p_1, \ldots, p_N$ on batch space $\mathbb{Y}$ is $d_\mathbb{Y}$-compatible when $\gamma(\boldsymbol{y}) > 0$ only if $\forall u > 1 : d_\mathbb{Y}(y_1, y_u) = d_\mathbb{Y}(\{y_1\}, \text{supp}(p_u))$ and $\forall u > t > 1 : d_\mathbb{Y}(y_u, y_t) = d_\mathbb{Y}(\text{supp}(p_t), \text{supp}(p_u))$, where $\text{supp}(p_u)$ is the support of $s_u$ and the distance between two sets is the minimum distance of their elements.

Essentially, a $d_\mathbb{Y}$-compatible coupling only assigns probability to a tuple of batches $\boldsymbol{y}$ when all pairs $y_i, y_j$ have the smallest possible distance to $y_1$ and each other while still being in the support of their distributions. In Appendix F, we prove that $d_\mathbb{Y}$-compatibility is sufficient for optimality. We further show that the optimal value has a canonical form whenever a $d_\mathbb{Y}$-compatible couplings exists.

**Summary.** To summarize, we propose the following three-step procedure for deriving subsampling guarantees: (1) Define two partitions of the batch space into $I$ and $J$ events. (2) Define a simultaneous coupling between the corresponding $I + J$ conditional distributions to obtain a bound in terms of divergences between small mixtures with $I$ and $J$ components. (3) Bound these mixture divergences by considering $I + J$ worst-case mixture components under pairwise batch distance constraints.

## 3.1 Tight mechanism-specific group privacy amplification

As a concrete example, and to illustrate the difference between mechanism-specific and -agnostic (see Section 3.2) bounds, let us consider the group privacy setting from Fig. 1. We have a group of $K$ users that can independently decide to contribute or not contribute their data, and the resulting dataset is Poisson subsampled. To provide formal privacy guarantees to the group, we need to prove that the distributions of $M(x), M(x')$ are almost indistinguishable for $x, x'$ with induced distance $d_{\pm,\mathbb{X}}(x, x') \leq K$. Let $x \simeq_{K_+, K_-, \mathbb{X}} x'$ be the relation that implies that $K_+$ records are inserted and $K_-$ records are removed to construct $x'$ from $x$. We can then show (full proof in Appendix M):

**Theorem 3.7.** *Let $M = B \circ S$, where $S$ is Poisson subsampling with rate $r$. Let $\simeq_{\mathbb{Y}}$ be the insertion/removal batch relation $\simeq_{\pm,\mathbb{Y}}$. Then, for all $x \simeq_{K_+, K_-, \mathbb{X}} x'$, $\Psi_\alpha(m_x || m_{x'})$ is l.e.q.*

$$\max_{\boldsymbol{y}} \Psi_\alpha \left( \sum_{i=1}^{K_-+1} b_{y_i^{(1)}} \cdot \mathrm{Binom}(i-1 \mid K_-, r) || \sum_{j=1}^{K_++1} b_{y_j^{(2)}} \cdot \mathrm{Binom}(j-1 \mid K_+, r) \right), \quad (4)$$

*subject to constraints $\boldsymbol{y} \in \mathbb{Y}^{K_-+K_++2}$, as well as $\forall l \in \{1, 2\}, \forall t, u : d_{\mathbb{Y}}(y_t^{(l)}, y_u^{(l)}) \leq |t - u|$, and $\forall t, u : d_{\mathbb{Y}}(y_t^{(1)}, y_u^{(2)}) \leq (t-1) + (u-1)$.*

*Proof sketch.* We let $A_i$ and $E_j$ be the events that $S(x)$ contains $i - 1$ deleted elements and $S(x')$ contains $j - 1$ inserted elements, respectively. Our coupling defines the following generative process: We sample $y_1^{(1)}$ from $s_x(\cdot \mid A_1)$ and let $y_1^{(2)} \leftarrow y_1^{(1)}$. We then iteratively generate the other $y_i^{(1)}$ by sampling a permutation in which we add the $K_-$ deleted elements to $y_1^{(1)}$. We then generate the other $y_j^{(2)}$ by sampling a permutation in which we add the $K_+$ inserted elements to $y_1^{(2)}$. The result then follows from the cost function bound in Proposition 3.5 and $d_{\mathbb{Y}}$-compatibility of the coupling. Batches $y_u^{(1)}$ and $y_t^{(1)}$ and have a distance bounded by $|u - t|$, because one can be obtained from the other by removing/inserting $|u - t|$ elements. The constraints for $y_u^{(2)}$ and $y_t^{(2)}$ are analogous. Batches $y_u^{(1)}$ and $y_t^{(2)}$ have a distance bounded by $(t - i) + (u - 1)$ because we need to remove $t - 1$ elements and insert $u - 1$ elements to construct one from the other. $\square$

Next, we can solve the constrained optimization problem in Theorem 3.7 – i.e., determine worst-case components – to obtain mechanism-specific guarantees. For instance (proof in Appendix M.2.1):

**Theorem 3.8.** *Let $M = B \circ S$, where $S$ is Poisson subsampling with rate $r$, and $B$ is the Gaussian mechanism $h + V$ with $h : \mathbb{Y} \to \mathbb{R}^D$ and $V \sim \mathcal{N}(\boldsymbol{0}, \sigma^2 \boldsymbol{I}_D)$. Define the $\ell_2$-sensitivity $L_2 = \max_{y \simeq_{\pm,\mathbb{Y}} y'} ||f(y) - f(y')||_2$. Then for all $x \simeq_{K_+, K_-, \mathbb{X}} x'$, $\Psi_\alpha(m_x || m_{x'})$ is l.e.q.*

$$\Psi_\alpha \left( \sum_{i=1}^{K_-+1} f_i^{(1)} \cdot \mathrm{Binom}(i-1 \mid K_-, r) || \sum_{j=1}^{K_++1} f_j^{(2)} \cdot \mathrm{Binom}(j-1 \mid K_+, r) \right),$$

*with univariate normal densities $f_i^{(1)} = \mathcal{N}(\cdot \mid (i-1), \sigma / L_2)$, $f_j^{(2)} = \mathcal{N}(\cdot \mid -(j-1), \sigma / L_2)$.*

Note that this bound is mechanism-specific in that it explictly depends on $B$ being a Gaussian mechanism with standard deviation $\sigma$ and sensitivity $L_2$. The bound can be numerically evaluated to arbitrary precision using standard techniques from privacy accounting literature (see Appendix M.4). In Appendix M.2 we derive similar guarantees for Laplace and randomized response mechanisms.

**Tightness.** These bounds are tight in a mechanism-specific sense: One cannot derive stronger ADP or RDP guarantees without additional information about the datasets $x, x' \in \mathbb{X}$ or the underlying function $h$ (proofs in Appendix M.3).

**Asymptotic bounds.** Our focus is on tight bounds that can be explicitly computed. However, some early works on RDP accounting (e.g.[6, 28]) also provided asymptotic versions of their bounds, e.g., as a function of divergence parameter $\alpha$. For completeness, we use Theorem 3.8 to generalize the asymptotic bounds of Abadi et al. [6] to the group privacy setting in Appendix N.5.

**Other contributions.** Our solutions to the optimization problem in Theorem 3.7 are of independent interest: The special case of $\Psi_\alpha = H_\alpha$, $K_- = 0$ or $K_+ = 0$, and Gaussian mechanisms was

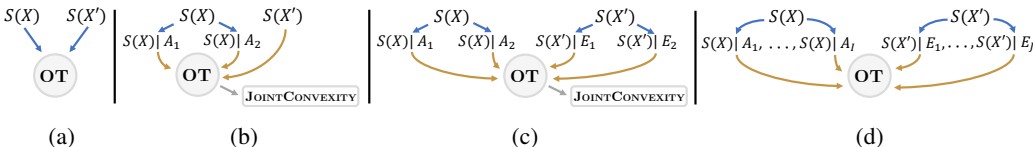

Figure 2: Mechanism-agnostic guarantees for (a) graph modification [40–42] (b) insertion/removal [14, 15, 29, 30] (c) substitution [43, 44, 15, 28] can be derived from (d) our proposed framework. In (b-c), events $A_i$ and $E_j$ indicate the presence of inserted or substituted elements.

derived to analyze matrix mechanisms in [24]. We significantly generalize it via an entirely different proof strategy that admits $\Psi_\alpha \in \{H_\alpha, \Lambda_\alpha\}$, arbitrary $K_+, K_- \in \mathbb{N}_0$, and non-Gaussian mechanisms (see Appendix O).

## 3.2 Tight mechanism-agnostic group privacy amplification

To further illustrate the difference between mechanism-specific and -agnostic bounds, let us apply the framework of Balle et al. [15] to the same setting: For $K_- = 0$ or $K_+ = 0$ and $\varepsilon \geq 0$, their ansatz shows that the subsampled mechanism $M = B \circ S$ is $(\varepsilon', \delta')$-DP with $\varepsilon' = \log(1 + (1 - \text{Binom}(0 \mid K, r)) \cdot (e^\varepsilon - 1))$ and $\delta' = \sum_{k=1}^K \text{Binom}(k \mid K, r) \cdot \delta_k$, with group privacy parameters $\delta_k = \max_{y,y'} H_{\exp(\varepsilon)}(b_y \| b_{y'})$ s.t. $d_{\mathbb{Y}}(y, y') \leq k$ (see Appendix N).

**Tightness.** This bound is tight in a mechanism-agnostic sense, i.e., for every $\varepsilon \geq 0$, one can construct *some* worst-case mechanism that exactly attains the bound (see Appendix N.2).

**Mechanism-specific vs mechanism-agnostic tightness.** An apparent advantage of this result is that it expresses $(\varepsilon', \delta')$ as a simple analytic formula of base mechanism DP parameters $\varepsilon, \delta_1, \ldots, \delta_K$. However, this simplicity comes at a cost: As we show in Appendix N.3, this guarantee implicitly upper-bounds the tight mixture divergence bound from Theorem 3.7 in terms of its component divergences using joint convexity.[2] As we experimentally demonstrate in Section 4, this relaxation leads to weaker privacy guarantees for group size $K \geq 2$. This gap has gone unnoticed in earlier work, because Theorem 3.8 happens to be identical to the bound of Balle et al. [15] for the special case of $K = 1$ (see proof of Proposition 30 in [10]). By studying the more complex group privacy setting, we demonstrate for the first time that *there is a qualitative difference between mechanism-agnostic and mechanism-specific tightness in privacy amplification.*

## 3.3 From mechanism-specific to mechanism-agnostic guarantees

The observed relation between tight mechanism-specific and mechanism-agnostic bounds does in fact extend beyond group privacy (see Fig. 2): As we demonstrate in Appendices G and H, mechanism-agnostic subsampling guarantees from prior work can equivalently be derived by (1) conditioning on at most 4 events indicating the presence of inserted / deleted / substituted elements, (2) defining a simultaneous coupling, and (3) using joint convexity to upper-bound the resultant mechanism-specific guarantee by component divergences. This includes the ADP guarantees of Balle et al. [15] (and thus prior work [14, 43, 44]) and the RDP guarantees from [28–30, 40].

**Subsumption of [15].** Going even further, we show in Appendix G.2 that our approach subsumes the entire framework of Balle et al. [15]: Any mechanism-agnostic guarantee derived via their ansatz can equivalently be derived by defining a (potentially suboptimal) simultaneous coupling between four subsampling distributions and upper-bounding the resultant guarantee via (advanced) joint convexity.

**Contribution.** Importantly, our contribution is not in the bounds themselves, but in the identification of this implicitly underlying pattern that unifies prior work. We further generalize this pattern through our proposed framework to enable tight analysis for more challenging scenarios like group privacy.

**Novel RDP accounting bounds.** In Appendix I, we use this observation to derive a variety of novel mechanism-agnostic RDP bounds for different combinations of $\simeq_{\mathbb{X}}, \simeq_{\mathbb{Y}}$, and subsampling schemes, such as insertion/removal and subsampling without replacement. We further derive a simple but tight mechanism-specific guarantee for subsampled randomized response under substitution (see Theorem I.3) that outperforms the best known mechanism-agnostic bound (see Section 4).

---

[2]In combination with "advanced joint convexity" [15].

### 3.4 From mechanism-specific guarantees to dominating pairs.

Because the mechanism-specific guarantees derived via our framework do not obfuscate the underlying distributions, they can be used to identify dominating pairs. These dominating pairs can then be combined with arbitrary (numerical) accountants to track privacy under composition. For example, we can immediately read off from Theorem 3.8 that the two Gaussian mixtures are a dominating pair for the "insert-$K_+$-remove-$K_-$" relation. In Appendix J, we describe a procedure for constructing dominating pairs when the bound is a weighted sum of multiple mixture divergences. We can thus determine dominating pairs for any bound derived via optimal transport. As we demonstrate in Appendix K, the dominating pairs for subsampled mechanisms derived by Zhu et al. [10] (special cases of which appear in [8, 17–19, 11]) can equivalently be proven via our framework.

**Subsampling with replacement.** As a novel contribution, we derive for the first time dominating pairs for subsampling *with* replacement (see Theorem L.5), which were posited but not proven in [8] (see discussion in Appendix L). This is enabled by our solution to the problem in Theorem 3.7. As we experimentally show in Appendix B.1.5, these bounds can be much stronger than those derived via the framework of Balle et al. [15]. These results thus demonstrate that *there is a qualitative difference between mechanism-specific and mechanism-agnostic tightness even for group size $K = 1$.*

### 3.5 Limitations and future work

**Limitations.** While our proposed framework can be applied to arbitrary subsampling distributions, conditioning on a finite set of events may be too restrictive in certain settings (e.g., continuous batch spaces). Also, while we found conditioning on the number of modified elements to be sufficient for all considered scenarios, an automated procedure for selecting events would be desirable. Maximal couplings fulfill this purpose in [15], but only yield pairs of distributions.

**Future work.** A natural direction is applying our ansatz to novel settings other than group privacy. In particular, it could potentially be used to provide epoch-level guarantees for correlated subsampling (e.g., batching via shuffling), similar to [38]. We present a preliminary result for 2-fold non-adaptive composition in Appendix P. Future work could also use our solutions to the problem in Theorem 3.7 to generalize the matrix mechanism analysis from [24] to substitutions and non-Gaussian distributions.

## 4 Experimental evaluation

The purpose of the following experiments is to verify that there can be a benefit to using mechanism-specific over mechanism-agnostic subsampling bounds, and that a joint analysis of subsampling and group privacy can offer stronger guarantees than post-hoc application of the generic group privacy property. In all figures, "specific" refers to our proposed mechanism-specific bounds, "agnostic" refers to (tight) mechanism-agnostic bounds, and "post-hoc" refers to applying the generic group privacy property to tight mechanism-specific bounds derived for group size 1. For all experiments, we assume $\ell_p$ sensitivities of 1. Further details on the experimental setup are provided in Appendix C. An implementation will be made available at https://cs.cit.tum.de/daml/group-amplification.

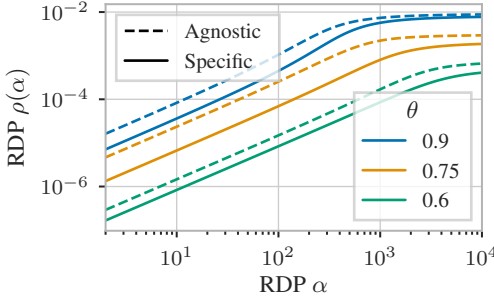
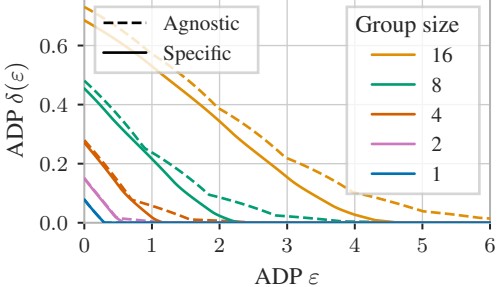

Figure 3: Randomized response with WOR subsampling $(q / N = 0.001)$, group size 1, and varying true response probability $\theta$.

Figure 4: Laplace mechanisms with scale $\lambda = 1$, Poisson subsampling $(r = 0.2)$, and varying group size.

## 4.1 Mechanism-agnostic and mechanism-specific guarantees

**Randomized response and RDP.** One potential source of looseness in mechanism-agnostic bounds is that they bound mixture divergences in terms of component divergences that can be summarized by a single privacy parameter. As an example, let us consider the best known guarantee for substitution, subsampling without replacement, and RDP from [28]. As discussed by the authors, it is only tight up to factors that are constant in $\alpha$. In Fig. 3 we compare it to our tight mechanism-specific bound for randomized response (see Theorem I.3) with batch-to-dataset ratio $q \,/\, N = 0.001$ and true response probability $\theta \in \{0.6, 0.75, 0.9\}$. In Appendix B.1.1 we consider other ratios. The tight guarantee eliminates the constant factors and thus achieves much smaller $\rho$ for a wide range of $\alpha \in (1, 10^4]$.

**Group privacy and ADP.** Another potential source of looseness in mechanism-agnostic bounds is that they stem from a binary partitioning of the batch space (recall Fig. 2), which may not be sufficient when there are multiple possible levels of privacy leakage (e.g. number of sampled group elements). We demonstrate this in Fig. 4, by comparing the tight mechanism-agnostic group privacy bound derived via the framework from [15] to our tight mechanism-specific bound for Laplace mechanisms (see Theorem M.2), ADP, scale $\lambda = 1$, subsampling rate $r = 0.2$, and varying group size. In Appendix B.1.2 we repeat the experiment with other mechanisms and parameter values, and also consider RDP. As discussed in Section 3.2, the mechanism-agnostic bound is identical to the mechanisms-specific bound for group size $K = 1$. For group sizes $K \geq 2$, however, the fine-grained partitioning underlying the mechanism-specific bound yields stronger privacy guarantees. These results confirm that we need to distinguish between mechanism-agnostic and mechanism-specific tightness when analyzing complex subsampling settings.

## 4.2 Post-hoc and mechanism-specific group privacy analysis

**Single-iteration group privacy.** Our next goal is to demonstrate the benefit of analyzing group privacy and subsampling jointly. In Fig. 5, we evaluate our tight guarantee for Gaussian mechanisms (see Theorem 3.8) with $\sigma = 2$, rate $r = 0.2$, and varying group size. As a baseline, we evaluate the same tight guarantee with $K_+ + K_- = 1$ and apply the generic group privacy property (see Appendix C.1.4) in a post-hoc manner. As can be seen, our tight analysis can lead to much stronger privacy guarantees. However, we interestingly observe in Appendix B.2.1 that the generic group privacy properties of ADP and RDP can serve as increasingly tight upper bounds when considering more private base mechanisms (e.g., $\sigma = 5$) and much smaller subsampling rates (e.g., $r = 10^{-3}$).

**Composed group privacy.** Nevertheless, even the gaps in tightness for very private mechanisms may quickly accumulate when repeatedly applying these mechanisms to a dataset. In Fig. 6, we use the dominating pairs derived via our framework to conduct tight PLD accounting [16, 8] with "connect the dots" [20] quantization using Google's `dp_accounting` library [45]. For $\sigma = 5$, $r = 10^{-3}$, and fixed $\varepsilon = 2$ (for other parameters and mechanisms, see Appendix B.2.3), the post-hoc analysis quickly diverges with increasing group size and number of iterations. For example, with group size 16 and privacy budget $\delta = 10^{-6}$, the post-hoc analysis allows less than 100 iterations of DP-SGD training [36, 6], whereas our tight analysis enables training for over 1000 iterations. In Appendix B.2.5 we train an image classification model on MNIST with PLD accounting to demonstrate that this increased number of training iterations can translate to much higher model utility.

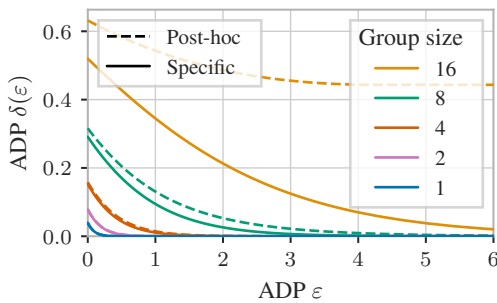
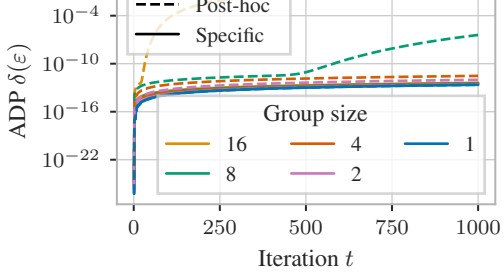

Figure 5: Gaussian mechanisms with standard deviation $\sigma = 2$, Poisson subsampling ($r = 0.2$), and varying group size.

Figure 6: PLD accounting for Gaussian mechanisms ($\sigma = 5$), Poisson subsampling ($r = 10^{-3}$), and varying group size at $\varepsilon = 2$.

## 5  Related Work

**Amplification by subsampling for ADP.** Using subsampling to strenghten $(\varepsilon, \delta)$-DP guarantees has a long history [13, 14] particularly in privacy-preserving machine learning [46, 47, 6]. For a thorough introduction to subsampling and its interaction with composition, we refer the reader to [48]. Balle et al. [15] ultimately proposed a framework for deriving tight mechanism-agnostic subsampling guarantees. Our work subsumes theirs and enables the derivation of mechanism-specific guarantees.

**Amplification for privacy accountants.** Abadi et al. [6]'s seminal work on moments accounting already considered subsampling. Similiar results were derived for the more general notion of RDP [7] in [28–30]. More recent works on accounting generally include a discussion of either Poisson subsampling or subsampling without replacement [17–19, 21, 11]. Subsampling with replacement is discussed in [8], albeit without complete proofs, which we provide in Appendix L. We demonstrate that amplification guarantees for the central notions of RDP [7] and dominating pairs [10] can – just like guarantees for ADP – be derived via optimal transport. Our novel approach not only unifies prior, but also enables a principled analysis of challenging scenarios like group privacy.

**Group privacy amplification.** A special case of group privacy amplification (which we use as an example to demonstrate the utility of our general framework) with hockey stick divergences, Gaussian mechanisms, and groups that collaboratively agree to contribute all their data was posited in [24] and proven in a recent follow-up note [22]. However, their result (and proof strategy) does not cover Rényi divergences, other mechanisms (Laplace, randomized response), and the standard notion of group privacy under which group members are not forced to collaborate (see, e.g., [3, 7, 9, 15, 23]).

**Hierarchical randomized smoothing.** Randomized smoothing [49–51] is a technique for constructing provably robust models via DP achieved through input perturbations (e.g. [52–68]). Similar to subsampling, Scholten et al. [69] proposed to only apply perturbations to randomly sampled parts of an input. We repurpose their technique of treating subsampling indicators as observable random variables in order to construct dominating pairs from weighted sums of divergences (see Appendix J.1).

**Unified amplification for $f$-DP.** Wang et al. [70] proved a form of joint concavity for the tradeoff functions underlying $f$-DP accounting [9, 34]. This enables them to analyze mixtures induced by random initialization and shuffling in a unified manner. However, they do *not* consider amplification by subsampling, and explicitly state that they need to address this problem in future work. Note that dominating pairs, which can be derived via our proposed framework, can be used in f-DP accounting due to duality of privacy profiles and tradeoff functions [9, 10].

## 6  Conclusion

The main purpose of this work is to provide a unified, principled framework for mechanism-specific subsampling analysis and subsampling analysis for privacy accountants. To this end, we proposed a three-step procedure based on optimal transport between conditional subsampling distributions. Beyond recovering known guarantees for Rényi DP and dominating pairs, this procedure lets us derive novel results that were previously only available for approximate DP, such as non-standard combinations of subsampling schemes and neighboring relations, or subsampling with replacement. We then applied this procedure to the problem of analyzing group privacy under subsampling. Our experimental evaluation demonstrates that our mechanism-specific group privacy amplification bounds are not only tight, but can also significantly outperform tight mechanism-agnostic bounds and traditional group privacy results – even under composition. On a higher level, these bounds represent a novel contribution to a larger body of work (e.g., [38, 71, 72]) that demonstrates the benefit of analyzing multiple properties of differential privacy jointly instead of independently.

## 7  Acknowledgements

We are grateful to Georgios Kaissis for valuable discussions and feedback on generalizing our work beyond Rényi differential privacy, Yan Scholten for pointing out connections between subsampled mechanisms and hierarchical randomized smoothing, as well as Leo Schwinn and Nicholas Gao for proofreading our manuscript. This research was funded by the German Research Foundation, grant GU 1409/4-1, and the Munich Data Science Institute (MDSI) at Technical University of Munich (TUM) via the Linde/MDSI Doctoral Fellowship program and the MDSI Seed Fund.

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

# A  Table of contents

# B  Additional experiments

## B.1  Mechanism-agnostic and mechanism-specific bounds

### B.1.1  Randomized response and RDP

In Fig. 7, we repeat the experiment from Fig. 3 with other ratios of batch and dataset sizes. That is, we compare our tight mechanism-specific RDP guarantee from Theorem I.3 to the best known mechanism-agnostic bound from [28] to illustrate that mechanism-agnostic bounds may loose privacy by bounding mixture divergences in terms of component divergences – even outside the group privacy setting. The tight guarantee eliminates constant factors and thus achieves much smaller $\rho$ for a wide range of $\alpha \in (1, 10^4]$, especially for small ratios.

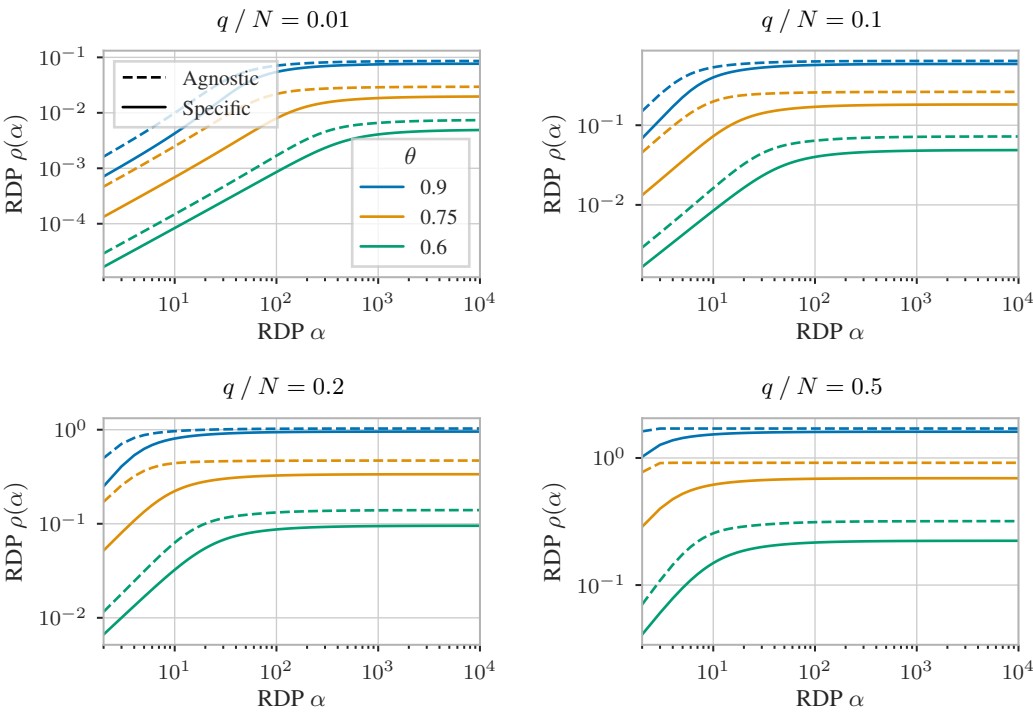

Figure 7: Randomized response under subsampling without replacement, with varying true response probability $\theta$ and batch-to-dataset ratio $q / N$. Theorem I.3 significantly improves upon the baseline for a wide range of $\alpha$.

### B.1.2  Group privacy and ADP

In Figs. 8 and 9 we repeat the experiment from Fig. 4 for other subsampling rates $r$, standard deviations $\sigma$, and also with Laplace mechanisms. That is, we compare our tight mechanism-specific group privacy guarantees to the tight mechanism-agnostic bounds derived via the framework of [15]. The results demonstrate that mechanism-specific tightness is a stronger property that can result in better privacy guarantees – although the difference in significantly more pronounced for Laplace mechanisms and/or larger group sizes.

In Fig. 10 we repeat the experiment with randomized response mechanisms. Here, both approaches yield identical guarantees. This verifies that randomized reseponse is indeed the worst-case mechanisms for which the mechanism-agnostic bound is optimal (see [15] and Appendix N.2).

**Gaussian mechanism**

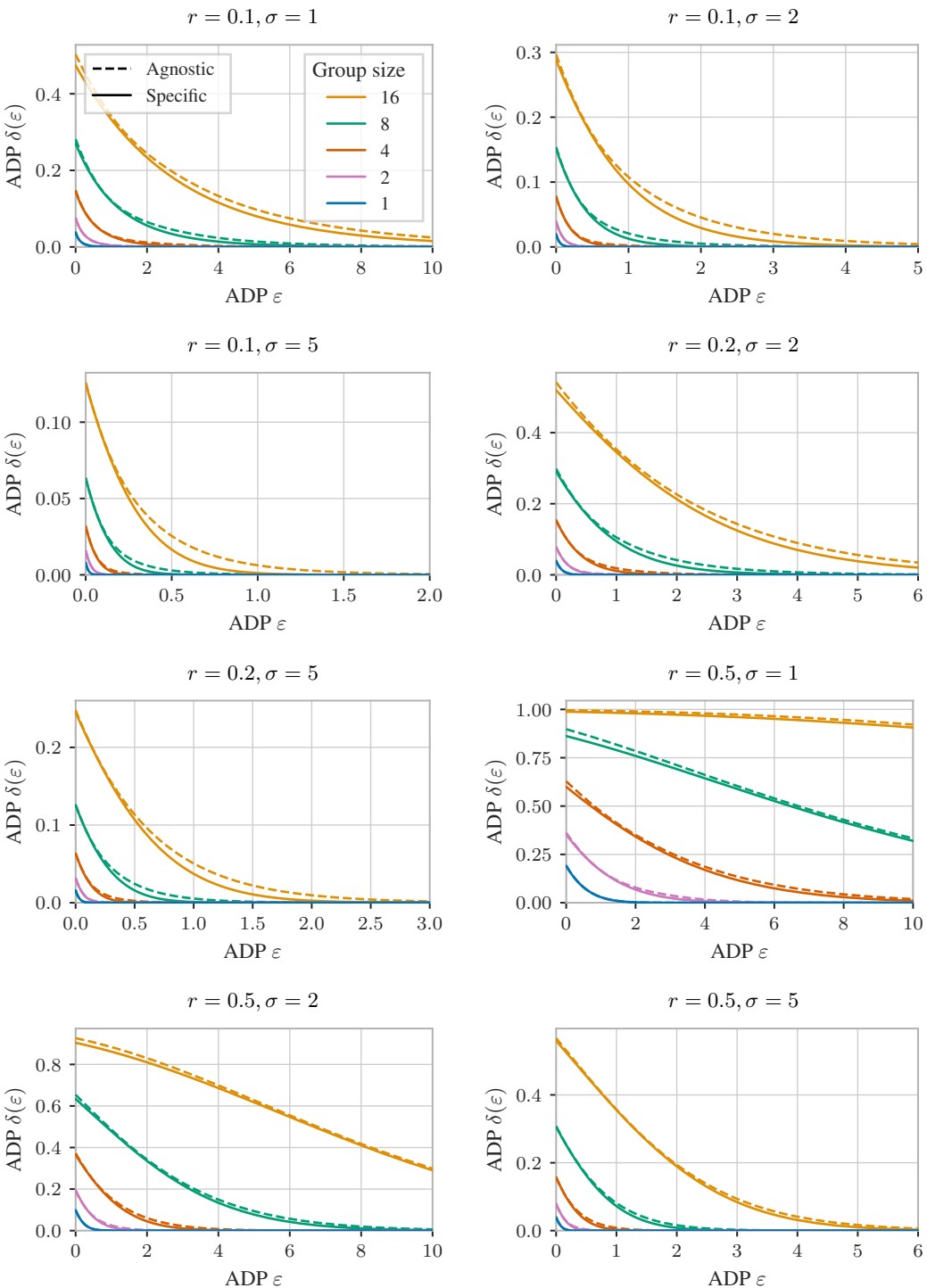

Figure 8: Gaussian mechanisms under Poisson subsampling, with varying standard deviation $\sigma$, subsampling rate $r$, and group size. The tight mechanism-specific guarantees are stronger than the tight mechanism-agnostic bounds, especially for larger group sizes and smaller subsampling rates.

**Laplace mechanism**

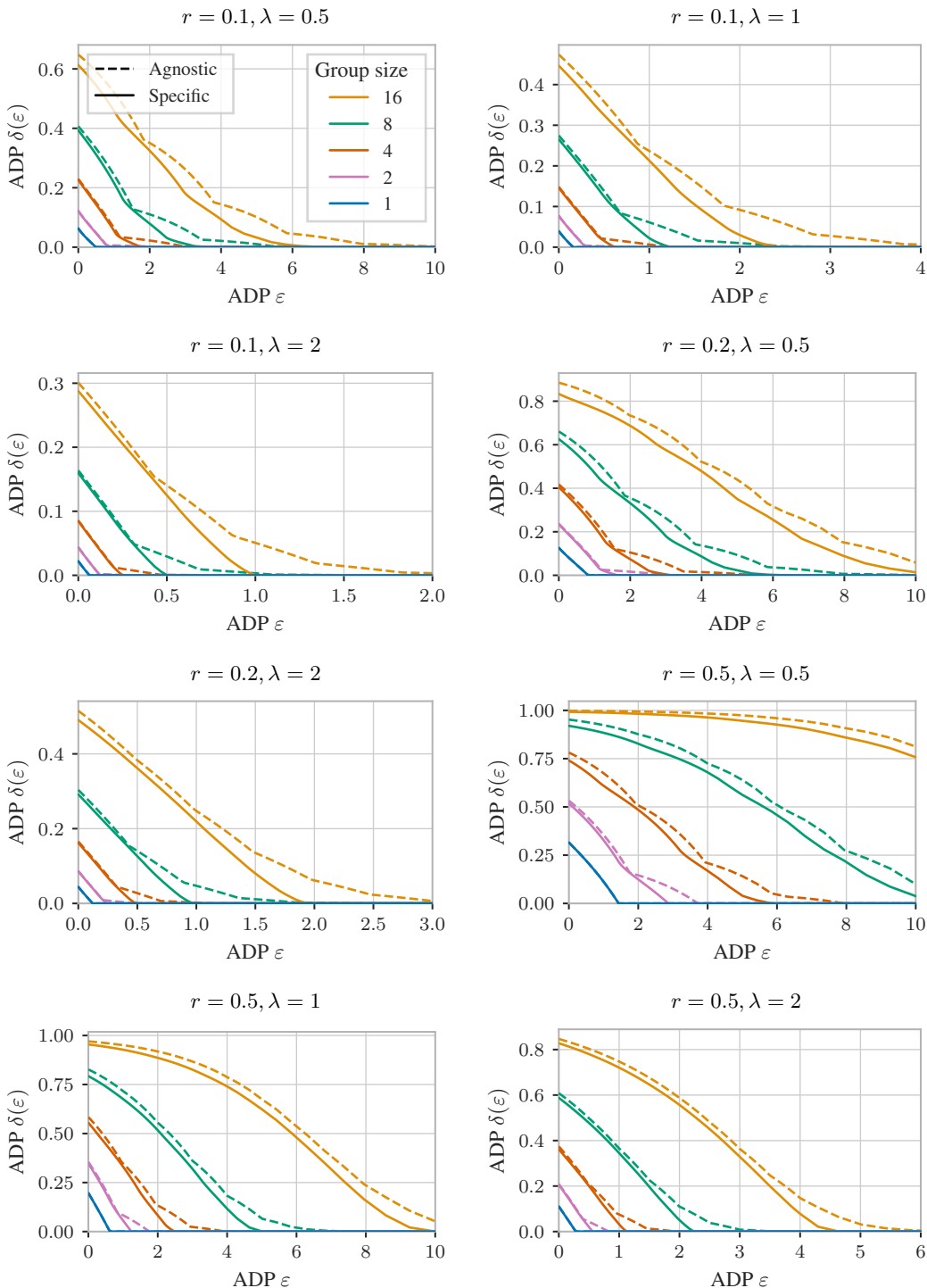

Figure 9: Laplace mechanisms under Poisson subsampling, with varying scale $\lambda$, subsampling rate $r$, and group size. The tight mechanism-specific guarantees are stronger than the tight mechanism-agnostic bounds, especially for larger group sizes.

**Randomized response mechanism**

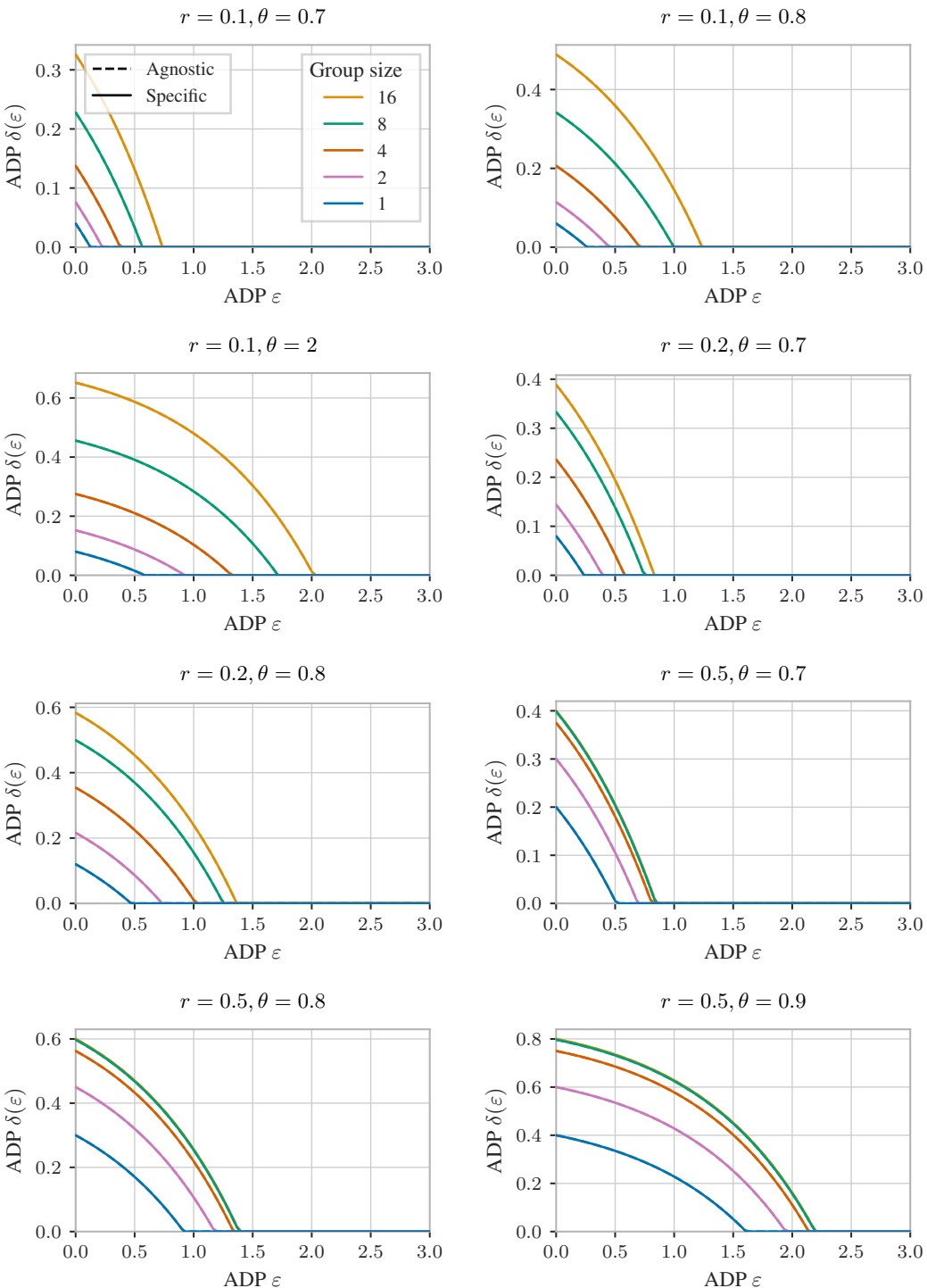

Figure 10: Randomized response under Poisson subsampling, with varying true response probability $\theta$, subsampling rate $r$, and group size. Randomized response is the worst-case mechanism for which the mechanism-agnostic guarantee is optimal, so both ansätze yield identical results.

### B.1.3 Group privacy and RDP

In Fig. 11 we repeat our comparison of mechanism-agnostic (Theorem N.5) and tight mechanism-specific group privacy amplification (Theorem 3.8) with RDP instead of ADP. We observe that the tight guarantee delays the phase transition from a high- to a low-privacy regime that was already observed in [30]. In other words, we have small $\rho$ for larger $\alpha$, which is beneficial for conversion to ADP after composition (see conversion formulae in [7, 31]).

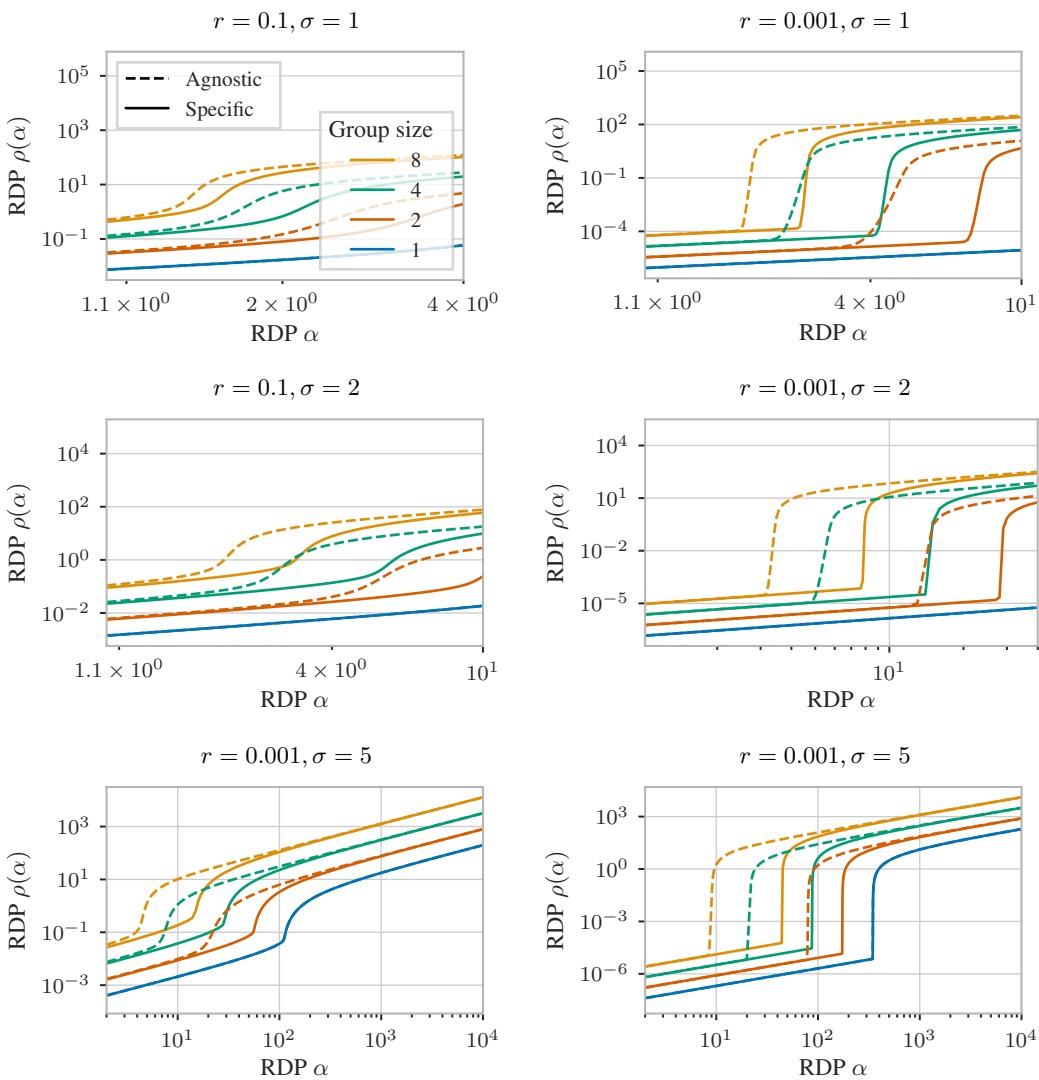

Figure 11: Gaussian mechanisms under Poisson subsampling, with varying standard deviation $\sigma$, subsampling rate $r$, and group size. The mechanism-specific guarantee delays the phase transition from high to low privacy.

### B.1.4 Benefit of conditioning

Next, we demonstrate the benefit of coupling multiple conditional distributions over coupling just two subsampling distributions in deriving amplification guarantees. Specifically, we evaluate Proposition H.7, which can be derived via Theorem 3.3, i.e., optimal transport without conditioning. Note that, unlike with the guarantees of Wang et al. [28], this bound depends on both the dataset size and the batch size, not just their ratio. We make the following observations: For large $\alpha$ Proposition H.7 converges to the baseline. Furthermore, this approach never outperforms the baseline for group size 1. Neither does it outperform the baseline for ratios $q / L \in \{0.01, 0.001\}$. But, for $q / L = 0.1$, there is a sweet spot of alphas between $10^1$ and $10^2$ in which it offers stronger guarantees. This further reinforces our claim that there is a benefit to treating group privacy and amplification jointly. Furthermore, we observe that in the case of $q = 1$, Proposition H.7 outperforms the baseline for large alpha, since it can capture that one cannot possibly include more than one substituted element in a singleton batch.

Overall, we can conclude that there is a benefit to jointly analyzing group privacy and subsampling in this manner, but that optimal transport without conditioning is not sufficient for tightly analyzing such complicated scenarios.

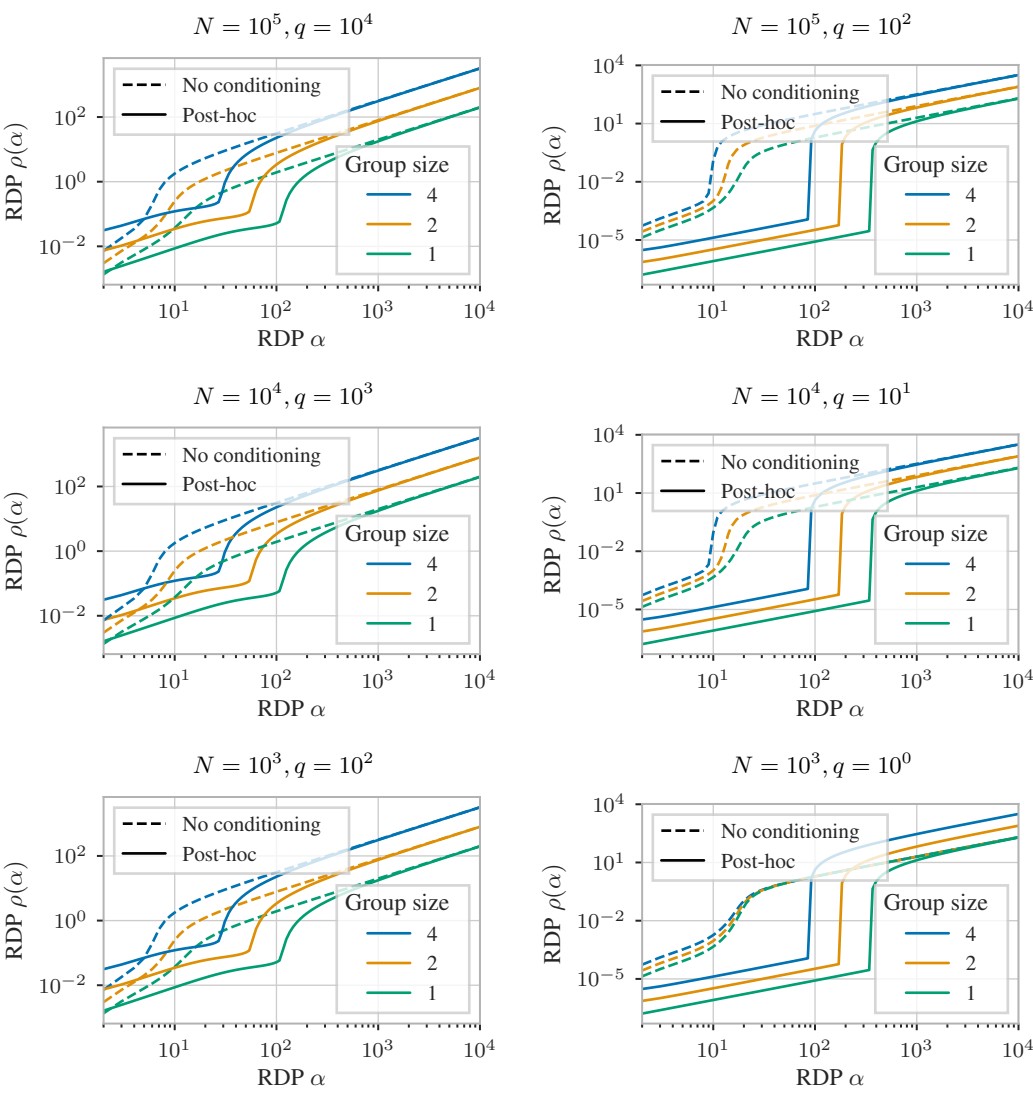

Figure 12: Proposition H.7 derived from Theorem 3.3 applied to Gaussian mechanism ($\sigma = 5.0$) under sampling without replacement for varying dataset size $N$, batch size $q$, and group size. Optimal transport without conditioning does not always improve upon the baseline.

### B.1.5 Subsampling with replacement

The previous experiments demonstrated that mechanism-specific analysis can improve upon non-tight mechanism-agnostic and – in the group privacy setting – tight mechanism-agnostic analysis. In the following, we demonstrate that mechanism-specific bounds can improve upon tight mechanism-agnostic bounds even for single-element relations, i.e., group size 1. To this end, we consider subsampling with replacement for Gaussian mechanisms under the substitution relation, comparing Theorem L.5 to Theorem 10 of Balle et al. [15].

Fig. 13 shows the resultant privacy profiles for standard deviation $\sigma = 1$, dataset size $N = 100$, and batch size $q = 8$. For $\varepsilon \geq 2$, the mechanism-specific bound is more than an order of magnitude smaller. Intuitively, this gap can be explained similarly to the gap in the group privacy setting: The single substituted element can be sampled 0, 1, 2, or up to $q$ times, with each case causing different levels of privacy leakage. Mechanism-agnostic bounds rely on a binary partitioning of the event space, which is not sufficient for capturing this granular behavior (recall Fig. 2.

Note that due to the relaxations of distance constraints we performed in deriving Theorem L.5, the mechanism-specific bound is not tight. A tight bound might lead to an even larger gap. Thus, this experiments further reinforces that there is a qualitative difference between mechanism-specific and mechanism-agnostic tightness.

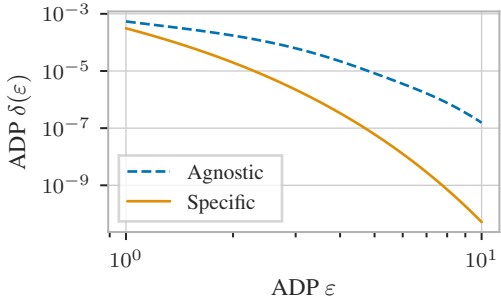

Figure 13: Gaussian mechanism ($\sigma = 1$) under sampling with replacement for single-element substitutions, dataset size $N = 100$, and batch size $q = 8$. Using the mechanism-specific bound results in stronger privacy guarantees.

## B.2 Post-hoc and tight group privacy

### B.2.1 Single-iteration ADP

In Figs. 14 to 15 we repeat the experiment from Fig. 5 for other subsampling rates $r$, standard deviations $\sigma$, and mechanisms. That is, we compare our tight mechanism-specific guarantees to post-hoc use of the group privacy property. For all mechanisms, the tight mechanism-specific analysis yields stronger privacy guarantees than the baseline – particularly for large group sizes. However, we interestingly see that with increasing base mechanisms noise level and decreasing subsampling rate, the post-hoc bound converges towards the tight bound.

**Gaussian mechanism**

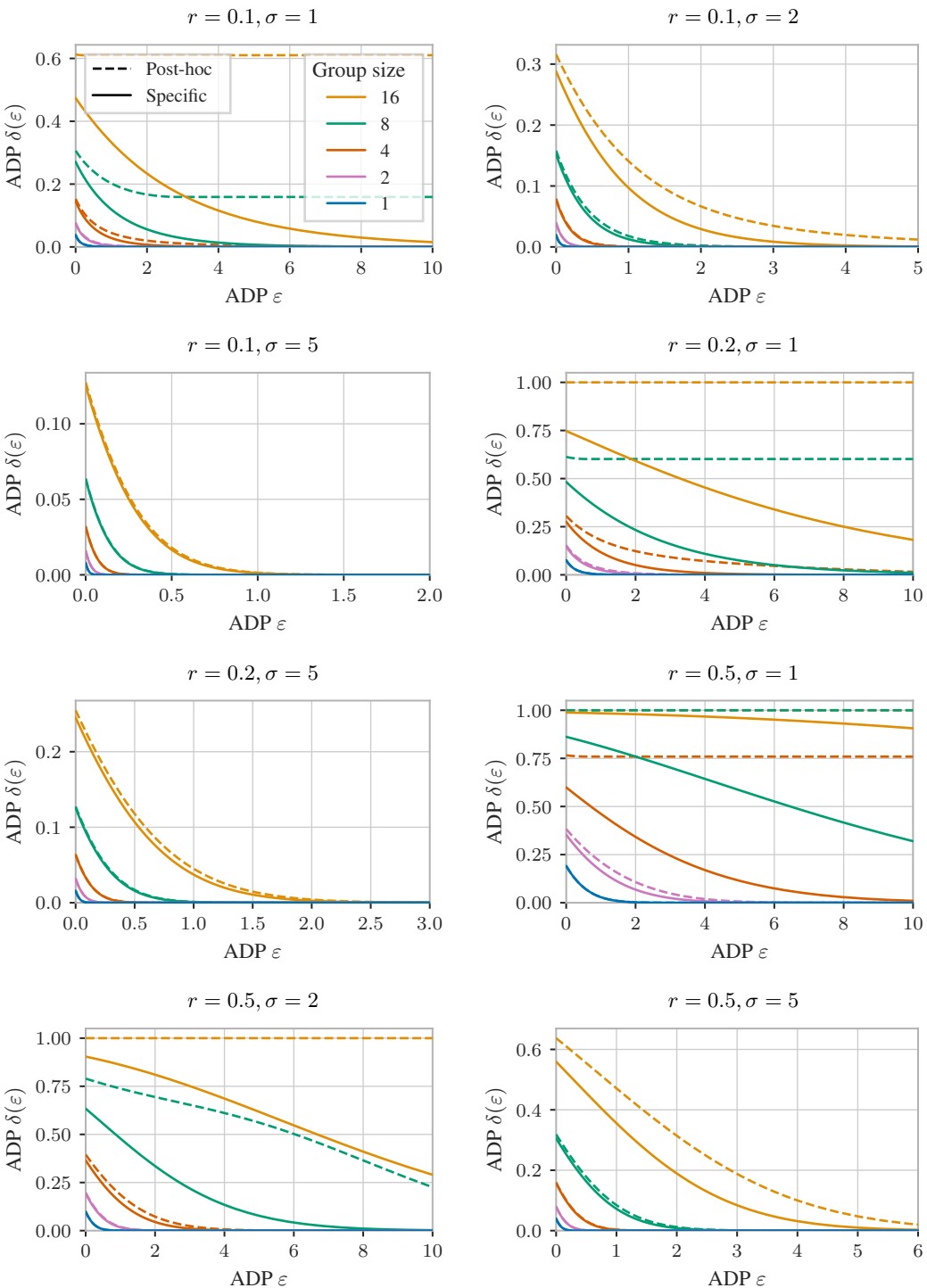

Figure 14: Gaussian mechanisms under Poisson subsampling, with varying standard deviation $\sigma$, subsampling rate $r$, and group size. Analyzing group privacy and subsampling jointly instead of in a post-hoc manner offers stronger guarantees.

**Laplace mechanism**

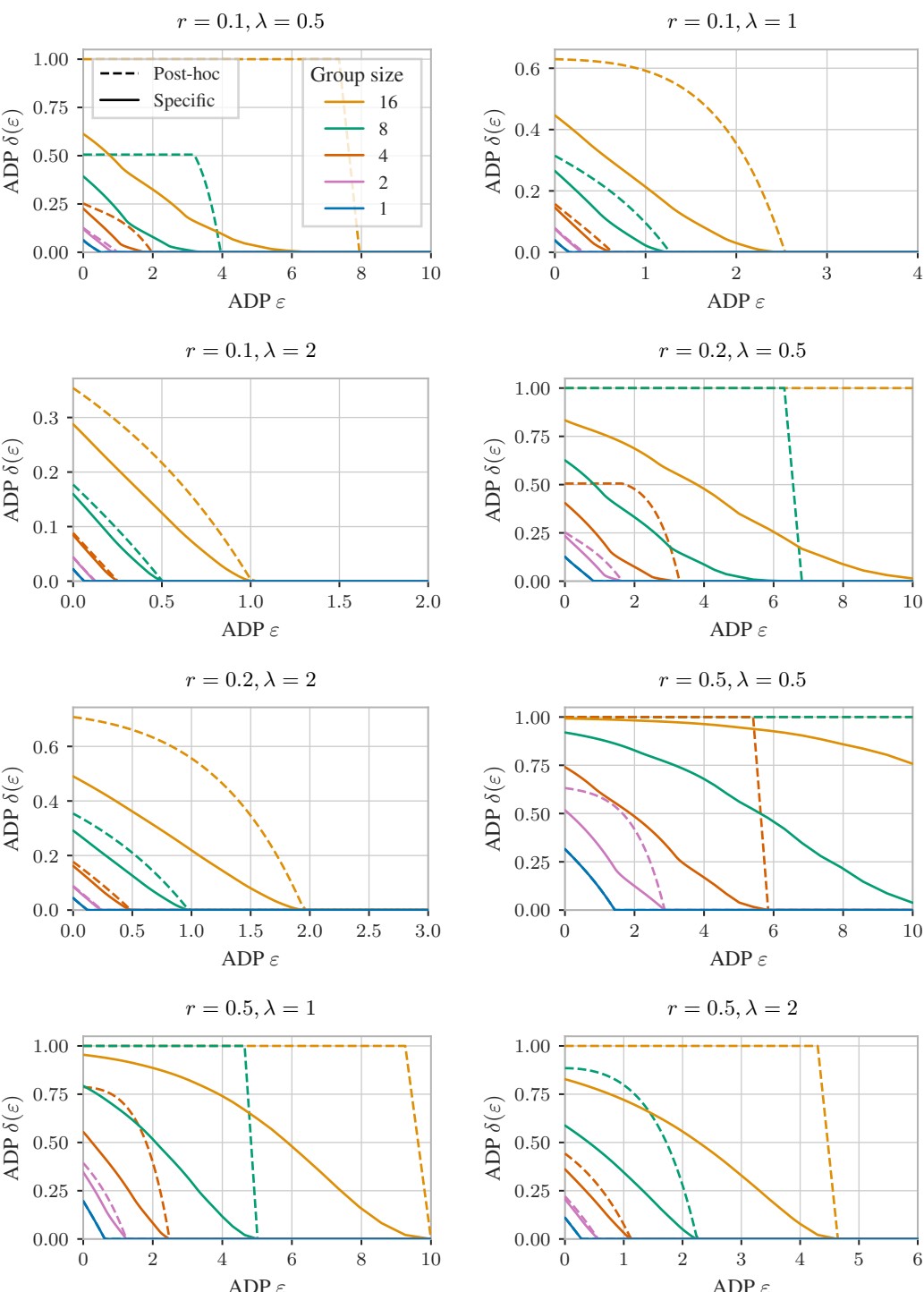

Figure 15: Laplace mechanisms under Poisson subsampling, with varying scale $\lambda$, subsampling rate $r$, and group size. Analyzing group privacy and subsampling jointly instead of in a post-hoc manner offers stronger guarantees.

**Randomized response mechanism**

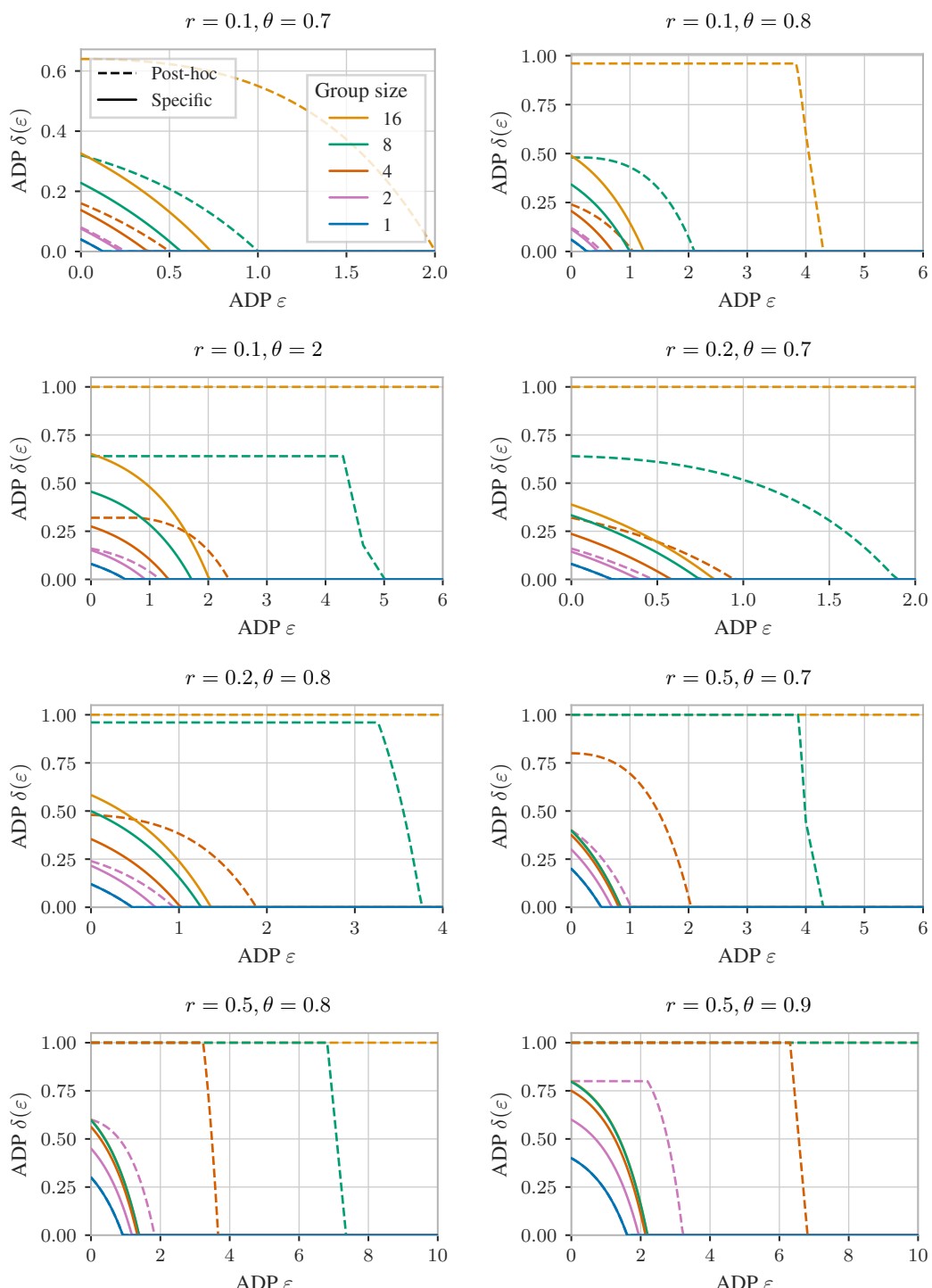

Figure 16: Randomized response mechanisms under Poisson subsampling, with varying true response probability $\theta$, subsampling rate $r$, and group size. Analyzing group privacy and subsampling jointly instead of in a post-hoc manner offers stronger guarantees.

### B.2.2 Single-iteration RDP

In Fig. 17 we repeat our comparison of tight mechanism-specific group privacy amplification and post-hoc group privacy for RDP instead of ADP. The tight analysis yields stronger guarantees and delays the phase transition from a high- to a low-privacy regime. However, as with ADP, the post-hoc bound can be a good upper bound for small subsampling rates and very private base mechanisms.

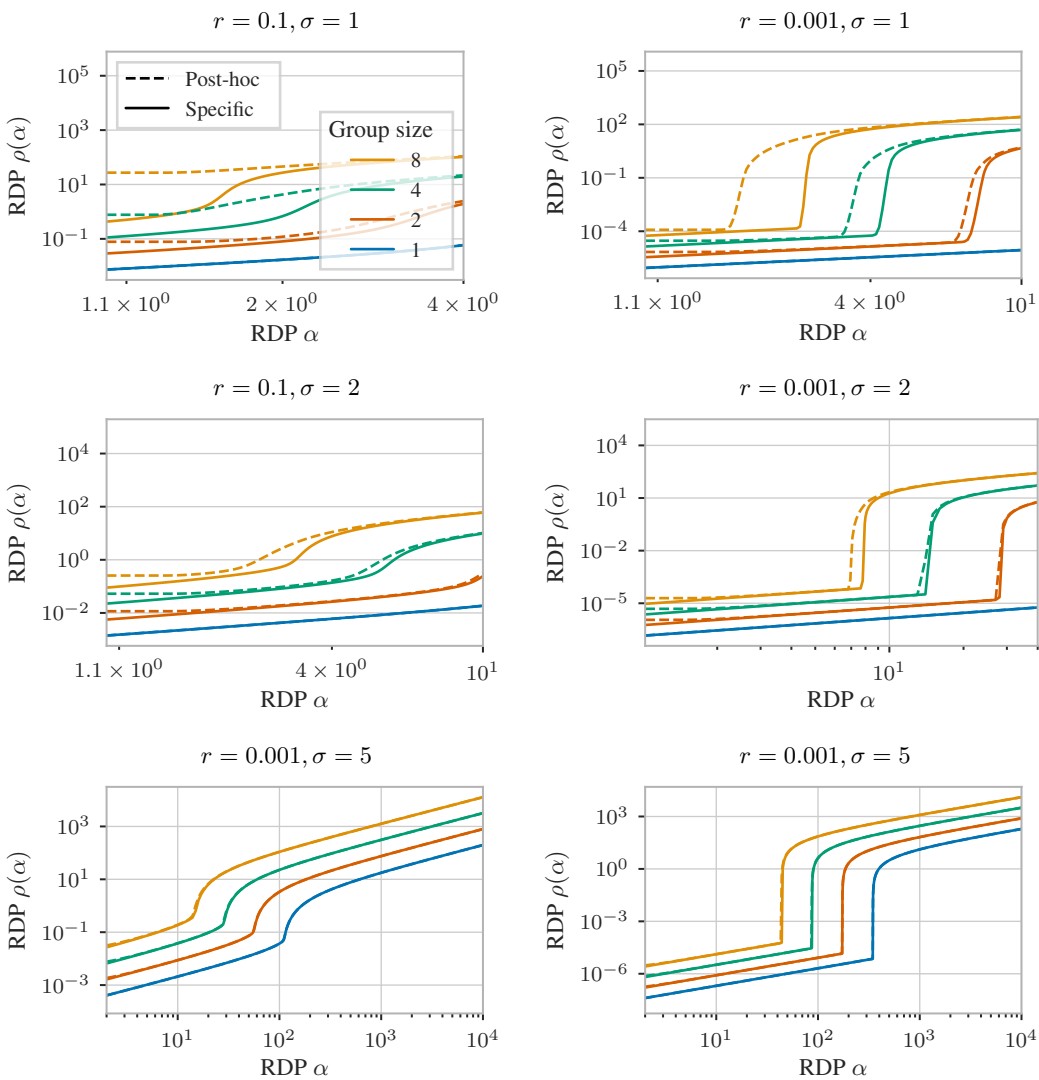

Figure 17: Gaussian mechanisms under Poisson subsampling, with varying standard deviation $\sigma$, subsampling rate $r$, and group size. Analyzing group privacy and subsampling jointly instead of in a post-hoc manner delays the phase transition from high to low privacy. For very private base mechanisms and small subsampling rates, the baseline is nevertheless a good upper bound.

### B.2.3 PLD Accounting

In Figs. 18 to 21 we repeat our comparison of tight mechanism-specific and post-hoc group privacy amplification guarantees under composition from. We consider different combinations of subsampling rate $r$, standard deviation $\sigma$, privacy parameter $\varepsilon$, as well as Laplace mechanisms. Even in high privacy scenarios, where the mechanism-specific and post-hoc analysis yield similar results on a single-iteration level, the mechanism-specific guarantees are significantly stronger under composition. Specifically, the post-hoc analysis diverges to much larger $\delta$ after some number of iterations. In settings with moderate privacy (e.g. $\sigma = 1$) or large group sizes (e.g. 16), the baseline diverges after less than 100 iterations.

**Gaussian mechanism ($r = 0.001$)**

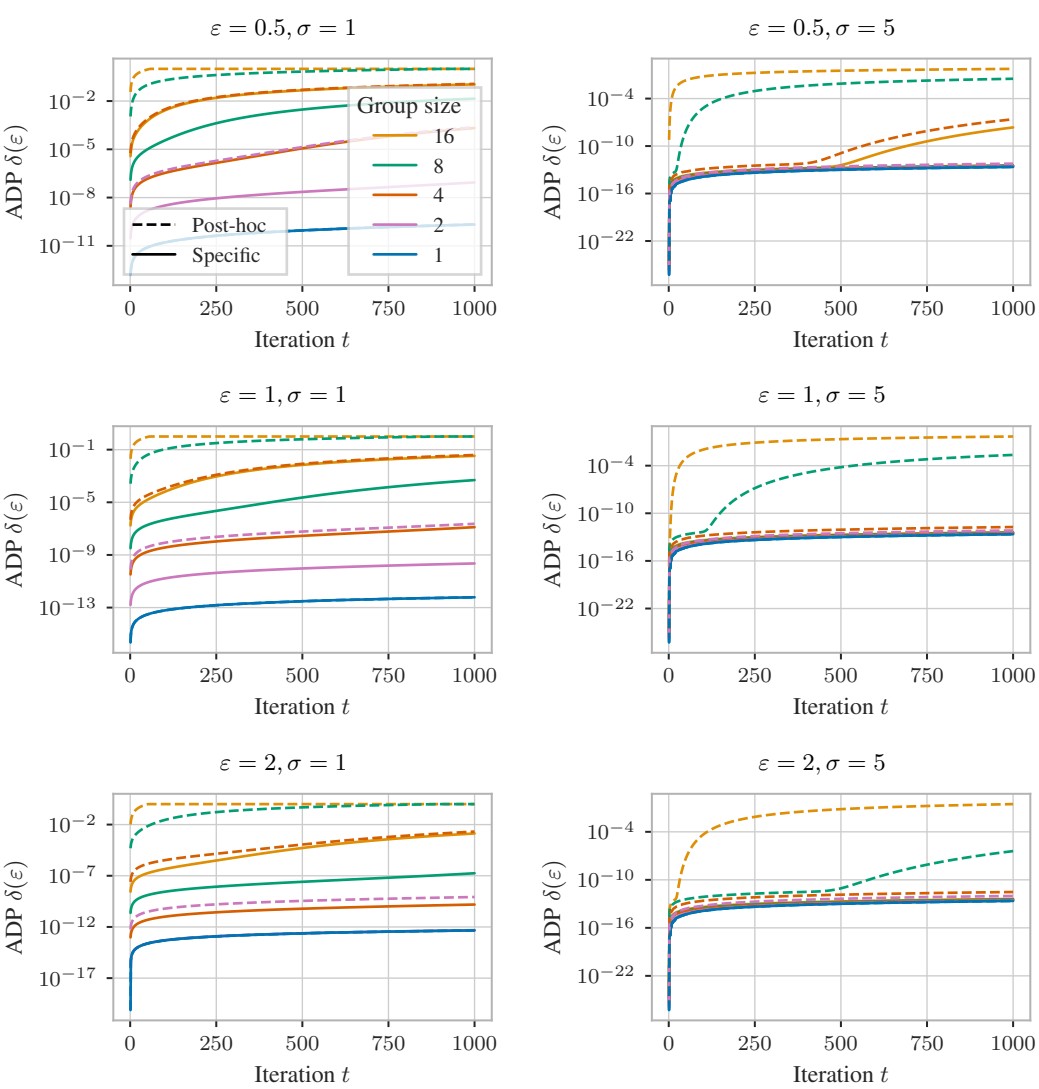

Figure 18: Self-composed Gaussian mechanisms under Poisson subsampling with subsampling rate $r = 0.001$ and varying standard deviation $\sigma$, privacy parameter $\varepsilon$ and group size. For sufficiently large group sizes, the post-hoc analysis divergence from the tight mechanism-specific guarantee within less than 1000 iterations.

**Gaussian mechanism ($r = 0.01$)**

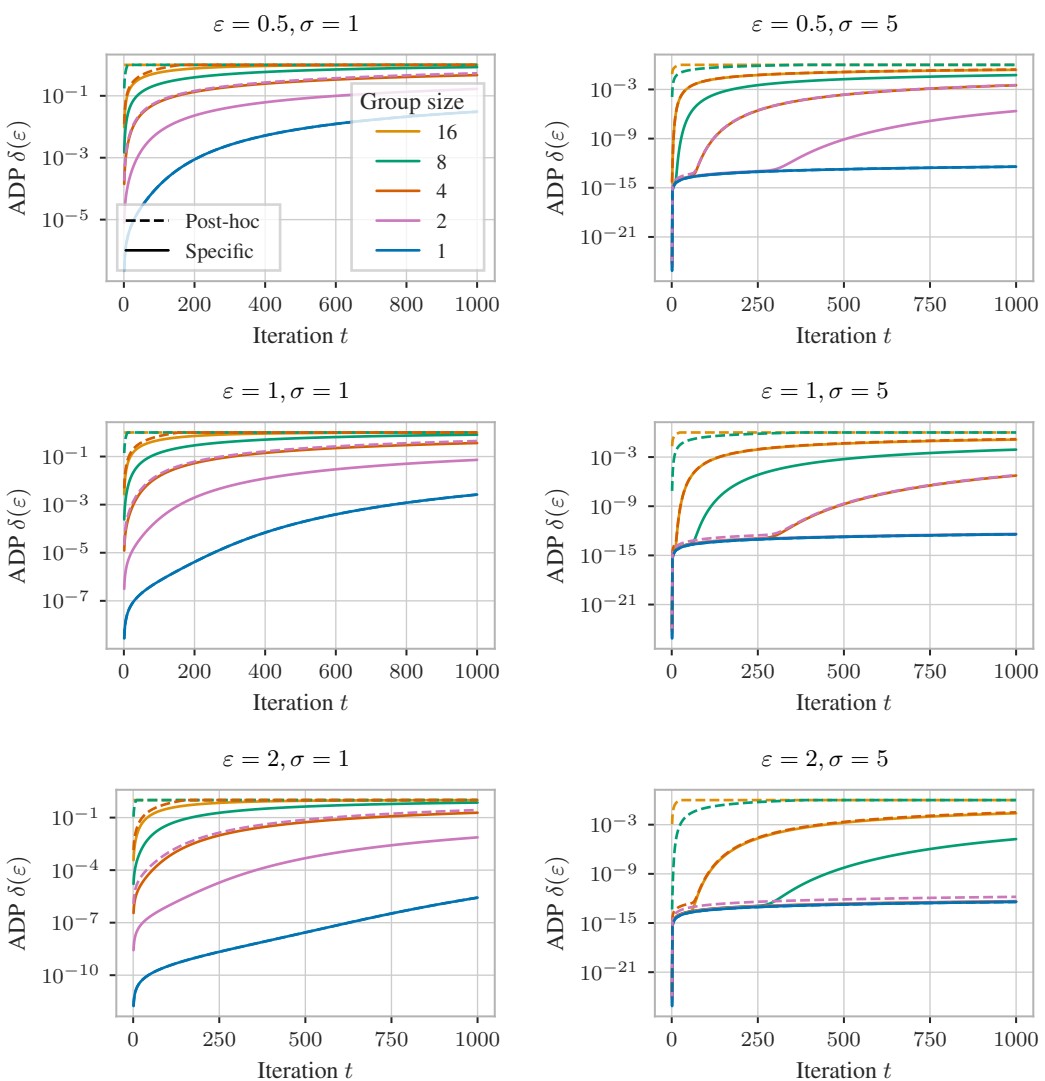

Figure 19: Self-composed Gaussian mechanisms under Poisson subsampling with subsampling rate $r = 0.01$ and varying standard deviation $\sigma$, privacy parameter $\varepsilon$ and group size. For sufficiently large group sizes, the post-hoc analysis divergence from the tight mechanism-specific guarantee within less than 1000 iterations.

**Laplace mechanism ($r = 0.001$)**

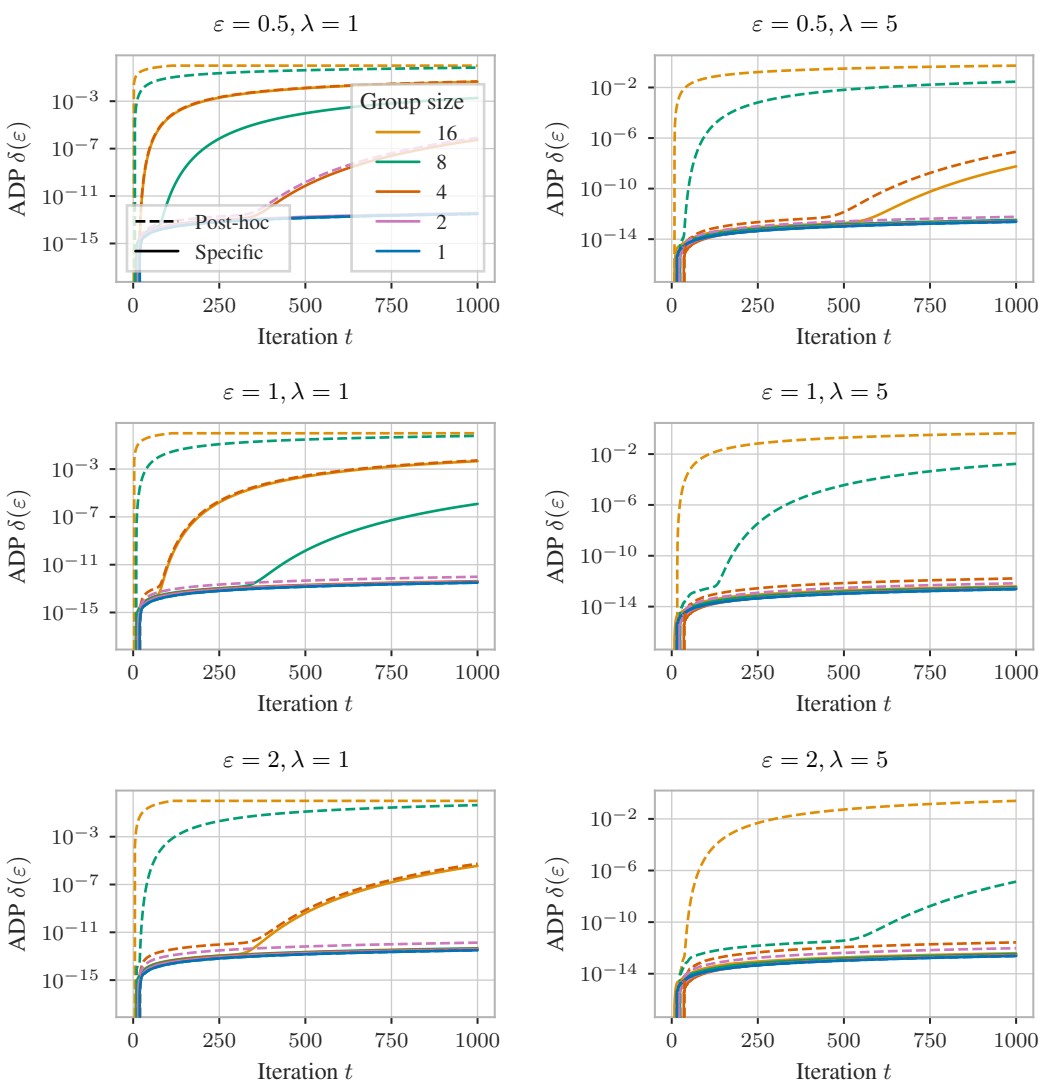

Figure 20: Self-composed Laplace mechanisms under Poisson subsampling with subsampling rate $r = 0.001$ and varying scale $\lambda$, privacy parameter $\varepsilon$ and group size. For sufficiently large group sizes, the post-hoc analysis divergence from the tight mechanism-specific guarantee within less than 1000 iterations.

**Laplace mechanism ($r = 0.01$)**

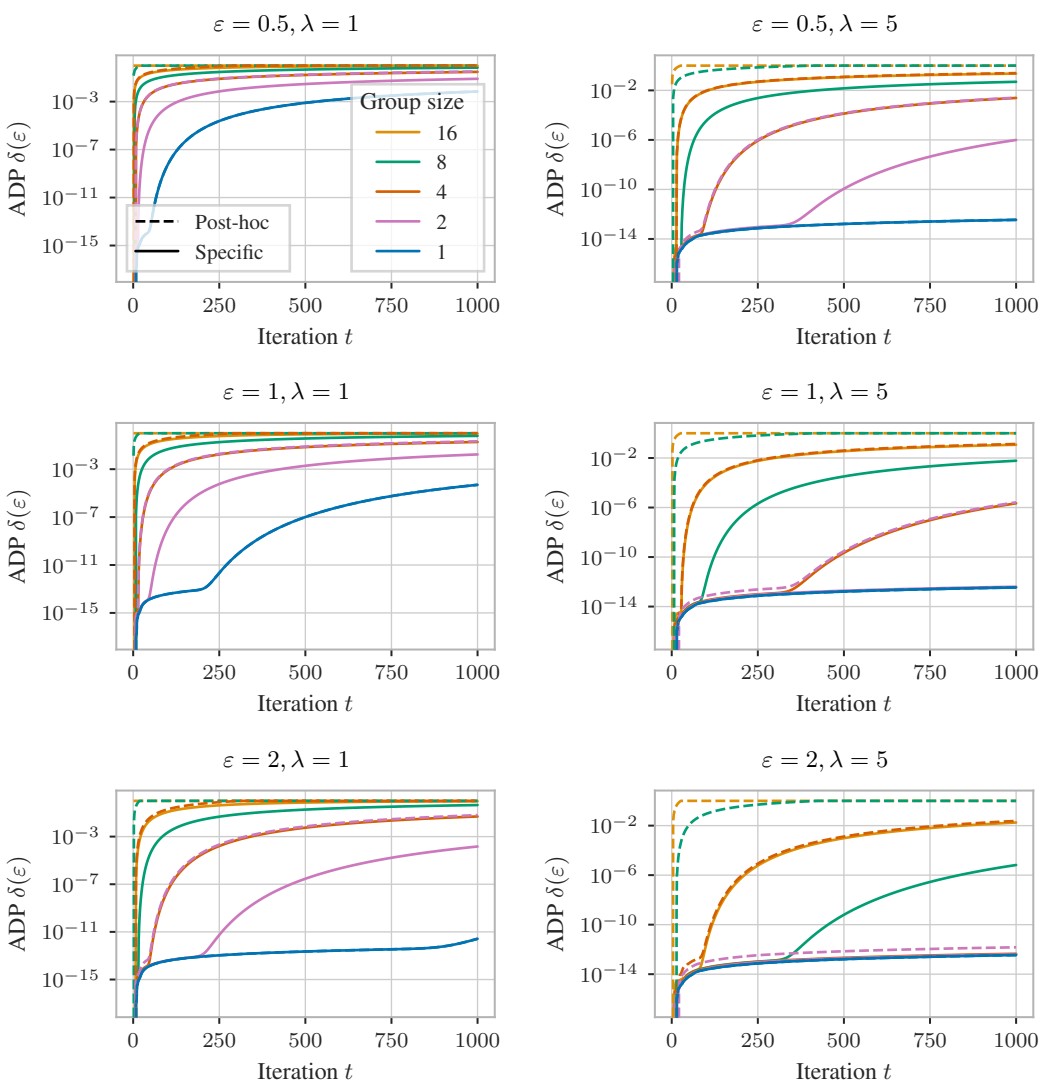

Figure 21: Self-composed Laplace mechanisms under Poisson subsampling with subsampling rate $r = 0.001$ and varying scale $\lambda$, privacy parameter $\varepsilon$ and group size. For sufficiently large group sizes, the post-hoc analysis divergence from the tight mechanism-specific guarantee within less than 1000 iterations.

### B.2.4 RDP accounting

Finally, we compare tight group privacy amplification to the post-hoc approach for RDP accounting. We begin with the Gaussian mechanism and subsampling rate $r = 0.001$ in Fig. 22. Our bound offers stronger group privacy guarantees for $\sigma = 1.0$, but both results are almost identical for very private base mechanisms with $\sigma = 5.0$. We suspect that this is because the phase transition from high to low privacy (recall Fig. 17), which the tight guarantee delays, gets shifted to very high $\alpha$ regions that are not useful for conversion to $(\varepsilon, \delta)$-DP.

Our observations for randomized response mechanisms (see Fig. 23 are also consistent with earlier results: For moderate subsampling rates $r = 0.1$, our guarantees are stronger, particularly for group size 8 and large numbers of iterations. But, when decreasing the subsampling rate to 0.001, both methods are almost identical. Nevertheless, group privacy amplification can demonstrably improve upon a direct combination of independently derived group privacy and amplification guarantees.

Note that these observations are mostly of theoretic interest. In practice, one would use PLD accounting, which tightly characterizes the composed mechanism's privacy leakage, instead of the looser RDP accounting. As shown in Figs. 18 to 21, the tight mechanisms-specific analysis drastically outperforms the post-hoc analysis for PLD accounting.

**Gaussian mechanism**

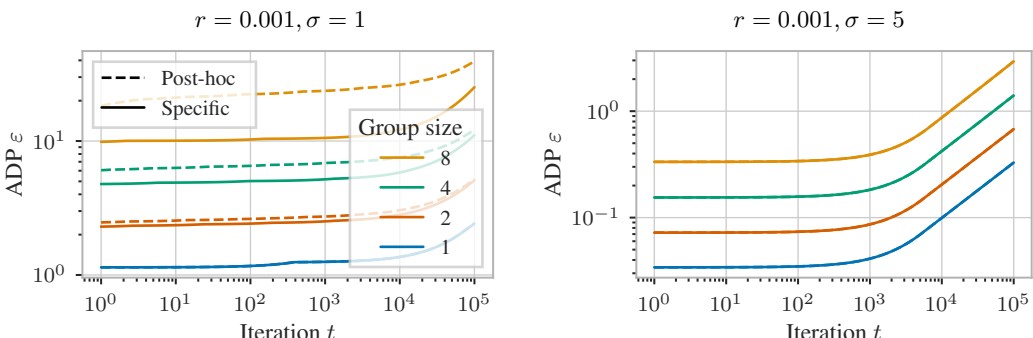

Figure 22: Self-composed Gaussian mechanisms under Poisson subsampling with privacy parameter $\delta = 10^{-8}$, subsampling rate $r = 0.001$, and varying standard deviation $\sigma$ and group size. The tight mechanism-specific analysis yields better privacy guarantees than the post-hoc baseline, except for large $\sigma$ where the baseline is a good upper bound.

**Randomized response mechanism**

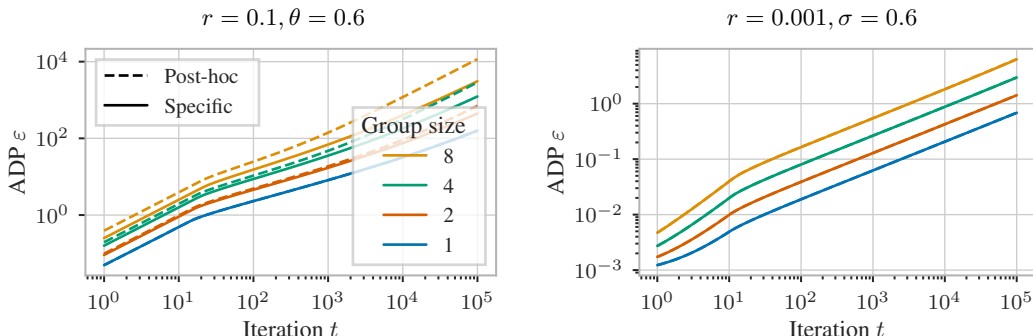

Figure 23: Self-composed randomized response mechanisms under Poisson subsampling with privacy parameter $\delta = 10^{-8}$, true response probability $\theta = 0.6$, and varying subsampling rate $r$ and group size. The tight mechanism-specific analysis yields better privacy guarantees than the post-hoc baseline, except for small $r$ where the baseline is a good upper bound.

### B.2.5 Model utility

The previous results demonstrated that handling group privacy via a tight mechanism-specific analysis allows for a larger number of compositions, i.e., iterations at a given privacy budget than post-hoc use of the group privacy property. In the following, we demonstrate that this increased number of iterations can in fact result in increased utility for group-private machine learning models.

We train a convolutional neural network (2 convolution layers with kernel sizes 3 and 32 / 64 channels, followed by two linear layers with hidden dimension 128) for image classification on MNIST (55000 training, 5000 validation, 10000 test samples). We set the gradient clipping norm of DP-SGD [6] to $C = 10^{-4}$, the Gaussian noise standard deviation to $0.6 \cdot C$, and the subsampling rate to $r = 64 / 55000$. The optimizer is ADAM with learning rate $1e - 3$. If the privacy budget is not used up earlier, training is terminated after 8 epochs, with $\lceil 55000 / 64 \rceil$ iterations per epoch.

Even with a large privacy budget of $\varepsilon = 8$ and $\delta = 1e - 5$, training with the post-hoc privacy analysis needs to terminate after less than 200 iterations. With the mechanism-specific analysis, $\delta$ does not even exceed $1e - 7$ after 8 epochs, i.e., the model could potentially be trained for even more iterations. The resultant validation accuracy's are $79.6\%$ and $91.2\%$, respectively. This showcases the superior privacy-utility trade-off that can be achieved by analyzing subsampling and group privacy jointly.

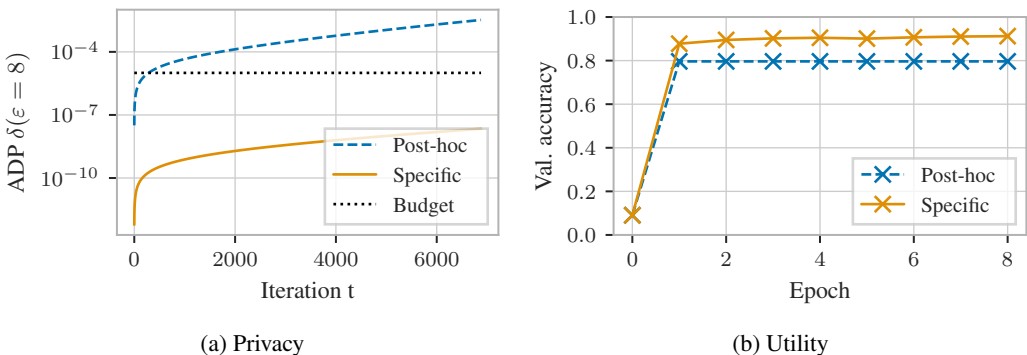

(a) Privacy          (b) Utility

Figure 24: Differentially private training of a 2-layer convolutional network on MNIST with PLD accounting for group size 2. Our tight mechanism-specific analysis allows us to train for significantly more epochs or to terminate training after 8 epochs with less privacy leakage and higher accuracy.

# C   Experimental setup

## C.1   Evaluation of privacy guarantees

In all experiments, we assume that all mechanisms have underlying sensitivity 1 w.r.t. $\ell_\infty$ (randomized response), $\ell_1$ (Laplace), or $\ell_2$ (Gaussian) output norms. We thus only specify noise parameters $\sigma$ or $\lambda$, rather than the ratio of $\sigma$ or $\lambda$ and the sensitivities.

### C.1.1   Single-iteration approximate differential privacy

**Privacy parameters.** We evaluate all guarantees for 121 equidistant values of $\varepsilon$ in $[0, 4]$ and 121 logspace-equidistant values in $10^{-3}, 10^1$, i.e., $\{10^x \mid x \in \{-3, -3 + \frac{4}{120}, \ldots, 1\}\}$. We clip values of $\delta(\varepsilon)$ that are larger than 1 to $[0, 1]$. For our baselines, we enforce that $\delta(\varepsilon)$ is monotonically decreasing by taking a running minimum from larger to smaller $\varepsilon$ (this may improve but never worsens the baselines).

**Mechanism-agnostic group privacy baseline.** As our mechanism-agnostic group privacy baselines, we use Proposition N.1. We take the maximum over the two cases $(K_+ = K, K_- = 0)$ and $(K_+ = 0, K_- = K)$, where $K$ is the group size. For Gaussian and Laplace mechanisms, we evaluate group privacy profiles $\delta_k(\varepsilon)$ by determining $\delta(\varepsilon)$ for the same mechanism with sensitivity $k$. For randomized response, we let $\delta_k = \delta_1$. This is referred to to as "white-box" group privacy in [15]). We evaluate this baseline analytically.

**Post-hoc group privacy baseline.** For our post-hoc baseline, we combine the tight Poisson subsampling guarantee for insertion/removal from [15] with the group privacy property of approximate differential privacy (see Appendix C.1.4 below). We evaluate this baseline analytically.

**Tight mechanism-specific group privacy.** In all ADP figures "specific" refers to our tight group privacy guarantees for Gaussian (Theorem 3.8), Laplace (Theorem M.2), or randomized response mechanisms (Theorem M.3). We take the maximum over all $K_+, K_- \in \mathbf{N}_0$ with $K_+ + K_- = K$, where $K$ is the group size. To evaluate the bounds for Gaussian and Laplace mechanisms, we pessimistically invert the privacy loss of dominating pairs $P, Q$ via binary search (see details in Appendix M.4), as implemented in the `dp_accounting` library [45]. We set the precision to $10^{-6}$, i.e., find the global optimum over multiples of $10^{-6}$. The bound for the randomized response mechanism is evaluated analytically in $\mathcal{O}(1)$.

### C.1.2   Single-iteration Rényi differential privacy

**Privacy parameters**. We evaluate all guarantees for $\alpha \in \{2, 3, \ldots, 1000\}$, as well as 121 logspace-equidistant values in $[0, 10^4]$ rounded to the next smallest integer (without 0 or 1), and 121 logspace-equidistance values in $(1, 10)$. For our baselines, we enforce that $\rho(\alpha)$ is monotonically increasing by taking a running minimum from smaller to larger $\alpha$ (this may improve but never worsens the baselines).

**Mechanism-agnostic group privacy baseline.** As our mechanism-agnostic group privacy baselines, we use Proposition N.1. We take the maximum over the two cases $(K_+ = K, K_- = 0)$ and $(K_+ = 0, K_- = K)$, where $K$ is the group size. For integer $\alpha$, we use the binomial expansion in Eq. (33) (without factor 2). For Gaussian and Laplace mechanisms, we evaluate group privacy profiles $\zeta_k(\alpha)$ by determining $\zeta(\alpha)$ for the same mechanism with sensitivity $k$. For randomized response, we let $\zeta_k = \zeta_1$. For continuous $\alpha$, we apply $\tanh$-sinh-quadrature with 50 digits of decimal precision to Eq. (33)

**Post-hoc group privacy baseline.** For our post-hoc baselines, we combine either tight Poisson subsampling guarantee for insertion/removal from [30] or the subsampling without replacement guarantee for insertion/removal from [28] with the group privacy property of approximate differential privacy (see Appendix C.1.4 below). For Poisson subsampling and integer $\alpha$, we use the binomial expansion from [30] (without factor 2). For continuous $\alpha$, we apply $\tanh$-sinh-quadrature with 50 digits of decimal precision. For subsampling without replacement, we use the improved self-consistency bound (Theorem 27 from [28]), which we evaluate via $\tanh$-sinh-quadrature with 50 digits of decimal precision due to the intractability of the nested binomial expansions for larger $\alpha$.

**Tight mechanism-specific group privacy.** In all RDP figures "specific" refers to our tight group privacy guarantees for Gaussian (Theorem 3.8), Laplace (Theorem M.2), or randomized response

mechanisms (Theorem M.3). We take the maximum over all $K_+, K_- \in \mathbf{N}_0$ with $K_+ + K_- = K$, where $K$ is the group size. To evaluate the bounds for Gaussian and Laplace mechanisms, we use $\tanh$-sinh-quadrature with 50 digits of decimal precision (see discussion in Appendix M.4). The bound for the randomized response mechanism is evaluated analytically in $\mathcal{O}(1)$.

### C.1.3 Privacy accounting

**RDP accounting.** For RDP accounting, we simply evaluate the single-iteration guarantees as described in Appendix C.1.2. We then multiply the $\rho(\alpha)$ with the number of iterations, apply the improved RDP-to-ADP formula from [31], and take the minimum over all obtained values. For the post-hoc baseline, we apply the group privacy property before composition (which is equivalent to applying it after composition, due to additivity of composition).

**PLD accounting.** For PLD accounting with dominating pairs, we use the implementation of the `dp_accounting` library [45]. To quantize the privacy loss distribution, we use "connect the dots" [20] with pessimistic estimates, a discretization interval size of $10^{-3}$, and truncation of $e^{-50}$ of the probability mass. During composition, we truncate $10^{-15}$ of the tail mass. For the post-hoc baseline, we apply the group privacy property after composition.

### C.1.4 Post-hoc group privacy

**ADP.** For RDP, we use the following result from the proof of Lemma 2.2 in [23], which provides tigher guarantees than Lemma 2.2 itself: Let $M$ be $(\varepsilon, \delta)$-DP under neighboring relation $\simeq_{\mathbb{X}}$. Then, $M$ is $(\varepsilon', \delta')$-DP with $\varepsilon' = \varepsilon \cdot K$ and $\delta' = \sum_{k=0}^{K-1} e^{k \cdot \varepsilon} \cdot \delta$.

**RDP.** For RDP, we use the following result from Corollary 4 of [7], which provides tighter guarantees than the upper bound in their Proposition 2. Let $D_\alpha$ be $\log(\Lambda_\alpha) / (\alpha - 1)$. This $D_\alpha$ fulfills the following triangle inequality:

$$D_\alpha(p||q) \leq \frac{\alpha - \frac{1}{2}}{\alpha - 1} D_{2\alpha}(p||r) + \frac{\alpha}{\alpha - 1} D_{2\alpha - 1}(r||q).$$

We recursively apply this bound $\log_2(K)$ times when evaluating the group privacy of our baselines for groups of size $K$.

### C.2 Computational resources

We conduct all experiments on a set of Xeon E5-2630 v4 CPUs @ 2.2 GHz.

We use one worker and job per subsampling theorem, base mechanism noise level, subsampling rate, and combination of group privacy parameters $K_-, K_+$. Per job, we allocate 2 CPU cores, 4 GB and 10 minutes of runtime (in practice, most jobs were completed in a few seconds). In total, we ran 13060 such jobs for ADP, 9731 for RDP, and 984 for RDP accounting. We estimate that over the course of the full research project twice as many jobs were executed.

### C.3 Assets and licenses

To perform high-precision quadrature for RDP guarantees, we use the $\tanh$-sinh quadrature implementation from the `mpmath` library (version 1.3.0.), which is available under the BSD-3-Clause license at `https://github.com/mpmath/mpmath`.

For PLD accounting and evaluation of ADP guarantees via bisection, we use and extend the `dp_accounting` library [45] (commit `0b109e959470c43e9f177d5411603b70a56cdc7a`), which is available under Apache-2.0 license at `https://github.com/google/differential-privacy`

For conversion from RDP to ADP guarantees, we use the `get_privacy_spent` method implemented in the `Opacus` library [73] (version 1.4.1), which is available under Apache-2.0 license at `https://github.com/pytorch/opacus`.

An implementation will be made available at https://cs.cit.tum.de/daml/group-amplification.

# D General setting and definitions

In the following, we generalize some of the definitions introduced in Sections 2 and 3 to their measure-theoretic equivalents. The purpose of this generalization is to handle both continuous and discrete spaces, base mechanisms, and subsampling schemes, without having to make constant case distinctions.

We use these more general definitions throughout the remaining appendix sections. The theoretic results presented in Section 3 follow as special cases.

## D.1 Spaces

Unlike before, we assume that dataset space $\mathbb{X}$ is some arbitrary space, whose elements do not have to be sets. Instead, datasets $x \in \mathbb{X}$ can also be graphs, sequences or any other data collection. We further assume that the batch space is a measurable space $(\mathbb{Y}, \mathcal{Y})$ with $\sigma$-algebra $\mathcal{Y}$. For example, $\mathbb{Y}$ can be composed of subsets, subgraphs, or subsequences. We also assume the output space to be a measurable space $(\mathbb{Z}, \mathcal{Z})$. Finally, we assume the existence of some measure $\lambda$ on the output space, such as the Lebesgue measure for continuous $\mathbb{Z}$ or the counting measure $\#$ for discrete $\mathbb{Z}$.

Whenever we consider Poisson subsampling, we assume that $\mathbb{X} \subseteq \mathcal{P}(\mathbb{A})$, $\mathbb{Y} = \{y \subseteq x \mid x \in \mathbb{X}\}$, $\mathcal{Y} = \mathcal{P}(\mathbb{Y})$, where $\mathbb{A}$ is some discrete, finite set and $\mathcal{P}(\cdot)$ is the powerset.

Whenever we consider subsampling without replacement with batch size $q$, we assume that $\mathbb{X} \subseteq \{x \in \mathcal{P}(\mathbb{A}) \mid |x| > q\}$, $\mathbb{Y} = \{y \subseteq x \mid x \in \mathbb{X}, |y| = q\}$, $\mathcal{Y} = \mathcal{P}(\mathbb{Y})$, where $\mathbb{A}$ is some discrete, finite set and $\mathcal{P}(\cdot)$ is the powerset.

## D.2 Neighboring relations

For set-valued datasets and batches, the insertion/removal relation and the substitution relation are formally defined as follows:

**Definition D.1.** Sets $x, x' \in \mathbb{X}$ are related by the insertion/removal relation $(x \simeq_{\pm} x')$ if $x \neq x'$ and there is some $a$ such that $x' = x \cup \{a\}$ or $x' = x \setminus \{a\}$.

**Definition D.2.** Sets $x, x' \in \mathbb{X}$ are related by the substitution relation $(x \simeq_{\Delta} x')$ if there is some $a \in x$ and $a' \notin x$ such that $x' = x \setminus \{a\} \cup \{a'\}$.

In our general setting, we also allow non-symmetric neighboring relations.

## D.3 Mechanisms and subsampling schemes

As before, the term "mechanism" refers to random functions that map to the output space $(\boldsymbol{Z}, \mathcal{Z})$. Formally, a random function $M : \mathbb{X} \to \mathbb{Z}$ is a family of random variables indexed by elements of $\mathbb{X}$.

**Definition D.3.** A random function $M : \mathbb{X} \to \mathbb{Z}$ is a function $M : \mathbb{X} \times \Omega \to \mathbb{Z}$, where $(\Omega, \mathcal{F}, P)$ is some probability space and all $M(x, \cdot) : \omega \mapsto M(x, \omega)$ are measurable.

We write $P_{M_x} : \mathcal{Z} \to [0, 1]$ for the distribution of random variable $M(x)$. We further assume that each $P_{M_x}$ is absolutely continuous w.r.t. the aforementioned output measure $\lambda$, i.e., $\forall x \in \mathbb{X} : P_{M_x} \ll \lambda$, and write $m_x : \mathbb{Z} \to \mathbb{R}_+$ for the corresponding Radon–Nikodym derivative $\mathrm{d}P_{M_x} / \mathrm{d}\lambda$. For example, when the output space $(\mathbb{Z}, \mathcal{Z})$ is continuous and $\lambda$ is the Lebesgue measure, then $m_x$ is the density.

Similarly, we define our base mechanism to be a random function $B : \mathbb{Y} \to \mathbb{Z}$ and write $P_{B_y} : \mathcal{Z} \to [0, 1]$ for the distribution of base mechanism outputs given a batch $y \in \mathbb{Y}$. We assume that each $P_{B_y}$ is absolutely continuous w.r.t. output measure $\lambda$ and write $b_y : \mathbb{Z} \to \mathbb{R}_+$ for $\mathrm{d}P_{B_y} / \mathrm{d}\lambda$.

Finally, we define our subsampling scheme to be a random function $S : \mathbb{X} \to \mathbb{Z}$ and write $P_{S_x} : \mathcal{Y} \to [0, 1]$ for the distribution of batches given a dataset $x \in \mathbb{X}$. We generally do not require $P_{S_x}$ to be absolutely continuous w.r.t. some other measure. When $P_{S_x}$ is absolutely continuous w.r.t. counting measure $\#$, we write $s_x : \mathbb{Y} \to [0, 1]$ for the corresponding probability mass function $\mathrm{d}P_{S_x} / \mathrm{d}\#$.

In particular, Poisson subsampling and subsampling without replacement are defined as follows:

**Definition D.4.** Poisson subsampling with rate $r \in [0, 1]$ has probability mass function $s_x(y) = r^{|y|}(1 - r)^{|x| - |y|}$ for batches $y \subseteq x$.

**Definition D.5.** Subsampling without replacement with batch size $q$ has probability mass function $s_x(y) = \binom{|x|}{q}^{-1}$ for batches $y \subseteq x$ with $|y| = q$.

As before, our goal is to provide privacy guarantees for subsampled mechanisms $M = B \circ S$. Similar to our discussion in Section 2, its distribution $P_{M_x}$ given a dataset $x \in \mathbb{X}$ is a mixture[3] with

$$m_x(z) = \int_{\mathbb{Y}} b_y(z) \, dP_{S_x}(y). \tag{5}$$

There is one component per batch $y$ from batch space $\mathbb{Y}$, and the weights depend on subsampling distribution $P_{S_x}$.

## D.4 Differential privacy notions

Since we no longer require the output space to be continuous, we also need to generalize the definitions of approximate differential privacy, Rényi differential privacy, dominating pairs, and the divergences underlying their definition. For this, recall that $\lambda$ is our assumed measure on output space $(\mathbb{Z}, \mathcal{Z})$.

**Definition D.6.** For $\varepsilon \geq 0$, a mechanism $M : \mathbb{X} \to \mathbb{Z}$ is $(\varepsilon, \delta)$-DP under relation $\simeq_{\mathbb{X}}$ if $\forall x \simeq_{\mathbb{X}} x'$ : $H_{\exp(\varepsilon)}(m_x||m_{x'}) \leq \delta$ and $H_{\exp(\varepsilon)}(m_{x'}||m_x) \leq \delta$ with hockey stick divergence

$$H_\alpha(m_x||m_{x'}) = \int_{\mathbb{Z}} \max\{m_x(z) \, / \, m_{x'}(z) - \alpha, 0\} \cdot m_{x'}(z) \, d\lambda(z). \tag{6}$$

**Definition D.7.** A pair of distributions $(P, Q)$ with $p = dP \, / \, d\lambda$ and $q = dQ \, / \, d\lambda$ is a dominating pair for mechanism $M$ under neighboring relation $\simeq_{\mathbb{X}}$, if $H_\alpha(m_x||m_{x'}) \leq H_\alpha(p||q)$ for all $x \simeq_{\mathbb{X}} x'$ and all $\alpha \geq 0$.

**Definition D.8.** For $\alpha \geq 1$, a mechanism $M : \mathbb{X} \to \mathbb{R}^D$ is $(\alpha, \rho)$-RDP under neighboring relation $\simeq_{\mathbb{X}}$ if $\forall x \simeq_{\mathbb{X}} x' : \log(\Lambda_\alpha(m_x||m_{x'})) \, / \, (\alpha - 1) \leq \rho$ and $\log(\Lambda_\alpha(m_{x'}||m_x)) \, / \, (\alpha - 1) \leq \rho$ with

$$\Lambda_\alpha(m_x||m_{x'}) = \int_{\mathbb{R}^D} m_x(z)^\alpha \cdot m_{x'}(z)^{1-\alpha} \, dz. \tag{7}$$

Importantly, note that $\Lambda_\alpha$ *is not the Rényi divergence* as used in [7]. It is its $\alpha$th moment, i.e., a scaled and exponentiated Rényi divergence. This is why a logarithm and quotient appears in Definition D.8. We use this definition, so that we can simultaneously discuss ADP and RDP without notational clutter.

## D.5 Joint convexity

The joint convexity of $\Psi_\alpha \in \{H_\alpha, \Lambda_\alpha\}$ is not limited to densitites, but also applies to other non-negative Radon–Nikodym derivatives of probability distributions [37, 38]:

**Lemma D.9.** *Consider arbitrary Radon–Nikodym derivatives* $f_1^{(1)}, f_2^{(1)}, f_1^{(2)}, f_2^{(2)} : \mathbb{Z} \to \mathbb{R}_+$ *and weight* $w \in [0, 1]$. *Then,*

$$\Psi_\alpha(wf_1^{(1)} + (1-w)f_2^{(1)}||wf_1^{(2)} + (1-w)f_2^{(2)}) \leq w\Psi_\alpha(f_1^{(1)}||f_1^{(2)}) + (1-w)\Psi_\alpha(f_2^{(1)}||f_2^{(2)}).$$

## D.6 Couplings

As with the discrete, finite-support subsampling distributions we considered in Section 3, the key tool we use for analyzing amplification are couplings. However, we no longer assume the subsampling scheme to always have a mass function. We thus use a more general notion of couplings between distributions instead of couplings between mass functions:

**Definition D.10.** A coupling between probability measures $P_1, \ldots, P_N$ on space $(\mathbb{Y}, \mathcal{Y})$ is a probability measure $\Gamma$ on product space $(\mathbb{Y}, \mathcal{Y})^N$, where the $n$th marginal is $P_n$, i.e., $\Gamma \circ \pi_n^{-1} = P_n$ with projection $\pi_n(\boldsymbol{y}) = y_n$.

Here, $\circ$ is the composition operator and $\pi_n^{-1}$ is the preimage (not necessarily the inverse) of the projection function. As before, when considering a coupling between two distributions $P_1, P_2$, the value $\Gamma(T, R)$ specifies for all events $T, R \in \mathcal{Y}$ how much probability should be transported from $P_1(T)$ to $P_2(R)$ to transform $P_1$ into $P_2$.

---

[3]assuming that $(y, Z) \mapsto P_{B_y}(Z)$ is a valid Markov kernel

# E   Proof of optimal transport bounds

## E.1   Proof of Theorem 3.3

In the following, we show a more general statement for the general setting introduced in Appendix D. Theorem 3.3 immediately follows in the special case where batch space $\mathbb{Y}$ is finite and discrete, and the subsampling distribution has a probability mass function $s_x$.

**Theorem E.1.** *Consider a subsampled mechanism $M = B \circ S$, and an arbitrary coupling $\Gamma$ between subsampling distributions $P_{S_x}$ and $P_{S_{x'}}$. Then*

$$\Psi_\alpha(m_x || m_{x'}) \leq \int_{\mathbb{Y}^2} c_\alpha(y^{(1)}, y^{(2)}) \, \mathrm{d}\Gamma((y^{(1)}, y^{(2)})) \tag{8}$$

*with cost function $c_\alpha(y^{(1)}, y^{(2)}) = \Psi_\alpha(b_{y^{(1)}} || b_{y^{(2)}})$.*

*Proof.* Recall that $m_x$ and $m_{x'}$ are mixtures with $m_x(z) = \int b_y(z) \, dP_{S_x}(y)$ and $m_{x'}(z) = \int b_y(z) \, dP_{S_{x'}}(y)$. Since $\Gamma$ is a coupling between $P_{S_x}$ and $P_{S_{x'}}$, we can use the projection $\pi_n(\boldsymbol{y}) = y_n$ and change of variables to rewrite these mixtures as

$$m_x(z) = \int_{\mathbb{Y}} b_y(z) \, d\left(\Gamma \circ \pi_1^{-1}\right)(y) = \int_{\mathbb{Y}^2} b_{\pi_1(\boldsymbol{y})}(z) \, d\Gamma(\boldsymbol{y}) = \int_{\mathbb{Y}^2} b_{y_1}(z) \, d\Gamma(\boldsymbol{y}),$$

$$m_{x'}(z) = \int_{\mathbb{Y}} b_y(z) \, d\left(\Gamma \circ \pi_2^{-1}\right)(y) = \int_{\mathbb{Y}^2} b_{\pi_2(\boldsymbol{y})}(z) \, d\Gamma(\boldsymbol{y}) = \int_{\mathbb{Y}^2} b_{y_2}(z) \, d\Gamma(\boldsymbol{y}).$$

Since $m_x(z)$ and $m_{x'}(z)$ are now expectations w.r.t. the same measure, we can use the joint convexity of $\Psi_\alpha$ (Lemma D.9) to show $\Psi_\alpha(m_x, m_{x'}) \leq \int_{\mathbb{Y}^2} \Psi_\alpha(b_{y_1} || b_{y_2}) \, \mathrm{d}\Gamma(\boldsymbol{y})$. $\qquad\square$

## E.2   Proof of Theorem 3.4

As before, we prove a more general statement from which Theorem 3.4 immediately follows. For this, recall that $\mathcal{Y}$ is the $\sigma$-algebra of batch space $(\mathbb{Y}, \mathcal{Y})$, and that $P(T \mid R) = P(T \cap R) / P(R)$.

**Theorem E.2.** *Consider a subsampled mechanism $M = B \circ S$. Further consider two disjoint partitionings $\bigcup_{i=1}^I A_i = \mathbb{Y}$ and $\bigcup_{j=1}^J E_j = \mathbb{Y}$ such that all $A_i, E_j$ are in $\mathcal{Y}$ and have non-zero measure under $S_x$ and $S_{x'}$, respectively. Let $\Gamma$ be an arbitrary coupling between $P_{S_x}(\cdot \mid A_1), \ldots, P_{S_x}(\cdot \mid A_I), P_{S_{x'}}(\cdot \mid E_1), \ldots, P_{S_{x'}}(\cdot \mid E_J)$. Then,*

$$\Psi_\alpha(m_x || m_{x'}) \leq \int_{\mathbb{Y}^{I+J}} c_\alpha(\boldsymbol{y}^{(1)}, \boldsymbol{y}^{(2)}) \, \mathrm{d}\Gamma((\boldsymbol{y}^{(1)}, \boldsymbol{y}^{(2)})),$$

*with cost function $c : \mathbb{Y}^I \times \mathbb{Y}^J \to \mathbb{R}_+$ defined by*

$$c_\alpha(\boldsymbol{y}^{(1)}, \boldsymbol{y}^{(2)}) = \Psi_\alpha\left(\sum_{i=1}^I b_{y_i^{(1)}} \cdot P_{S_x}(A_i) \middle\| \sum_{j=1}^J b_{y_j^{(2)}} \cdot P_{S_{x'}}(E_j)\right). \tag{9}$$

*Proof.* Using the law of total expectation, linearity of integration, and change of variables with projection $\pi_n(\boldsymbol{y}) = y_n$ shows that

$$m_x(z) = \sum_{i=1}^I \left(\int_{\mathbb{Y}} b_y(z) \, dP_{S_x}(y \mid A_i)\right) P_{S_x}(A_i)$$

$$= \int_{\mathbb{Y}} \sum_{i=1}^I b_y(z) \cdot P_{S_x}(A_i) \, dP_{S_x}(y \mid A_i)$$

$$= \int_{\mathbb{Y}} \sum_{i=1}^I b_y(z) \cdot P_{S_x}(A_i) \, d(\Gamma \circ \pi_i^{-1})(y)$$

$$= \int_{\mathbb{Y}^{I+J}} \left(\sum_{i=1}^I b_{y_i}(z) \cdot P_{S_x}(A_i)\right) d\Gamma(\boldsymbol{y})$$

and

$$m_{x'}(z) = \int_{\mathbb{Y}^{I+J}} \left( \sum_{j=1}^{J} b_{y_{(j+I)}}(z) \cdot P_{S_x}(E_j) \right) \, d\Gamma(\boldsymbol{y}).$$

Since $m_x(z)$ and $m_{x'}(z)$ are now expectations w.r.t. the same measure, we can use the joint convexity of $\Psi_\alpha$ (Lemma D.9) to show

$$\Psi_\alpha(m_x \| m_{x'}) \leq \int_{\mathbb{Y}^{I+J}} \Psi_\alpha \left( \sum_{i=1}^{I} b_{y_i}(z) \cdot P_{S_x}(A_i) \| \sum_{j=1}^{J} b_{y_{(j+I)}}(z) \cdot P_{S_x}(E_j) \right) \, d\Gamma(\boldsymbol{y}).$$

A change of indexing via $\boldsymbol{y}_i^{(1)} = \boldsymbol{y}_i$ and $\boldsymbol{y}_j^{(2)} = \boldsymbol{y}_{(j+I)}$ concludes our proof. $\qquad\square$

### E.3 Proof of Proposition 3.5

**Proposition 3.5.** *Consider $\boldsymbol{y}^{(1)} \in \mathbb{Y}^I, \boldsymbol{y}^{(2)} \in \mathbb{Y}^J$, and cost function $c$ defined in Eq. (2). Let $d_\mathbb{Y}$ be the distance induced by $\simeq_\mathbb{Y}$ (see Definition 2.4). Then, $c_\alpha(\boldsymbol{y}^{(1)}, \boldsymbol{y}^{(2)}) \leq \hat{c}_\alpha(\boldsymbol{y}^{(1)}, \boldsymbol{y}^{(2)})$, with*

$$\hat{c}_\alpha(\boldsymbol{y}^{(1)}, \boldsymbol{y}^{(2)}) = \max_{\hat{\boldsymbol{y}}^{(1)}, \hat{\boldsymbol{y}}^{(2)}} c_\alpha(\hat{\boldsymbol{y}}^{(1)}, \hat{\boldsymbol{y}}^{(2)}) \tag{3}$$

*subject to $\forall k, l \in \{1, 2\}, \forall t, u : d_\mathbb{Y}(\hat{y}_t^{(k)}, \hat{y}_u^{(l)}) \leq d_\mathbb{Y}(y_t^{(k)}, y_u^{(l)})$ and $\hat{\boldsymbol{y}}^{(1)} \in \mathbb{Y}^I, \hat{\boldsymbol{y}}^{(2)} \in \mathbb{Y}^J$.*

*Proof.* The original tuples of batches $\boldsymbol{y}^{(1)} \in \mathbb{Y}^I$ and $\boldsymbol{y}^{(2)} \in \mathbb{Y}^J$ constitute a feasible solution to the maximization problem, since they fulfill the constraints with equality, i.e., $\forall k, l, t, u : d_\mathbb{Y}(y_t^{(k)}, y_u^{(l)}) = d_\mathbb{Y}(y_t^{(k)}, y_u^{(l)})$. The value of any feasible solution to a maximization problem is l.e.q. its optimal value. $\qquad\square$

# F    Distance-compatible couplings of multiple distributions

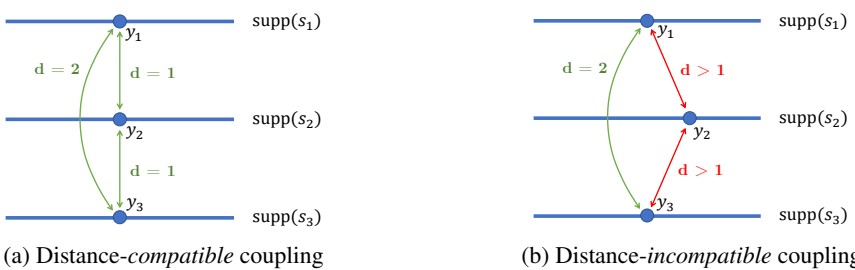

(a) Distance-*compatible* coupling      (b) Distance-*incompatible* coupling

Figure 25: Example of a distance-compatible and a distance-incompatible coupling

In the following, we generalize the notion of distance-compatible couplings proposed in [15] from two distributions to an arbitrary number of distributions. This provides a sufficient optimality condition for Theorem E.2.

Note that we use a more general, measure-theoretic definition of distance-compatibility (see Definition F.4). Definition 3.6 is a special case for subsampling schemes with finite, discrete support.

As discussed in Section 3, a $d_\mathbb{Y}$-compatible coupling between (conditional) subsampling distributions only assigns probability to tuples of batches $\boldsymbol{y}$ when all pairs $y_i, y_j$ have the smallest possible distance to $y_1$ and to each other, while still being in the support of their respective distributions.

To avoid having to make additional assumptions about the topology of batch space $(\mathbb{Y}, \mathcal{Y})$, we will assume that all subsampling schemes have densities and reason about the support of these densities.

**Definition F.1.** The support of a function $p : \mathbb{Y} \to \mathbb{R}_+$ is $\mathrm{supp}(p) = \{y \in \mathbb{Y} \mid p(y) > 0\}$.

Based on this notion of support, we can define a notion of distance between a batch $y \in \mathbb{Y}$ and the support of a density:

**Definition F.2.** Consider a distance $d_\mathbb{Y}$ induced by a neighboring relation $\simeq_\mathbb{Y}$. The distance between an element $y \in \mathbb{Y}$ and the support of a function $p : \mathbb{Y} \to \mathbb{R}_+$ is defined as $d_\mathbb{Y}(y, \mathrm{supp}(p)) = \min_{y' \in \mathrm{supp}(p)} d_\mathbb{Y}(y, y')$.

Furthermore, we can define a notion of distance between the support of two different densities:

**Definition F.3.** Consider a distance $d_\mathbb{Y}$ induced by a neighboring relation $\simeq_\mathbb{Y}$. The distance between the support of functions $p_1, p_2 : \mathbb{Y} \to \mathbb{R}_+$ is defined as

$$d_\mathbb{Y}(\mathrm{supp}(p_1), \mathrm{supp}(p_2)) = \min_{y_1, y_2} d_\mathbb{Y}(y_1, y_2) \quad \text{s.t.} \quad \forall i \in \{1, 2\} : y_i \in \mathrm{supp}(p_i).$$

Based on these definitions, we can now formally define distance-compatible couplings between multiple distributions.

**Definition F.4.** Consider a coupling $\Gamma$ between probability measures $P_1, \ldots, P_N$ on measurable space $(\mathbb{Y}, \mathcal{Y})$ with symmetric neighboring relation $\simeq_\mathbb{Y}$ and induced distance $d_\mathbb{Y}$. Assume that $\forall n : P_n \ll \nu_n$ for some measures $\nu_1, \ldots, \nu_N$ and define $p_n = \mathrm{d}P_n / \mathrm{d}\nu_n$. Further assume that $\Gamma \ll \prod_{n=1}^N \nu_n$ with product measure $\prod_{n=1}^N \nu_n$, and define $\gamma = \mathrm{d}\Gamma / \mathrm{d}\prod \nu_n$. Then, $\Gamma$ is a $d_\mathbb{Y}$-compatible coupling if

$$(\boldsymbol{y} \in \mathrm{supp}(\Gamma) \implies \forall u > 1 : d_\mathbb{Y}(y_1, y_u) = d_\mathbb{Y}(y_1, \mathrm{supp}(s_u)))$$
$$\wedge (\boldsymbol{y} \in \mathrm{supp}(\Gamma) \implies \forall u > t > 1 : d_\mathbb{Y}(y_u, y_t) = d_\mathbb{Y}(\mathrm{supp}(s_t), \mathrm{supp}(s_u))).$$

That is, $\Gamma$ only assigns probability to a tuple of batches $\boldsymbol{y}$ if all $y_t$ and $y_u$ are as close as possible to $y_1$ and as close as possible to each other, while still being in the support of their corresponding densities. Note that our choice of focusing on $y_1$ is arbitrary, and $d_\mathbb{Y}$-compatibility could also be defined for any other reference index $n \in \{1, \ldots, N\}$.

We shall now prove that $d_\mathbb{Y}$-compatibility is a sufficient optimality condition for our optimal transport problem. For this proof, we will use the following lemma, which immediately follows from Definitions F.2 and F.3:

**Lemma F.5.** *Consider a distance $d_\mathbb{Y}$ induced by a relation $\simeq_\mathbb{Y}$ and two functions $p_1, p_2 : \mathbb{Y} \to \mathbb{R}_+$. Then, for all $y_1 \in \mathrm{supp}(p_1), y_2 \in \mathrm{supp}(p_2)$,*

$$d_\mathbb{Y}(y_1, y_2) \geq d_\mathbb{Y}(y_1, \mathrm{supp}(p_2)) \geq d_\mathbb{Y}(\mathrm{supp}(p_1), \mathrm{supp}(p_2)).$$

**Theorem F.6.** *Consider a subsampled mechanism $M = B \circ S$. Further consider two finite partitions $A_1, \ldots, A_I \in \mathcal{Y}$ and $E_1, \ldots, E_J \in \mathcal{Y}$ of $\mathbb{Y}$ such that all $A_i$ and $E_j$ have non-zero measure under $S_x$ and $S_{x'}$, respectively. Let $d_\mathbb{Y}$ be the distance induced by a symmetric neighboring relation $\simeq_\mathbb{Y}$. Let $\Gamma^*$ be a $d_\mathbb{Y}$-compatible coupling between $P_{S_x}(\cdot \mid A_1), \ldots, P_{S_x}(\cdot \mid A_I), P_{S_{x'}}(\cdot \mid E_1), \ldots, P_{S_{x'}}(\cdot \mid E_J)$, which have Radon–Nikodym derivatives $s_1, \ldots, s_{I+J}$. Then, for all $\alpha > 1$,*

$$\Gamma^* \in \arg\min_{\Gamma \in \mathbb{G}} \leq \int_{\mathbb{Y}^{I+J}} \hat{c}_\alpha(\boldsymbol{y}^{(1)}, \boldsymbol{y}^{(2)}) \, \mathrm{d}\Gamma((\boldsymbol{y}^{(1)}, \boldsymbol{y}^{(2)})). \tag{10}$$

*where $\mathbb{G}$ is the set of valid couplings between the $I + J$ measures, and $\hat{c}_\alpha : \mathbb{Y}^I \times \mathbb{Y}^J \to \mathbb{R}_+$ is the cost function upper bound defined in Proposition 3.5.*

*Proof.* Consider an arbitrary, not necessarily $d_\mathbb{Y}$-compatible coupling $\Gamma$. By definition of $\hat{c}$ and symmetry of $\simeq$, we have

$$\int_{\mathbb{Y}^{I+J}} \hat{c}_\alpha(\boldsymbol{y}^{(1)}, \boldsymbol{y}^{(2)}) \, \mathrm{d}\Gamma((\boldsymbol{y}^{(1)}, \boldsymbol{y}^{(2)}))$$

$$= \int_{\mathbb{Y}^{I+J}} \left( \max_{\hat{y} \in \mathbb{Y}^{I+J}} c_\alpha(\hat{\boldsymbol{y}}_{:I}, \hat{\boldsymbol{y}}_{I:}) \text{ s.t. } \forall t < u : d_\mathbb{Y}(\hat{y}_t, \hat{y}_u) \leq d_\mathbb{Y}(y_t, y_u) \right) \, \mathrm{d}\Gamma(\boldsymbol{y}),$$

with original cost function $c_\alpha : \mathbb{Y}^I \times \mathbb{Y}^J \to \mathbb{R}_+$ defined in Theorem E.2.

We can now use Lemma F.5 to tighten the constraints of the optimization problem inside the integrand and thus lower-bound its optimal value for all $\boldsymbol{y} \in \mathrm{supp}(\Gamma)$:

$$\max_{\hat{y} \in \mathbb{Y}^{I+J}} c_\alpha(\hat{\boldsymbol{y}}_{:I}, \hat{\boldsymbol{y}}_{I:}) \text{ s.t. } \forall t < u : d_\mathbb{Y}(\hat{y}_t, \hat{y}_u) \leq d_\mathbb{Y}(y_t, y_u)$$

$$\geq \max_{\hat{y} \in \mathbb{Y}^{I+J}} c_\alpha(\hat{\boldsymbol{y}}_{:I}, \hat{\boldsymbol{y}}_{I:})$$

$$\text{s.t. } \forall u > 1 : d_\mathbb{Y}(\hat{y}_1, \hat{y}_u) \leq d_\mathbb{Y}(y_1, \mathrm{supp}(s_u)),$$
$$\forall u > t > 1 : d_\mathbb{Y}(\hat{y}_t, \hat{y}_u) \leq d_\mathbb{Y}(\mathrm{supp}(s_t), \mathrm{supp}(s_u)).$$

We notice that the lower bound only depends on $y_1$ and shall thus refer to it as $\kappa_\alpha(y_1)$. Further note that any $\boldsymbol{y} \notin \mathrm{supp}(\Gamma)$ does not contribute to the integral. Since $\Gamma$ is a valid coupling, we can marginalize out all variables except $y_1$ via projection $\pi_1(\boldsymbol{y}) = y_1$ to show

$$\int_{\mathbb{Y}^{I+J}} \hat{c}_\alpha(\boldsymbol{y}^{(1)}, \boldsymbol{y}^{(2)}) \, \mathrm{d}\Gamma((\boldsymbol{y}^{(1)}, \boldsymbol{y}^{(2)}))$$

$$\geq \int_{\mathbb{Y}^{I+J}} \kappa_\alpha(\pi_1(\boldsymbol{y})) \, \mathrm{d}\Gamma(\boldsymbol{y}) = \int_\mathbb{Y} \kappa_\alpha(y_1) \, \mathrm{d}(\Gamma \circ \pi_1^{-1})(y_1) = \int_\mathbb{Y} \kappa_\alpha(y_1) \, \mathrm{d}P_{S_x}(y_1 \mid A_1).$$

By construction of $\kappa_\alpha$, this holds with equality whenever $\Gamma^*$ is a $d_\mathbb{Y}$-compatible coupling. $\square$

Finally, the exact derivations we used for Theorem F.6 can also be used to show that the optimal value of our transport problem has a simple, canonical form whenever a $d_\mathbb{Y}$-compatible coupling exists:

**Corollary F.7.** *Consider a subsampled mechanism $M = B \circ S$. Further consider two finite partitions $A_1, \ldots, A_I \in \mathcal{Y}$ and $E_1, \ldots, E_J \in \mathcal{Y}$ of $\mathbb{Y}$ such that all $A_i$ and $E_j$ have non-zero measure under $S_x$ and $S_{x'}$, respectively. Let $d_\mathbb{Y}$ be the distance induced by a symmetric neighboring relation $\simeq_\mathbb{Y}$. Assume that a $d_\mathbb{Y}$-compatible coupling between $P_{S_x}(\cdot \mid A_1), \ldots, P_{S_x}(\cdot \mid A_I), P_{S_{x'}}(\cdot \mid E_1), \ldots, P_{S_{x'}}(\cdot \mid E_J)$ with Radon–Nikodym derivatives $s_1, \ldots, s_{I+J}$ exists. Then, for all $\alpha > 1$,*

$$\min_{\Gamma \in \mathbb{G}} \int_{\mathbb{Y}^{I+J}} \hat{c}_\alpha(\boldsymbol{y}^{(1)}, \boldsymbol{y}^{(2)}) \, \mathrm{d}\Gamma((\boldsymbol{y}^{(1)}, \boldsymbol{y}^{(2)})) = \int_\mathbb{Y} \kappa_\alpha(y_1) \, \mathrm{d}P_{S_x}(y_1 \mid A_1), \tag{11}$$

*where $\mathbb{G}$ is the space of valid couplings between the $I + J$ measures,*

$$\kappa_\alpha(y_1) = \max_{\hat{y} \in \mathbb{Y}^{I+J}} c_\alpha(\hat{\boldsymbol{y}}_{:I}, \hat{\boldsymbol{y}}_{I:}) \quad s.t. \quad \forall u > 1 : d_\mathbb{Y}(\hat{y}_1, \hat{y}_u) \leq d_\mathbb{Y}(y_1, \mathrm{supp}(s_u)),$$

$$\forall u > t > 1 : d_\mathbb{Y}(\hat{y}_t, \hat{y}_u) \leq d_\mathbb{Y}(\mathrm{supp}(s_t), \mathrm{supp}(s_u)),$$

*and $c_\alpha : \mathbb{Y}^I \times \mathbb{Y}^J \to \mathbb{R}_+$ is the original cost function defined in Theorem E.2.*

Thus, we can focus on constructing distance-compatible couplings when trying to derive existing or novel amplification by subsampling guarantees. Note that these result also generalize to asymmetric neighboring relations $\simeq_{\mathbb{Y}}$. We just wanted to avoid further complicating the indexing. Future work may want to generalize these results to a more general, topological notion of support that does not rely on the existence of subsampling densities.

## G Recovering known mechanism-agnostic ADP guarantees

In the following, we first provide an overview of the framework for deriving mechanism-agnostic ADP guarantees proposed by Balle et al. [15]. We then show that the same guarantees can be derived via optimal transport between multiple subsampling distributions. Finally, we use observations made during the proof to explain why the mechanism-agnostic guarantees derived via this approach can be suboptimal.

### G.1 Overview of Balle et al. framework

The framework from [15] uses four steps to derive tight mechanism-agnostic guarantees for subsampled mechanisms: (1) Partitioning the subsampling distributions via maximal couplings, (2) applying advanced joint convexity, (3) applying joint convexity, and (4) defining two couplings involving two distributions.

Maximal couplings are a construction that makes it possible to partition the subsampling probability mass functions as

$$s_x(y) = (1 - w)p_x(y) + wq_x(y)$$
$$s_{x'}(y) = (1 - w)p_x(y) + wq_{x'}(y),$$

with probability mass functions $p_x, q_x, q_{x'} : \mathbb{Y} \to [0, 1]$ chosen such that $q_x$ and $q_{x'}$ have disjoint support and $w \in [0, 1]$ is as small as possible.

Note that, when considering typical subsampling schemes (Poisson, without replacement, with replacement) and single-element neighboring relations (insertion/removal, substitution), this partition is simply equivalent to

$$s_x(y) = \Pr[S_x \in A_1] \cdot s_x(y \mid A_1) + \Pr[S_x \in A_2] \cdot s_x(y \mid A_2)$$
$$s_{x'}(y) = \Pr[S_{x'} \in E_1] \cdot s_{x'}(y \mid E_1) + \Pr[S_x \in E_2] \cdot s_{x'}(y \mid E_2),$$

where $A_1$ and $E_1$ are the event that the inserted/removed or substituted element is not sampled, and $A_2, E_2$ are their complements (see Appendix B in [15]).

Using this construction, the densities $m_x$ and $m_{x'}$ of subsampled mechanisms $M = B \circ S$ can be rewritten as

$$m_x(z) = (1 - w) \sum_{y \in \mathbb{Y}} b_y(z)p_x(y) + w \sum_{y \in \mathbb{Y}} b_y(z)q_x(y) \tag{12}$$

$$m_{x'}(z) = (1 - w) \sum_{y \in \mathbb{Y}} b_y(z)p_x(y) + w \sum_{y \in \mathbb{Y}} b_y(z)q_{x'}(y) \tag{13}$$

Next, one can rewrite the divergence $H_\alpha(m_x || m_{x'})$ via the following property:

**Proposition G.1** (Advanced joint convexity [15]). *Let $m_x, m'_x : \mathbb{Z} \to [0, 1]$ be probability mass functions satisfying $m_x(z) = (1 - w)f(z) + wg(z)$ and $m_{x'}(z) = (1 - w)f(z) + wg'(z)$ for some $w \in [0, 1]$, $f, g, g' : \mathbb{Z} \to [0, 1]$. Given $\alpha \geq 1$, let $\alpha' = 1 + w(\alpha - 1)$ and $\beta = \alpha' / \alpha$. Then the following holds:*

$$D_{\alpha'}(m_x || m_{x'}) = wD_\alpha(g || (1 - \beta)f + \beta g').$$

Applying advanced joint convexity, followed by joint convexity to Eqs. (12) and (13) shows that

$$H_{\alpha'}(m_x || m_{x'}) \leq (1 - \beta)H_\alpha \left( \sum_{y \in \mathbb{Y}} b_y q_x(y) || \sum_{y \in \mathbb{Y}} b_y p_x(y) \right)$$

$$+ \beta H_\alpha \left( \sum_{y \in \mathbb{Y}} b_y q_x(y) || \sum_{y \in \mathbb{Y}} b_y q_{x'}(y) \right)$$

Finally, one can construct a coupling $\gamma : \mathbb{Y}^2 \to [0, 1]$ between $q_x$ and $p_x$, as well as a coupling $\gamma' : \mathbb{Y}^2 \to [0, 1]$ between $q_x$ and $q_{x'}$. One can then invoke a special case of Theorem 3.3 with $\Psi_\alpha = H_\alpha$ and the cost function upper bound from Proposition 3.5 to show

$$H_{\alpha'}(m_x || m_{x'}) \leq (1 - \beta) \sum_{\boldsymbol{y} \in \mathbb{Y}^2} \hat{c}_\alpha(y^{(1)}, y^{(2)})\gamma(y^{(1)}, y^{(2)}) + \beta \sum_{\boldsymbol{y} \in \mathbb{Y}^2} \hat{c}_\alpha(y^{(1)}, y^{(2)})\gamma'(y^{(1)}, y^{(2)}).$$

Since we are only considering pairs of batches, we have

$$\hat{c}_\alpha(y^{(1)}, y^{(2)}) = \max_{\hat{y}} H_\alpha(b_{\hat{y}^{(1)}} || b_{\hat{y}^{(1)}}) \text{ s.t. } d_{\mathbb{Y}}(y^{(1)}, y^{(2)}) \leq d_{\mathbb{Y}}(\hat{y}^{(1)}, \hat{y}^{(2)})$$

with induced distance $d_{\mathbb{Y}}$ from Definition 2.4. This specific cost function bound is referred to as "group privacy profile" in [15].

## G.2 Subsumption of Balle et al. framework

Next, we show that we can always obtain the same guarantee by defining a (potentially suboptimal) coupling between four subsampling distributions, and then pessimistically upper-bounding the guarantee we would obtain through Theorem 3.4:

**Theorem G.2.** *Consider a subsampled mechanism $M = B \circ S$ and some $x, x' \in \mathbb{X}$. Assume that subsampling pmfs $s_x, s_{x'} : \mathbb{Y} \to [0, 1]$ satisfy $s_x(z) = (1 - w)p_x(z) + wq_x(z)$ and $s_{x'}(z) = (1 - w)p_x(z) + wq_{x'}(z)$ for some $w \in [0, 1]$ and $p_x, q_x, q_{x'} : \mathbb{Y} \to [0, 1]$. Let $\gamma : \mathbb{Y}^2 \to [0, 1]$ be an arbitrary coupling of $q_x, p_x$, and $\gamma' : \mathbb{Y}^2 \to [0, 1]$ be an arbitrary coupling of $q_x, p_{x'}$. Then, there is a coupling $\tilde{\gamma} : \mathbb{Y}^4 \to [0, 1]$ of $p_x, q_x, p_x, q_{x'}$ such that for all $\alpha \geq 1$*

$$H_{\alpha'}(m_x || m_{x'}) \leq \sum_{\boldsymbol{y} \in \mathbb{Y}^{2+2}} \hat{c}_{\alpha'}(\boldsymbol{y}^{(1)}, \boldsymbol{y}^{(2)})\tilde{\gamma}(\boldsymbol{y}^{(1)}, \boldsymbol{y}^{(1)})$$

$$\leq (1 - \beta) \sum_{\boldsymbol{y} \in \mathbb{Y}^{1+1}} \hat{c}_\alpha(y^{(1)}, y^{(2)})\gamma(y^{(1)}, y^{(2)}) + \beta \sum_{\boldsymbol{y} \in \mathbb{Y}^{1+1}} \hat{c}_\alpha(y^{(1)}, y^{(2)})\gamma'(y^{(1)}, y^{(2)}),$$

*with cost function upper bound $\hat{c}_\alpha$ defined in Proposition 3.5, $\alpha' = 1 + w(\alpha - 1)$, and $\beta = \alpha' / \alpha$.*

*Proof.* The main idea is to invoke Theorem 3.4 with a specifically crafted coupling, and then apply advanced joint convexity and joint convexity.

Specifically, we define a coupling $\tilde{\gamma} : \mathbb{Y} \to [0, 1]$ that corresponds to the following generative process: We first sample $y_2^{(1)}$ and $y_2^{(2)}$ from the coupling $\gamma'$ (recall that a coupling is a joint mass function). This gives us two elements from the support of $q_x$ and $q_{x'}$, respectively. Then, we sample $y_1^{(1)}$ from $\gamma$ conditioned on $y_2^{(1)}$. This gives us an element from the support of $p_x$. Finally, we let $y_1^{(2)} \leftarrow y_1^{(1)}$. Formally, this coupling can be defined as

$$\tilde{\gamma}(\boldsymbol{y}^{(1)}, \boldsymbol{y}^{(2)}) = \gamma'(y_2^{(1)}, y_2^{(2)}) \cdot \frac{\gamma(y_2^{(1)}, y_1^{(1)})}{q_x(y_2^{(1)})} \cdot \mathbb{1}\left[y_1^{(2)} = y_1^{(1)}\right].$$

**Validity of coupling.** Before proceeding, we need to verify that this is a valid coupling, i.e., its four marginals are $p_x, q_x, p_x, q_{x'}$. For any $y_1^{(1)} \in \mathbb{Y}$, we have

$$\sum_{y_2^{(1)}, y_1^{(2)}, y_2^{(2)} \in \mathbb{Y}^3} \tilde{\gamma}(y_1^{(1)}, y_2^{(1)}, y_1^{(2)}, y_2^{(2)})$$

$$= \sum_{y_2^{(1)}, y_2^{(2)} \in \mathbb{Y}^2} \gamma'(y_2^{(1)}, y_2^{(2)}) \cdot \frac{\gamma(y_2^{(1)}, y_1^{(1)})}{q_x(y_2^{(1)})}$$

$$= \sum_{y_2^{(1)} \in \mathbb{Y}} q_x(y_2^{(1)}) \cdot \frac{\gamma(y_2^{(1)}, y_1^{(1)})}{q_x(y_2^{(1)})}$$

$$= \sum_{y_2^{(1)} \in \mathbb{Y}} \gamma(y_2^{(1)}, y_1^{(1)})$$

$$= p_x(y_1^{(1)}).$$

where the first inequality is due to the indicator function, the second equality follows from marginalizing $\gamma'$, and the last equality follows from marginalizing $\gamma$.

The proof for any $y_1^{(2)} \in \mathbb{Y}$ is analogous.

For any $y_2^{(1)} \in \mathbb{Y}$, we similarly have

$$\sum_{y_1^{(1)}, y_1^{(2)}, y_2^{(2)} \in \mathbb{Y}^3} \tilde{\gamma}(y_1^{(1)}, y_2^{(1)}, y_1^{(2)}, y_2^{(2)}) \tag{14}$$

$$= \sum_{y_1^{(1)}, y_2^{(2)} \in \mathbb{Y}^2} \gamma'(y_2^{(1)}, y_2^{(2)}) \cdot \frac{\gamma(y_2^{(1)}, y_1^{(1)})}{q_x(y_2^{(1)})} \tag{15}$$

$$= \sum_{y_1^{(1)} \in \mathbb{Y}^1} \gamma(y_2^{(1)}, y_1^{(1)}) \tag{16}$$

$$= q_x(y_2^{(1)}), \tag{17}$$

and for any $y_2^{(2)}$ we have

$$\sum_{y_1^{(1)}, y_2^{(1)}, y_1^{(2)} \in \mathbb{Y}^3} \tilde{\gamma}(y_1^{(1)}, y_2^{(1)}, y_1^{(2)}, y_2^{(2)}) \tag{18}$$

$$= \sum_{y_1^{(1)}, y_2^{(1)} \in \mathbb{Y}^2} \gamma'(y_2^{(1)}, y_2^{(2)}) \cdot \frac{\gamma(y_2^{(1)}, y_1^{(1)})}{q_x(y_2^{(1)})} \tag{19}$$

$$= \sum_{y_2^{(1)} \in \mathbb{Y}} \gamma'(y_2^{(1)}, y_2^{(2)}) \cdot \frac{q_x(y_2^{(1)})}{q_x(y_2^{(1)})} \tag{20}$$

$$= q_{x'}(y_2^{(2)}). \tag{21}$$

**First inequality.** Now that we have a valid coupling, we can use the same joint-convexity argument as in our proof of Theorem 3.4, combined with our cost function bound $\hat{c}_\alpha$ to show

$$H_{\alpha'}(m_x || m_{x'}) \leq \sum_{\mathbb{Y}^{2+2}} \max_{\hat{\boldsymbol{y}}} H_{\alpha'} \left( (1-w)b_{\hat{y}^{(1)}} + wb_{\hat{y}^{(1)}_2} || 1-w)b_{\hat{y}^{(2)}_1} + wb_{\hat{y}^{(2)}_2} \right) \cdot \gamma(\boldsymbol{y}^{(1)}, \boldsymbol{y}^{(2)}),$$

with each of the $|\mathbb{Y}^{2+2}|$ optimization problems being constrained by $\forall k, l, t, u : d_\mathbb{Y}(\hat{y}_t^{(k)}, \hat{y}_u^{(l)}) \leq d_\mathbb{Y}(y_t^{(k)}, y_u^{(l)})$ and $\hat{\boldsymbol{y}}^{(1)} \in \mathbb{Y}^2, \hat{\boldsymbol{y}}^{(2)} \in \mathbb{Y}^2$. This corresponds to the first equality in our Theorem.

**Second inequality.** Due to construction of our coupling, we always have $d_\mathbb{Y}(y_1^{(1)}, y_1^{(2)}) = 0$, i.e., $y_1^{(1)} = y_1^{(2)}$. We can thus use advanced joint convexity, joint convexity, and linearity of summation to obtain a looser bound via

$$H_{\alpha'}(m_x || m_{x'}) \leq (1-\beta) \sum_{\mathbb{Y}^{2+2}} \max_{\hat{\boldsymbol{y}}} H_\alpha \left( b_{\hat{y}^{(1)}_2} || b_{\hat{y}^{(1)}_1} \right) \cdot \gamma(\boldsymbol{y}^{(1)}, \boldsymbol{y}^{(2)})$$

$$+ \beta \sum_{\mathbb{Y}^{2+2}} \max_{\hat{\boldsymbol{y}}} H_\alpha \left( b_{\hat{y}^{(1)}_2} || b_{\hat{y}^{(2)}_2} \right) \cdot \gamma(\boldsymbol{y}^{(1)}, \boldsymbol{y}^{(2)}),$$

with each of the $2 \cdot |\mathbb{Y}^{2+2}|$ optimization problems being constrained by $\forall k, l, t, u : d_\mathbb{Y}(\hat{y}_t^{(k)}, \hat{y}_u^{(l)}) \leq d_\mathbb{Y}(y_t^{(k)}, y_u^{(l)})$ and $\hat{\boldsymbol{y}}^{(1)} \in \mathbb{Y}^2, \hat{\boldsymbol{y}}^{(2)} \in \mathbb{Y}^2$.

Next, we can further loosen this bound by dropping all constraints involving $y_1^{(2)}$ and $y_2^{(2)}$ in the first optimization problem. We can also drop all constraints involving $y_1^{(1)}$ and $y_1^{(2)}$ in the second optimization problem. Thus, by definition of the group privacy profile, we have

$$H_{\alpha'}(m_x || m_{x'}) \leq (1-\beta) \sum_{\mathbb{Y}^{2+2}} \hat{c}_\alpha \left( \boldsymbol{y}_2^{(1)}, \boldsymbol{y}_1^{(1)} \right) \cdot \gamma(\boldsymbol{y}^{(1)}, \boldsymbol{y}^{(2)})$$

$$+ \beta \sum_{\mathbb{Y}^{2+2}} \hat{c}_\alpha \left( \boldsymbol{y}_2^{(1)}, \boldsymbol{y}_2^{(2)} \right) \cdot \gamma(\boldsymbol{y}^{(1)}, \boldsymbol{y}^{(2)}).$$

Note that the first cost function term does not depend on $y_1^{(2)}$ nor $y_2^{(2)}$, and the second cost function term does not depend on $y_1^{(1)}$ nor $y_1^{(2)}$. We can thus marginalize out these variables (recall Eq. (16) and Eq. (20)) to conclude our proof. □

The same argument also applies to the general problem setting defined in Appendix D: Given two couplings $\Gamma, \Gamma'$, we can define a product coupling $\tilde{\Gamma} \propto \Gamma \cdot \Gamma'$, invoke Theorem 3.4, and then apply (advanced) joint convexity to pessimistically upper-bound the guarantee that would be obtained through our proposed framework.

### G.3 Deficiencies of mechanism-agnostic ADP bounds.

Following our discussion, we can identify three potential sources for looseness in this approach for deriving mechanism-agnostic bounds. Firstly, it only uses a binary partitioning of subsampling pmfs $s_x$ and $s_{x'}$. The resultant mechanism-specific guarantee only depends on divergences between two-component mixtures, which may not be sufficient to tightly bound the overall divergence in complicated scenarios like group privacy. Secondly, it neglects the pairwise distances of $y_1^{(1)}$ and $y_2^{(2)}$, resulting in potentially very large values of the cost function. Thirdly, the bound might be further loosened by applying joint convexity once more.

# H  Recovering known mechanism-agnostic RDP guarantees

In the following, we demonstrate that existing amplification by subsampling guarantees for Rényi-DP can be derived by instantiating our proposed framework (see Fig. 2). Specifically, we demonstrate that these guarantees can be derived via the procedure discussed in Section 3.3 and shown in Figs. 2b and 2c: (1) Conditioning on at most 4 events indicating the presence of inserted / deleted / substituted elements, (2) defining a simultaneous coupling, and (3) using joint convexity to upper-bound the resultant mechanism-specific guarantee by component divergences.

These guarantees are mechanism-agnostic in the sense that they express the subsampled mechanism's privacy parameters $(\alpha, \rho)$ as a function of the base mechanism's privacy parameters. We derive guarantees for the general measure-theoretic problem setting introduced in Appendix D, where the base mechanism can be either discrete or continuous.

The reader may notice that parts of the proofs in Appendices H.1 and H.2 are very similar to those in [28, 30], safe for the discussion of couplings and $d_{\mathbb{Y}}$-compatibility. That is precisely the point: There is an optimal transport problem that implicitly underlies results from prior work, which we have identified and can now generalize to more challenging scenarios like group privacy amplification.

For this section, recall that $\Delta_\alpha$ is not the Rényi divergence, but its $\alpha$th moment, i.e., a scaled and exponentiated Rényi divergence (see Definition D.8).

## H.1  Subsampling without replacement and substitution

For subsampling without replacement and substitution relation $\simeq_\Delta$ we first show a more general result Theorem H.1. We then demonstrate that it can be upper-bounded via joint convexity of exponentiated Rényi divergence $\Delta_\alpha$ to recover the guarantee from [28].

**Theorem H.1.** *Let $M = B \circ S$ be a subsampled mechanism, where $S$ is subsampling without replacement with batch size $q$. Let $\simeq_{\mathbb{Y}}$ be the substitution relation $\simeq_{\Delta,\mathbb{Y}}$. Then, for $\alpha > 1$ and all $x \simeq_{\Delta,\mathbb{X}} x'$ of size $N$,*

$$\Delta_\alpha(m_x || m_{x'}) \leq \max_{\hat{y}} \Delta_\alpha((1-w) \cdot b_{y_1^{(1)}} + w \cdot b_{y_2^{(1)}} || (1-w) \cdot b_{y_1^{(2)}} + w \cdot b_{y_2^{(2)}})$$

*subject to $d_{\Delta,\mathbb{Y}}(y_1^{(1)}, y_2^{(1)}) \leq 1$, $d_{\Delta,\mathbb{Y}}(y_1^{(1)}, y_2^{(2)}) \leq 1$, $d_{\Delta,\mathbb{Y}}(y_2^{(1)}, y_2^{(2)}) \leq 1$, $y_1^{(1)} = y_1^{(2)}$, and with $w = q / N$.*

*Proof.* Consider arbitrary $x \simeq_{\Delta,\mathbb{X}} x'$. By definition of $\simeq_\Delta$, there must be some $a \in x$, $a' \in x'$ such that $x' = x \setminus \{a\} \cup \{a'\}$. We thus define both $A_1$ and $E_1$ from Theorem E.2 to be the event that neither $a$ nor $a'$ is sampled, i.e., $A_1 = E_1 = \{y \in \mathbb{Y} \mid y \cap \{a, a'\} = \varnothing\}$. We further define $A_2$ and $E_2$ to be the event that $a$ or $a'$ is sampled, i.e., $A_2 = \overline{A_1}$ and $E_2 = \overline{E_1}$.

By definition of subsampling without replacement, we have

$$P_{S_x}(A_1) = P_{S_{x'}}(E_1) = \text{HyperGeom}(0 \mid N, 1, q) = 1 - \frac{q}{N},$$

$$P_{S_x}(A_2) = P_{S_{x'}}(E_2) = \text{HyperGeom}(1 \mid N, 1, q) = \frac{q}{N},$$

which corresponds to the weights $(1-w)$ and $w$, respectively. We further have

$$s_x(y \mid A_1) = \begin{cases} \binom{|x|-1}{q}^{-1} & \text{if } y \subseteq x \wedge a \notin y \\ 0 & \text{otherwise} \end{cases}, \quad s_x(y \mid A_2) = \begin{cases} \binom{|x|-1}{q-1}^{-1} & \text{if } y \subseteq x \wedge a \in y \\ 0 & \text{otherwise} \end{cases},$$

and

$$s_{x'}(y \mid E_1) = \begin{cases} \binom{|x|-1}{q}^{-1} & \text{if } y \subseteq x' \wedge a' \notin y \\ 0 & \text{otherwise} \end{cases}, \quad s_x(y \mid E_2) = \begin{cases} \binom{|x|-1}{q-1}^{-1} & \text{if } y \subseteq x' \wedge a' \in y \\ 0 & \text{otherwise} \end{cases}.$$

**Coupling.** We now define a coupling $\gamma : \mathbb{Y}^{2+2} \to \mathbb{R}_+$ that corresponds to the following generative process: We first generate $y_1^{(1)}$ by sampling a batch that does not contain $a$ uniformly at random from

x. We then let $y_1^{(2)} \leftarrow y_1^{(1)}$ Finally, we pick a random element $\tilde{a}$ of $y_1^{(1)}$ and replace it with $a$ and $a'$ to generate $y_2^{(1)}$ and $y_2^{(2)}$, respectively. More formally:

$$
\gamma(\boldsymbol{y}^{(1)}, \boldsymbol{y}^{(2)}) = \begin{cases} s_x(y_1^{(1)} \mid A_1) \cdot \frac{1}{q} & \text{if } \boldsymbol{y}^{(1)}, \boldsymbol{y}^{(2)} \text{ fulfills H.2} \\ 0 & \text{otherwise.} \end{cases}
$$

**Condition H.2.** A tuple $\boldsymbol{y}^{(1)} \in \mathbb{Y}^2$, $\boldsymbol{y}^{(2)} \in \mathbb{Y}^2$ fulfills this condition when $y_1^{(2)} = y_1^{(1)}$ and $\exists \tilde{a} \in y_1^{(1)} : \left( y_2^{(1)} = y_1^{(1)} \setminus \{\tilde{a}\} \cup \{a\} \wedge y_2^{(2)} = y_1^{(1)} \setminus \{\tilde{a}\} \cup \{a'\} \right)$.

**Validity.** We now show that this constitutes a valid coupling. Consider $y_1^{(1)}$. If and only if $s_x(y_1^{(1)} \mid A_1) > 0$, there are exactly $q$ combinations of $y_2^{(1)}, y_1^{(2)}, y_2^{(2)}$ for which $\gamma(\boldsymbol{y})$ is non-zero. Thus,

$$
\sum_{y_2^{(1)}, y_1^{(2)}, y_2^{(2)} \in \mathbb{Y}^3} \gamma(\boldsymbol{y}) = q \cdot s_x(y_1^{(1)} \mid A_1).
$$

The proof for $y_1^{(2)}$ is analogous.

Next, consider $y_2^{(1)}$. If and only if $s_x(y_2^{(1)} \mid A_1) > 0$, there are exactly $|x| - q$ elements that could have been replaced by $a$ to generate $y_2^{(1)}$ from $y_2^{(1)}$. Specifically, these elements are all elements that do not appear in $y_2^{(1)}$. We thus have

$$
\sum_{y_1^{(1)}, y_1^{(2)}, y_2^{(2)} \in \mathbb{Y}^3} \gamma(\boldsymbol{y}) = (|x| - q) \cdot \frac{(|x| - 1 - q)! \cdot q!}{|x| - 1} \cdot \frac{1}{q} = \binom{|x| - 1}{q - 1}^{-1}.
$$

The proof for $y_2^{(2)}$ is analogous.

**Compatibility.** Finally, we show that $\gamma$ is a $d_\mathbb{Y}$-compatible coupling (see Appendix F). Whenever $\gamma(\boldsymbol{y}) > 0$, then

$$
d_\mathbb{Y}(y_1^{(1)}, y_2^{(1)}) = d_\mathbb{Y}\left( y_1^{(1)}, \operatorname{supp}(s_x(\cdot \mid A_2)) \right) = 1,
$$
$$
d_\mathbb{Y}(y_1^{(1)}, y_1^{(2)}) = d_\mathbb{Y}\left( y_1^{(1)}, \operatorname{supp}(s_{x'}(\cdot \mid E_1)) \right) = 0,
$$
$$
d_\mathbb{Y}(y_1^{(1)}, y_2^{(2)}) = d_\mathbb{Y}\left( y_1^{(1)}, \operatorname{supp}(s_{x'}(\cdot \mid E_2)) \right) = 1.
$$

Similarly, the pairwise distances between all $y_t, y_u$ with $u > t > 1$ are identical to the distance of their respective supports: The batches have a distance of 1 because one can transform one into another using a single substitution. The supports have a distance of 1 because one can transition from one to another using a single substitution.

The result then immediately follows from Corollary F.7. $\qquad \square$

Next, we can derive the upper bound from [28]. For this derivation, we will use the following Lemma, which is proven in Appendix B of [6]:

**Lemma H.3.** *Consider two probability measures $P, Q$ on output measure space $(\mathbb{Z}, \mathcal{Z}, \lambda)$. Define $p = \mathrm{d}P \,/\, \mathrm{d}\lambda$ and $q = \mathrm{d}Q \,/\, \mathrm{d}\lambda$. Then*

$$
\Delta_\alpha(p \| q) = 1 + \sum_{l=2}^{\alpha} \binom{\alpha}{l} \int (p(z) - q(z))^l q(z)^{1-l} d\lambda(z).
$$

The following proof essentially follows that of [28], but skips their Appendix B.2, since we have already successfully decomposed mixtures $m_x$ and $m_{x'}$ into small terms that only involve base mechanism densities.

**Proposition H.4** (Wang et al. [28]). *Let $M = B \circ S$ be a subsampled mechanism, where $S$ is subsampling without replacement with batch size $q$. Let $\simeq_{\mathbb{Y}}$ be the substitution relation $\simeq_{\Delta,\mathbb{Y}}$. Then, for $\alpha > 1$ and all $x \simeq_{\Delta,\mathbb{X}} x'$ of size $N$, $\Delta_\alpha(m_x || m_{x'})$ is l.e.q.*

$$1 + 2\sum_{l=2}^{\alpha} \binom{\alpha}{l} w^l \max_{y \simeq_{\Delta,\mathbb{Y}} y'} \Delta_l(b_y || b_{y'}),$$

*with $w = q \,/\, N$.*

*Proof.* Using the constraint $y_1^{(1)} = y_1^{(2)}$ in Theorem H.1, we can rewrite its objective as

$$\max_{y_1^{(1)}, y_2^{(1)}, y_2^{(2)}} \Delta_\alpha((1-w) \cdot b_{y_1^{(1)}} + w \cdot b_{y_2^{(1)}} || (1-w) \cdot b_{y_1^{(1)}} + w \cdot b_{y_2^{(2)}}).$$

Using Lemma H.3, we can upper-bound its optimal value via

$$\max_{y_1^{(1)}, y_2^{(1)}, y_2^{(2)}} \Delta_\alpha((1-w) \cdot b_{y_1^{(1)}} + w \cdot b_{y_2^{(1)}} || (1-w) \cdot b_{y_1^{(1)}} + w \cdot b_{y_2^{(2)}})$$

$$= \max_{y_1^{(1)}, y_2^{(1)}, y_2^{(2)}} 1 + \sum_{l=2}^{\alpha} \binom{\alpha}{l} \int \frac{(w \cdot b_{y_2^{(1)}} - w \cdot b_{y_2^{(2)}})^l}{\left((1-w) \cdot b_{y_1^{(1)}} + w \cdot b_{y_2^{(2)}}\right)^{l-1}} d\lambda(z)$$

$$\leq \max_{y_1^{(1)}, y_2^{(1)}, y_2^{(2)}} 1 + \sum_{l=2}^{\alpha} \binom{\alpha}{l} w^l \int \frac{|b_{y_2^{(1)}} - b_{y_2^{(2)}}|^l}{\left((1-w) \cdot b_{y_1^{(1)}} + w \cdot b_{y_2^{(2)}}\right)^{l-1}} d\lambda(z)$$

$$\leq \sum_{l=0}^{\alpha} \binom{\alpha}{l} w^l \max_{y_1^{(1)}, y_2^{(1)}, y_2^{(2)}} \int \frac{|b_{y_2^{(1)}} - b_{y_2^{(2)}}|^l}{\left((1-w) \cdot b_{y_1^{(1)}} + w \cdot b_{y_2^{(2)}}\right)^{l-1}} d\lambda(z),$$

where each of the $\alpha + 1$ optimization problems is independently constrained by $d_{\mathbb{Y}}(y_1^{(1)}, y_2^{(1)}) \leq 1$, $d_{\mathbb{Y}}(y_1^{(1)}, y_2^{(2)}) \leq 1$, and $d_{\mathbb{Y}}(y_2^{(1)}, y_2^{(2)}) \leq 1$.

Next, we bound the optimal value of each of the $\alpha + 1$ optimization problems. Using the joint convexity of $x, y \mapsto x^l \cdot y^{1-l}$, which implies convexity in the second component, shows that

$$\max_{y_1^{(1)}, y_2^{(1)}, y_2^{(2)}} \int \frac{|b_{y_2^{(1)}} - b_{y_2^{(2)}}|^l}{\left((1-w) \cdot b_{y_1^{(1)}} + w \cdot b_{y_2^{(2)}}\right)^{l-1}} d\lambda(z)$$

$$\leq \max_{y_1^{(1)}, y_2^{(1)}, y_2^{(2)}} (1-w) \cdot \int \frac{|b_{y_2^{(1)}} - b_{y_2^{(2)}}|^l}{b_{y_1^{(1)}}^{l-1}} d\lambda(z) + w \cdot \int \frac{|b_{y_2^{(1)}} - b_{y_2^{(2)}}|^l}{b_{y_2^{(2)}}^{l-1}} d\lambda(z)$$

$$\leq (1-w) \cdot \left( \max_{y_1^{(1)}, y_2^{(1)}, y_2^{(2)}} \int \frac{|b_{y_2^{(1)}} - b_{y_2^{(2)}}|^l}{b_{y_1^{(1)}}^{l-1}} d\lambda(z) \right) + w \cdot \left( \max_{y_1^{(1)}, y_2^{(1)}, y_2^{(2)}} \int \frac{|b_{y_2^{(1)}} - b_{y_2^{(2)}}|^l}{b_{y_2^{(2)}}^{l-1}} d\lambda(z) \right)$$

$$\leq (1-w) \cdot \left( \max_{y_1^{(1)}, y_2^{(1)}, y_2^{(2)}} \int \frac{|b_{y_2^{(1)}} - b_{y_2^{(2)}}|^l}{b_{y_1^{(1)}}^{l-1}} d\lambda(z) \right) + w \cdot \left( \max_{y_1^{(1)}, y_2^{(1)}, y_2^{(2)}} \int \frac{|b_{y_2^{(1)}} - b_{y_2^{(2)}}|^l}{b_{y_1^{(1)}}^{l-1}} d\lambda(z) \right)$$

$$= \max_{y_1^{(1)}, y_2^{(1)}, y_2^{(2)}} \int \frac{|b_{y_2^{(1)}} - b_{y_2^{(2)}}|^l}{b_{y_1^{(1)}}^{l-1}} d\lambda(z),$$

where all optimization problems are independent, with each one being independently constrained by $d_{\mathbb{Y}}(y_1^{(1)}, y_2^{(1)}) \leq 1$, $d_{\mathbb{Y}}(y_1^{(1)}, y_2^{(2)}) \leq 1$, and $d_{\mathbb{Y}}(y_2^{(1)}, y_2^{(2)}) \leq 1$. Note that, for the last inequality, we replaced a $y_2^{(2)}$ in the second maximization with a $y_1^{(1)}$, which essentially adds a degree of freedom and thus leads to an upper bound.

In [28], the optimal value of the final problem is referred to as the ternary-$|\chi|^l$-divergence of $b_2$, $b_{y_2^{(2)}}$, and $b_{y_1^{(1)}}$. As shown in their Lemma 19, it can be upper bounded via $2 \cdot \max_{y \simeq_{\Delta,\mathbb{Y}} y'} \Delta_l(b_y || b_{y'})$, which concludes our proof. $\qquad\square$

**Additional terms.** Note that [28] derive three additional bounds (see their Lemma 17, Lemma 19, and Theorem 27) on the the ternary-$|\chi|^l$-divergence. This introduces additional terms, but does not change the fact that their result is lower-bounded by Theorem H.1. We use their full theorem as a baseline in our experiments.

## H.2  Poisson subsampling and insertion/removal

Next, we show that the Poisson subsampling guarantees in [30, 7] follow from another optimal transport problem. For our proof, we will use the following Lemma, which corresponds to the "novel alternative decomposition" in Appendix A.1 of [30].

**Lemma H.5.** *Consider $K + 1 \in \mathbb{N}$ distributions $P, Q_1, \ldots, Q_K$ on output measure space $(\mathbb{Z}, \mathcal{Z}, \lambda)$, and define $p = \mathrm{d}P / \mathrm{d}\lambda$, $q_k = \mathrm{d}Q_k / \mathrm{d}\lambda$. Further consider some $w_1, \ldots, w_K \in [0, 1]$ with $\sum_{k=1}^{K} w_k = 1$. Then,*

$$
\Delta_\alpha \left( p \| \sum_{k=1}^{K} w_k \cdot q_k \right) \leq \sum_{k=1}^{K} w_k \cdot \Delta_\alpha \left( q_k + p - \sum_{l=1}^{K} w_l \cdot q_l \| q_k \right).
$$

*Proof.* Based on the definition of $\Delta_\alpha$, we have

$$
\Delta_\alpha \left( p \| \sum_{k=1}^{K} w_k \cdot q_k \right)
$$

$$
= \int \frac{p(z)^\alpha}{\left( \sum_{k=1}^{K} w_k \cdot q_k(z) \right)^{\alpha-1}} \, \mathrm{d}\lambda(z)
$$

$$
= \int \frac{\left( \left( \sum_{k=1}^{K} w_k \cdot q_k(z) \right) + p(z) - \left( \sum_{l=1}^{K} w_l \cdot q_l(z) \right) \right)^\alpha}{\left( \sum_{k=1}^{K} w_k \cdot q_k(z) \right)^{\alpha-1}} \, \mathrm{d}\lambda(z)
$$

$$
= \int \frac{\left( \sum_{k=1}^{K} w_k \cdot \left( q_k(z) + p(z) - \sum_{l=1}^{K} w_l \cdot q_l(z) \right) \right)^\alpha}{\left( \sum_{k=1}^{K} w_k \cdot q_k(z) \right)^{\alpha-1}} \, \mathrm{d}\lambda(z)
$$

The result then follows from joint convexity of $\Delta_\alpha$.  $\square$

Note that the proof of this lemma is very similar to the proof strategy we used in deriving Proposition H.4 using Lemma H.3: We add $0 = c - c$ with some $c$ to the numerator (which leads to the binomial expansion in Lemma H.3) and then apply joint convexity to obtain an upper bound.

**Proposition H.6** (Zhu and Wang [30]). *Let $M = B \circ S$ be a subsampled mechanism, where $S$ is Poisson subsampling with rate $r$. Let $\simeq_\mathbb{Y}$ be the insertion/removal relation $\simeq_{\pm,\mathbb{Y}}$. Then, for $\alpha > 1$ and all $x \simeq_{\pm,\mathbb{X}} x'$, $\Delta_\alpha(m_x \| m_{x'})$ is l.e.q.*

$$
2 \cdot \sum_{l=0}^{\alpha} \binom{\alpha}{l} r^l (1-r)^{\alpha-l} \max_{y \simeq_{\pm,\mathbb{Y}} y'} \Delta_l(b_y \| b_{y'}).
$$

*Proof.* Since we are concerned with insertion/removal, we need to consider two cases:

**Case 1: Removal.** In this case, there is some $a \in x$ such that $x' = x \setminus \{a\}$. We let $A_1$ be the event that $a$ is not sampled, i.e. $A_1 = \{y \in \mathbb{Y} \mid a \notin y\}$, and let $A_2 = \overline{A_1}$. We let $E_1 = \mathbb{Y}$, i.e., do not condition on any particular event.

By definition of Poisson subsampling, we have $P_{S_x}(A_1) = 1 - r$, $P_{S_x}(A_2) = r$, and $P_{S_{x'}}(E_1) = 1$. We further have

$$
s_x(y \mid A_1) = \begin{cases} r^{|y|}(1-r)^{|x|-|y|-1} & \text{if } y \subseteq x \wedge a \notin y \\ 0 & \text{otherwise} \end{cases}
$$

$$
s_x(y \mid A_2) = \begin{cases} r^{|y|-1}(1-r)^{|x|-|y|} & \text{if } y \subseteq x \wedge a \in y \\ 0 & \text{otherwise} \end{cases},
$$

and

$$s_{x'}(y \mid E_1) = \begin{cases} r^{|y|}(1-r)^{|x'|-|y|} & \text{if } y \subseteq x' \\ 0 & \text{otherwise} \end{cases} = \begin{cases} r^{|y|}(1-r)^{|x|-|y|-1} & \text{if } y \subseteq x \wedge a \notin y \\ 0 & \text{otherwise} \end{cases}.$$

Note that $s_x(y \mid A_1) = s_{x'}(y \mid E_1)$.

**Coupling.** We now define a coupling $\gamma : \mathbb{Y}^{2+1} \to \mathbb{R}_+$ that corresponds to the following generative process: We first generate $y_1^{(1)}$ by sampling a batch that does not contain $a$ via Poisson subsampling from $x \setminus \{a\}$. We then let $y_1^{(2)} \leftarrow y_1^{(1)}$. Finally, we deterministically insert $a$ to generate $y_2^{(1)}$. This can be formally defined via

$$\gamma(\boldsymbol{y}^{(1)}, \boldsymbol{y}^{(2)}) = \begin{cases} s_x(y_1^{(1)} \mid A_1) & \text{if } y_1^{(1)} = y_1^{(2)} \wedge y_2^{(1)} = y_1^{(1)} \cup \{a\} \\ 0 & \text{otherwise.} \end{cases}$$

**Validity.** We can verify the validity of this coupling as follows: For every $y_1^{(1)}$ with $s_x(y_1^{(1)} \mid A_1) > 0$, there is exactly one combination $y_2^{(1)}, y_1^{(2)}$ for which $\gamma(\boldsymbol{y}) > 0$, namely $y_1^{(2)} = y_1^{(1)}$ and $y_2^{(1)} = y_1^{(1)} \cup \{a\}$. We thus have $\sum_{y_2^{(1)}, y_1^{(2)} \in \mathbb{Y}^2} \gamma(y_1^{(1)}, y_2^{(1)}, y_1^{(2)}) = s_x(y_1^{(1)} \mid A_1)$. The proof for $y_1^{(2)}$ is analogous. For every $y_2^{(1)}$, we have exactly one combination of $y_1^{(1)}, y_1^{(2)}$ for which $\gamma(\boldsymbol{y}) > 0$, namely $y_1^{(1)} = y_1^{(2)} = y_2^{(1)} \setminus \{a\}$. We thus have

$$\sum_{y_1^{(1)}, y_1^{(2)} \in \mathbb{Y}^2} \gamma(y_1^{(1)}, y_2^{(1)}, y_1^{(2)})$$

$$= s_x(y_2^{(1)} \setminus \{a\} \mid A_1) = r^{|y_2^{(1)}|-1}(1-r)^{|x|-(|y|-1)-1} = r^{|y_2^{(1)}|-1}(1-r)^{|x|-|y|} = s_x(y_2^{(1)} \mid A_2).$$

**Compatibility.** Finally, it can be easily shown that $\gamma$ is a $d_{\mathbb{Y}}$-compatible coupling (see Appendix F). Whenever $\gamma(\boldsymbol{y}) > 0$, then

$$d_{\mathbb{Y}}(y_1^{(1)}, y_2^{(1)}) = d_{\mathbb{Y}}\left(y_1^{(1)}, \mathrm{supp}(s_x(\cdot \mid A_2))\right) = 1,$$

$$d_{\mathbb{Y}}(y_1^{(1)}, y_1^{(2)}) = d_{\mathbb{Y}}\left(y_1^{(1)}, \mathrm{supp}(s_{x'}(\cdot \mid E_1))\right) = 0.$$

Similarly, the pairwise distance between $y_2^{(1)}$ and $y_1^{(2)}$ is always 1. This is identical to the distance of their supports, because one can always transition between them via a single insertion/removal.

It thus follows from Corollary F.7 that

$$\Delta_\alpha(m_x || m_{x'}) \le \max_{y_1^{(1)}, y_2^{(1)}, y_1^{(2)}} \Delta_\alpha((1-r) \cdot b_{y^{(1)}} + r \cdot b_{y_2^{(1)}} || b_{y_1^{(2)}})$$

$$\text{s.t.} \quad d_{\pm,\mathbb{Y}}(y_1^{(1)}, y_2^{(1)}) \le 1, y_1^{(1)} = y_1^{(2)}, d_{\pm,\mathbb{Y}}(y_2^{(1)}, y_1^{(2)}) \le 1$$

Because $y_1^{(1)} = y_1^{(2)}$ and because $d_{\pm,\mathbb{Y}}(y_1^{(1)}, y_2^{(1)}) \le 1$ is equivalent to $y_1^{(1)} \simeq_{\pm,\mathbb{Y}} y_2^{(1)}$, this bound can be restated as

$$\Delta_\alpha(m_x || m_{x'}) \le \max_{y \simeq_{\pm,\mathbb{Y}} y'} \Delta_\alpha((1-r) \cdot b_y + r \cdot b_{y'} || b_y).$$

Finally, one can use the definition of $\Delta_\alpha$ and binomial expansion to show

$$\Delta_\alpha(m_x || m_{x'}) \le \max_{y \simeq_{\pm,\mathbb{Y}} y'} \int_{\mathbb{Z}} \frac{((1-r)b_y(z) + rb_{y'}(z))^\alpha}{b_y(z)^{\alpha-1}} \, \mathrm{d}\lambda(z)$$

$$= \max_{y \simeq_{\pm,\mathbb{Y}} y'} \int_{\mathbb{Z}} \left(\frac{(1-r)b_y(z) + rb_{y'}(z)}{b_y(z)}\right)^\alpha b_y(z) \, \mathrm{d}\lambda(z)$$

$$= \max_{y \simeq_{\pm,\mathbb{Y}} y'} \int_{\mathbb{Z}} \left((1-r) + r\frac{b_{y'}(z)}{b_y(z)}\right)^\alpha b_y(z) \, \mathrm{d}\lambda(z)$$

$$= \max_{y \simeq_{\pm,\mathbb{Y}} y'} \sum_{l=0}^{\alpha} \binom{\alpha}{l} r^l (1-r)^{\alpha-l} \Delta_l(b_y || b_{y'})$$

$$\le \sum_{l=0}^{\alpha} \binom{\alpha}{l} r^l (1-r)^{\alpha-l} \max_{y \simeq_{\pm,\mathbb{Y}} y'} \Delta_l(b_y || b_{y'}).$$

Note that this is smaller than the term in Proposition H.6 by a factor of 2.

**Case 2: Insertion** In this case, there is some $a \in x'$ such that $x = x' \setminus \{a\}$. Very similar to before, we let $A_1 = \mathbb{Y}$, i.e., we do not condition on any particular event. We further let $E_1$ be the event that $a$ is not sampled, i.e. $E_1 = \{y \in \mathbb{Y} \mid a \notin y\}$, and let $E_2 = \overline{E_1}$. We notice that we have exactly the same conditional distributions as in the first case. We can thus define exactly the same coupling (up to changes in indexing) to show that

$$\Delta_\alpha(m_x || m_{x'}) \leq \max_{y \simeq_{\pm, \mathbb{Y}} y'} \Delta_\alpha(b_y || (1-r)b_y + rb_{y'}).$$

Next, applying Lemma H.5 and using the definition of $\Delta_\alpha$ shows that

$$\Delta_\alpha(m_x || m_{x'})$$
$$\leq \max_{y \simeq_{\pm, \mathbb{Y}} y'} \Delta_\alpha(b_y || (1-r)b_y + rb_{y'})$$
$$\leq \max_{y \simeq_{\pm, \mathbb{Y}} y'} (1-r)\Delta_\alpha(b_y + b_y - ((1-r)b_y + rb_{y'})||b_y) + r\Delta_\alpha(b_{y'} + b_y - ((1-r)b_y + rb_{y'})||b_{y'})$$
$$= \max_{y \simeq_{\pm, \mathbb{Y}} y'} (1-r)\Delta_\alpha((1+r)b_y - rb_{y'}||b_y) + r\Delta_\alpha((1-r)b_{y'} + rb_y||b_{y'})$$

This corresponds exactly to Eq. 6 of the "novel alternative decomposition" in Appendix A.1 of [30]. One can then through the remaining steps in their Appendix A to show that this term is at most two times larger than the one we derived in the deletion case. □

As already mentioned at the beginning of this section, the proof is very similar to that in [30], except for the discussion of couplings and $d_\mathbb{Y}$-compatibility. We emphasize that the crucial aspect, which is not discussed in any prior work on Rényi-DP amplification, is that there is an implicit, underlying optimal transport problem between conditional subsampling distributions. Now that we have identified this connection, we can apply this more general optimal transport principle to a much broader range of amplification by subsampling scenarios for Rényi-DP.

**Further tightening.** The factor 2 in Proposition H.6 can be eliminated for distributions with particular symmetries (see Theorem 5 in [29]) or bounded Pearson-Vajda $\chi^l$-pseudo-divergence (see Theorem 8 in [30]). We thus do not include the factor 2 when using this amplification guarantee as a baseline in our experiments.

### H.3 Graph subsampling without replacement and node modification

Daigavane et al. [40] consider a setting that differs from the usual insertional/removal into datasets: Node-level privacy for graphs. There, the dataset space is the set of all directed, attributed graphs $\mathbb{X} = \bigcup_{N,D\in\mathbb{N}} \mathbb{R}^{N\times D} \times \{0,1\}^{N\times N}$, which are composed of a continuous feature matrix and a discrete adjacency matrix. Two graphs $x, x'$ are related by dataset relation $\simeq_\mathbb{X}$ if $x'$ can be constructed by inserting a node (including new edges) into $x$, or removing a node (including its edges) from $x$.

To analyze this problem setting, they perform a preprocessing step (see their Algorithm 2) to represent each graph by a set of subgraphs, with each subgraph corresponding to a node in the graph. Via this construction, they return to the traditional setting where $\mathbb{X} \subseteq \mathcal{P}(\mathbb{A})$ for some set $\mathbb{A}$, and the modification of a node's features and edges corresponds to the substitution of $K$ elements,[4] i.e.

$$\simeq_{K\Delta, \mathbb{X}} = \{(x, x') \in \mathbb{X} \mid \exists g \in x \setminus x', g' \in x' \setminus x : x' = x \setminus g \cup g' \wedge |g| = |g'| = K\}, \quad (22)$$

where $K$ depends on the maximum considered graph degree.

Irrespective of the graph neural network specific details, this discussion suggests that group privacy and differentially private learning for graphs are related.

We shall now demonstrate that their result for subsampling without replacement can be derived from Theorem E.1, i.e. optimal transport without conditioning.

**Proposition H.7** (Daigavane et al. [40])**.** *Let $M = B \circ S$ be a subsampled mechanism, where $S$ is subsampling without replacement with batch size $q$. Let $\simeq_\mathbb{Y}$ be the substitution relation $\simeq_{\Delta, \mathbb{Y}}$. Let*

---

[4]Note that they do not actually consider insertion/removal, because their analysis assumes the number of subgraphs to remain constant.

$\simeq_{K\Delta,\mathbb{X}}$ *be the K-fold substitution relation defined in Eq. (22). Then, for $\alpha > 1$ and all $x \simeq_{K\Delta,\mathbb{X}} x'$ of size $N$,*

$$\Delta_\alpha(m_x||m_{x'}) \leq \sum_{k=0}^{K} w_k \cdot \max_{y \simeq_{k\Delta,\mathbb{Y}} y'} \Delta_k(b_y||b_{y'}).$$

*with $w_k = \text{HyperGeom}(k \mid N, K, q)$.*

*Proof.* Consider arbitrary $x \simeq_{\Delta,\mathbb{X}} x'$. By definition of $\simeq_\Delta$, there must be some $g \in x \setminus x'$, $g' \in x' \setminus x$ such that $x' = x \setminus g \cup g'$ and $|g| = |g'| = K$.

We condition on the events $A_1 = \mathbb{Y}$ and $E_1 = \mathbb{Y}$, meaning

$$s_x(y \mid A_1) = \begin{cases} \binom{N}{q}^{-1} & \text{if } y \subseteq x \\ 0 & \text{otherwise} \end{cases} \quad \text{and} \quad s_{x'}(y \mid E_1) = \begin{cases} \binom{N}{q}^{-1} & \text{if } y \subseteq x' \\ 0 & \text{otherwise} \end{cases}.$$

**Coupling.** We now define a coupling via $\gamma : \mathbb{Y}^{1+1} \to \mathbb{R}_+$ that define the following generative process: We first define $y^{(1)}$ by sampling a batch of size $q$ from $x$ without replacement. We then remove all elements that are in $g$ and insert an equal number of elements, which are chosen uniformly at random from $g'$. This coupling can be formally defined as

$$\gamma(y^{(1)}, y^{(2)}) = \begin{cases} s_x(y^{(1)} \mid A_1) \cdot \binom{K}{|y^{(1)} \cap g|}^{-1} & \text{if } y^{(1)}, y^{(2)} \text{ fulfills Condition H.8,.} \\ 0 & \text{otherwise.} \end{cases}$$

**Condition H.8.** A tuple $y^{(1)} \in \mathbb{Y}$, $y^{(2)} \in \mathbb{Y}$ fulfills this condition when there exists a $\tilde{g} \subseteq g$ such that $y^{(2)} = y^{(1)} \setminus g \cup \tilde{g}$ and $|y^{(2)} \cap \tilde{g}| = |y^{(1)} \cap g|$.

**Validity.** We now show that this constitutes a valid coupling. Consider any $y^{(1)}$ with $s_x(y^{(1)} \mid A_1) > 0$ and $|y^{(1)} \cap g| = k$. There are exactly $\binom{K}{k}$ different $y^{(2)}$ for which $\gamma(\boldsymbol{y})$ is non-zero, and each one has the same probability. Thus,

$$\sum_{y^{(2)} \in \mathbb{Y}} \gamma(\boldsymbol{y}) = \binom{K}{k} \cdot s_x(y^{(1)} \mid A_1) \cdot \binom{K}{k}^{-1} = s_x(y^{(1)} \mid A_1) \cdot \binom{|x-1|}{q}.$$

The other case is analogous.

**Compatibility.** Evidently, this is a $d_\mathbb{Y}$-compatible coupling, because (a) whenever $y^{(1)}$ contains $k$ elements from $g$, it has a distance of $k$ from $s_{x'}(y \mid E_1)$ and (b) whenever $y^{(1)}$ contains $k$ elements from $g$, we generate $y^{(2)}$ by substituting exactly $k$ elements.

It thus follows from Corollary F.7 that

$$\Delta_\alpha(m_x||m_{x'}) \leq \sum_y s_x(y^{(1)} \mid A_1)\kappa_\alpha(y^{(1)})$$

with

$$\kappa_\alpha(y^{(1)}) = \max_{y,y'} \Delta_\alpha(b_y||b_{y'}) \quad \text{s.t.} \quad d_\mathbb{Y}(y, y') \leq |y^{(1)} \cap g|.$$

The result then follows from the definition of induced distance, the definition of the K-fold substitution relation $\simeq_{K\Delta,\mathbb{X}}$ and the fact that $|y^{(1)} \cap g|$ is a random variable with distribution $\text{HyperGeom}(N, K, q)$. $\qquad\square$

Note that [40] instantiate this result with Gaussian mechanisms with sensitivity $C$, for which $\max_{y \simeq_{k\Delta,\mathbb{Y}} y'} \Delta_\alpha(b_y||b_{y'}) = \exp(\alpha \cdot (\alpha - 1) \cdot \frac{k^2 C^2}{\sigma^2})$.

In Appendix B.1.4, we experimentally demonstrate that this result can improve upon the baseline of combining [28] with the traditional group privacy property, but is not sufficiently tight to consistently outperform it across a wide range of parameters. This emphasizes the benefit of considering optimal transport between proper conditional distributions when deriving amplification guarantees.

# I  Novel RDP guarantees

Now that we have a framework that enables generic subsampling analysis for Rényi differential privacy, we can derive a variety of novel guarantees that were previously only available for approximate differential privacy.

In the following, we first demonstrate that RDP guarantees for a dataset relation $\simeq_{\mathbb{X}}$ can be derived from a base mechanism that is only known to be RDP under a different batch relation $\simeq_{\mathbb{Y}}$, i.e., "hybrid neighboring relations" [15]. We then show that we can derive RDP guarantees for non-standard combinations of subsampling schemes and neighboring relations, such as subsampling without replacement and insertion/removal. Finally, we derive a simple, tight mechanism-specific subsampling guarantee for randomized response. We use this guarantee in Section 4 to demonstrate the limitations of using mechanism-agnostic RDP bounds instead of mechanism-specific RDP bounds.

For this section, recall again that $\Lambda_\alpha$ is not the Rényi divergence, but its $\alpha$th moment, i.e., a scaled and exponentiated Rényi divergence (see Definition D.8).

## I.1  Hybrid neighboring relations

Hybrid neighboring relations have thus far not been discussed in the context of Rényi-DP. Analyzing such scenarios may be particularly useful when the dataset space $\mathbb{X}$ and the batch space $(\mathbb{Y}, \mathcal{Y})$ are different from each other, e.g., when mapping from large text corpora to short sequences of token embeddings. Note that hybrid relations may also be useful for noisy stochastic gradient descent with fixed batch sizes (see end of Appendix I.2).

As an example, we consider the following scenario: Subsampling without replacement, with substitution relation $\simeq_{\Delta,\mathbb{X}}$ for datasets and insertion/removal relation $\simeq_{\pm,\mathbb{Y}}$ for batches. The proof is largely identical to that of Theorem H.1, but uses the fact that a substitution can be represented by an insertion, followed by a removal.

**Theorem I.1.** *Let $M = B \circ S$ be a subsampled mechanism, where $S$ is subsampling without replacement with batch size $q$. Let $\simeq_{\mathbb{Y}}$ be the insertion/removal relation $\simeq_{\pm,\mathbb{Y}}$ and $\simeq_{\mathbb{X}}$ be the substitution relation $\simeq_{\Delta,\mathbb{X}}$. Then, for $\alpha > 1$ and all $x \simeq_{\Delta,\mathbb{X}} x'$ of size $N$,*

$$\Lambda_\alpha(m_x || m_{x'}) \leq \max_{\boldsymbol{y}} \Lambda_\alpha((1-w) \cdot b_{y_1^{(1)}} + w \cdot b_{y_2^{(1)}} || (1-w) \cdot b_{y_1^{(2)}} + w \cdot b_{y_2^{(2)}})$$

*with $\boldsymbol{y} \in \mathbb{Y}^{2+2}$ subject to $y_1^{(1)} = y_1^{(2)}$, $d_{\pm,\mathbb{Y}}(y_1^{(1)}, y_2^{(1)}) \leq 2$, $d_{\pm,\mathbb{Y}}(y_1^{(1)}, y_2^{(2)}) \leq 2$, $d_{\pm,\mathbb{Y}}(y_2^{(1)}, y_2^{(2)}) \leq 2$, and with $w = q / N$.*

*Proof.* Consider arbitrary $x \simeq_{\Delta,\mathbb{X}} x'$. By definition of $\simeq_\Delta$, there must be some $a \in x$, $a' \in x'$ such that $x' = x \setminus \{a\} \cup \{a'\}$. We thus define both $A_1$ and $E_1$ from Theorem E.2 to be the event that neither $a$ nor $a'$ is sampled, i.e., $A_1 = E_1 = \{y \in \mathbb{Y} \mid y \cap \{a, a'\} = \varnothing\}$. We further define $A_2$ and $E_2$ to be the event that $a$ or $a'$ is sampled, i.e., $A_2 = \overline{A_1}$ and $E_2 = \overline{E_1}$.

By definition of subsampling without replacement, we have

$$P_{S_x}(A_1) = P_{S_{x'}}(E_1) = \text{HyperGeom}(0 \mid N, 1, q) = 1 - \frac{q}{N},$$

$$P_{S_x}(A_2) = P_{S_{x'}}(E_2) = \text{HyperGeom}(1 \mid N, 1, q) = \frac{q}{N},$$

which corresponds to the weights $(1-w)$ and $w$, respectively.

**Coupling.** We now define a coupling via $\gamma : \mathbb{Y}^4 \to \mathbb{R}_+$:

$$\gamma(\boldsymbol{y}^{(1)}, \boldsymbol{y}^{(2)}) = \begin{cases} s_x(y_1^{(1)} \mid A_1) \cdot \frac{1}{q} & \text{if } \boldsymbol{y}^{(1)}, \boldsymbol{y}^{(2)} \text{ fulfill Condition H.2,} \\ 0 & \text{otherwise.} \end{cases}$$

**Condition H.2.** A tuple $\boldsymbol{y}^{(1)} \in \mathbb{Y}^2$, $\boldsymbol{y}^{(2)} \in \mathbb{Y}^2$ fulfills this condition when $y_1^{(2)} = y_1^{(1)}$ and $\exists \tilde{a} \in y_1^{(1)} : \left( y_2^{(1)} = y_1^{(1)} \setminus \{\tilde{a}\} \cup \{a\} \land y_2^{(2)} = y_1^{(1)} \setminus \{\tilde{a}\} \cup \{a'\} \right)$.

As explained before, $\gamma$ defines the following generative process: We first generate $y_1^{(1)}$ by sampling a batch that does not contain $a$ uniformly at random from x. We then let $y_1^{(2)} \leftarrow y_1^{(1)}$ Finally, we pick a random element $\tilde{a}$ of $y_1^{(1)}$ and replace it with $a$ and $a'$ to generate $y_2^{(1)}$ and $y_2^{(2)}$, respectively. Note that each of these substitution corresponds to exactly one insertion and one removal.

**Validity.** The validity of this coupling has already been demonstrated in our proof of Theorem H.1, since validity of a coupling does not depend on batch neighboring relation $\simeq_{\mathbb{Y}}$.

**Compatibility.** We can thus focus on showing that $\gamma$ is a $d_{\mathbb{Y}}$-compatible coupling (see Appendix F). Whenever $\gamma(\boldsymbol{y}) > 0$, then

$$d_{\mathbb{Y}}(y_1^{(1)}, y_2^{(1)}) = d_{\mathbb{Y}}\left(y_1^{(1)}, \mathrm{supp}(s_x(\cdot \mid A_2))\right) = 2,$$

$$d_{\mathbb{Y}}(y_1^{(1)}, y_1^{(2)}) = d_{\mathbb{Y}}\left(y_1^{(1)}, \mathrm{supp}(s_{x'}(\cdot \mid E_1))\right) = 0,$$

$$d_{\mathbb{Y}}(y_1^{(1)}, y_2^{(2)}) = d_{\mathbb{Y}}\left(y_1^{(1)}, \mathrm{supp}(s_{x'}(\cdot \mid E_2))\right) = 2.$$

We see that the pairwise distances between all $y_t, y_u$ with $u > t > 1$ are identical to the distance of their respective supports: The batches have a distance of 2 because one can transform one into another using a single substitution, i.e. an insertion and a removal. The supports also have a distance of 2 because one can transition from one to another using a single substitution.

The result then immediately follows from Corollary F.7. $\qquad\qquad\square$

If we wanted to express this result in terms of the Rényi-DP parameters of the base mechanism, we could go through the same derivations as in our proof of Proposition H.4 to show that

$$\Lambda_\alpha(m_x||m_{x'}) \leq 1 + 2\sum_{l=2}^{\alpha} \binom{\alpha}{l} w^l \left(\max_{y,y'} \Lambda_l(b_y||b_{y'}) \quad \text{s.t.} \quad d_{\pm,\mathbb{Y}}(y,y') \leq 2\right),$$

with $w = q \,/\, N$. The inner term can, for instance, be evaluated using the traditional group privacy property from [31].

### I.2 Subsampling without replacement under insertion/removal

To demonstrate that our framework is not limited to analyzing the usually considered combination of subsampling without replacement and substitution or Poisson subsampling and insertion/removal, we consider the following non-standard combination: Subsampling without replacement and insertion/removal.

We show that we can derive a guarantee that is qualitatively similar to that of [30], while preserving a fixed batch size. Such results could be useful when implementing noisy stochastic gradient descent in deep learning frameworks with static computation graphs, where the variable batch sizes resulting from Poisson subsampling may be problematic. However, note that this guarantee only guarantees privacy for spaces of datasets whose elements are larger than some $Q \in \mathbb{N}$ with $Q > q$, where q is the batch size.

For the following proof, we condition on the same events as in our proof of Proposition H.6, but need to construct a different $d_{\mathbb{Y}}$-compatible coupling, since we have a different subsampling distribution.

**Theorem I.2.** *Assume a dataset space and batch space defined by $\mathbb{X} \subseteq \{x \in \mathcal{P}(\mathbb{A}) \mid |x| > Q\}$, $\mathbb{Y} = \{y \subseteq x \mid x \in \mathbb{X}, |y| = q\}$, $\mathcal{Y} = \mathcal{P}(\mathbb{Y})$ and finite set $\mathbb{A}$, with $Q, q \in \mathbb{N}$ and $Q > q$. Let $M = B \circ S$ be a subsampled mechanism, where $S$ is subsampling without replacement with batch size q r. Let $\simeq_{\mathbb{Y}}$ be the insertion/removal relation $\simeq_{\pm,\mathbb{Y}}$. Then, for $\alpha > 1$ and all $x \simeq_{\pm,\mathbb{X}} x'$,*

$$\Lambda_\alpha(m_x||m_{x'}) \leq \max_{N \in \mathbb{N}} \left(2 \cdot \sum_{l=0}^{\alpha} \binom{\alpha}{l} w^l (1-w)^{\alpha-l} \left(\max_{y,y'} \Lambda_l(b_y||b_{y'}) \; s.t. \; d_{\mathbb{Y}}(y,y') \leq 2\right)\right)$$

*subject to $N > Q$ and with $w = q \,/\, N$.*

*Proof.* Like in our proof of Proposition H.6, we need to distinguish between insertion and removal.

**Case 1: Removal.** In this case, there is some $a \in x$ such that $x' = x \setminus \{a\}$. We define $N = |x'|$, which fulfills $N > Q$ by definition of dataset space $\mathbb{X}$.

We let $A_1$ be the event that $a$ is not sampled, i.e. $A_1 = \{y \in \mathbb{Y} \mid a \notin y\}$, and let $A_2 = \overline{A_1}$. We let $E_1 = \mathbb{Y}$, i.e., do not condition on any particular event.

By definition of subsampling without replacement, we have $P_{S_x}(A_1) = 1 - qN$, $P_{S_x}(A_2) = q \,/\, N$, and $P_{S_{x'}}(E_1) = 1$. We further have

$$s_x(y \mid A_1) = \begin{cases} \binom{N}{q}^{-1} & \text{if } y \subseteq x \wedge a \notin y \\ 0 & \text{otherwise} \end{cases}, \qquad s_x(y \mid A_2) = \begin{cases} \binom{N}{q-1}^{-1} & \text{if } y \subseteq x \wedge a \in y \\ 0 & \text{otherwise} \end{cases},$$

and

$$s_{x'}(y \mid E_1) = \begin{cases} \binom{N}{q}^{-1} & \text{if } y \subseteq x' \\ 0 & \text{otherwise} \end{cases}.$$

Note that $s_x(y \mid A_1) = s_{x'}(y \mid E_1)$ and that all three conditional distribution are subsampling without replacement from sets of size $N$, not of size $N+1$ or $N-1$. Further note that, for event $A_1$, we only need to sample $q-1$ elements, since one element is always fixed to be $a$.

**Coupling.** We now define a coupling via $\gamma : \mathbb{Y}^{2+1} \to \mathbb{R}_+$:

$$\gamma(\boldsymbol{y}^{(1)}, \boldsymbol{y}^{(2)}) = \begin{cases} s_x(y_1^{(1)} \mid A_1) \cdot \frac{1}{q} & \text{if } y_1^{(1)} = y_1^{(2)} \wedge y_2^{(1)} = y_1^{(1)} \cup \{a\} \\ 0 & \text{otherwise.} \end{cases}$$

Simply put, $\gamma$ defines the following generative process: We first generate $y_1^{(1)}$ by sampling a batch that does not contain $a$ via subsampling without replacement from $x \setminus \{a\}$. We then let $y_1^{(2)} \leftarrow y_1^{(1)}$. We then randomly replace one of the $q$ batch elements with $a$ to generate $y_2^{(1)}$.

**Validity.** We can verify the validity of this coupling as follows: For every $y_1^{(1)}$ with $s_x(y_1^{(1)} \mid A_1) > 0$, there are exactly $q$ combination $y_2^{(1)}, y_1^{(2)}$ for which $\gamma(\boldsymbol{y}) > 0$. We thus have $\sum_{y_2^{(1)}, y_1^{(2)} \in \mathbb{Y}^2} \gamma(y_1^{(1)}, y_2^{(1)}, y_1^{(2)}) = s_x(y_1^{(1)} \mid A_1)$. The proof for $y_1^{(2)}$ is analogous.

For every $y_2^{(1)}$, there are exactly $N - (q-1)$ combinations of $y_1^{(1)}, y_1^{(2)}$ for which $\gamma(\boldsymbol{y}) > 0$. This is because $a$ must have replaced one of the $N - (q-1)$ elements of $x'$ that are not in $y_1^{(2)}$.

We thus have

$$\sum_{y_1^{(1)}, y_1^{(2)} \in \mathbb{Y}^2} \gamma(y_1^{(1)}, y_2^{(1)}, y_1^{(2)})$$

$$= (N - (q-1)) \cdot \binom{N}{q}^{-1} \cdot \frac{1}{q} = \left( \frac{q}{N-q+1} \cdot \binom{N}{q} \right)^{-1} = \binom{N}{q-1}^{-1}.$$

**Compatibility.** Finally, it can be easily shown that $\gamma$ is a $d_{\mathbb{Y}}$-compatible coupling. Whenever $\gamma(\boldsymbol{y}) > 0$, then

$$d_{\mathbb{Y}}(y_1^{(1)}, y_2^{(1)}) = d_{\mathbb{Y}}\left( y_1^{(1)}, \operatorname{supp}(s_x(\cdot \mid A_2)) \right) = 2,$$

$$d_{\mathbb{Y}}(y_1^{(1)}, y_1^{(2)}) = d_{\mathbb{Y}}\left( y_1^{(1)}, \operatorname{supp}(s_{x'}(\cdot \mid E_1)) \right) = 0.$$

Similarly, the pairwise distance between $y_2^{(1)}$ and $y_1^{(2)}$ is always 2. This is identical to the distance of their supports, because one can always transition between them via a substitution, i.e., an insertion and a removal.

It thus follows from Corollary F.7 that

$$\Lambda_\alpha(m_x \| m_{x'}) \le \max_{y_1^{(1)}, y_2^{(1)}, y_1^{(2)}} \Lambda_\alpha((1 - q \,/\, N) \cdot b_{y_1^{(1)}} + q \,/\, N \cdot b_{y_2^{(1)}} \| b_{y_1^{(2)}})$$

$$\text{s.t.} d_{\pm, \mathbb{Y}}(y_1^{(1)}, y_2^{(1)}) \le 1, y_1^{(1)} = y_1^{(2)}, d_{\pm, \mathbb{Y}}(y_2^{(1)}, y_1^{(2)}) \le 2$$

As in our proof of Proposition H.6 one can use the definition of $\Lambda_\alpha$ and binomial expansion to finally show that

$$\Lambda_\alpha(m_x||m_{x'}) \leq \sum_{l=0}^{\alpha} \binom{\alpha}{l} w^l (1-w)^{\alpha-l} \left( \max_{y,y'} \Lambda_l(b_y||b_{y'}) \quad \text{s.t.} \quad d_{\mathbb{Y}} y, y' \leq 2. \right)$$

with $w = q / N$. Note that this is smaller than the term in Proposition H.6 by a factor of 2.

**Case 2: Insertion** In this case, there is some $a \in x'$ such that $x = x' \setminus \{a\}$. We can thus define the same events and the same coupling in a symmetric manner to show

$$\Lambda_\alpha(m_x||m_{x'}) \leq \max_{y,y'}(1-w) \int_{\mathbb{Z}} \left( \frac{(1+w)b_y(z) - wb_{y'}(z)}{b_y(z)} \right)^\alpha b_y(z) \, d\lambda(z)$$
$$+ w \int_{\mathbb{Z}} \left( \frac{(1-w)b_{y'}(z) + wb_y(z)}{b_{y'}(z)} \right)^\alpha b_{y'}(z) \, d\lambda(z),$$

subject to $d_{\mathbb{Y}}(y, y') \leq 2$. with $w = q / N$. Again, this corresponds exactly to Eq. 6 of the "novel alternative decomposition" in Appendix A.1 of [30]. One can then through the remaining steps in their Appendix A to show that this term is at most two times larger than the one we derived in the deletion case.

Finally, since this bound depends on $N$, and we want the guarantee to hold for all $x \simeq_{\pm,\mathbb{X}} x'$, we need to determine the $N$ that maximizes the bound. $\quad\square$

**Hybrid Relations.** Note that we could have also gone through the above derivations with batch substitution relation $\simeq_{\Delta,\mathbb{Y}}$. Then, each substitution would correspond to an actual substitution, instead of a pair of insertion and deletion, and we would obtain

$$\Lambda_\alpha(m_x||m_{x'}) \leq \max_{N \in \mathbb{N}} \left( 2 \cdot \sum_{l=0}^{\alpha} \binom{\alpha}{l} w^l (1-w)^{\alpha-l} \max_{y \simeq_{\Delta,\mathbb{Y}} y'} \Lambda_l(b_y||b_{y'}) \right) \quad \text{s.t.} \quad N > Q,$$

with $w = q / N$.

## I.3 Tight mechanism-specific subsampling without replacement for randomized response

As mentioned earlier, the following result is meant as a simple example to demonstrate the benefit of using mechanism-specific over mechanism-agnostic RDP bounds in Section 4. For the following discussion, recall Theorem H.1, which we proved in the previous section.

**Theorem H.1.** *Let $M = B \circ S$ be a subsampled mechanism, where $S$ is subsampling without replacement with batch size $q$. Let $\simeq_{\mathbb{Y}}$ be the substitution relation $\simeq_{\Delta,\mathbb{Y}}$. Then, for $\alpha > 1$ and all $x \simeq_{\Delta,\mathbb{X}} x'$ of size $N$,*

$$\Delta_\alpha(m_x||m_{x'}) \leq \max_{\hat{y}} \Delta_\alpha((1-w) \cdot b_{y_1^{(1)}} + w \cdot b_{y_2^{(1)}} || (1-w) \cdot b_{y_1^{(2)}} + w \cdot b_{y_2^{(2)}})$$

*subject to $d_{\Delta,\mathbb{Y}}(y_1^{(1)}, y_2^{(1)}) \leq 1$, $d_{\Delta,\mathbb{Y}}(y_1^{(1)}, y_2^{(2)}) \leq 1$, $d_{\Delta,\mathbb{Y}}(y_2^{(1)}, y_2^{(2)}) \leq 1$, $y_1^{(1)} = y_1^{(2)}$, and with $w = q / N$.*

We shall now solve Theorem H.1 for randomized response and prove the tightness of the resultant guarantee.

**Theorem I.3.** *Let $B$ be the randomized response mechanism $|h - (1 - V)|$ with $h : \mathbb{Y} \to \{0, 1\}$, $V \sim \text{Bern}(\theta)$, and true response probability $\theta \in [0, 1]$. Let $S$ be subsampling without replacement with batch size $q$. Then, for all $x \simeq_{\Delta,\mathbb{X}} x'$ of size $N$, $\Lambda_\alpha(m_x||m_{x'})$ is l.e.q.*

$$\max_\tau \Lambda_\alpha((1-w)\text{Bern}(\cdot \mid \theta) + w\text{Bern}(\cdot \mid \tau) || (1-w)\text{Bern}(\cdot \mid \theta) + w\text{Bern}(\cdot \mid 1 - \tau))$$

*subject to $\tau \in \{\theta, 1 - \theta\}$ and with $w = q / N$.*

*Proof.* Because we have no further information, we must consider the worst possible underlying function $h : \mathbb{Y} \to \{0, 1\}$.

Due to the last constraint in Theorem H.1, we must have $h(y_1^{(1)}) = h(y_1^{(2)})$. Due to symmetry in $\Lambda_\alpha$ (we are summing over $z \in \{0,1\}$), we can assume w.l.o.g. that $h(y_1^{(1)}) = h(y_1^{(2)}) = 1$ and thus $b_{y_1^{(1)}}(z) = b_{y_1^{(1)}}(z) = \text{Bern}(z \mid \theta)$.

Because the batches $y_2^{(1)}$ and $y_2^{(1)}$ can differ from $y_1^{(1)}$, we can choose the corresponding function values $h(y_2^{(1)})$ and $h(y_2^{(2)})$ arbitrarily. To make $\Lambda_\alpha$ greater than 1, the two mixtures must be different from each other. Thus, we must choose either $h(y_2^{(1)}) = 1$ and $h(y_2^{(2)}) = 0$, or $h(y_2^{(1)}) = 0$ and $h(y_2^{(2)}) = 1$. This corresponds to $b_{y_2^{(1)}}(z) = \text{Bern}(z \mid \tau)$ and $b_{y_2^{(2)}}(z) = \text{Bern}(z \mid 1 - \tau)$ with $\tau \in \{\theta, 1 - \theta\}$. Maximizing over both options yields our result. $\square$

Note that this guarantee can be evaluated in $\mathcal{O}(1)$: We need to iterate over two possible values of $\tau$, and evaluating the corresponding $\Lambda_\alpha$ requires summing over two values $z \in \{0,1\}$.

Next, we prove the mechanism-specific tightness of this result, meaning it is not possible to derive stronger guarantees without additional information about dataset space $\mathbb{X}$ and the function $h$ underlying the randomized response mechanism.

**Theorem I.4.** *Let $S$ be subsampling without replacement with arbitrary batch size $q \in \mathbb{N}$ and $\theta \in [0,1]$ be some true response probability. There exists a dataset space $\mathbb{X}$ and a batch space $(\mathbb{Y}, \mathcal{Y})$ fulfilling the constraints in Definition D.5, as well a pair of datasets $x \simeq_{\Delta, \mathbb{X}}$ of size $N$, and a function $h : \mathbb{Y} \to \{0,1\}$, such that the corresponding subsampled randomized response mechanism $M = B \circ S$ fulfills*

$$\Lambda_\alpha(m_x || m_{x'})$$
$$= \max_\tau \Lambda_\alpha((1-w)\text{Bern}(\cdot \mid \theta) + w\text{Bern}(\cdot \mid \tau) || (1-w)\text{Bern}(\cdot \mid \theta) + w\text{Bern}(\cdot \mid 1 - \tau))$$

*subject to $\tau \in \{\theta, 1 - \theta\}$ and with $w = q \, / \, N$.*

*Proof.* Let $\mathbb{X} = \{x \subseteq \mathbb{N} \mid |x| > q\}$. Consider an arbitrary $x \in \mathbb{X}$ and select arbitrary elements $a \in x$, $a' \in \mathbb{N} \setminus x$. Define $x' = x \setminus \{a\} \cup \{a'\}$.

Assume w.l.o.g. that the divergence is maximized by $\tau = \theta$. We now construct an indicator function $h : \mathbb{Y} \to \{0,1\}$ for $a$ and $a'$ that leads to the largest possible divergence.

$$h(y) = \begin{cases} 1 & \text{if } y \cap \{a, a'\} = \emptyset \\ 1 & \text{if } a \in y \\ 0 & \text{if } a' \in y \end{cases}$$

By construction of $h$, the corresponding base mechanism pmf is

$$b_y(z) = \begin{cases} \text{Bern}(z \mid \theta) & \text{if } y \cap \{a, a'\} = \emptyset \\ \text{Bern}(z \mid \theta) & \text{if } a \in y \\ \text{Bern}(z \mid 1 - \theta) & \text{if } a' \in y \end{cases}$$

Under the distribution of $S(x)$, the first case occurs with probability $1 - w$ and the second case occurs with probability $w$. Under the distribution of $S(x)$, the first case occurs with probability $1 - w$ and the second case occurs with probability $w$. We thus have

$$m_x(z) = (1-w)\text{Bern}(z \mid \theta) + w\text{Bern}(z \mid \theta),$$
$$m_{x'}(z) = (1-w)\text{Bern}(z \mid \theta) + w\text{Bern}(z \mid 1 - \theta),$$

which exactly attains the desired divergence when $\tau = \theta$ is the optimal value. $\square$

## J From mechanism-specific guarantees to dominating pairs

Our proposed optimal transport approach lets us derive mechanism-specific guarantees, which upper-bound the hockey stick divergence between the distribution of $M(x)$ and $M(x')$ with $x \simeq_{\mathbb{X}} x'$ via a weighted sum [5] of mixture divergences, i.e.,

$$H_\alpha(m_x||m_{x'}) \leq \sum_{k=1}^{K} w_{\alpha,k}^{(x,x')} \cdot H_\alpha(p_{\alpha,k}^{(x,x')}||q_{\alpha,k}^{(x,x')}) \tag{23}$$

Here, the $w_{\alpha,k}^{(x,x')}$ are weights with $\sum_{k=1}^{K} w_{\alpha,k}^{(x,x')} = 1$, which depend on the chosen coupling $\gamma$ and are indexed by $\alpha \geq 0$, $1 \leq k \leq K$, and $x, x' \in \mathbb{X}$. The $p_{\alpha,k}^{(x,x')}$ and $p_{\alpha,K}^{(x,x')}$ are densities of mixture distributions $P_{\alpha,k}^{(x,x')}$ and $q_{\alpha,K}^{(x,x')}$ on the output space $\mathbb{R}^D$, [6] which are also indexed by $\alpha \geq 0$, $1 \leq k \leq K$, and $x, x' \in \mathbb{X}$.

Our goal is to construct dominating pairs from such bounds, so that we can then use them for privacy accounting. For this discussion, recall the definition of dominating pairs:

**Definition 2.3.** A pair of distributions $(P, Q)$ with densities $(p, q)$ is a dominating pair for mechanism $M$ under neighboring relation $\simeq_{\mathbb{X}}$, if $H_\alpha(m_x||m_{x'}) \leq H_\alpha(p||q)$ for all $x \simeq_{\mathbb{X}} x'$ and all $\alpha \geq 0$.

If the densities on the r.h.s. of Eq. (23) are constant in $\alpha$, $k$, $x$, and $x'$, i.e.., $p_{\alpha,k}^{(x,x')} = \tilde{p}$ and $q_{\alpha,k}^{(x,x')} = \tilde{q}$ with some densitities $\tilde{p}, \tilde{q}$, then we can immediately identify that the corresponding distributions $\tilde{P}$ and $\tilde{Q}$ are a dominating pair. For instance, in Section 3.4 we could immediately determine that the two Gaussian mixtures in Theorem 3.8 are a dominating pair for Poisson subsampling and the "insert-$K_+$-remove-$K_-$" relation.

However, this is generally not the case. In the following, we discuss a three-step procedure that let us (1) reduce the weighted sum of divergences in Eq. (23) to a single divergence (2) resolve non-constancy in the dataset pairs $(x, x')$, and (3) resolve non-constancy in the divergence order $\alpha$.

Note that, depending on the considered setting, it may be possible to skip one or multiple of these steps (like in our group privacy example).

### J.1 Step 1: Eliminating weighted sums

Consider some fixed order $\alpha$ and fixed datasets $x \simeq_{\mathbb{X}} x'$. Let us thus omit the corresponding indexes. In this step, we construct a pair of distributions $\tilde{P}, \tilde{Q}$ with densities $\tilde{p}$ and $\tilde{q}$ such that

$$\sum_{k=1}^{K} w_k \cdot H_\alpha(p_k||q_k) = H_\alpha(\tilde{p}||\tilde{q}).$$

To achieve this, we notice that there is no requirement for dominating pairs to be distributions on the same space $\mathbb{R}^D$. Taking inspiration from hierarchical randomized smoothing [69], we thus construct $\tilde{p}$ and $\tilde{q}$ that are mixtures of the $p_k$ and $q_k$ and additionally release their component indices:

**Lemma J.1.** *Consider arbitrary densitities $p_1, \ldots, p_K, q_1, \ldots, q_K : \mathbb{R}^D \to \mathbb{R}_+$ and arbitrary weights $w_1, \ldots, w_K \in [0, 1]$ with $\sum_{k=1}^{K} w_k = 1$. Further let $r : \{1, \ldots, K\} \to [0, 1]$ be the probability mass function of* $\text{Categorical}(w_1, \ldots, w_K)$. *Define $\tilde{p}, \tilde{q} : \mathbb{R}^D \times \{1, \ldots, K\} \to \mathbb{R}_+$ as $\tilde{p}(z, k) = p_k(z) \cdot r(k)$ and $\tilde{q}(z, k) = q_k(z) \cdot r(k)$. Then, $H_\alpha(\tilde{p}||\tilde{q}) = \sum_{k=1}^{K} w_k \cdot H_\alpha(p_k||q_k)$.*

---

[5] assuming that the batch space is finite and discrete

[6] the arguments in this section also apply to the general problem setting from Appendix D, where we allow arbitrary output spaces

*Proof.* By the definition of hockey stick divergence and $\tilde{p}, \tilde{q}$, we have

$$
\begin{aligned}
H_\alpha(\tilde{p}||\tilde{q}) &= \sum_{k=1}^{K} \int_{\mathbb{R}^D} \max\left\{ \frac{\tilde{p}(z,v)}{\tilde{q}(z,v)}, 0 \right\} \cdot \tilde{q}(z,v)\,\mathrm{d}z \\
&= \sum_{k=1}^{K} \int_{\mathbb{R}^D} \max\left\{ \frac{p_k(z) \cdot r(k)}{q_k(z) \cdot r(k)}, 0 \right\} \cdot q_k(z) \cdot r(k)\,\mathrm{d}z \\
&= \sum_{k=1}^{K} r(k) \cdot \int_{\mathbb{R}^D} \max\left\{ \frac{p_k(z)}{q_k(z)}, 0 \right\} \cdot q_k(z)\,\mathrm{d}z \\
&= \sum_{k=1}^{K} w_k \cdot H_\alpha(p_k||q_k).
\end{aligned}
$$

$\square$

Note that one could equivalently construct continuous distributions $\tilde{P}$ and $\tilde{Q}$ by defining the categorical distribution as a transformation of a uniform distribution. Further note that the distribution of privacy loss random variable $\log(\tilde{p}(Z)\,/\,\tilde{q})$ with $Z \sim \tilde{P}$ is simply a weighted sum of the components' privacy loss distributions, which can be easily shown via law of total probability. This makes this construction amenable to numerical privacy accounting.

## J.2 Step 2: Resolving non-constancy in dataset pairs

After applying the construction from the previous step, we have bounds of the form $H_\alpha(m_x||m_{x'}) \leq H_\alpha(\tilde{p}_\alpha^{(x,x')}||\tilde{q}_\alpha^{(x,x')})$. If these bounds are not constant in the datasets $x, x' \in \mathbb{X}$, we cannot simply read off a dominating pair.

However, the ultimate goal behind identifying dominating pairs is to use them in a privacy accounting method to algorithmically derive guarantees for all possible pair $x \simeq_{\mathbb{X}} x'$ under composition. Providing guarantees for all possible pair does not require that we derive guarantees for all pairs simultaneously. To formalize this, recall that the neighboring relation $\simeq_{\mathbb{X}}$ is a set of tuples from $\mathbb{X}^2$.

**Lemma J.2.** *Consider any $\alpha \geq 0$, mechanism $M$ and $L$ different neighboring relations $\simeq^{(1)}, \ldots, \simeq^{(L)} \subseteq \mathbb{X}^2$. Assume that there are $L$ constants $c^{(1)}, \ldots, c^{(L)}$ such that $H_\alpha(m_x||m_{x'}) \leq c^{(l)}$ for all $l \in \{1, \ldots, L\}$ and all $x \simeq^{(l)} x'$. If the neighboring relations partition $\simeq_{\mathbb{X}}$, i.e., $\simeq_{\mathbb{X}} = \bigcup_{l=1}^{L} \simeq^{(l)}$, then*

$$
\forall x \simeq_{\mathbb{X}} x' : H_\alpha(m_x||m_{x'}) \leq \max_l c^{(l)}.
$$

*Proof.* Consider any $x \simeq_{\mathbb{X}} x'$. Because the neighboring relations partition $\simeq_{\mathbb{X}}$, there must be some $l^*$ such that $x \simeq^{(l^*)} x'$ and thus $H_\alpha(m_x||m_{x'}) \leq c^{(l^*)} \leq \max_l c^{(l)}$. $\square$

For the purposes of privacy accounting, it is thus sufficient to determine some sufficiently fine-grained partition $\simeq^{(1)}, \ldots, \simeq^{(L)}$ such that for all $l \in \{1, \ldots, L\}$ and all $\alpha \geq 0$

$$
\forall x \simeq^{(l)} x' : H_\alpha(\tilde{p}_\alpha^{(x,x')}||\tilde{q}_\alpha^{(x,x')}) = H_\alpha(\hat{p}_\alpha^{(l)}||\hat{q}_\alpha^{(l)}),
$$

where $\hat{p}_\alpha^l$ and $\hat{q}_\alpha^{(l)}$ are families of densities indexed by partition index $l$ and divergence order $\alpha$. One can then perform privacy accounting independently for each of the relations.

This goal can always be achieved by having one atomic relation per possible pair of neighboring elements. In practice, however, much more coarse-grained partitions may be sufficient. Zhu et al. [10] applied this ansatz to the insertion/removal relation, by deriving dominating pairs for insertion and removal separately. In our work, we partition the group insertion/removal relation for groups of size $K$ into $K+1$ different "insert-$K_+$-remove-$K_-$" relations with $K_+ + K_- = K$.

### J.3 Step 3: Resolving non-constancy in divergence order

Consider some fixed partition index $l \in \{1, \ldots, L\}$, which we omit in the following. We are now left with a bound of the form $\forall x \simeq x' : H_\alpha(m_x || m_{x'}) \leq H_\alpha(\hat{p}_\alpha || \hat{q}_\alpha)$. If the families of densitities $\hat{p}_\alpha$, $\hat{q}_\alpha$ is constant in $\alpha$, i.e.

$$\forall \alpha : H_\alpha(\hat{p}_\alpha || \hat{q}_\alpha) = H_\alpha(\check{p} || \check{q}),$$

with some $\check{p}, \check{q}$, then the corresponding distributions $\check{P}, \check{Q}$ are dominating pairs. Importantly, the numeric value of the bound can vary with $\alpha$. We just want the numeric value to be identical to the divergence $H_\alpha$ of two specific distributions for all $\alpha$.

If this is not the case, there are two options to construct a dominating pair. The first option uses a characterization of dominating pairs as a function of privacy profiles. It was applied by Zhu et al. [10] to determine dominating pairs for subsampling without replacement and the substitution relation. The second option is enabled by our novel optimal transport and constrained optimization perspective on subsampling.

#### J.3.1 Option 1: Convex conjugation of privacy profiles

The following result from [10] establishes a correspondence between privacy profiles and dominating pairs, as well as providing a formula for constructing dominating pairs from privacy profiles:

**Proposition J.3** (Zhu et al. [10]). *For a given $f : \mathbb{R}_+ \to \mathbb{R}$, there exists $\check{P}, \check{Q}$ such that $\forall \alpha \geq 0 :$ $H(\alpha) = H_\alpha(P, Q)$ if and only if $f \in \mathcal{F}$ where*

$$\mathcal{F} = \{f : \mathbb{R}_+ \to \mathbb{R} \mid f \text{ is convex, decreasing, } f(0) = 1 \text{ and } f(x) \geq \max\{1 - x, 0\}\}. \quad (24)$$

*Moreoever, one can explicitly construct such $P$ and $Q$: $P$ has $CDF$ $1 + f^*(x - 1)$ in $[0, 1)$ and $Q = \mathrm{Uniform}([0, 1])$, where $f^*$ is the convex conjugate of $f$.*

Next, we show how to construct a function $f(\alpha) \in \mathcal{F}$ from our families of densities $\hat{q}_\alpha, \hat{q}_\alpha$ indexed by $\alpha$ that allows us to determine dominating pairs.

**Lemma J.4.** *Consider two families of densities $\hat{p}_\alpha, \hat{q}_\alpha$ indexed by $\alpha \geq 0$ such that $\forall x \simeq x' :$ $H_\alpha(m_x || m_{x'}) \leq H_\alpha(\hat{p}_\alpha || \hat{q}_\alpha)$. Define the function $f : \mathbb{R}_+ \to \mathbb{R}$ with*

$$f(\alpha) = \max_{\beta \geq 0} H_\alpha(\hat{p}_\beta || \hat{q}_\beta).$$

*Then $f$ is a valid privacy profile, i.e., $f \in \mathcal{F}$ with $\mathcal{H}$ defined in Eq. (24).*

*Proof.* For every $\beta \geq 0$, let $f_\beta(\alpha) = H_\alpha(\hat{p}_\beta || \hat{q}_\beta)$ be the privacy profile associated with a specific pair from the considered families. All these functions are elements of $\mathcal{F}$ as per the first part of Proposition J.3. We thus know that each $f_\beta$ is convex, decreasing, and has value 1 at $\alpha = 0$. Naturally, their maximum is also convex, decreasing, and has value 1 at $\alpha = 0$. We further know that, for all $\beta \geq 0$, $f_\beta(x) \geq \max\{1 - x, 0\}$. Thus, we have $f(x) \geq f_\beta(x) \geq (1 - x)$ for any $\beta \geq 0$. $\square$

We can then invoke the second part of Proposition J.3 to define our dominating pair.

As mentioned earlier, Zhu et al. [10] applied this principle to subsampling without replacement by taking a maximum over two different pairs of distributions. Lemma J.4 generalizes this principle to larger families.

#### J.3.2 Option 2: Relaxation of cost function constraints

A novel solution is enabled by our proposed framework for deriving mechanism-specific subsampling guarantees: Recall that the densities $\hat{p}_\alpha$ and $\hat{q}_\alpha$ are the result of identifying worst-case mixture components under pairwise batch distance constraints (Proposition 3.5). In Appendix L, we demonstrate that it is possible to relax some of these constraints to obtain a single pair of densities that bounds $H_\alpha(m_x || m_{x'})$ for all $\alpha$ and pairs of datasets $x \simeq x'$. Specifically, we apply this principle to the substitution relation and subsampling without replacement, as well as subsampling with replacement. Due to the relaxation, these dominating pairs are not necessarily tight. They may however be easier to use in practice than convex conjugation, which requires solving an optimization problem.

# K  Recovering known dominating pairs

In this section, we demonstrate that known dominating pairs for the standard combinations of Poisson subsampling under insertion/removal and subsampling without replacement under substitution, as well as subsampling without replacement under insertion/removal, can also be derived via our proposed optimal transport framework.

As before, our contribution are not the guarantees themselves. Our contribution is identifying that, just like mechanism-agnostic guarantees for ADP and RDP, these guarantees can be derived by defining an (optimal) coupling and then identifying worst-case mixture components under pairwise batch distance constraints.

## K.1  Poisson subsampling and insertion/removal

As discussed in Appendix J, it is beneficial to partition the insertion removal $\simeq_{\pm,\mathbb{X}}$ into two relations: The insertion relation $\simeq_{+,\mathbb{X}}$, where $x \simeq_{+,\mathbb{X}} x'$ implies that there is some $a \notin x$ such that $x' = x \cup \{a\}$ and the removal relation $\simeq_{-,\mathbb{X}}$, where $x \simeq_{-,\mathbb{X}}$ implies that there is some $a \in x$ such that $x' = x \setminus \{a\}$.

While we could derive dominating pairs for both relations from scratch, we can use the following result from [10] to reduce our workload:

**Lemma K.1.** *If distributions $(P, Q)$ are a dominating pair for mechanism $M$ under the insertion relation $\simeq_+$, then $(Q, P)$ are a dominating pair under the removal relation $\simeq_-$.*

Next, we use our proposed framework to derive the following guarantee in a very natural manner that does not require reasoning about tradeoff functions and optimal testing rules (c.f. proof of Lemma 29 in [10]).

**Proposition K.2** (Zhu et al. [10]). *Consider a subsampled mechanism $M = B \circ S$, where $S$ is Poisson subsampling with subsampling rate $r \in [0, 1]$. assume that $(P, Q)$ are a dominating pair for base mechanism $B$ under the batch insertion relation $\simeq_{+,\mathbb{Y}}$. Then, $(P, (1-r)P + rQ)$ are a dominating pair for $M$ under the dataset insertion relation $\simeq_{+,\mathbb{X}}$ and $((1-r)P + rQ, Q)$ are a dominating pair for $M$ under the dataset removal relation $\simeq_{-,\mathbb{X}}$.*

*Proof.* We begin by deriving the dominating pair for the removal relation $\simeq_{-,\mathbb{X}}$. Consider an arbitrary pair $x \simeq_- x'$ and an arbitrary $\alpha \geq 0$. Recall from our proof of the mechanism-agnostic Poisson subsampling guarantee Proposition H.6 that we can construct a $d_{\mathbb{Y}}$-compatible coupling to show that

$$H_\alpha(m_x || m_{x'}) \leq \max_{\boldsymbol{y}} H_\alpha((1-r)b_{y_1^{(1)}} + rb_{y_2^{(1)}} || b_{y_1^{(2)}})$$

subject to $\boldsymbol{y} \in \mathbb{Y}^{2+1}$, $y_1^{(1)} = y_1^{(2)}$, $d_{\mathbb{Y}}(y_2^{(1)}, y_1^{(1)}) \leq 1$, $d_{\mathbb{Y}}(y_1^{(1)}, y_2^{(1)}) \leq \infty$.

The last constraint is because there is no sequence of deletions that will result in an insertion. It can be ignored, meaning our optimization problem is equivalent to

$$\max_{y \simeq_- y'} H_\alpha((1-r)b_{y'} + rb_y || b_{y'}).$$

Using the definition of hockey stick divergence,[7] this can be restated as

$$\max_{y \simeq_- y'} H_\alpha((1-r)b_{y'} + rb_y || b_{y'})$$

$$= \max_{y \simeq_- y'} \int_{\mathbb{R}^D} \max\left(\frac{(1-r)b_{y'}(z) + rb_y(z)}{b_{y'}(z)} - \alpha, 0\right) \cdot b_{y'}(z) \, \mathrm{d}z$$

$$= \max_{y \simeq_- y'} \int_{\mathbb{R}^D} \max\left\{(1-r) + r\frac{b_y(z)}{b_{y'}(z)} - \alpha, 0\right\} \cdot b_{y'}(z) \, \mathrm{d}z$$

$$= \max_{y \simeq_- y'} r \cdot \int_{\mathbb{R}^D} \max\left\{\frac{b_y(z)}{b_{y'}(z)} - \left(\alpha - \frac{1-r}{r}\right), 0\right\} \cdot b_{y'}(z) \, \mathrm{d}z$$

---

[7]For simplicity, we assume that we have continuous-valued mechanisms, but the same proof strategy can also be used for the more general definition of hockey stick divergence from Appendix D by calculating $\mathrm{d}((1-r)P_{B_{y'}} + rP_{B_y}) / \mathrm{d}P_{B_{y'}}$.

We can now distinguish two cases, based on the value of $\alpha' = \alpha - (1-r)/r$.

**Case 1:** Assume that $\alpha' < 0$. Then, the integrand is always non-negative, and the objective function evaluates to a constant $r(1 - a')$. We can thus choose $y, y'$ arbitrarily, because

$$\max_{y \simeq_+ y'} H_\alpha((1-r)b_{y'} + rb_y || b_{y'}) = r(1-a') = H_\alpha((1-r)q + rp || q),$$

where $p$ and $q$ are the densities of dominating pair $(P, Q)$.

**Case 2:** Assume that $\alpha' \geq 0$. Because $(P, Q)$ is a dominating pair for the batch insertion relation $\simeq_{+,\mathbb{Y}}$, we know from Lemma K.1 that $(Q, P)$ is a dominating pair for the batch removal relation $\simeq_{-,\mathbb{Y}}$. Thus

$$\max_{y \simeq_- y'} H_\alpha((1-r)b_{y'} + rb_y || b_{y'})$$
$$= \max_{y \simeq_- y'} r \cdot H_{\alpha'}(b_y || b_{y'})$$
$$\leq r \cdot H_{\alpha'}(q || p)$$
$$= H_\alpha((1-r)p + rq || q)$$

This shows that $((1-r)P + rQ, Q)$ is a dominating pair for the removal relation $\simeq_{-,\mathbb{X}}$. It then follows from Lemma K.1 that $(Q, (1-r)P + rQ)$ is a dominating pair for the insertion relation $\simeq_{+,\mathbb{X}}$. $\qquad\square$

Note that case 1 (but not case 2) of Theorem 11 in [10] states that $((1-r)Q + rP, P)$ were a dominating pair for the removal relation. This does however appear to be a typographical error, since the authors use the same symmetry argument (Lemma K.1) for their proof.

## K.2 Subsampling without replacement and insertion/removal

The proof for the following result is virtually identical to the previous one. They only differ in the coupling which we need to construct, which again highlights the generality and usefulness of our proposed framework.

**Proposition K.3** (Zhu et al. [10]). *Assume a dataset space and batch space defined by $\mathbb{X} \subseteq \{x \in \mathcal{P}(\mathbb{A}) \mid |x| \in \{N, N-1\}$, $\mathbb{Y} = \{y \subseteq x \mid x \in \mathbb{X}, |y| = q\}$, and finite set $\mathbb{A}$, with $q < N$. Consider a subsampled mechanism $M = B \circ S$, where $S$ is subsampling without replacement with batch size $q$, and define batch-to-dataset ratio $w = q / N$. assume that $(P, Q)$ are a dominating pair for base mechanism $B$ under the batch substitution relation $\simeq_{\Delta,\mathbb{Y}}$. Then, $(P, (1-w)P + wQ)$ are a dominating pair for $M$ under the dataset insertion relation $\simeq_{+,\mathbb{X}}$ and $((1-w)P + wQ, Q)$ are a dominating pair for $M$ under the dataset removal relation $\simeq_{-,\mathbb{X}}$.*

*Proof.* As in the previous proof, we begin with removal relation $\simeq_{-,\mathbb{X}}$. Consider an arbitrary pair $x \simeq_- x'$ and an arbitrary $\alpha \geq 0$. Recall from our novel proof of the mechanism-agnostic RDP guarantee for subsampling without replacement and insertion/removal (see Theorem I.2) that we can construct a $d_\mathbb{Y}$-compatible coupling to show that

$$H_\alpha(m_x || m_{x'}) \leq \max_{\boldsymbol{y}} H_\alpha((1-w)b_{y_1^{(1)}} + wb_{y_2^{(1)}} || b_{y_1^{(2)}})$$

subject to $\boldsymbol{y} \in \mathbb{Y}^{2+1}$, $y_1^{(1)} = y_1^{(2)}$, $d_\mathbb{Y}(y_2^{(1)}, y_1^{(1)}) \leq 1$, $d_\mathbb{Y}(y_1^{(1)}, y_2^{(1)}) \leq 1$.

By definition of the induced distance, this optimization problem is equivalent to

$$\max_{y \simeq_\Delta y'} H_\alpha((1-r)b_{y'} + rb_y || b_{y'}).$$

We can now go through exactly the same steps as in our derivation of Proposition K.2, replacing every occurrence of "$\simeq_-$" with "$\simeq_\Delta''$", to conclude our proof. $\qquad\square$

## K.3 Subsampling without replacement and substitution

In the case of subsampling without replacement and substitution (as well as Poisson subsampling and insertion/removal without partitioning of the neighboring relation) we do not need to provide a

new proof. This is because Zhu et al. [10] prove this result by invoking a tight mechanism-agnostic subsampling guarantee derived by Balle et al. [15] and then applying advanced joint convexity in reverse order (see proof of Proposition 30 in [10]).

As we discussed in Appendix G.2 and formalized in Theorem G.2, any guarantee that is derived via the framework from [15] can be equivalently derived by defining a (potentially suboptimal) coupling between multiple conditional subsampling distributions. Thus, we know that the guarantee could have equivalently been derived via our proposed framework

**Proposition K.4** (Zhu et al. [10]). *Consider a subsampled mechanism $M = B \circ S$, where $S$ is subsampling without replacement with batch size $q \in \mathbb{N}$. Assume that $(P, Q)$ are a dominating pair for base mechanism $B$ under the batch substitution relation $\simeq_{\Delta,\mathbb{Y}}$. Consider arbitrary $x \simeq_{\Delta,\mathbb{X}} x'$ of size $N$ and define batch-to-dataset ratio $w = q \mathbin{/} N$. Then*

$$H_\alpha(m_x\|m_{x'}) \leq \begin{cases} H_\alpha((1-w)q + wp\|q) & \text{for } \alpha \geq 1 \\ H_\alpha(p, (1-w)p + wq\|q) & \text{for } 0 < \alpha < 1 \end{cases}$$

Note that this result does not immediately specify a dominating pair. However, as shown in [10], one can construct a dominating pair via convex conjugation (recall Appendix J.3.1 and Proposition J.3).

# L Novel results for dominating pairs

In the following, we formally derive dominating pairs for subsampling with replacement under the substitution relation. We also derive an alternative dominating pair for subsampling without replacement under substitution which does not require convex conjugation (c.f. Appendix K.3). These results are novel in three respects.

**Novel contribution 1 – Formal guarantees.** Using the dominating pairs we shall shortly derive for privacy accounting was already proposed in [8]. However, the authors did not provide a formal proof for why they should lead to valid privacy guarantees. They only proved that if two pairs of multivariate Gaussians with colinear means were to be a dominating pair[8], then one could use a change of variable and marginalization to reduce them to a univariate dominating pair (see Appendix B.1 in [8]). They did not prove why this assumption should hold. A formal proof of this assumption is now enabled by the analysis we conducted for solving the group privacy optimization problem in Theorem M.6 (see Appendix O).

**Novel contribution 2 – Other base mechanisms.** The generality of our solution to the group privacy optimization problem in Theorem M.6 lets us not only derive dominating pairs for Gaussian mechanisms, but also for Laplace mechanisms and randomized response.

**Novel contribution 3 – Resolving non-constancy via constraint relaxation.** Perhaps most importantly, our proposed framework offers a novel perspective on an issue encountered when analyzing subsampling without replacement: While one can derive a tight mechanism-specific guarantee, this guarantee depends on different mixture distributions for $\alpha < 1$ and $\alpha \geq 1$ (see Proposition K.4 [10] and our more general discussion in Appendix J.3). Thus, one cannot simply read off a dominating pair from this mechanism-specific guarantee, as we did with our group privacy bounds. However, as we shall demonstrate, this issue no longer appears when we relax some of the batch distance constraints in our proposed cost function bound from Proposition 3.5. This suggests that this could be a more general approach for addressing this non-constancy issue. While the resultant dominating pairs are not necessarily tight, they may be easier to use in practice than the convex conjugation construction from [10].

## L.1 Subsampling without replacement and substitution

We begin with subsampling without replacement to demonstrate the potential benefit of relaxing batch distance constraints to determine tractable dominating pairs.

**Theorem L.1.** *Let $M = B \circ S$, where $S$ is subsampling without replacement with batch size $q$, and $B$ is the Gaussian mechanism $h + V$ with $h : \mathbb{Y} \to \mathbb{R}^D$ and $V \sim \mathcal{N}(\mathbf{0}, \sigma^2 \mathbf{I}_D)$. Define the $\ell_2$-sensitivity $L_2 = \max_{y \simeq_\Delta, \mathbb{Y}y'} ||f(y) - f(y')||_2$, where $\simeq_\Delta$ is the substitution relation. Consider an arbitrary dataset size $N \in \mathbb{N}$ and define batch-to-dataset ratio $w = \frac{q}{N}$. Then, for any pair $x \simeq_{\Delta, \mathbb{X}} x'$ of size $N$,*

$$H_\alpha(m_x || m_{x'}) \leq H_\alpha\left( f_1^{(1)} \cdot (1-w) + f_2^{(1)} \cdot w || f_1^{(2)} \cdot (1-w) + f_2^{(2)} \cdot w \right), \qquad (25)$$

*with univariate normal densities $f_i^{(1)} = \mathcal{N}(\cdot \mid (i-1), \sigma / L_2)$, $f_j^{(2)} = \mathcal{N}(\cdot \mid -(j-1), \sigma / L_2)$.*

*Proof.* Recall from our proof of the mechanism-agnostic subsampling without replacement guarantee Proposition H.4 that we can construct a $d_\mathbb{Y}$-compatible coupling to show that

$$H_\alpha(m_x || m_{x'}) \leq \max_{\boldsymbol{y}} H_\alpha((1-w)b_{y_1^{(1)}} + w b_{y_2^{(1)}} || (1-w)b_{y_1^{(2)}} + w b_{y_2^{(2)}})$$

subject to $\boldsymbol{y} \in \mathbb{Y}^{2+1}$, $y_1^{(1)} = y_1^{(2)}$, $d_\mathbb{Y}(y_1^{(1)}, y_2^{(1)}) \leq 1$, $d_\mathbb{Y}(y_1^{(1)}, y_2^{(2)}) \leq 1$, $d_\mathbb{Y}(y_1^{(2)}, y_1^{(2)}) \leq 1$.

Next, we relax the constraint between the two components that contain a substituted element: $d_\mathbb{Y}(y_1^{(2)}, y_1^{(2)}) \leq 2$.

We now observe that this optimization problem is identical to the optimization problem from our group privacy bound (Theorem 3.7) with one insertion ($K_+ = 1$), one deletion ($K_- = 1$), and subsampling rate $r = w$. The result thus follows immediately from our mechanism-specific group privacy amplification bound for Gaussian mechanism (Theorem 3.8). □

---

[8]Technically, the notion of dominating pairs[10] had not been introduced at this point, which is why the authors discuss a vaguer notion of "worst-case distributions", similar to other early work on numerical accounting.

We can generalize this result to other mechanisms by using group privacy amplification bounds for Laplace mechanisms (Theorem M.2) and randomized response mechanisms (Theorem M.2).

We see that the upper bound in Eq. (25) depends on the same pair of mixture distributions for all $\alpha \geq 0$. Thus, this pair of mixture distributions is a dominating pair for subsampling without replacement under substitution.

## L.2 Subsampling with replacement and substitution

Next, we provide the first formal derivation of dominating pairs for subsampling with replacement.

In subsampling with replacement, a single element may appear in a batch multiple times. To formalize this, we assume a dataset space $\mathbb{X} \subseteq \mathcal{P}(\mathbb{A})$ that is composed of sets (not multisets) of elements from an underlying set $\mathbb{A}$. We further assume a batch space $\mathbb{Y} \subseteq \mathcal{P}_{\text{multi}}(\mathbb{A})$ that is composed of multisets of elements from $\mathbb{A}$. Given some $y \in \mathbb{Y}$, we write $\xi_y(a)$ for the number of times that $a$ appears in $y$. We further define the support of batch $y \in \mathbb{Y}$ as $\text{supp}(y) = \{a \in \mathbb{Y} \mid \xi_y(a) \geq 0\}$.

**Definition L.2.** Subsampling with replacement with batch size $q$ has probability mass function $s_x(y) = \binom{|x|+q-1}{q}^{-1}$ for batches $y \in \mathbb{Y}$ with $\text{supp}(y) \subseteq x$ and $|y| = q$.

Given this subsampling scheme, we can use optimal transport to derive the following constrained optimization problem, which we shall then relax and solve:

**Theorem L.3.** *Let $M = B \circ S$, where $S$ is subsampling without replacement with batch size $q$, and $B$ is the Gaussian mechanism $h + V$ with $h : \mathbb{Y} \to \mathbb{R}^D$ and $V \sim \mathcal{N}(\mathbf{0}, \sigma^2 \mathbf{I}_D)$. Define the $\ell_2$-sensitivity $L_2 = \max_{y \simeq_\Delta \mathbb{Y} y'} ||f(y) - f(y')||_2$, where $\simeq_\Delta$ is the substitution relation. Consider an arbitrary dataset size $N \in \mathbb{N}$ and define batch-to-dataset ratio $w = \frac{q}{N}$. Then, for any pair $x \simeq_{\Delta, \mathbb{X}} x'$ of size $N$,*

$$H_\alpha(m_x || m_{x'}) \leq \max_{\boldsymbol{y}} H_\alpha \left( \sum_{i=1}^{q+1} b_{y_i^{(1)}} \cdot w_i || \sum_{j=1}^{q+1} b_{y_j^{(2)}} \cdot w_j \right),$$

*subject to $\boldsymbol{y} \in \mathbb{Y}^{(q+1)+(q+1)}, \forall l, t, u : d_{\mathbb{Y}}(y_t^{(l)}, y_u^{(l)}) \leq |t - u|, \forall t, u : d_{\mathbb{Y}}(y_t^{(1)}, y_u^{(2)}) \leq \max\{t, u\}$, and with weights $w_i = \left(\frac{1}{N}\right)^i \cdot \left(1 - \frac{1}{N}\right)^{q-i}$.*

*Proof.* By definition of the substitution relation $\simeq_\Delta$ there is some $a \in x$ with $a \notin x'$ and some $a' \in x'$ with $a' \notin x$ such that $x' = x \setminus \{a\} \cup \{a'\}$.

To simplify indexing, we define zero-based indexed events $A_0, \ldots, A_q$ and $E_0, \ldots, E_q$. We define $A_i$ to be the event that $a$ is sampled $i$ times, i.e., $\xi_y(a) = i$. We define $E_j$ to be the event that $a'$ is sampled $j$ times, i.e., $\xi_y(a') = j$.

By definition subsampling with replacement, we have $P_{S_x}(A_i) = w_i$, and $P_{S_x}(E_j) = w_j$ with weights $w_i = \left(\frac{1}{N}\right)^i \cdot \left(1 - \frac{1}{N}\right)^{q-i}$.

For dataset size $N$, we have

$$s_x(y \mid A_i) = \begin{cases} \binom{(N-1)+(q-i)-1}{q-i}^{-1} & \text{if } \text{supp}(y) \subseteq x \land \xi_y(a) = i \\ 0 & \text{otherwise} \end{cases}$$

$$= \begin{cases} \left(\frac{1}{N-1}\right)^{q-i}(q-i)! \prod_{\tilde{a} \neq a} \frac{1}{\xi_y(\tilde{a})!} & \text{if } \text{supp}(y) \subseteq x \land \xi_y(a) = i \\ 0 & \text{otherwise} \end{cases}$$

and

$$s_{x'}(y \mid E_j) = \begin{cases} \binom{(N-1)+(q-i)-1}{q-i}^{-1} & \text{if } \text{supp}(y) \subseteq x' \land \xi_y(a') = j \\ 0 & \text{otherwise} \end{cases}$$

$$= \begin{cases} \left(\frac{1}{N-1}\right)^{q-i}(q-i)! \prod_{\tilde{a} \neq a'} \frac{1}{\xi_y(\tilde{a})!} & \text{if } \text{supp}(y) \subseteq x' \land \xi_y(a') = j \\ 0 & \text{otherwise} \end{cases}$$

Note that $s_x(y \mid A_0) = s_{x'}(y \mid E_0)$. Further note that $s_x(y \mid A_q) = \mathbb{1}[\text{supp}(y) = \{a\} \land |y| = q]$ and $s_{x'}(y \mid E_q) = \mathbb{1}[\text{supp}(y) = \{a'\} \land |y| = q]$.

**Coupling.** We now define a coupling $\gamma : \mathbb{Y}^{(q+1)+(q+1)} \to \mathbb{R}_+$ that corresponds to the following generative process: We first generate $y_q^{(1)}$ by constructing a batch of size $q$ whose elements are all $a$. Beginning at $i \leftarrow q$, we then iteratively generate $y_{i-1}^{(1)}$ from $y_{i-1}^{(1)}$ by replacing one instance of $a$ with an element $\tilde{a} \in x \setminus a$ that is sampled uniformly at random. Finally, we construct the $y_j^{(2)}$ by replacing every occurence of $a$ in $y_j^{(1)}$ with $a'$. More formally,

$$\gamma(\boldsymbol{y}^{(1)}, \boldsymbol{y}^{(2)}) = \begin{cases} \left(\frac{1}{N-1}\right)^q & \text{if } \boldsymbol{y}^{(1)}, \boldsymbol{y}^{(2)} \text{ fulfills Condition L.4} \\ 0 & \text{otherwise.} \end{cases}$$

**Condition L.4.** A tuple $\boldsymbol{y}^{(1)} \in \mathbb{Y}^{q+1}$, $\boldsymbol{y}^{(q+1)} \in \mathbb{Y}^{K_++1}$ fulfills this condition when $\xi_{y_q^{(1)}}(a) = \xi_{y_q^{(2)}}(a') = q$ and $\forall i, \exists \tilde{a} \in x \setminus \{a\} : y_{i-1}^{(1)} = y_{i-1}^{(1)} \setminus \{a\} \cup \{\tilde{a}\}$ and $\forall i, \forall \tilde{a} \notin \{a, a'\}) : \xi_{y_q^{(1)}}(\tilde{a}) = \xi_{y_q^{(2)}}(\tilde{a})$ and $\xi_{y_q^{(1)}}(a) = \xi_{y_q^{(2)}}(a')$.

**Validity.** To verify the validity of this coupling, consider an arbitrary $i$ and an arbitrary $\boldsymbol{y}$ such that $\gamma(\boldsymbol{y}^{(1)}, \boldsymbol{y}^{(2)}) > 0$. We know that there are $(q-i)! \prod_{\tilde{a} \neq a} \frac{1}{\xi_y(\tilde{a})!}$ sequences of $y_q^{(1)}, y_{q-1}^{(1)}, \dots, y_i^{(1)}$ that could have generated $y_i^{(1)}$. We further know that there are $\left(\frac{1}{N-1}\right)^i$ possible sequences of $y_{i-1}^{(1)}, \dots, y_0^{(1)}$ that can be generated from $y_i^{(1)}$. All $y_j^{(2)}$ are deterministically determined by $y_j^{(1)}$. We thus have

$$\sum_{\boldsymbol{y} \in \mathbb{Y}^{(q+1)+(q+1)}} \mathbb{1}[y_i^{(1)} = y] \gamma(\boldsymbol{y}^{(1)}, \boldsymbol{y}^{(2)})$$

$$= (q-i)! \prod_{\tilde{a} \neq a} \frac{1}{\xi_y(\tilde{a})!} \left(\frac{1}{N-1}\right)^i \left(\frac{1}{N-1}\right)^q$$

$$= \left(\frac{1}{N-1}\right)^{q-i} (q-i)! \prod_{\tilde{a} \neq a} \frac{1}{\xi_y(\tilde{a})!}$$

$$= s_x(y \mid A_i).$$

The proof for the $\boldsymbol{y}_j^{(2)}$ is analogous.

**Compatibility.** Finally, it can be easily shown that $\gamma$ is a $d_{\mathbb{Y}}$-compatible coupling (recall Appendix F). Whenever $\gamma(\boldsymbol{y}) > 0$, then

$$\forall 1 \leq i \leq K_- : d_{\mathbb{Y}}(y_0^{(1)}, y_i^{(1)}) = d_{\mathbb{Y}}\left(y_0^{(1)}, \text{supp}(s_x(\cdot \mid A_i))\right) = i,$$

$$\forall 0 \leq j \leq K_+ : d_{\mathbb{Y}}(y_0^{(1)}, y_i^{(2)}) = d_{\mathbb{Y}}\left(y_0^{(1)}, \text{supp}(s_{x'}(\cdot \mid E_j))\right) = j$$

because we generate the $y_i^{(1)}$ from $y_0^{(1)}$ by substituting exactly $i$ elements, and construct the $y_j^{(2)}$ by substituting exactly $j$ elements. Similarly, we have for the pairwise distances that do not involve $y_0^{(1)}$

$$\forall 0 < t < u \leq K_- : d_{\mathbb{Y}}(y_t^{(1)}, y_u^{(1)}) = d_{\mathbb{Y}}\left(\text{supp}(s_x(\cdot \mid A_t)), \text{supp}(s_x(\cdot \mid A_u))\right) = |t - u|,$$

$$\forall 0 < t < u \leq K_- : d_{\mathbb{Y}}(y_t^{(2)}, y_u^{(2)}) = d_{\mathbb{Y}}\left(\text{supp}(s_{x'}(\cdot \mid E_t)), \text{supp}(s_{x'}(\cdot \mid E_u))\right) = |t - u|.$$

$$\forall 0 \leq t < u \leq K_- : d_{\mathbb{Y}}(y_t^{(1)}, y_u^{(2)}) = d_{\mathbb{Y}}\left(\text{supp}(s_x(\cdot \mid A_t)), \text{supp}(s_{x'}(\cdot \mid E_u))\right) = \max\{t, u\}.$$

The last equality holds for $t \leq u$ because to construct $y_u^{(2)}$ from $y_t^{(1)}$ we need to substitute all occurrences of $a$ with $a'$ and then substitute $u - t$ additional elements with $a'$ It also holds for $t > u$, because then we need to replace $u$ occurences of $a$ with $a'$ and then replace another $t - u$ occurences with values other than $a'$.

Applying Corollary F.7, with $P_{S_x}(A_i) = w_i$, $P_{S_{x'}}(E_j) = w_j$, shows that $\qquad \square$

Finally, we can relax and solve Theorem L.3 for specific mechanisms, such as the Gaussian mechanism:

**Theorem L.5.** *Let $M = B \circ S$, where $S$ is subsampling with replacement with batch size $q$, and $B$ is the Gaussian mechanism $h + V$ with $h : \mathbb{Y} \to \mathbb{R}^D$ and $V \sim \mathcal{N}(\mathbf{0}, \sigma^2 \mathbf{I}_D)$. Define the $\ell_2$-sensitivity $L_2 = \max_{y \simeq_{\Delta, \mathbb{Y}} y'} \|f(y) - f(y')\|_2$, where $\simeq_\Delta$ is the substitution relation. Consider an arbitrary dataset size $N \in \mathbb{N}$. Then, for any pair $x \simeq_{\Delta, \mathbb{X}} x'$ of size $N$,*

$$H_\alpha(m_x \| m_{x'}) \leq H_\alpha \left( \sum_{i=1}^{q} f_i^{(1)} \cdot w_i \| \sum_{j=1}^{q} f_j^{(2)} \cdot w_j \right), \tag{26}$$

*with univariate normal densities $f_i^{(1)} = \mathcal{N}(\cdot \mid (i-1), \sigma / L_2)$, $f_j^{(2)} = \mathcal{N}(\cdot \mid -(j-1), \sigma / L_2)$, and weights $w_i = \left( \frac{1}{N} \right)^q \cdot \left( 1 - \frac{1}{N} \right)^{q-i}$.*

*Proof.* We first relax the optimization problem in Theorem L.3 by replacing the last constraint $\forall t, u : d_{\mathbb{Y}}(y_t^{(1)}, y_u^{(2)}) \leq \max\{t, u\}$ with a new constraint $\forall t, u : d_{\mathbb{Y}}(y_t^{(1)}, y_u^{(2)}) \leq t + u$.

This leaves us with

$$\max_{\boldsymbol{y}} H_\alpha \left( \sum_{i=1}^{q+1} b_{y_i^{(1)}} \cdot w_i \| \sum_{j=1}^{q+1} b_{y_i^{(2)}} \cdot w_j \right),$$

subject to $\boldsymbol{y} \in \mathbb{Y}^{(q+1)+(q+1)}$, $\forall l, t, u : d_{\mathbb{Y}}(y_t^{(l)}, y_u^{(l)}) \leq |t - u|$, $\forall t, u : d_{\mathbb{Y}}(y_t^{(1)}, y_u^{(2)}) \leq t + u$, and with weights $w_i = \left( \frac{1}{N} \right)^i \cdot \left( 1 - \frac{1}{N} \right)^{q-i}$.

We now observe that this optimization problem is identical to the optimization problem from our group privacy bound (Theorem 3.7) with $q$ insertions ($K_+ = q$), $q$ deletions ($K_- = q$), and a different set of weights. The result thus follows immediately from our solution to the group privacy amplification problem from Theorem 3.7, which did not make any specific assumptions about the weights. $\qquad \square$

We see that the upper bound in Eq. (25) depends on the same pair of mixture distributions for all $\alpha \geq 0$. Thus, this pair of mixture distributions is a dominating pair for subsampling without replacement under substitution.

## M  Tight mechanism-specific group privacy amplification

In the following, we first prove our generic group privacy amplification guarantee for Poisson subsampling and insertion/removal, which gives us a constrained optimization problem whose optimal value bounds $\Psi_\alpha(m_x\|m_{x'})$. We then solve this constrained optimization problem for Gaussian, Laplace, and randomized response mechanisms. After that, we prove the mechanism-specific tightness for the resultant guarantees. Finally, we discuss how to evaluate these guarantees numerically.

### M.1  Proof of Theorem 3.7

**Theorem 3.7.** *Let* $M = B \circ S$, *where* $S$ *is Poisson subsampling with rate* $r$. *Let* $\simeq_\mathbb{Y}$ *be the insertion/removal batch relation* $\simeq_{\pm,\mathbb{Y}}$. *Then, for all* $x \simeq_{K_+,K_-,\mathbb{X}} x'$, $\Psi_\alpha(m_x\|m_{x'})$ *is l.e.q.*

$$
\max_{\boldsymbol{y}} \Psi_\alpha \left( \sum_{i=1}^{K_-+1} b_{y_i^{(1)}} \cdot \mathrm{Binom}(i-1 \mid K_-, r) \Big\| \sum_{j=1}^{K_++1} b_{y_j^{(2)}} \cdot \mathrm{Binom}(j-1 \mid K_+, r) \right), \quad (4)
$$

*subject to constraints* $\boldsymbol{y} \in \mathbb{Y}^{K_-+K_++2}$, *as well as* $\forall l \in \{1,2\}, \forall t,u : d_\mathbb{Y}(y_t^{(l)}, y_u^{(l)}) \le |t-u|$, *and* $\forall t,u : d_\mathbb{Y}(y_t^{(1)}, y_u^{(2)}) \le (t-1) + (u-1)$.

*Proof.* By definition of the "insert $K_+$, remove $K_-$" relation $\simeq_{K_+,K_-,\mathbb{X}}$ there is some inserted set $g_+ \subseteq x'$ of size $K_+$ with $g_+ \cap x = \emptyset$ and some removed set $g_- \subseteq x$ of size $K_-$ with $g_- \cap x' = \emptyset$ such that $x' = x \setminus g_- \cup g_+$.

To simplify indexing, we define zero-based indexed events $A_0, \dots, A_{K_-}$ and $E_0, \dots, E_{K_+}$. We define $A_i$ to be the event that $i$ removed elements are sampled, i.e. $|y \cap g_-| = i$. We define $E_j$ to be the event that $j$ inserted elements are sampled, i.e. $|y \cap g_+| = j$.

By definition of Poisson subsampling, we have $P_{S_x}(A_i) = \mathrm{Binom}(i \mid K_-, r)$, and $P_{S_x}(E_j) = \mathrm{Binom}(j \mid K_+, r)$. We further have

$$
s_x(y \mid A_i) = \begin{cases} r^{|y|}(1-r)^{|x|-|y|} \cdot \mathrm{Binom}(i \mid K_-, r)^{-1} & \text{if } y \subseteq x \wedge |g_- \cap y| = i \\ 0 & \text{otherwise} \end{cases}
$$

$$
= \begin{cases} r^{|y|-i}(1-r)^{|x|-|y|-(K_--i)} \cdot \binom{K_-}{i}^{-1} & \text{if } y \subseteq x \wedge |g_- \cap y| = i \\ 0 & \text{otherwise} \end{cases}
$$

and

$$
s_{x'}(y \mid E_j) = \begin{cases} r^{|y|}(1-r)^{|x|-|y|} \cdot \mathrm{Binom}(i \mid K_+, r)^{-1} & \text{if } y \subseteq x' \wedge |g_+ \cap y| = i \\ 0 & \text{otherwise} \end{cases}
$$

$$
= \begin{cases} r^{|y|-j}(1-r)^{|x|-|y|-(K_+-j)} \cdot \binom{K_+}{j}^{-1} & \text{if } y \subseteq x' \wedge |g_+ \cap y| = i \\ 0 & \text{otherwise} \end{cases}
$$

Note that $s_{x'}(y \mid E_0) = s_x(y \mid A_0)$.

**Coupling.** We now define a coupling $\gamma : \mathbb{Y}^{(K_++1)+(K_-+1)} \to \mathbb{R}_+$ that corresponds to the following generative process: We first generate $y_0^{(1)}$ by sampling a batch that does not contain any inserted or removed elements via Poisson subsampling from $x \setminus g_-$. We then let $y_0^{(2)} \leftarrow y_0^{(1)}$. Finally, we sample uniformly at random an order in which we include removed elements from $g_-$ and inserted elements from $g_+$ to iteratively construct batches from $(A_i)_{i>1}$ and $(E_j)_{j>1}$, respectively. More formally,

$$
\gamma(\boldsymbol{y}^{(1)}, \boldsymbol{y}^{(2)}) = \begin{cases} s_x(y_0^{(1)} \mid A_0) \cdot (K_+! \cdot K_-!)^{-1} & \text{if } \boldsymbol{y}^{(1)}, \boldsymbol{y}^{(2)} \text{ fulfills Condition M.1} \\ 0 & \text{otherwise.} \end{cases}
$$

**Condition M.1.** A tuple $\boldsymbol{y}^{(1)} \in \mathbb{Y}^{K_-+1}$, $\boldsymbol{y}^{(2)} \in \mathbb{Y}^{K_++1}$ fulfills this condition when $y_0^{(1)} = y_0^{(2)}$, and $\forall i : \exists a_- \in g_- \setminus y_i^{(1)} : y_{i+1}^{(1)} = y_i^{(1)} \cup \{a_-\}$, and $\forall j : \exists a_+ \in g_+ \setminus y_j^{(2)} : y_{j+1}^{(2)} = y_j^{(2)} \cup \{a_-\}$.

**Validity.** We can verify the validity of this coupling as follows:

For $A_0$ and every $y$ with $s_x(y \mid A_0) > 0$, there are exactly $K_+! \cdot K_-!$ combinations with $y_0^{(1)} = y$ for which $\gamma(\boldsymbol{y}) > 0$. We thus have $\sum_{\boldsymbol{y} \in \mathbb{Y}^{K_++K_-+2}} \mathbb{1}[y_0^{(1)} = y]\gamma(\boldsymbol{v}) = s_x(y \mid A_0)$. The proof for $E_0$ is analogous.

Next, consider an arbitrary $A_i$ with $0 < i \le K_-$. For every $y$ with $s_x(y \mid A_0) > 0$, there are exactly $i! \cdot (K_- - i)! \cdot K_+!$ combinations with $y_i^{(1)} = y$ for which $\gamma(\boldsymbol{y}) > 0$, because there are (a) $i!$ orders in which the removed elements leading up to $i$ could have been sampled (b) $(K_- - i!)$ permutations for the remaining removed elements (c) $K_+!$ permutations for the inserted elements that are only relevant for $\boldsymbol{y}^{(2)}$. We thus have

$$\sum_{\boldsymbol{y} \in \mathbb{Y}^{K_++K_-+2}} \mathbb{1}[y_i^{(1)} = y]\gamma(\boldsymbol{y})$$

$$= (i! \cdot (K_- - i)! \cdot K_+!) \cdot s_x(y_0^{(1)} \mid A_0) \cdot (K_+! \cdot K_-!)^{-1}$$

$$= (i! \cdot (K_- - i)! \cdot K_+!) \cdot r^{|y_0^{(1)}|}(1 - r)^{|x|-|y_0^{(1)}|-(K_-)} \cdot (K_+! \cdot K_-!)^{-1}$$

$$= i!(K_- - i)! \cdot r^{|y_0^{(1)}|}(1 - r)^{|x|-|y_0^{(1)}|-(K_-)} \cdot (K_-!)^{-1}$$

$$= r^{|y_0^{(1)}|}(1 - r)^{|x|-|y_0^{(1)}|-(K_-)} \cdot \binom{K_-}{i}^{-1}$$

$$= r^{|y|-i}(1 - r)^{|x|-|y|-(K_--i)} \cdot \binom{K_-}{i}^{-1}$$

For the last step, we have used that $y_0^{(1)}$ contains 0 of the removed elements from $g_-$, whereas $y$ contains $i$ of the removed elements.

The proof for $E_j$ with $0 < j \le K_+$ is analogous.

**Compatibility.** Finally, it can be easily shown that $\gamma$ is a $d_{\mathbb{Y}}$-compatible coupling (recall Appendix F). Whenever $\gamma(\boldsymbol{y}) > 0$, then

$$\forall 1 \le i \le K_- : d_{\mathbb{Y}}(y_0^{(1)}, y_i^{(1)}) = d_{\mathbb{Y}}\left(y_0^{(1)}, \text{supp}(s_x(\cdot \mid A_i))\right) = i,$$

$$\forall 0 \le j \le K_+ : d_{\mathbb{Y}}(y_0^{(1)}, y_i^{(2)}) = d_{\mathbb{Y}}\left(y_0^{(1)}, \text{supp}(s_{x'}(\cdot \mid E_j))\right) = j$$

because we generate the $y_i^{(1)}$ from $y_0^{(1)}$ by inserting exactly $i$ elements, and construct the $y_j^{(2)}$ by inserting exactly $j$ elements. Similarly, we have for the pairwise distances that do not involve $y_0^{(1)}$

$$\forall 0 < t < u \le K_- : d_{\mathbb{Y}}(y_t^{(1)}, y_u^{(1)}) = d_{\mathbb{Y}}(\text{supp}(s_x(\cdot \mid A_t), \text{supp}(s_x(\cdot \mid A_u))) = |t - u|,$$

$$\forall 0 < t < u \le K_- : d_{\mathbb{Y}}(y_t^{(2)}, y_u^{(2)}) = d_{\mathbb{Y}}(\text{supp}(s_{x'}(\cdot \mid E_t), \text{supp}(s_{x'}(\cdot \mid E_u))) = |t - u|.$$

$$\forall 0 \le t < u \le K_- : d_{\mathbb{Y}}(y_t^{(1)}, y_u^{(2)}) = d_{\mathbb{Y}}(\text{supp}(s_x(\cdot \mid A_t), \text{supp}(s_{x'}(\cdot \mid E_u))) = t + u.$$

The last equality holds because to construct $y_u^{(2)}$ from $y_t^{(1)}$ we need to remove $t$ elements and insert $u$ elements.

The result then follows from Corollary F.7, $P_{S_x}(A_i) = \text{Binom}(i \mid K_-, r)$, $P_{S_{x'}}(E_j) = \text{Binom}(j \mid K_+, r)$, and reverting back to one-based indexing. $\qquad \square$

### M.2 Instantiations

Next, we can solve the derived optimization problem. In this section, we only provide proof sketches in which we reduce the optimization problem over batches to an optimization problem over the means of multiple Gaussian, Laplace, or Bernoulli random variables. Because solving these problems requires some further exposition, we then refer the reader to Appendix O.

#### M.2.1 Gaussian mechanism guarantee

**Theorem 3.8.** *Let $M = B \circ S$, where $S$ is Poisson subsampling with rate $r$, and $B$ is the Gaussian mechanism $h + V$ with $h : \mathbb{Y} \to \mathbb{R}^D$ and $V \sim \mathcal{N}(\boldsymbol{0}, \sigma^2 \boldsymbol{I}_D)$. Define the $\ell_2$-sensitivity $L_2 = $*

$\max_{y \simeq_{\pm, \mathbb{Y}} y'} ||f(y) - f(y')||_2$. *Then for all* $x \simeq_{K_+, K_-, \mathbb{X}} x'$, $\Psi_\alpha(m_x || m_{x'})$ *is l.e.q.*

$$\Psi_\alpha \left( \sum_{i=1}^{K_-+1} f_i^{(1)} \cdot \text{Binom}(i-1 \mid K_-, r) || \sum_{j=1}^{K_++1} f_j^{(2)} \cdot \text{Binom}(j-1 \mid K_+, r) \right),$$

*with univariate normal densities* $f_i^{(1)} = \mathcal{N}(\cdot \mid (i-1), \sigma / L_2)$, $f_j^{(2)} = \mathcal{N}(\cdot \mid -(j-1), \sigma / L_2)$.

*Proof sketch.* Recall from Theorem 3.7 that we need to solve the optimization problem

$$\max_{\boldsymbol{y}} \Psi_\alpha \left( \sum_{i=1}^{K_-+1} w_i^{(1)} \cdot b_{y_i^{(1)}} || \sum_{j=1}^{K_++1} w_j^{(2)} \cdot b_{y_j^{(2)}} \right),$$

subject to $\boldsymbol{y} \in \mathbb{Y}^{(K_++1)+(K_-+1)}, \forall l, t, u : d_{\mathbb{Y}}(y_t^{(l)}, y_u^{(l)}) \leq |t-u|, \forall t, u : d_{\mathbb{Y}}(y_t^{(1)}, y_u^{(2)}) \leq (t-1) + (u-1)$, and with $w_i^{(1)} = \text{Binom}(i-1 \mid K_-, r), w_j^{(2)} = \text{Binom}(j-1 \mid K_+, r)$.

Let $\boldsymbol{\mu}_i^{(1)} = h(y_i^{(1)})$ and $\boldsymbol{\mu}_j^{(2)} = h(y_j^{(2)})$. Since we do not have any additional information about $h$ beyond its $\ell_2$-sensitivity, we have to make the worst-case assumption that the $\boldsymbol{\mu}_i^{(1)}, \boldsymbol{\mu}_j^{(2)}$ are arbitrary vectors constrained by $\forall l, t, u : ||\boldsymbol{\mu}_t^{(l)} - \boldsymbol{\mu}_u^{(l)}||_2 \leq L_2|t-u|, \forall t, u : ||\boldsymbol{\mu}_t^{(1)} - \boldsymbol{\mu}_u^{(2)}||_2 \leq L_2((t-1) + (u-1))$,

Thus, the optimization problem is equivalent to

$$\max_{\boldsymbol{\mu}^{(1)}, \boldsymbol{\mu}^{(2)}} \left( \Psi_\alpha \left( \sum_{i=1}^{K_-+1} w_i^{(1)} \mathcal{N}(\cdot \mid \boldsymbol{\mu}_i^{(1)}, \sigma^2 \boldsymbol{I}_D / L_2^2) || \sum_{j=1}^{K_++1} w_j^{(2)} \mathcal{N}(\cdot \mid \boldsymbol{\mu}_j^{(2)}, \sigma^2 \boldsymbol{I}_D / L_2^2) \right) \right)$$

subject to $\forall l, t, u : ||\boldsymbol{\mu}_t^{(l)} - \boldsymbol{\mu}_u^{(l)}||_2 \leq |t-u|, \forall t, u : ||\boldsymbol{\mu}_t^{(1)} - \boldsymbol{\mu}_u^{(2)}||_2 \leq (t-1) + (u-1)$.

In Appendix O, we rigorously prove that the maximum is attained by co-linear, equidistant means that fulfill these distance constraints exactly. Thus, we can perform a coordinate transformation such that $\boldsymbol{\mu}_1 = \boldsymbol{0}$ and $\boldsymbol{\mu}_i^{(1)} = (i-1)e_1$ and $\boldsymbol{\mu}_j^{(2)} = -(j-1)e_1$ with first-component indicator vector $\boldsymbol{e}_1 \in \mathbb{R}^D$. Since the likelihood ratio in $\Psi_\alpha$ is Gaussian with zero mean in all but the first dimension, we can marginalize all other dimensions out to obtain our one-dimensional result (cf. Appendix O). $\square$

### M.2.2 Laplace mechanism guarantee

**Theorem M.2.** *Let* $M = B \circ S$, *where* $S$ *is Poisson subsampling with rate* $r$, *and* $B$ *is the Laplacian mechanism* $h + V$ *with* $h : \mathbb{Y} \to \mathbb{R}^D$ *and* $V \sim \text{Lap}(\boldsymbol{0}, \sigma^2 \boldsymbol{I}_D)$, *with location* $\boldsymbol{0} \in \mathbb{R}^D$ *and diagonal scale matrix* $\lambda \boldsymbol{I}_D \in \mathbb{R}_+^{D \times D}$, *which adds independent Laplacian noise to each dimension. Define the* $\ell_1$-sensitivity $L_1 = \max_{y \simeq_{\pm, \mathbb{Y}} y'} ||f(y) - f(y')||_1$. *Then, for all* $x \simeq_{K_+, K_-, \mathbb{X}} x'$, $\Psi_\alpha(m_x || m_{x'})$ *is l.e.q.*

$$\Psi_\alpha \left( \sum_{i=1}^{K_-+1} f_i^{(1)} \cdot \text{Binom}(i-1 \mid K_-, r) || \sum_{j=1}^{K_++1} f_j^{(2)} \cdot \text{Binom}(j-1 \mid K_+, r) \right),$$

*with univariate Laplace densities* $f_i^{(1)} = \text{Lap}(\cdot \mid (i-1), \lambda / L_1)$, $f_j^{(2)} = \text{Lap}(\cdot \mid -(j-1), \lambda / L_1)$.

*Proof sketch.* Recall from Theorem 3.7 that we need to solve the optimization problem

$$\Psi_\alpha \left( \sum_{i=1}^{K_-+1} b_{y_i^{(1)}} \cdot w_i^{(1)} || \sum_{j=1}^{K_++1} b_{y_j^{(2)}} \cdot w_j^{(2)} \right),$$

subject to $\boldsymbol{y} \in \mathbb{Y}^{K_++K_-}, \forall l, t, u : d_{\mathbb{Y}}(y_t^{(l)}, y_u^{(l)}) \leq |t-u|, \forall t, u : d_{\mathbb{Y}}(y_t^{(1)}, y_u^{(2)}) \leq (t-1) + (u-1)$, and with $w_i^{(1)} = \text{Binom}(i-1 \mid K_-, r), w_j^{(2)} = \text{Binom}(j-1 \mid K_+, r)$.

Let $\boldsymbol{\mu}_i^{(1)} = h(y_i^{(1)})$ and $\boldsymbol{\mu}_j^{(2)} = h(y_j^{(2)})$. Since we do not have any additional information about $h$ beyond its $\ell_1$-sensitivity, we have to make the worst-case assumption that the $\boldsymbol{\mu}_i^{(1)}, \boldsymbol{\mu}_j^{(2)}$ are arbitrary vectors constrained by $\forall l, t, u : ||\boldsymbol{\mu}_t^{(l)} - \boldsymbol{\mu}_u^{(l)}||_1 \leq L_1|t - u|, \forall t, u : ||\boldsymbol{\mu}_t^{(1)} - \boldsymbol{\mu}_u^{(2)}||_1 \leq L_1\left((t - 1) + (u - 1)\right)$,

Thus, the optimization problem is equivalent to.

$$\max_{\boldsymbol{\mu}^{(1)}, \boldsymbol{\mu}^{(2)}} \left( \Psi_\alpha \left( \sum_{i=1}^{K_-+1} \mathrm{Lap}(\cdot \mid \boldsymbol{\mu}_i^{(1)}, \lambda \boldsymbol{I}_D / L_1) \cdot w_i^{(1)} || \sum_{j=1}^{K_++1} \mathrm{Lap}(\cdot \mid \boldsymbol{\mu}_j^{(2)}, \lambda \boldsymbol{I}_D / L_1) \cdot w_j^{(2)} \right) \right)$$

subject to $\forall l, t, u : ||\boldsymbol{\mu}_t^{(l)} - \boldsymbol{\mu}_u^{(l)}||_1 \leq |t - u|, \forall t, u : ||\boldsymbol{\mu}_t^{(1)} - \boldsymbol{\mu}_u^{(2)}||_1 \leq (t - 1) + (u - 1)$.

In Appendix O, we rigorously show that the maximum is attained by collinear, equidistant vectors along a single coordinate axis that leave no slack on the pairwise distance constraints. This then allows us to marginalize out all remaining dimensions to obtain our guarantee in terms of univariate Laplace densities (see Appendix O). $\qquad\square$

### M.2.3 Randomized response mechanism guarantee

**Theorem M.3.** *Let $B$ be the randomized response mechanism $|h - (1 - V)|$ with $h : \mathbb{Y} \to \{0, 1\}$, $V \sim \mathrm{Bernoulli}(\theta)$, and true response probability $\theta \in [0, 1]$. Let $M = B \circ S$ be the corresponding subsampled mechanism, where $S$ is Poisson subsampling with rate $r$. Finally, let $\simeq_\mathbb{Y}$ be the insertion/removal relation $\simeq_{\pm,\mathbb{Y}}$. Then, for all $x \simeq_{K_+, K_-, \mathbb{X}} x'$, $\Psi_\alpha(m_x || m_{x'})$ is l.e.q.*

$$\max_\tau \Psi_\alpha((1 - w^{(1)})\mathrm{Bern}(\cdot \mid \theta) + w^{(1)}\mathrm{Bern}(\cdot \mid \tau) || (1 - w^{(2)})\mathrm{Bern}(\cdot \mid \theta) + w^{(2)}\mathrm{Bern}(\cdot \mid 1 - \tau))$$

*subject to $\tau \in \{\theta, 1 - \theta\}$, and with $w^{(1)} = 1 - (1 - r)^{K_-}$ and $w^{(2)} = 1 - (1 - r)^{K_+}$.*

*Proof sketch.* Recall from Theorem 3.7 that we need to solve the optimization problem

$$\Psi_\alpha \left( \sum_{i=1}^{K_-+1} b_{y_i^{(1)}} \cdot w_i^{(1)} || \sum_{j=1}^{K_++1} b_{y_j^{(2)}} \cdot w_j^{(2)} \right),$$

subject to $\boldsymbol{y} \in \mathbb{Y}^{K_++K_-}, \forall l, t, u : d_\mathbb{Y}(y_t^{(l)}, y_u^{(l)}) \leq |t - u|, \forall t, u : d_\mathbb{Y}(y_t^{(1)}, y_u^{(2)}) \leq (t - 1) + (u - 1)$, and with $w_i^{(1)} = \mathrm{Binom}(i - 1 \mid K_-, r), w_j^{(2)} = \mathrm{Binom}(j - 1 \mid K_+, r)$.

Let $\mu_i^{(1)} = h(y_i^{(1)})$ and $\mu_j^{(2)} = h(y_j^{(2)})$. Since we do not have any additional information about $h$ beyond it mapping to $\{0, 1\}$, we have to make the worst-case assumption that the $\mu_i^{(1)}, \mu_j^{(2)}$ are only constrained by $\mu_2^{(1)} = \mu_1^{(2)}$.

Thus, the optimization problem is equivalent to.

$$\max_{\boldsymbol{\mu}^{(1)}, \boldsymbol{\mu}^{(2)}} \left( \Psi_\alpha \left( \sum_{i=1}^{K_-+1} \mathrm{Bern}(\cdot \mid \mu_i^{(1)}) \cdot w_i^{(1)} || \sum_{j=1}^{K_++1} \mathrm{Bern}(\cdot \mid \mu_j^{(2)}) \cdot w_j^{(2)} \right) \right)$$

subject to $\mu_1^{(1)} = \mu_1^{(2)}$.

In Appendix O, we rigorously show that the maximum is attained whenever all $\mu_i^{(1)}$ with $i > 1$ are simultaneously set to either 0 or 1, and all $\mu_j^{(2)}$ with $j > 1$ are set to the opposite value. $\qquad\square$

Note that this guarantee can be evaluated in $\mathcal{O}(1)$: We need to iterate over two possible values of $\tau$, and evaluating the corresponding $\Psi_\alpha$ requires summing over two values $z \in \{0, 1\}$.

### M.3 Tightness

Next, we prove tightness by constructing datasets and underlying functions $h$ such that the corresponding base mechanism $B$ exactly attains the different bounds under subsampling scheme $S$.

### M.3.1 Gaussian mechanism tightness

**Theorem M.4.** *Let $S$ be Poisson subsampling with rate $r \in [0, 1]$ and $\theta \in [0, 1]$ be some true response probability. There exists a dataset space $\mathbb{X}$ and a batch space $(\mathbb{Y}, \mathcal{Y})$ fulfilling the constraints in Definition D.4, as well a pair of datasets $x \simeq_{K_+, K_-, \mathbb{X}} x'$, and a function $h : \mathbb{Y} \to \mathbb{R}^D$ with $\ell_2$-sensitivity $L_2$, such that the corresponding subsampled Gaussian mechanism $M = B \circ S$ fulfills*

$$\Psi_\alpha(m_x || m_{x'}) = \Psi_\alpha \left( \sum_{i=1}^{K_-+1} f_i^{(1)} \cdot \mathrm{Binom}(i - 1 \mid K_-, r) || \sum_{j=1}^{K_++1} f_j^{(2)} \cdot \mathrm{Binom}(j - 1 \mid K_+, r) \right),$$

*with univariate normal densities $f_i^{(1)} = \mathcal{N}(\cdot \mid (i - 1), \sigma / L_2)$, $f_j^{(2)} = \mathcal{N}(\cdot \mid -(j - 1), \sigma / L_2)$.*

*Proof.* Let $\mathbb{X} = \mathcal{P}(\mathbb{N})$. Consider an arbitrary $x \in \mathbb{X}$ and select arbitrary deleted elements $g \subseteq x$, with $|g| = K_-$ and arbitrary inserted elements $g \subseteq \mathbb{X} \setminus x$. Define $x' = x \setminus g_- \cup g_+$.

We now construct a counting function for $h$ that leads to the largest possible divergence under the sensitivity constraint. We define function $h$ as follows:

$$h(y) = \begin{cases} (i - 1) \cdot \boldsymbol{e}_1 L_2 & \text{if } |g_- \cap y| = i - 1 \\ -(j - 1) \cdot \boldsymbol{e}_1 L_2 & \text{if } |g_+ \cap y| = j - 1 \end{cases}$$

with first-component indicator vector $\boldsymbol{e}_1 \in \mathbb{R}^D$.

By construction, $P_{M_x}$ is a mixture distributions with $K + 1$ components, each corresponding to a size of a subset of $g_-$ that is included in batch $y$. These cases each have probability $\mathrm{Binom}(i - 1 \mid K_-, r)$ under subsampling distributions $P_{S_x}$. Each component has distribution $\mathcal{N}(\cdot \mid (i - 1), \boldsymbol{e}_1 L_2, \sigma^2 \boldsymbol{I}_D)$.

Analogously, $P_{m_{x'}}$ is the other desired mixture of Gaussians.

As in our previous proof, we can now notice that the likelihood ratio in $\Psi_\alpha$ is constant in all but the first dimension. We can thus marginalize out the remaining dimensions to obtain our result. $\square$

### M.3.2 Laplace mechanism tightness

**Theorem M.5.** *Let $S$ be Poisson subsampling with rate $r \in [0, 1]$ and $\theta \in [0, 1]$ be some true response probability. There exists a dataset space $\mathbb{X}$ and a batch space $(\mathbb{Y}, \mathcal{Y})$ fulfilling the constraints in Definition D.4, as well a pair of datasets $x \simeq_{K_+, K_-, \mathbb{X}} x'$, and a function $h : \mathbb{Y} \to \mathbb{R}^D$ with $\ell_1$-sensitivity $L_1$, such that the corresponding subsampled Laplace mechanism $M = B \circ S$ fulfills*

$$\Psi_\alpha(m_x || m_{x'}) = \Psi_\alpha \left( \sum_{i=1}^{K_-+1} f_i^{(1)} \cdot \mathrm{Binom}(i - 1 \mid K_-, r) || \sum_{j=1}^{K_++1} f_j^{(2)} \cdot \mathrm{Binom}(j - 1 \mid K_+, r) \right),$$

*with univariate Laplace densities $f_i^{(1)} = \mathrm{Lap}(\cdot \mid (i - 1), \lambda / L_1)$, $f_j^{(2)} = \mathrm{Lap}(\cdot \mid -(j - 1), \lambda / L_1)$.*

*Proof.* The proof is identical to that of the tightness guarantee from Theorem M.4. We can construct exactly the same counting function that indicates the number of inserted or removed element that appear in a batch sampled from $S_x$ and $S_{x'}$, respectively.

As in our previous proof, we can now notice that the likelihood ratio in $\Psi_\alpha$ is constant in all but the first dimension. We can thus marginalize out the remaining dimensions to obtain our result. $\square$

### M.3.3 Randomized response mechanism tightness

**Theorem M.6.** *Let $S$ be Poisson subsampling with rate $r \in [0, 1]$ and $\theta \in [0, 1]$ be some true response probability. There exists a dataset space $\mathbb{X}$ and a batch space $(\mathbb{Y}, \mathcal{Y})$ fulfilling the constraints in Definition D.4, as well a pair of datasets $x \simeq_{K_+, K_-, \mathbb{X}} x'$, and a function $h : \mathbb{Y} \to 0, 1$, such that the corresponding subsampled randomized response mechanism $M = B \circ S$ fulfills*

$$\Psi_\alpha(m_x || m_x')$$
$$= \max_\tau \Psi_\alpha((1 - w^{(1)})\mathrm{Bern}(\cdot \mid \theta) + w^{(1)}\mathrm{Bern}(\cdot \mid \tau) || (1 - w^{(2)})\mathrm{Bern}(\cdot \mid \theta) + w^{(2)}\mathrm{Bern}(\cdot \mid 1 - \tau))$$

*subject to $\tau \in \{\theta, 1 - \theta\}$, and with $w^{(1)} = 1 - (1 - r)^{K_-}$ and $w^{(2)} = 1 - (1 - r)^{K_+}$.*

*Proof.* The following proof is largely identical to our randomized response tightness proof for subsampling without replacement from Appendix I.3.

Let $\mathbb{X} = \mathcal{P}(\mathbb{N})$. Consider an arbitrary $x \in \mathbb{X}$ and select arbitrary deleted elements $g \subseteq x$, with $|g| = K_-$ and arbitrary inserted elements $g \subseteq \mathbb{X} \setminus x$. Define $x' = x \setminus g_- \cup g_+$.

Assume w.l.o.g. that the divergence is maximized by $\tau = \theta$.

We now construct an indicator function $h : \mathbb{Y} \to \{0, 1\}$ for $a$ and $a'$ that leads to the largest possible divergence.

$$h(y) = \begin{cases} 1 & \text{if } y \cap (g_- \cup g_+) = \emptyset \\ 1 & \text{if } y \cap g_- \neq \emptyset \\ 0 & \text{otherwise.} \end{cases}$$

By construction of $h$, the corresponding base mechanism pmf is

$$b_y(z) = \begin{cases} \text{Bern}(z \mid \theta) & \text{if } y \cap (g_- \cup g_+) = \emptyset \\ \text{Bern}(z \mid \theta) & \text{if } y \cap g_- \neq \emptyset \\ \text{Bern}(z \mid 1 - \theta) & \text{otherwise.} \end{cases}$$

Under the distribution of $S(x)$, the first case occurs with probability $(1 - r)^{K_-}$ and the second case occurs with probability $1 - (1 - r)^{K_-}$. Under the distribution of $S(x')$, the first case occurs with probability $(1 - r)^{K_+}$ and the second case occurs with probability $1 - (1 - r)^{K_+}$. We thus have

$$m_x(z) = (1 - w^{(1)})\text{Bern}(z \mid \theta) + w^{(1)}\text{Bern}(z \mid \theta),$$

$$m_{x'}(z) = (1 - w^{(2)})\text{Bern}(z \mid \theta) + w^{(2)}\text{Bern}(z \mid 1 - \theta),$$

which exactly attains the desired divergence when $\tau = \theta$ is the optimal value. $\square$

### M.4 Numerical evaluation

Evidently, we do not have a closed-form analytical expression for our tight mechanism-specific group privacy bounds. However, we can evaluate them to arbitrary precision using standard techniques from privacy accounting literature.

#### M.4.1 Gaussian mechanism

**ADP.** To evaluate the guarantee from Theorem 3.8, we can use Lemma 5 from [10], which generalizes an alternative characterization of privacy profiles from [74] to dominating pairs. Let $P, Q$ be the dominating pair of two univariate Gaussian mixtures. Define privacy loss random variables $L_{P,Q} = \frac{p(Z)}{q(Z)}$ with $Z \sim P$ and $L_{Q,P} = \frac{q(Z)}{p(Z)}$ with $Z \sim Q$. Then

$$H_\alpha(p||q) \leq \Pr[L_{P,Q} > \log(\alpha)] - \alpha \Pr[L_{Q,P} < -\log(\alpha)].$$

Because the privacy loss between the two Gaussian mixtures is monotonically increasing in $z$ [8], one can perform a change of variables via a binary search for a $z^*$ such that $\log(\frac{p(z^*)}{q(z^*)}) \approx \log(\alpha)$. By picking one of the two search boundaries, one can either over- or under-approximate the hockey stick divergence (see, e.g., [8, 24]).

**RDP via quadrature.** To evaluate the guarantee for RDP, we can simply use numerical quadrature. This can be done efficiently because we only need to integrate over univariate Gaussians. This approach was already proposed and used in Abadi et al. [6]'s work on moments accounting.

**RDP via expansion.** For the special case of $K_- = K$ and $K_+ = 0$, one can also use multinomial expansion (similar to prior work on RDP subsampling from [28–30]): We have

$$\left( \sum_{k=0}^{K} w_k \cdot \mathcal{N}(z \mid \mu_k, \sigma^2 \boldsymbol{I}) \right)^\alpha$$

$$= \sum_{l_0 + \cdots + l_K = \alpha} \binom{\alpha}{l_0, \ldots, l_K} \left( \prod_{k=0}^{K} w_k^{l_k} \right) \left( \prod_{k=0}^{K} \mathcal{N}(z \mid \mu_k, \sigma^2 \boldsymbol{I})^{l_k} \right)$$

$$= \sum_{l_0 + \cdots + l_K = \alpha} \binom{\alpha}{l_0, \ldots, l_K} \left( \prod_{k=0}^{K} w_k^{l_k} \right) \left( \prod_{k=0}^{K} \mathcal{N}(z \mid \mu_k, \sigma^2 \boldsymbol{I})^{l_k/\alpha} \right)^\alpha$$

Using quadratic expansion, we have

$$\prod_{k=0}^{K} \mathcal{N}(z \mid \mu_k, \sigma^2 \boldsymbol{I})^{l_k/\alpha}$$

$$= \mathcal{N}\left(z \mid \sum_k \frac{l_k}{\alpha} \mu_k, \sigma^2 \boldsymbol{I}\right) \cdot \exp\left(-\frac{1}{2\sigma^2} \sum_k \frac{l_k}{\alpha} ||\mu_k||_2^2\right) \cdot \exp\left(\frac{1}{2\sigma^2} \sum_k ||(\frac{l_k}{\alpha} \mu_k)||_2^2\right)$$

Since only the first factor depends on $z$, our problem reduces to computing the divergence

$$\Psi_\alpha \left( \mathcal{N}\left(z \mid \sum_k \frac{l_k}{\alpha} \mu_k, \sigma^2 \boldsymbol{I}\right) || \mathcal{N}\left(z \mid \mathbf{0}, \sigma^2 \boldsymbol{I}\right) \right)$$

for different $l_k$. This can be done in closed form, as shown in [7].

### M.4.2 Laplace mechanism

**ADP.** Because the privacy loss is constant on $(-\infty, -K_+)$, monotonically increasing on $[-K_+, K_-]$ and constant on $(K_-, \infty)$, we can again use the same bisection method.

**RDP.** As with the Gaussian mechanism, we can evaluate the bound via univariate numerical quadrature. Because the privacy loss is non-smooth at the component means $\{-K_+, -K_+ + 1, \ldots, 0, 1, \ldots, K_-\}$, we partition $\mathbb{R}$ at these means and integrate over each interval separately.

### M.4.3 Randomized response mechanism

The guarantee for randomized smoothing can be evaluated exactly in $\mathcal{O}(1)$. We just need to iterate over the two options $\tau \in \{0, 1\}$, and for each one evaluate the divergence on space $\{0, 1\}$, which only requires evaluating two fractions and two sums.

# N Tight mechanism-agnostic group privacy amplification

In this section, we apply the framework from [15], which we summarized in Appendix G.1, to the group privacy setting. We then show the tightness of the resultant guarantees, using the same proof strategy as in Section 5 of [15]. We then demonstrate that it is directly related to our tight mechanism-specific guarantee through joint convexity. Finally, we derive a qualitatively similar guarantee for RDP.

For this discussion, we focus on the special case where all of the $K$ group members collaboratively agree to insert their data ($K_+ = K, K_- = 0$) or delete their data ($K_+ = 0, K_- = K$), so that the partition induced by the maximal coupling remains interpretable. We define the corresponding neighboring relation as

$$\simeq_{K\pm,\mathbb{X}} = \{(x, x') \in \mathbb{X}^2 \mid x \subset x' \wedge |x'| = |x| + K\} \cup \{(x, x') \in \mathbb{X}^2 \mid x \supset x' \wedge |x'| = |x| - K\}. \quad (27)$$

In our experiments, we only evaluate the baseline for ($K_+ = K, K_- = 0$) and ($K_+ = 0, K_- = K$), while we evaluate our method for all $K_+ + K_- = K$ and take the maximum. This evaluation favors the baseline.

## N.1 ADP guarantee

**Proposition N.1.** *Let $M = B \circ S$, where $S$ is Poisson subsampling with rate $r$. Let $\simeq_\mathbb{Y}$ be the insertion/removal batch relation $\simeq_{\pm,\mathbb{Y}}$. Then, for all $x \simeq_{K\pm,\mathbb{X}} x'$ and all $\varepsilon \geq 0$*

$$H_{\exp(\varepsilon')}(m_x || m_{x'}) \leq \sum_{k=1}^{K} \mathrm{Binom}(k \mid K, r) \cdot \delta_k$$

*with $\varepsilon' = \log(1 + (1 - (1-r)^K) \cdot (e^\varepsilon - 1))$ and group privacy parameters*

$$\delta_k = \max_{y,y'} H_{\exp(\varepsilon)}(b_y || b_{y'}) \quad s.t. \quad d_\mathbb{Y}(y, y') \leq k.$$

*Proof.* **Case 1: Deletion.** We first consider the case where $K$ elements are deleted, i.e., there is some deleted set $g \subseteq x$ of size $K$ with $g_- \cap x' = \emptyset$ such that $x' = x \setminus g$.

The partition induced by the maximal coupling [9] is

$$s_x(y) = (1-w)s_x(y \mid A_2) + ws_x(y \mid A_1)$$
$$s_{x'}(y) = (1-w)s_x(y \mid A_2) + ws_x(y \mid A_2),$$

where $A_2 = \{y \in \mathbb{Y} \mid y \cap g = \emptyset$ is the event that no element of $g$ is sampled, $A_1$ is its complement, and $w = (1 - (1-r)^K) = P_{S_x}(A_1)$. We use these indices for $A_2$ and $A_1$ because advanced joint convexity will later reverse their order.

Applying advanced joint convexity (see Proposition G.1) and joint convexity shows that

$$H_{\exp(\varepsilon')}(m_x || m_{x'}) \leq w \cdot H_{\exp(\varepsilon)} \left( \sum_{y \in \mathbb{Y}} b_y \cdot s_x(y \mid A_1) || \sum_{y \in \mathbb{Y}} b_y \cdot s_x(y \mid A_2) \right). \quad (28)$$

Next, we can bound this mixture divergence by constructing a coupling invoking Theorem 3.3 with the special case of $\Psi_\alpha = H_{\exp(\varepsilon)}$. For this, notice that

$$s_x(y \mid A_1) = \begin{cases} \frac{1}{w} \cdot r^{|y|} \cdot (1-r)^{|x|-|y|} & \text{if } y \in A_1 \wedge y \subseteq x \\ 0 & \text{otherwise.} \end{cases}$$

and

$$s_x(y \mid A_2) = s_{x'}(y) = \begin{cases} r^{|y|} \cdot (1-r)^{|x'|-|y|} & \text{if } y \in A_2 \wedge y \subseteq x' \\ 0 & \text{otherwise.} \end{cases}$$

---

[9] as can be seen by the fact that $w$ is the total variation distance of $s_x$ and $s_{x'}$, and that $s_x(y \mid A_2)$ and $s_x(y \mid A_1)$ have disjoint support

**Coupling.** We define a coupling $\gamma : \mathbb{Y}^2 \to [0, 1]$ that corresponds to the following generative process: We first sample $y^{(2)}$ from $s_x(y \mid A_2)$. We then sample from a truncated binomial distribution how many elements $k$ from group $g$ should be included. Given this $k$, we sample uniformly at random a $\tilde{g} \subseteq g$ from all size-$k$ subsets of $g$ and let $y^{(1)} \leftarrow y^{(2)} \cup \tilde{g}$. Formally this is defined by

$$\gamma(y^{(1)}, y^{(2)}) = \begin{cases} s_x(y^{(2)} \mid A_2) \cdot \frac{1}{w} \cdot \mathrm{Binom}(|y^{(1)} \cap g| \mid K, r) \cdot \binom{K}{|y^{(1)} \cap g|}^{-1} & \text{under Condition N.2} \\ 0 & \text{otherwise.} \end{cases}$$

**Condition N.2.** A tuple $y^{(1)} \in \mathbb{Y}$, $y^{(2)} \in \mathbb{Y}$ fulfills this condition when there exists a $\tilde{g} \subseteq g$ with $|\tilde{g}| \geq 1$ such that $y^{(1)} = y^{(2)} \cup \tilde{g}$.

**Validity.** Next, we verify the validity of this coupling. For every $y^{(1)}$, there is exactly one $y^{(2)}$ such that $\gamma(y^{(1)}, y^{(2)}) > 0$, namely $y^{(2)} = y^{(1)} \setminus g'$. Thus,

$$\sum_{y^{(2)}} \gamma(y, y^{(2)}) = s_x(y \setminus g \mid A_2) \cdot \frac{1}{w} \cdot \mathrm{Binom}(|y \cap g| \mid K, r) \cdot \binom{K}{y \cap g}^{-1}$$

$$= r^{|y \setminus g|} \cdot (1 - r)^{|x'| - |y \setminus g|} \cdot \frac{1}{w} \cdot r^{|y \cap g|} \cdot (1 - r)^{K - |y \cap g|}$$

$$= s_x(y \mid A_1).$$

For every $y^{(2)}$, there are exactly $\binom{K}{k}$ batches $y^{(1)}$ such that $\gamma(y^{(1)}, y^{(2)}) > 0$ and $|y^{(1)} \cap g = k|$. Thus,

$$\sum_{y^{(1)}} \gamma(y^{(1)}, y) = \sum_{k=1}^{K} \binom{K}{k} s_x(y \mid A_2) \cdot \frac{1}{w} \cdot \mathrm{Binom}(k \mid K, r) \cdot \binom{K}{k}^{-1}$$

$$= s_x(y \mid A_2) \cdot \frac{1}{w} \cdot \sum_{k=1}^{K} \mathrm{Binom}(k \mid K, r)$$

$$= s_x(y \mid A_2).$$

Now that we have proven the validity of the coupling, we can apply the optimal transport bound (Theorem 3.3) to Eq. (28) in order to prove

$$H_{\exp(\varepsilon')}(m_x \| m_{x'}) \leq w \cdot \sum_{y^{(1)}, y^{(2)}} \gamma(y^{(1)}, y^{(2)}) \delta_{d_{\mathbb{Y}}(y^{(1)}, y^{(2)})}$$

$$= w \cdot \sum_{k=1}^{K} \binom{K}{k} \frac{1}{w} \mathrm{Binom}(k \mid K, r) \binom{K}{k}^{-1} \cdot \delta_k$$

$$= \sum_{k=1}^{K} \mathrm{Binom}(k \mid K, r) \cdot \delta_k.$$

**Case 2: Insertion.** In this case, the partition induced by the maximal coupling is

$$s_x(y) = (1 - w)s_x(y \mid A_1) + w s_x(y \mid A_2)$$
$$s_{x'}(y) = (1 - w)s_x(y \mid A_1) + w s_x(y \mid A_1),$$

which is identical to the previous partition, up to symmetry. The proof is identical, except for changes in indexing.

Taking the maximum over both guarantees yields the result. □

## N.2 Tightness of ADP guarantee

**Proposition N.3.** *Let $S$ be Poisson subsampling with rate $r$. Let $\simeq_{\mathbb{Y}}$ be the insertion/removal batch relation $\simeq_{\pm, \mathbb{Y}}$. Then, for all $x \simeq_{K\pm, \mathbb{X}} x'$ and all $\varepsilon \geq 0$, there exists a worst-case base mechanism $B$*

such that the corresponding subsampled mechanism $M = B \circ S$ fulfills

$$H_{\exp(\varepsilon')}(m_x||m_{x'}) = \sum_{k=1}^{K} \mathrm{Binom}(k \mid K, r) \cdot \delta_k \tag{29}$$

with $\varepsilon' = \log(1 + (1 - (1 - r)^K) \cdot (e^\varepsilon - 1))$ and group privacy parameters

$$\delta_k = \max_{y, y'} H_{\exp(\varepsilon)}(b_y||b_{y'}) \quad s.t. \quad d_{\mathbb{Y}}(y, y') \le k.$$

*Proof.* To show tightness of the bound, we notice that the bound $\sum_{k=1}^{K} \mathrm{Binom}(k \mid K, r) \cdot \delta_k$ is identical to the bound for subsampling with replacement from [15], except for the numeric value of the weights in the weighted sum. We can thus use exactly the same proof strategy.

Assume w.l.o.g. that we are in the insertion case, i.e., there is some $g$ of size $K$ with $g \cap x = \emptyset$ and $x' = x \cup g$. Consider an arbitrary $\varepsilon$ and define an arbitrary true response probability $\theta \in [0, 1]$. Further define the randomized membership base mechanism

$$B(y) = |\mathbb{1}[|y \cap g| > 0] - (1 - V)|$$

with $V \sim \mathrm{Bern}(\theta)$. It is easy to verify [15] that $\delta_k = \psi_\theta(\varepsilon) = \max\{\theta - e^\varepsilon(1 - p), 0\}$. We thus have for the r.h.s. of Eq. (29)

$$\sum_{k=1}^{K} \mathrm{Binom}(k \mid K, r) \cdot \delta_k = \sum_{k=1}^{K} \mathrm{Binom}(k \mid K, r) \cdot \psi_\theta(\varepsilon) := w \cdot \psi_\theta(\varepsilon).$$

Because Poisson subsampling is a "natural subsampling"[15] scheme that only yields elements from the dataset it is applied to, it follows from Lemma 12 of [15] that also

$$H_{\exp(\varepsilon')}(m_x||m_{x'}) = w \cdot \psi_\theta(\varepsilon).$$

$\square$

### N.3 Relation to tight mechanism-specific bound

In the following, we show that this mechanism-agnostic guarantee implicitly upper bounds our tight mechanism-specific guarantee via joint convexity. Note that this is qualitatively different from our discussion in Appendix G. There we showed, that the mechanism-agnostic guarantees can be derived from mechanism-specific bounds that only use a binary partitioning of the batch space (unlike the mechanism-specific guarante considered here). We show this result w.l.o.g. for $K_- = 0$ and $K_+ = K$.

**Theorem N.4.** *Let $B : \mathbb{Y} \to \mathbb{Z}$ be an arbtirary base mechanism. Let $\mathbf{y}^{(1)} \in \mathbb{Y}^1$ and $, \mathbf{y}^{(2)} \in \mathbb{Y}^{K+1}$ be arbitrary tuples of batches that fulfill $\forall l, t, u : d_{\mathbb{Y}}(y_t^{(l)}, y_u^{(l)}) \le |t - u|$, and $\forall t, u : d_{\mathbb{Y}}(y_t^{(1)}, y_u^{(2)}) \le (t - 1) + (u - 1)$. Then, for $\alpha \ge 1$*

$$H_{\alpha'}\left(b_{y_1^{(1)}} || \sum_{j=1}^{K+1} b_{y_j^{(2)}} \cdot \mathrm{Binom}(j - 1 \mid K, r)\right) \le \sum_{k=1}^{K} \mathrm{Binom}(k \mid K, r) \cdot \delta_k(\alpha)$$

with $w = (1 - (1 - r)^K)$, $\alpha' = 1 + w(\alpha - 1)$ and group privacy parameters

$$\delta_k(\alpha) = \max_{y, y'} H_\alpha(b_y||b_{y'}) \quad s.t. \quad d_{\mathbb{Y}}(y, y') \le k.$$

*Proof.* We can apply joint convexity to show

$$H_{\alpha'}\left(b_{y_1^{(1)}} \middle|\middle| \sum_{j=1}^{K+1} b_{y_j^{(2)}} \cdot \mathrm{Binom}(j-1 \mid K, r)\right) \tag{30}$$

$$=H_{\alpha'}\left(b_{y_1^{(1)}} \middle|\middle| (1-w) \cdot b_{y_1^{(2)}} + \sum_{j=2}^{K+1} w \cdot b_{y_j^{(2)}} \cdot \left(\frac{1}{w} \cdot \mathrm{Binom}(j-1 \mid K, r)\right)\right) \tag{31}$$

$$=H_{\alpha'}\left(b_{y_1^{(1)}} \middle|\middle| \sum_{j=2}^{K+1} \left((1-w) \cdot b_{y_1^{(2)}} + w \cdot b_{y_j^{(2)}}\right) \cdot \left(\frac{1}{w} \cdot \mathrm{Binom}(j-1 \mid K, r)\right)\right) \tag{32}$$

$$\leq \sum_{j=2}^{K+1} \frac{1}{w} \cdot \mathrm{Binom}(j-1 \mid K, r) \cdot H_{\alpha'}\left(b_{y_1^{(1)}} \middle|\middle| \left((1-w) \cdot b_{y_1^{(2)}} + w \cdot b_{y_j^{(2)}}\right)\right) \tag{33}$$

The result then immediately follows from advanced joint convexity (see Proposition G.1) and joint convexity. $\square$

## N.4   RDP guarantee

For RDP, we can obtain a qualitatively similar guarantee by simply applying the known mechanism-agnostic RDP subsampling guarantee for Poisson subsampling from [29, 30] to each of the $K$ divergences (although we could also derive this RDP guarantee from scratch via optimal transport) in Eq. (33) of the previous derivation.

**Theorem N.5.** *Let* $M = B \circ S$*, where* $S$ *is Poisson subsampling with rate* $r$*. Let* $\simeq_{\mathbb{Y}}$ *be the insertion/removal batch relation* $\simeq_{\pm, \mathbb{Y}}$*. Then, for all* $x \simeq_{K\pm, \mathbb{X}} x'$ *and all* $\alpha > 0$

$$\Lambda_\alpha(m_x || m_{x'}) \leq \sum_{k=1}^{K} \frac{1}{w} \cdot \mathrm{Binom}(k \mid K, r) \cdot 2 \cdot \sum_{l=0}^{\alpha} \binom{\alpha}{l} w^l (1-w)^{\alpha-l} \zeta_k(l)$$

*with* $w = (1 - (1-r)^K)$ *and group privacy parameters*

$$\zeta_k(l) = \max_{y, y'} \Lambda_l(b_y || b_{y'}) \quad s.t. \quad d_{\mathbb{Y}}(y, y') \leq k.$$

The factor 2 in Theorem N.5 can be eliminated for distributions with particular symmetries (see Theorem 5 in [29]) or bounded Pearson-Vajda $\chi^l$-pseudo-divergence (see Theorem 8 in [30]). We thus do not include the factor 2 when using this amplification guarantee as a baseline in our experiments.

## N.5   Asymptotic RDP guarantees

As mentioned in Section 3.1, our focus is on tight bounds that can be explicitly computed. However, analyzing the asymptotic behavior of these bounds can provide a potentially useful high-level picture of their behavior. As discussed in the previous section, and as can be seen from Eq. (33), Rényi divergence in the group privacy setting is bounded by a weighted sum of Rényi divergences between a single distribution and a mixture of two distributions. For the special case of additive Gaussian mechanisms with global sensitivity $L$, this bound is equivalent (see Appendix A of [30]) to

$$\sum_{k=1}^{K} \frac{1}{w} \cdot \mathrm{Binom}(k \mid K, r) \cdot 2 \cdot \Lambda_\alpha\left(\mathcal{N}(0, \sigma) || (1-w)N(0, \sigma) + w \cdot N(1, \sigma / (k \cdot L))\right). \tag{34}$$

Asymptotic bounds on Rényi divergences with one or two components have been derived in prior work on privacy accounting [6, 28, 48]. We can apply these asymptotic bounds to each summand. For instance (see Lemma 5 in [6]):

**Proposition N.6.** *Abadi et al. [6] Let* $\sigma \geq 1$ *and* $q \leq \frac{1}{16\sigma}$*. Then, for any positive integer* $\alpha \leq \sigma^2 \ln\left(\frac{1}{q\sigma}\right) - 1$,

$$\Lambda_\alpha\left(\mathcal{N}(0, \sigma) || (1-q)N(0, \sigma) + q \cdot N(1, \sigma)\right) \leq \frac{q^2 \alpha(\alpha-1)}{(1-q)\sigma^2} + \mathcal{O}(q^3 \alpha^3 / \sigma^3).$$

Applying Proposition N.6 to Eq. (34) yields the following asymptotic bound for RDP group privacy:

**Theorem N.7.** *Let $M = B \circ S$, where $S$ is Poisson subsampling with rate $r$ and $B$ is an additive gaussian mechanism with global sensitivity $L$ under the insertion/removal relation $\simeq_{\pm}$. Consider arbitrary datasets $x \simeq_{K\pm,\mathbb{X}} x'$. Define weight $w = (1 - (1 - r)^K)$. If $\sigma \geq k \cdot L$ and $w \leq \frac{1}{16\sigma}$, then it holds for any positive integer $\alpha \leq \frac{\sigma^2}{k^2 \cdot L^2} \ln\left(\frac{1}{w\sigma}\right) - 1$ that*

$$\Lambda_\alpha(m_x || m_{x'}) \leq \sum_{k=1}^{K} \frac{1}{w} \cdot \mathrm{Binom}(k \mid K, r) \cdot 2 \cdot \frac{k^2 L^2 q^2 \alpha(\alpha - 1)}{(1 - w)\sigma^2} + \mathcal{O}(k^3 L^3 w^3 \alpha^3 \,/\, \sigma^3).$$

Alternatively, one could apply the asymptotic bound from Theorem 38 from [48] to each summand. If we were to instead consider group privacy under dataset-level substitutions, where each summand would include a divergence between two mixtures with two components, we could instead use the asymptotic bounds from Appendix C of [28].

## O  Worst-case mixture components

### O.1  Gaussian and Laplacian mixtures

We outline a self-contained and extendable proving strategy which we use to find dominating pairs of Gaussian and Laplacian mixtures given divergences of the form $\Psi_\alpha(P\|Q) = \int_{\mathbb{R}^D} f\left(P(\boldsymbol{x}), -Q(\boldsymbol{x})\right) d\boldsymbol{x}$, where $f$ is (not necessarily strictly) convex and increasing in both arguments. Our two main examples of the hockey stick divergence $\Psi_\alpha = H_\alpha$ and (scaled and exponentiated) Rényi divergence $\Psi_\alpha = \Lambda_\alpha$ are special cases with $f(x, y) = \max\{x + \alpha y, 0\}$ and $f(x, y) = x^\alpha(-y)^{1-\alpha}$, respectively. Throughout the subsection, we use $M$, $N$ to denote the sets of means of the two mixtures $P, Q$. We start with two general lemmata before treating the Gaussian and Laplacian mixture cases in particular. Together, they provide a constructive toolset to connect any mixture pair with means $(M, N)$ to a dominating mixture pair with means $(M^*, N^*)$ via a path of geometric transformations $(M, N) \longmapsto (M^{(1)}, N^{(1)}) \longmapsto \cdots \longmapsto (M^*, N^*)$: concretely, these are

- mirroring the means of the two mixtures onto opposite sides of a hyperplane, and

- pushing such hyperplane-separated means further away along the hyperplane normal.

In the first part of this section, we prove that the divergence can only stay equal or grow under any such transformation. Afterwards, we construct an explicit path of transformations that maps any Gaussian, as well as any Laplacian mixture onto a dominating pair which is feasible under the pairwise distance constraints discussed in Appendix M.2.

**Lemma O.1.** *Given $p \in \{1, 2\}$, consider two mixtures of the form $P_M(\boldsymbol{x}) = \sum_{k=0}^K w_k \rho(\|\boldsymbol{x} - \mu_k\|_p)$, $Q_N(\boldsymbol{x}) = \sum_{\kappa=0}^{\mathcal{K}} \omega_\kappa \rho(\|\boldsymbol{x} - \nu_\kappa\|_p)$ with means in $M := \{\mu_0, \ldots, \mu_K\} \subset \mathbb{R}^D$, $N := \{\nu_0, \ldots, \nu_\mathcal{K}\} \subset \mathbb{R}^D$ and a decreasing $\rho \colon \mathbb{R}_0^+ \to [0, 1]$. Consider all hyperplanes which contain zero and are normal to $L^2$-unit vectors $\hat{\boldsymbol{n}} \in \mathcal{H}_p$, where*

*(1) $\mathcal{H}_1 = \{\sigma \hat{e}_i \mid 0 \leq i \leq K, \sigma \in \{+, -\}\} \cup \{\frac{1}{\sqrt{2}}(\sigma_1 \hat{e}_i + \sigma_2 \hat{e}_j) \mid 0 \leq i \neq j \leq K, \sigma_1, \sigma_2 \in \{+, -\}\}$,*
*(2) $\mathcal{H}_2 = S^D$, the unit $D$-sphere,*

*and define the lower half-space $\mathbb{R}_{\hat{\boldsymbol{n}}}^- = \{\boldsymbol{x} \in \mathbb{R}^D \mid \left(\boldsymbol{x}^T \hat{\boldsymbol{n}}\right) < 0\}$ and upper half-space $\mathbb{R}_{\hat{\boldsymbol{n}}}^+ = \{\boldsymbol{x} \in \mathbb{R}^D \mid \left(\boldsymbol{x}^T \hat{\boldsymbol{n}}\right) > 0\}$.*
*Consider the map*

$$(\,\cdot\,)_{\hat{\boldsymbol{n}}}' \colon \mathbb{R}^D \longrightarrow \mathbb{R}^D \setminus \mathbb{R}_{\hat{\boldsymbol{n}}}^-, \quad \boldsymbol{x} \longmapsto \boldsymbol{x} - \mathbb{1}_{\mathbb{R}_{\hat{\boldsymbol{n}}}^-}(\boldsymbol{x})(2\hat{\boldsymbol{n}}^T \boldsymbol{x})\hat{\boldsymbol{n}}$$

*which mirror-reflects the lower into the upper half-space, and its image sets $M_{\hat{\boldsymbol{n}}}' = \mathbb{R}^D \setminus \mathbb{R}_{\hat{\boldsymbol{n}}}^-$ and $N_{-\hat{\boldsymbol{n}}}' \subset \mathbb{R}^D \setminus \mathbb{R}_{\hat{\boldsymbol{n}}}^+$.*
*The reflected pair of mixtures has equal or greater divergence:*

$$\Psi_\alpha\left(P_{M_{\hat{\boldsymbol{n}}}'}\|Q_{N_{-\hat{\boldsymbol{n}}}'}\right) \geq \Psi_\alpha\left(P_M\|Q_N\right).$$

*Remark* O.2. Lemma O.1 applies to Laplacian and Gaussian mixtures with $p = 1$ and $p = 2$, respectively.

*Proof.* We will need the following lemma.

**Lemma O.3.** *Given $p \in \{1, 2\}$, a hyperplane normal to $\hat{\boldsymbol{n}} \in \mathcal{H}_p$, and points $\boldsymbol{x}_1, \boldsymbol{x}_2 \in \mathbb{R}^D$, we have*

$$\|(\boldsymbol{x}_1)_{\hat{\boldsymbol{n}}}' - (\boldsymbol{x}_2)_{\hat{\boldsymbol{n}}}'\|_p \leq \|\boldsymbol{x}_1 - \boldsymbol{x}_2\|_p \quad if \quad (\boldsymbol{x}_1 \in \mathbb{R}_{\hat{\boldsymbol{n}}}^+ \wedge \boldsymbol{x}_2 \in \mathbb{R}_{\hat{\boldsymbol{n}}}^-) \quad \vee \quad (\boldsymbol{x}_1 \in \mathbb{R}_{\hat{\boldsymbol{n}}}^- \wedge \boldsymbol{x}_2 \in \mathbb{R}_{\hat{\boldsymbol{n}}}^+)$$
$$\|(\boldsymbol{x}_1)_{\hat{\boldsymbol{n}}}' - (\boldsymbol{x}_2)_{\hat{\boldsymbol{n}}}'\|_p = \|\boldsymbol{x}_1 - \boldsymbol{x}_2\|_p \quad else.$$

**Corollary O.4.** *If the sets $M$, $N$ are feasible under pairwise $p$-norm distance constraints, then so are $M_{\hat{\boldsymbol{n}}}'$, $N_{-\hat{\boldsymbol{n}}}'$.*

*Proof of Lemma O.3.* If $\text{sign}(\boldsymbol{n}^T \boldsymbol{x}_1) = \text{sign}(\boldsymbol{n}^T \boldsymbol{x}_2)$, then $(\,\cdot\,)_{\hat{\boldsymbol{n}}}'$ acts uniformly on both vectors and the condition clearly holds. Else, assume w.l.o.g. that $\boldsymbol{x}_2 \in \mathbb{R}_{\hat{\boldsymbol{n}}}^-$. We start with the case $p = 1$.

First, assume $\hat{\boldsymbol{n}} \in \{\sigma \hat{\boldsymbol{e}}_i \mid 0 \le i \le K, \sigma_1 \in \{+, -\}\}$. Since $(\,\cdot\,)'_{\hat{\boldsymbol{n}}}$ now acts only on vector component $i$, we can write $\|(\boldsymbol{x}_1)'_{\hat{\boldsymbol{n}}} - (\boldsymbol{x}_2)'_{\hat{\boldsymbol{n}}}\|_1 - \|\boldsymbol{x}_1 - \boldsymbol{x}_2\|_1 = \|\hat{\boldsymbol{e}}_i^T \boldsymbol{x}_1| - |\hat{\boldsymbol{e}}_i^T \boldsymbol{x}_2\| - |\hat{\boldsymbol{e}}_i^T \boldsymbol{x}_1 - \hat{\boldsymbol{e}}_i^T \boldsymbol{x}_2| \le 0$ by the inverse triangle inequality.

Now assume instead that $\hat{\boldsymbol{n}} \in \{\frac{1}{\sqrt{2}}(\sigma_1 \hat{\boldsymbol{e}}_i + \sigma_2 \hat{\boldsymbol{e}}_j) \mid 1 \le i \ne j \le D, \sigma_1, \sigma_2 \in \{+1, -1\}\}$. Now, $(\,\cdot\,)'_{\hat{\boldsymbol{n}}}$ acts only on vector components $i$ and $j$. Thus, $\|(\boldsymbol{x}_1)'_{\hat{\boldsymbol{n}}} - (\boldsymbol{x}_2)'_{\hat{\boldsymbol{n}}}\|_1 - \|\boldsymbol{x}_1 - \boldsymbol{x}_2\|_1 = \|\pi_{i,j}\left[(\boldsymbol{x}_1)'_{\hat{\boldsymbol{n}}} - (\boldsymbol{x}_2)'_{\hat{\boldsymbol{n}}}\right]\|_1 - \|\pi^{(i,j)}\left[\boldsymbol{x}_1 - \boldsymbol{x}_2\right]\|_1$ where $\pi^{(i,j)}$ denotes projection onto the subspace spanned by the basis vectors $\hat{\boldsymbol{e}}_i, \hat{\boldsymbol{e}}_j$. It is useful to write

$$\|\pi_{i,j}\left[(\boldsymbol{x}_1)'_{\hat{\boldsymbol{n}}} - (\boldsymbol{x}_2)'_{\hat{\boldsymbol{n}}}\right]\|_1 = \max_{\boldsymbol{\sigma} \in \{-1,+1\}^D} |\boldsymbol{\sigma}^T \pi_{i,j}\left[(\boldsymbol{x}_1)'_{\hat{\boldsymbol{n}}} - (\boldsymbol{x}_2)'_{\hat{\boldsymbol{n}}}\right]|$$

$$= \max_{\boldsymbol{\sigma} \in \{-1,+1\}^D} |\boldsymbol{\sigma}^T \pi_{i,j}\left[\boldsymbol{x}_1 - \boldsymbol{x}_2 + 2(\hat{\boldsymbol{n}}^T \boldsymbol{x}_2)\hat{\boldsymbol{n}}\right]|.$$

The argument of the max function admits two cases. Either, $\boldsymbol{\sigma}^T(\pi_{i,j}\hat{\boldsymbol{n}}) = 0$, in which case $|\boldsymbol{\sigma}^T \pi_{i,j}\left[(\boldsymbol{x}_1)'_{\hat{\boldsymbol{n}}} - (\boldsymbol{x}_2)'_{\hat{\boldsymbol{n}}}\right]| = |\boldsymbol{\sigma}^T \pi_{i,j}\left[\boldsymbol{x}_1 - \boldsymbol{x}_2\right]|$. In the other case, $\pi_{i,j}\boldsymbol{\sigma} = \pm\frac{2}{\sqrt{2}}\pi_{i,j}\hat{\boldsymbol{n}}$. Then, by the inverse triangle inequality,

$$|\boldsymbol{\sigma}^T \pi_{i,j}\left[(\boldsymbol{x}_1)'_{\hat{\boldsymbol{n}}} - (\boldsymbol{x}_2)'_{\hat{\boldsymbol{n}}}\right]| = \frac{2}{\sqrt{2}}\left|\pm\hat{\boldsymbol{n}}^T \boldsymbol{x}_1 \mp \hat{\boldsymbol{n}}^T \boldsymbol{x}_2 \pm 2\hat{\boldsymbol{n}}^T \boldsymbol{x}_2\right| = \frac{2}{\sqrt{2}}\left|\pm\hat{\boldsymbol{n}}^T \boldsymbol{x}_1 \pm \hat{\boldsymbol{n}}^T \boldsymbol{x}_2\right|$$

$$= \frac{2}{\sqrt{2}}\left|\pm\left(|\hat{\boldsymbol{n}}^T \boldsymbol{x}_1| - |\hat{\boldsymbol{n}}^T \boldsymbol{x}_2|\right)\right| \le \left|\frac{2}{\sqrt{2}}\hat{\boldsymbol{n}}^T \left(\boldsymbol{x}_1 - \boldsymbol{x}_2\right)\right| = |\boldsymbol{\sigma}^T \pi_{i,j}\left[\boldsymbol{x}_1 - \boldsymbol{x}_2\right]|.$$

For completeness, we also prove $p = 2$. By invariance of the scalar product under mirror reflection, $\|\boldsymbol{x}_1' - \boldsymbol{x}_2'\|_2 = \boldsymbol{x}_1^T \boldsymbol{x}_1 + \boldsymbol{x}_2'^T \boldsymbol{x}_2' - 2\boldsymbol{x}_1^T \boldsymbol{x}_2' = \boldsymbol{x}_1^T \boldsymbol{x}_1 + \boldsymbol{x}_2^T \boldsymbol{x}_2 - 2\boldsymbol{x}_1^T \boldsymbol{x}_2 + 2(\boldsymbol{n}^T \boldsymbol{x}_2)(\boldsymbol{n}^T \boldsymbol{x}_1) = \|\boldsymbol{x}_1 - \boldsymbol{x}_2\|_2 + 2(\boldsymbol{n}^T \boldsymbol{x}_2)(\boldsymbol{n}^T \boldsymbol{x}_1) \le \|\boldsymbol{x}_1 - \boldsymbol{x}_2\|_2$. $\qquad\square$

We recall the notation introduced at the start of the section, $\Psi_\alpha(P\|Q) = \int_{\mathbb{R}^D} f\left(P(\boldsymbol{x}), -Q(\boldsymbol{x})\right) d\boldsymbol{x}$, where we assume an integrand $f$ that is convex in both arguments. The statement of Lemma O.1 will follow from the next lemma, which informally states that at any point $\boldsymbol{x}$ in the upper half-space, the mirror-reflection of means contained in $M_{\hat{\boldsymbol{n}}}^-$ causes an increase of the integrand $f\left(P_M(\boldsymbol{x}), -Q_N(\boldsymbol{x})\right)$ at $\boldsymbol{x}$ which dominates the corresponding decrease at the mirror image point $\boldsymbol{x}'$ in the lower half-space.

**Lemma O.5.** *Given a hyperplane normal to $\hat{\boldsymbol{n}} \in \mathcal{H}_p$ as defined in Lemma O.1 as well as any point $\boldsymbol{x} \in \mathbb{R}_{\hat{\boldsymbol{n}}}^+$ along with its mirror image $\boldsymbol{x}' = \boldsymbol{x} - 2\left(\boldsymbol{x}^T \hat{\boldsymbol{n}}\right)\hat{\boldsymbol{n}} \in \mathbb{R}_{\hat{\boldsymbol{n}}}^-$, the change of $f$ satisfies*

$$\left[f\left(P_{M_{\hat{\boldsymbol{n}}}'}(\boldsymbol{x}), -Q_{N_{-\hat{\boldsymbol{n}}}'}(\boldsymbol{x})\right) - f\left(P_M(\boldsymbol{x}), -Q_N(\boldsymbol{x})\right)\right]$$

$$+ \left[f\left(P_{M_{\hat{\boldsymbol{n}}}'}(\boldsymbol{x}'), -Q_{N_{-\hat{\boldsymbol{n}}}'}(\boldsymbol{x}')\right) - f\left(P_M(\boldsymbol{x}'), -Q_N(\boldsymbol{x}')\right)\right] \ge 0.$$

*Proof of Lemma O.5.* We introduce the following notation,

$$P_0(\boldsymbol{x}) := \sum_{\mu_k \in M \setminus \mathbb{R}_{\hat{\boldsymbol{n}}}^-} w_k \rho(\|\boldsymbol{x} - \mu_k\|_p), \qquad Q_0(\boldsymbol{x}) := \sum_{\nu_k \in N \setminus \mathbb{R}_{-\hat{\boldsymbol{n}}}^-} \omega_k \rho(\|\boldsymbol{x} - \nu_\kappa\|_p),$$

$$\delta P(\boldsymbol{x}) := \sum_{\mu_k \in M \cap \mathbb{R}_{\hat{\boldsymbol{n}}}^-} w_k \rho(\|\boldsymbol{x} - \mu_k\|_p), \qquad \delta Q(\boldsymbol{x}) := \sum_{\nu_k \in N \cap \mathbb{R}_{-\hat{\boldsymbol{n}}}^-} \omega_k \rho(\|\boldsymbol{x} - \nu_\kappa\|_p),$$

thus decomposing the densities into a part that stays fixed and a part that gets mirror-reflected. Using Lemma O.3 and the monotonicity requirement on $\rho$, we can directly verify that

$$P_0(\boldsymbol{x}) \ge P_0(\boldsymbol{x}'), \qquad \delta P(\boldsymbol{x}) \le \delta P(\boldsymbol{x}'), \tag{35}$$

$$-Q_0(\boldsymbol{x}) \ge -Q_0(\boldsymbol{x}'), \qquad -\delta Q(\boldsymbol{x}) \le -\delta Q(\boldsymbol{x}'). \tag{36}$$

Using invariance of distances under simultaneous mirroring of both vectors, we rewrite the expression of the lemma as

$$[f\left(P_0(\boldsymbol{x}) + \delta P(\boldsymbol{x}'), -Q_0(\boldsymbol{x}) - \delta Q(\boldsymbol{x}')\right) - f\left(P_0(\boldsymbol{x}) + \delta P(\boldsymbol{x}), -Q_0(\boldsymbol{x}) - \delta Q(\boldsymbol{x})\right)]$$

$$- [f\left(P_0(\boldsymbol{x}') + \delta P(\boldsymbol{x}'), -Q_0(\boldsymbol{x}') - \delta Q(\boldsymbol{x}')\right) - f\left(P_0(\boldsymbol{x}') + \delta P(\boldsymbol{x}), -Q_0(\boldsymbol{x}') - \delta Q(\boldsymbol{x})\right)]$$

The statement of the lemma follows from convexity of $f$ in both arguments by Jensen's inequality.

$$\square$$

We now proceed to prove Lemma O.1. The difference of divergences can be rewritten as

$$
\Psi_\alpha \left( P_{M'_{\hat{n}}} \| Q_{N'_{-\hat{n}}} \right) - \Psi_\alpha \left( P_M \| Q_N \right)
$$

$$
= \int_{\mathbb{R}^D} \left[ f \left( P_{M'_{\hat{n}}}(\boldsymbol{x}), -Q_{N'_{-\hat{n}}}(\boldsymbol{x}) \right) - f \left( P_M(\boldsymbol{x}), -Q_N(\boldsymbol{x}) \right) \right] d\boldsymbol{x}
$$

$$
= \int_{\mathbb{R}^-_{\hat{n}}} \left[ f \left( P_{M'_{\hat{n}}}(\boldsymbol{x}), -Q_{N'_{-\hat{n}}}(\boldsymbol{x}) \right) - f \left( P_M(\boldsymbol{x}), -Q_N(\boldsymbol{x}) \right) \right] d\boldsymbol{x}
$$

$$
+ \int_{\mathbb{R}^+_{\hat{n}}} \left[ f \left( P_{M'_{\hat{n}}}(\boldsymbol{x}), -Q_{N'_{-\hat{n}}}(\boldsymbol{x}) \right) - f \left( P_M(\boldsymbol{x}), -Q_N(\boldsymbol{x}) \right) \right] d\boldsymbol{x}
$$

$$
= \int_{\mathbb{R}^+_{\hat{n}}} \left[ f \left( P_{M'_{\hat{n}}}(\boldsymbol{x}'), -Q_{N'_{-\hat{n}}}(\boldsymbol{x}') \right) - f \left( P_M(\boldsymbol{x}'), -Q_N(\boldsymbol{x}') \right) \right]
$$

$$
+ \left[ f \left( P_{M'_{\hat{n}}}(\boldsymbol{x}), -Q_{N'_{-\hat{n}}}(\boldsymbol{x}) \right) - f \left( P_M(\boldsymbol{x}), -Q_N(\boldsymbol{x}) \right) \right] d\boldsymbol{x}
$$

$$
\geq \quad 0
$$

Apart from Lemma O.5, we used the fact that mirror reflection has a unit absolute Jacobian determinant, and that the hyperplane, a null set, does not contribute to the integral. $\qquad\square$

Lemma O.1 shows that we can construct a dominating mixture by mirroring all means onto opposite sides of a suitable hyperplane. This is exactly the condition under which the follow-up transform discussed in the next lemma yields equal or greater divergence. We slightly change perspective and consider the means of both mixtures as a joint vector. Given two sets of means $M = \{\mu_0, \ldots, \mu_K\}$, $N = \{\nu_0, \ldots, \nu_{\mathcal{K}}\}$ we define (in arbitrary ordering), $\boldsymbol{\mu} = (\mu_0, \ldots, \mu_K, \nu_0, \ldots, \nu_{\mathcal{K}}) \in \mathcal{F} \subset \mathbb{R}^{D \times (K+\mathcal{K})}$ where $\mathcal{F}$ denotes the region feasible under the distance constraints. With this implicit mapping $(M, N) \mapsto \boldsymbol{\mu}$ between sets and vectors of means, we can treat the divergence as a function $\Psi_\alpha \colon \boldsymbol{\mu} \mapsto \int_{\mathbb{R}^D} \tilde{f}(\boldsymbol{x}, \boldsymbol{\mu}) d\boldsymbol{x}$ with integrand $\tilde{f}(\boldsymbol{x}, \boldsymbol{\mu}) := f(P_M(\boldsymbol{x}), -Q(\boldsymbol{x}))$. Now we can state:

**Lemma O.6.** *For $p \in \{1, 2\}$, let $\hat{\boldsymbol{n}} \in \mathcal{H}_p$ be a normal vector, $M$, $N$ with corresponding vector $\boldsymbol{\mu}$ as above be two sets of means for which $M \cap \mathbb{R}^-_{\hat{\boldsymbol{n}}}$ and $N \cap \mathbb{R}^-_{-\hat{\boldsymbol{n}}}$ are empty (i.e., the hyperplane separates the two sets), and let $\Psi_\alpha$ denote either $\Lambda_\alpha$ or $H_\alpha$.*

*Then, the normal directional derivatives of the divergence with respect to the means $\mu_l$ and $\nu_\iota$ are nonnegative for all $l \in \{1, \ldots, K\}$, $\iota \in \{1, \ldots, \mathcal{K}\}$,*

$$
d^{(\mu_l)}_{\hat{\boldsymbol{n}}} \Psi_\alpha (\boldsymbol{\mu}) := \lim_{\varepsilon \to 0} \frac{1}{\varepsilon} \left[ \Psi_\alpha \left( (\mu_0, \ldots, \mu_l + \varepsilon \hat{\boldsymbol{n}}, \ldots, \nu_{\mathcal{K}}) \right) - \Psi_\alpha (\boldsymbol{\mu}) \right] \geq 0
$$

$$
d^{(\nu_\iota)}_{-\hat{\boldsymbol{n}}} \Psi_\alpha (\boldsymbol{\mu}) := \quad \lim_{\varepsilon \to 0} \frac{1}{\varepsilon} \left[ \Psi_\alpha \left( (\mu_0, \ldots, \nu_\iota - \varepsilon \hat{\boldsymbol{n}}, \ldots, \nu_{\mathcal{K}}) \right) - \Psi_\alpha (\boldsymbol{\mu}) \right] \geq 0.
$$

*In the case $\Psi_\alpha = H_\alpha$, the above expressions are instead defined in the sense of weak derivatives.*

The lemma is quite intuitive: once both sets of means are already separated by a hyperplane, pushing them apart in the normal direction, thereby increasing the margin to the hyperplane, never decreases the divergence. As we will only ever use the directional derivatives within line integrals, the weak sense in which the lemma holds for $\Psi_\alpha = H_\alpha$ is sufficient for our purposes.

*Proof of Lemma O.6.* We first check that for all $\boldsymbol{x} \in \mathbb{R}^D$ and $\boldsymbol{\mu} \in \mathcal{F}$, the derivative $\partial_{\boldsymbol{\mu}} \tilde{f}$ (possibly in the weak sense) exists and is dominated by an integrable function. In the case of Gaussian or Laplacian mixtures and $\Psi_\alpha = \Lambda_\alpha$, existence of the derivative is clear. For Gaussian mixtures, the integrability condition holds since $\partial_{\boldsymbol{\mu}} \tilde{f} = \mathcal{O}(\exp(-\|\boldsymbol{x}\|_2^2))$, for Laplacian mixtures $\partial_{\boldsymbol{\mu}} \tilde{f} = \mathcal{O}(\exp(-\|\boldsymbol{x}\|_1))$ since the privacy loss factor in the integrand eventually becomes constant. In the case $\Psi_\alpha = H_\alpha$, $\tilde{f}$ has a weak derivative $\partial_{\boldsymbol{\mu}} \tilde{f} = \mathbb{1}_{f \geq 0} \partial_{\boldsymbol{\mu}} (P - \alpha Q)$, which agrees with its derivative except where the latter is not defined (i.e., on the null set of means and $\boldsymbol{x}$ at the boundary of exact DP). Integrability

follows again from $\partial_{\boldsymbol{\mu}}\tilde{f} = \mathcal{O}(\exp(-\|\boldsymbol{x}\|_1))$. In the case of $\Psi_\alpha = \Lambda_\alpha$,

$$d_{\hat{\boldsymbol{n}}}^{(\mu_l)}\Psi_\alpha(\boldsymbol{\mu})$$

$$= \int_{\mathbb{R}^D} d_{\hat{\boldsymbol{n}}}^{(\mu_l)}\left[\left(\sum_{k=0}^K w_k\rho(\|\boldsymbol{x}-\mu_k\|_p)\right)^\alpha\right]\left(\sum_{\kappa=0}^{\mathcal{K}}\omega_\kappa\rho(\|\boldsymbol{x}-\nu_\kappa\|_p)\right)^{1-\alpha} d\boldsymbol{x}$$

$$= \alpha w_l \int_{\mathbb{R}^D}\left(\frac{P_M(\boldsymbol{x})}{Q_N(\boldsymbol{x})}\right)^{\alpha-1} d_{\hat{\boldsymbol{n}}}^{(\mu_l)}\rho(\|\boldsymbol{x}-\mu_l\|_p)d\boldsymbol{x}$$

$$= \alpha w_l \int_{\mathbb{R}^D}\left(\frac{P_M(\boldsymbol{x}+\mu_l)}{Q_N(\boldsymbol{x}+\mu_l)}\right)^{\alpha-1} d_{\hat{\boldsymbol{n}}}^{(\mu_l)}\rho(\|\boldsymbol{x}\|_p)\,d\boldsymbol{x}$$

Consider $\boldsymbol{x}\in\mathbb{R}^+$ and its mirror image $\boldsymbol{x}'\in\mathbb{R}^-$. By Lemma O.3 and the assumption that the hyperplane separates $M$ and $N$, $\frac{P_M(\boldsymbol{x}+\mu_l)}{Q_N(\boldsymbol{x}+\mu_l)} = \frac{\frac{\rho(\|\boldsymbol{x}+\mu_l\|_p)}{\rho(\|\boldsymbol{x}\|_p)}P_M(\boldsymbol{x})}{\frac{\rho(\|\boldsymbol{x}+\mu_l\|_p)}{\rho(\|\boldsymbol{x}\|_p)}Q_N(\boldsymbol{x})} = \frac{P_M(\boldsymbol{x})}{Q_N(\boldsymbol{x})} \geq \frac{P_M(\boldsymbol{x}')}{Q_N(\boldsymbol{x}')} = \frac{P_M(\boldsymbol{x}'+\mu_l)}{Q_N(\boldsymbol{x}'+\mu_l)}$. Moreover, also by Lemma O.3, $\rho(\|\boldsymbol{x}\|_p) = \rho(\|\boldsymbol{x}'\|_p)$, and therefore $d_{\hat{\boldsymbol{n}}}^{(\mu_l)}\rho(\|\boldsymbol{x}\|_p) = d_{-\hat{\boldsymbol{n}}}^{(\mu_l)}\rho(\|\boldsymbol{x}'\|_p) = -d_{\hat{\boldsymbol{n}}}^{(\mu_l)}\rho(\|\boldsymbol{x}'\|_p)$.

Similarly, we observe how in

$$d_{-\hat{\boldsymbol{n}}}^{(\nu_\iota)}\Psi_\alpha(\boldsymbol{\mu})$$

$$= \int_{\mathbb{R}^D}\left(\sum_{k=0}^K w_k\rho(\|\boldsymbol{x}-\mu_k\|_p)\right)^\alpha d_{-\hat{\boldsymbol{n}}}^{(\nu_\iota)}\left[\left(\sum_{\kappa=0}^{\mathcal{K}}\omega_\kappa\rho(\|\boldsymbol{x}-\nu_\kappa\|_p)\right)^{1-\alpha}\right] d\boldsymbol{x}$$

$$= (-1)(1-\alpha)\omega_\iota\int_{\mathbb{R}^D}\left(\frac{P_M(\boldsymbol{x})}{Q_N(\boldsymbol{x})}\right)^\alpha d_{\hat{\boldsymbol{n}}}^{(\nu_\iota)}\rho(\|\boldsymbol{x}-\nu_\iota\|_p)d\boldsymbol{x}$$

$$= (\alpha-1)\omega_\iota\int_{\mathbb{R}^D}\left(\frac{P_M(\boldsymbol{x}+\nu_\iota)}{Q_N(\boldsymbol{x}+\nu_\iota)}\right)^\alpha d_{\hat{\boldsymbol{n}}}^{(\nu_\iota)}\rho(\|\boldsymbol{x}\|_p)\,d\boldsymbol{x}$$

the negative signs of $-\hat{\boldsymbol{n}}$ and $(1-\alpha)$ cancel and apply analogous reasoning thereafter.

The statement for $\Psi_\alpha = \Lambda_\alpha$ now follows in analogy to the proof of Lemma O.1 via Lemma O.5.

For $\Psi_\alpha = H_\alpha$, we find

$$d_{\hat{\boldsymbol{n}}}^{(\mu_l)}\Psi_\alpha(\boldsymbol{\mu}) = w_l\int_{\mathbb{R}^D}\mathbb{1}_{\{P(\boldsymbol{x})-\alpha Q(\boldsymbol{x})\geq 0\}}d_{\hat{\boldsymbol{n}}}^{(\mu_l)}\rho(\|\boldsymbol{x}-\mu_l\|_p)d\boldsymbol{x}$$

$$= w_l\int_{\mathbb{R}^D}\mathbb{1}_{\left\{\frac{P_M(\boldsymbol{x}+\mu_l)}{Q_N(\boldsymbol{x}+\mu_l)}\geq\alpha\right\}}d_{\hat{\boldsymbol{n}}}^{(\mu_l)}\rho(\|\boldsymbol{x}\|_p)\,d\boldsymbol{x}$$

$$d_{\hat{\boldsymbol{n}}}^{(\nu_\iota)}\Psi_\alpha(\boldsymbol{\mu}) = -w_\iota\int_{\mathbb{R}^D}\mathbb{1}_{\{P_M(\boldsymbol{x})-\alpha Q_N(\boldsymbol{x})\geq 0\}}\left(-d_{\hat{\boldsymbol{n}}}^{(\mu_l)}\rho(\|\boldsymbol{x}-\nu_\iota\|_p)\right)d\boldsymbol{x}$$

$$= w_\iota\int_{\mathbb{R}^D}\mathbb{1}_{\left\{\frac{P_M(\boldsymbol{x}+\mu_l)}{Q_N(\boldsymbol{x}+\mu_l)}\geq\alpha\right\}}d_{\hat{\boldsymbol{n}}}^{(\mu_l)}\rho(\|\boldsymbol{x}\|_p)\,d\boldsymbol{x}$$

and can proceed by analogous reasoning from here on.

$\square$

We are now equipped to construct the sequence of transformations taking two arbitrary sets $M, N$ of Gaussian or Laplacian mixture means to the means $M^*, N^*$ of a dominating pair. We start with the Gaussian case.

### O.1.1 Dominating pair of Gaussian mixtures

**Theorem O.7.** *Let $\Psi_\alpha = H_\alpha$ or $\Psi_\alpha = \Lambda_\alpha$. Any pair of Gaussian mixtures with means satisfying distance constraints of the form*

$$\mu_0 = \nu_0 = 0 \qquad \|\mu_i-\mu_j\|_2 \leq |i-j| \quad \forall 0\leq i,j\leq K, \qquad \|\nu_\iota-\nu_\tau\|_2 \leq |i-j| \quad \forall 0\leq\iota,\tau\leq\mathcal{K},$$

*is dominated by a mixture with means*

$$\mu_k = k\cdot\hat{\boldsymbol{e}} \quad\forall k\in\{0,\ldots,K\}, \qquad \nu_\kappa = -\kappa\cdot\hat{\boldsymbol{e}} \quad\forall\kappa\in\{0,\ldots,\mathcal{K}\}, \qquad \text{with } \hat{\boldsymbol{e}}\in\mathbb{R}^D, \|\hat{\boldsymbol{e}}\|_2 = 1.$$

*Proof.* We start by proving the following lemma.

**Lemma O.8.** *Any pair of Gaussian mixtures with means at a fixed set of radii,*

$$\|\mu_k\|_2 = r_k, \; \|\nu_\kappa\|_2 = \rho_\kappa, \quad r_0 = \rho_0 = 0, \quad r_k, \rho_\kappa \in \mathbb{R}_0^+ \quad \forall k \in \{1, \dots K\} \, \forall \kappa \in \{1, \dots \mathcal{K}\},$$

*and satisfying the constraints of Theorem O.7 is dominated by a feasible pair with means at equal radii that are collinear on diametral half-lines through zero, i.e.,*

$$\mu_k = r_k \cdot \hat{e}, \quad \nu_\kappa = -\rho_\kappa \cdot \hat{e} \quad \forall k \in \{0, \dots, K\} \, \forall \kappa \in \{0, \dots, \mathcal{K}\} \quad \text{with } \hat{e} \in \mathbb{R}^D, \, \|\hat{e}\|_2 = 1.$$

*Proof of Lemma O.8.* Without loss of generality, we can pick as the collinearity direction from the lemma $\hat{e} = e_1$, the first canonical basis vector, since the divergence is rotation invariant. Pick any orthogonal direction, say, the second basis vector $e_2$. By Lemma O.1, we can mirror the two mixtures onto opposite sides of the hyperplanes with normal vectors $\hat{e}_1$ and $\hat{e}_2$ such that $\hat{e}_1^T \mu_k \geq 0 \geq \hat{e}_2^T \mu_k$ for all $k$ and $\hat{e}_1^T \nu_\kappa \geq 0 \geq \hat{e}_2^T \nu_\kappa$ for all $\kappa$.

Now, consider a hyperplane with normal vector $\hat{n}(\theta) = \hat{e}_1 \cos(\theta) + \hat{e}_2 \sin(\theta)$, $\theta \in [0, \pi/2]$. Intuitively, we let this hyperplane undergo a full rotation and "scoop up" all the means which are not in the hyperplane normal to $\hat{e}_2$, giving a new set of means which are. Formally, we find that by the above sign condition, there is an angle $\theta_k \in [0, \pi/2]$ for every $k$ such that $\text{sign}(\hat{n}(\theta)^T \hat{\mu}_k) = \text{sign}(\theta_k - \theta)$. Namely, using $\pi_{12}$ to denote the projection onto the subspace spanned by vectors $\hat{e}_1$ and $\hat{e}_2$, $\pi_{12}\mu_k = \|\pi_{12}\mu_k\|_2 (\sin(\theta_k)\hat{e}_1 - \cos(\theta_k)\hat{e}_2))$. Consider the paths $\gamma_k \colon [0, \pi/2] \to \mathbb{R}^D$, where

$$\gamma_k(\theta) = \mathbb{1}_{\{\theta_k - \theta \geq 0\}} \mu_k + \mathbb{1}_{\{\theta_k - \theta < 0\}} \left((\mathbb{1} - \pi_{12})\mu_k + \|\pi_{12}\mu_k\|_2 (\sin(\theta)\hat{e}_1 - \cos(\theta)\hat{e}_2)\right)$$

and a sign-flipped construction with angles $\theta_\kappa$ and curves $\zeta_\kappa$ for the means $\nu_\kappa$. For the derivatives along the curves, we find $\gamma_k'(\theta) = \|\pi_{12}\mu_k\|_2 \hat{n}(\theta)$ and $\zeta_\kappa'(\theta) = -\|\pi_{12}\zeta_\kappa\|_2 n(\theta)$. Also, by construction, $\hat{n}(\theta)^T \hat{\gamma}_k(\theta) \geq 0 \geq \hat{n}(\theta)^T \hat{\zeta}_\kappa(\theta)$. Hence, the prerequisites for Lemma O.6 are fulfilled at every $\theta \in [0, \pi/2]$. By evaluating the path integral along these curves and invoking Lemma O.6, we find that the divergence along the path $M, N \mapsto \tilde{M}, \tilde{N}$ cannot decrease:

$$\Phi_\alpha(P_{\tilde{M}}\|Q_{\tilde{N}}) - \Phi_\alpha(P_M\|Q_N)$$

$$= \int_0^{\pi/2} \sum_{l=1}^K d_{\hat{n}}^{(\mu_l)} \Psi_\alpha\left((\gamma_1, \dots, \gamma_K, \zeta_1, \dots, \zeta_\mathcal{K})(\theta)\right)$$

$$+ \sum_{\iota=1}^\mathcal{K} d_{-\hat{n}}^{(\nu_\iota)} \Psi_\alpha\left((\gamma_1, \dots, \gamma_K, \zeta_1, \dots, \zeta_\mathcal{K})(\theta)\right) d\theta \geq 0$$

Clearly, the paths also preserve the radius of each mean. The final set of means $\tilde{M} = \{\gamma_1(\pi/2), \dots, \gamma_K(\pi_2)\}, \tilde{N} = \{\zeta_1(\pi/2), \dots, \zeta_\mathcal{K}(\pi_2)\}$ is contained in the hyperplane normal to $\hat{e}_2$. We can now simply repeat this procedure with all basis vectors orthogonal to $\hat{e}_1$ to find the set from the lemma. Since it is collinear with the same radii as $M, N$, and the means $M, N$ are feasible, the new set is also feasible by the Cauchy-Schwarz inequality. $\square$

Since we can map any set of Gaussian mixture means onto one with the same radii that is collinear without decreasing the divergence, we can from now on study the problem restricted to a single radial dimension, using

$$\Psi_\alpha\Big|_{\mathbb{R}\hat{e}} \colon \prod_{k=0}^K \mathbb{R}_0^+ \times \prod_{\kappa=0}^\mathcal{K} \mathbb{R}_0^+ \longrightarrow \mathbb{R},$$

$$(r_0, \dots, r_K, \rho_0, \dots, \rho_\mathcal{K}) \longmapsto \Psi_\alpha\left(\sum_{\kappa=0}^\mathcal{K} \omega_\kappa \mathcal{N}(-\rho_\kappa \hat{e}, \sigma^2 I) \| \sum_{k=0}^K w_k \mathcal{N}(r_k \hat{e}, \sigma^2 I)\right)$$

We can now state the next lemma which implies Theorem O.7.

**Theorem O.9.** $(0, 1, \dots, K, 0, \dots, \mathcal{K})$ *is a global maximizer of* $\Psi_\alpha\Big|_{\mathbb{R}\hat{e}}$ *under the constraints* $r_0 = \rho_0 = 0, \, |r_i - r_j| \leq |i - j| \; \forall i, j \in \{0, \dots, K\}, \, |\rho_\iota - \rho_\tau| \leq |\iota - \tau| \; \forall \iota, \tau \in \{0, \dots, \mathcal{K}\}.$

*Proof of Theorem O.9.* To avoid clutter, we constrain ourselves to the case with only means $\mu_k$ and a single $\nu_0 = 0$, since the proof involving several $\nu_\kappa$ is completely analogous. Consider the line integral of the gradient

$$\nabla\Psi_\alpha\Big|_{\mathbb{R}_0^+\hat{e}} : \prod_{k=0}^{K}\mathbb{R}_0^+ \longrightarrow \mathbb{R}^K, \quad (r_0,\ldots,r_K) \longmapsto \left(\frac{\partial}{\partial r_0}\Psi_\alpha\Big|_{\mathbb{R}_0^+\hat{e}}, \ldots, \frac{\partial}{\partial r_K}\Psi_\alpha\Big|_{\mathbb{R}_0^+\hat{e}}\right)$$

between any feasible $(r_0,\ldots,r_K)$ and the point $(0,\ldots,K)$ along the connecting path

$$\gamma_{r_0,\ldots,r_K} : [0,1] \longrightarrow \prod_{k=0}^{K}\mathbb{R}_0^+,$$

$$t \longmapsto (r_0,\ldots,r_K) + t\boldsymbol{v} := (r_0,\ldots,r_K) + t\left[(0,\ldots,K) - (r_0,\ldots,r_K)\right].$$

First, observe that $r_k \leq k \,\forall k \in \{0,\ldots,K\}$ anywhere in the feasible region, since $r_k = \sum_{j=0}^{k-1}(r_{j+1} - r_j) \leq \sum_{j=0}^{k-1}|r_{j+1} - r_j| \leq \sum_{j=0}^{k-1}1 = k$. This means that $\boldsymbol{v}$ is componentwise-nonnegative and vanishes only if $(r_0,\ldots,r_K) = (0,\ldots,K)$. By Lemma O.6, the gradient $\nabla\Psi_\alpha\Big|_{\mathbb{R}_0^+\hat{e}}$ is componentwise-nonnegative anywhere along $\gamma_{r_0,\ldots,r_K}$. We may thus conclude that

$$\Psi_\alpha\Big|_{\mathbb{R}_0^+\hat{e}}(0,\ldots,K) - \Psi_\alpha\Big|_{\mathbb{R}_0^+\hat{e}}(r_0,\ldots,r_K)$$

$$= \int_{\gamma_{(r_0,\ldots,r_K)}} \nabla\Psi_\alpha(\boldsymbol{r})\,d\boldsymbol{r} = \int_0^1 \nabla\Psi_\alpha(\gamma(t))^T\boldsymbol{v}(t)dt \geq 0.$$

$\square$

This concludes the proof of Theorem O.7. $\square$

### O.1.2  Dominating pair of Laplacian mixtures

We also find an analogous result for Laplacian mixtures.

**Theorem O.10.** *Let $\Psi_\alpha = H_\alpha$ or $\Psi_\alpha = \Lambda_\alpha$. Any pair of Laplacian mixtures with means in $M = \{\mu_0, \ldots, \mu_K\}, N = \{\nu_0, \ldots, \nu_\mathcal{K}\} \subset \mathbb{R}^D$ satisfying pairwise $L^1$ distance constraints*

$$r_0 = \rho_0 = 0, \|\mu_i - \mu_j\|_1 \leq |i - j| \,\forall i,j \in \{0,\ldots,K\}, \|\nu_\iota - \nu_\tau\|_1 \leq |\iota - \tau| \,\forall \iota,\tau \in \{0,\ldots,\mathcal{K}\}.$$

*is dominated by a pair for $M^*, N^*$ of the form*

$$\mu_k^* = k\cdot\hat{e}, \quad \nu_\kappa^* = -\kappa\cdot\hat{e} \quad \forall k \in \{0,\ldots,K\}\,\forall\kappa \in \{0,\ldots,\mathcal{K}\} \quad \text{with } \hat{e} = \pm\hat{e}_i,$$

*where $1 \leq i \leq D$ such that $\hat{e}_i$ is the $i$-th canonical basis vector of $\mathbb{R}^D$.*

*Proof.*

**Lemma O.11.** *There is a dominating pair $M^*, N^*$ located in diametral quadrants of $\mathbb{R}^D$: given the component-wise sign function,*

$$\exists\boldsymbol{\sigma} \in \{+,-\}^D : \ sign(\mu_k) = \boldsymbol{\sigma} = -sign(\nu_\kappa) \quad \forall\mu_k \in M^*, \ \nu_\kappa \in N^*.$$

*Proof.* This is an immediate consequence of considering the canonical basis vectors as normal vectors in Lemma O.1. $\square$

**Lemma O.12.** *For $\mu_k \in M$, $\nu_\kappa \in N$, define the offset vectors $\delta\mu_k := \mu_k - \mu_{k-1}$, $\delta\nu_\kappa := \nu_\kappa - \nu_{\kappa-1}$ for $1 \leq k \leq K$, $1 \leq \kappa \leq \mathcal{K}$. A dominating pair $M^*, N^*$ has offset vectors located on diametral simplices of the 1-sphere: $\exists\boldsymbol{\sigma} \in \{+1,-1\}^D : \delta\mu_k^* \in S_{\boldsymbol{\sigma}}, \delta\nu_\kappa^* \in S_{-\boldsymbol{\sigma}}$, with $S_{\boldsymbol{\sigma}} := \{\boldsymbol{x} \in \mathbb{R}^D \mid \boldsymbol{\sigma}^T\boldsymbol{x} = 1\}, \forall k \in \{1,\ldots,K\}\,\forall\kappa \in \{1,\ldots,\mathcal{K}\}.$*

*Proof.* For any pair $M, N$ of means not satisfying this condition, we will construct a new pair that does and has at least equal divergence. By Lemma O.11, we can assume without generality loss that $\exists \boldsymbol{\sigma}^{(D)} \in \{+1, -1\}^D \colon \operatorname{sign}(\mu_k) = \boldsymbol{\sigma}^{(D)} = -\operatorname{sign}(\nu_\kappa) \quad \forall \mu_k \in M, \nu_\kappa \in N$, where we put a superscript $(D)$ above $\boldsymbol{\sigma}$ for later reasons. At a later stage of the proof, we will invoke that shifting the means along the basis vector directions with signs prescribed by $\boldsymbol{\sigma}^{(D)}$ are positive almost anywhere as a consequence of Lemma O.6.

It is useful to recast the optimization constraints in terms of the offset vectors: $\forall\, k \in \{1, \dots, K\}\, \forall\, \kappa \in \{1, \dots, \mathcal{K}\}$,

$$\mu_0 = \nu_0 = 0, \quad \|\delta\mu_k\|_1 = \max_{\boldsymbol{\sigma} \in \{-1,+1\}^D} \left(\boldsymbol{\sigma}^T \delta\mu_k\right) \leq 1, \quad \|\nu_\kappa\|_1 = \max_{\boldsymbol{\sigma} \in \{-1,+1\}^D} \left(\boldsymbol{\sigma}^T \delta\nu_\kappa\right) \leq 1,$$

where the constraints between non-ascending and non-neighboring index pairs are implied by the symmetry and triangle inequality of the norm. For $0 \leq k \leq K$, and $0 \leq \kappa \leq \mathcal{K}$, set $\mu_k^{(0)} := \mu_k$, $\nu_\kappa^{(0)} := \nu_\kappa$, and define (recursively for $1 \leq i \leq D$) the paths

$$\gamma_k^{(i)} \colon [0,1] \longrightarrow \mathbb{R}^D, \quad \gamma_k^{(i)}(t) = \mu_k^{(i-1)} + t \sum_{l=1}^{k} \left(1 - (\boldsymbol{\sigma}_l^{(i)})^T \delta\mu_l^{(i-1)}\right) \left((\boldsymbol{\sigma}^{(D)})^T \hat{\boldsymbol{e}}_i\right) \hat{\boldsymbol{e}}_i,$$

$$\delta\mu_l^{(i-1)} = \mu_l^{(i-1)} - \mu_{l-1}^{(i-1)}, \quad \boldsymbol{\sigma}_l^{(i)} = \operatorname*{arg\,max}_{\boldsymbol{\sigma} \in \{-1,+1\}^D,\, \boldsymbol{\sigma}^T \hat{\boldsymbol{e}}_i = (\boldsymbol{\sigma}^{(D)})^T \hat{\boldsymbol{e}}_i} \left(\boldsymbol{\sigma}^T \delta\mu_l^{(i-1)}\right), \quad \mu_k^{(i)} = \gamma_k^{(i)}(1),$$

$$\zeta_\kappa^{(i)} \colon [0,1] \longrightarrow \mathbb{R}^D, \quad \zeta_\kappa^{(i)}(t) = \nu_\kappa^{(i-1)} + t \sum_{\iota=1}^{\kappa} \left(1 - (\boldsymbol{\sigma}_\iota^{(i)})^T \delta\nu_\iota^{(i-1)}\right) \left(-(\boldsymbol{\sigma}^{(D)})^T \hat{\boldsymbol{e}}_i\right) \hat{\boldsymbol{e}}_i,$$

$$\delta\nu_\iota^{(i-1)} = \nu_\iota^{(i-1)} - \nu_{\iota-1}^{(i-1)}, \quad \boldsymbol{\sigma}_\iota^{(i)} = \operatorname*{arg\,max}_{\boldsymbol{\sigma} \in \{-1,+1\}^D,\, \boldsymbol{\sigma}^T \hat{\boldsymbol{e}}_i = -(\boldsymbol{\sigma}^{(D)})^T \hat{\boldsymbol{e}}_i} \left(\boldsymbol{\sigma}^T \delta\nu_\iota^{(i-1)}\right), \quad \nu_\kappa^{(i)} = \zeta_\kappa^{(i)}(1).$$

By construction,

$$\delta\mu_k^{(i)} = \delta\mu_k^{(i-1)} + \left(1 - (\boldsymbol{\sigma}_k^{(i)})^T \delta\mu_k^{(i-1)}\right) \left((\boldsymbol{\sigma}^{(D)})^T \hat{\boldsymbol{e}}_i\right) \hat{\boldsymbol{e}}_i,$$

$$\delta\nu_\kappa^{(i)} = \delta\nu_\kappa^{(i-1)} + \left(1 - (\boldsymbol{\sigma}_\kappa^{(i)})^T \delta\nu_\kappa^{(i-1)}\right) \left(-(\boldsymbol{\sigma}^{(D)})^T \hat{\boldsymbol{e}}_i\right) \hat{\boldsymbol{e}}_i.$$

We can easily check that $\delta\mu_k^{(i)} \in S_{\boldsymbol{\sigma}_k^{(i)}}, \delta\nu_\kappa^{(i)} \in S_{\boldsymbol{\sigma}_\kappa^{(i)}}$, for $1 \leq k \leq K$, $1 \leq \kappa \leq \mathcal{K}$.

We can moreover verify that, based on the definitions of $\sigma_k^{(i)}$ and $\sigma_\kappa^{(i)}$, the feasibility conditions $\|\delta\mu_k^{(i-1)}\|_1 \leq 1$ and $\|\delta\nu_\kappa^{(i-1)}\|_1 \leq 1$ imply $\|\delta\mu_k^{(i)}\|_1 = \max_{\boldsymbol{\sigma} \in \{-1,+1\}^D} \left(\boldsymbol{\sigma}^T \delta\mu_k^{(i)}\right) \leq 1$ and $\|\delta\nu_\kappa^{(i)}\|_1 = \max_{\boldsymbol{\sigma} \in \{-1,+1\}^D} \left(\boldsymbol{\sigma}^T \delta\nu_\kappa^{(i)}\right) \leq 1$. Since, furthermore, $\mu_0^{(i)} = \mu_0 = 0, \nu_0^{(i)} = \nu_0 = 0$ for $1 \leq i \leq D$, and $M, N$ are feasible by assumption, all pairs of $M^{(i)} := \{\mu_0^{(i)}, \dots, \mu_K^{(i)}\}, N^{(i)} := \{\nu_0^{(i)}, \dots, \nu_\mathcal{K}^{(i)}\}$ are also feasible by induction.

Finally, the above simplex and feasibility properties together imply that $\forall\, k \in \{1, \dots, K\}, \forall\, \kappa \in \{1, \dots \mathcal{K}\}$, $\operatorname{sign}\left(\hat{\boldsymbol{e}}_i^T \delta\mu_k^{(j)}\right) = (\boldsymbol{\sigma}^{(D)})^T \hat{\boldsymbol{e}}_i$ and $\operatorname{sign}\left(\hat{\boldsymbol{e}}_i^T \delta\nu_\kappa^{(j)}\right) = -(\boldsymbol{\sigma}^{(D)})^T \hat{\boldsymbol{e}}_i$ if $j \geq i$. But this means in particular that we can identify $\sigma_k^{(D)} = -\sigma_\kappa^{(D)} = \sigma^{(D)}$ for $1 \leq k \leq K$ and $1 \leq \kappa \leq \mathcal{K}$.

Putting everything together, we conclude that the pair $M^{(D)}, N^{(D)}$ obtained from $M, N$ by concatenation of all paths is feasible and has offset vectors on diametral simplices, $\delta\mu_k^{(D)} \in S_{\boldsymbol{\sigma}^{(D)}}$, $\delta\nu_\kappa^{(D)} \in S_{-\boldsymbol{\sigma}^{(D)}} \forall\, k \in \{1, \dots, K\}, \forall\, \kappa \in \{1, \dots \mathcal{K}\}$. What is left to show is that $M^{(D)}, N^{(D)}$ has at least the same divergence as $M, N$. To this end, define the product curves

$$(\tilde{M}^{(i)} \times \tilde{N}^{(i)}) \colon [0,1] \longrightarrow \mathbb{R}^{(K+\mathcal{K}) \times D}, \quad t \longmapsto (\gamma_0^{(i)}(t), \dots, \gamma_K^{(i)}(t), \zeta_0^{(i)}(t), \dots, \zeta_\mathcal{K}^{(i)}(t)), \quad 1 \leq i \leq D$$

connecting the endpoints $(M^{(i-1)}, N^{(i-1)})$ and $(M^{(i)}, N^{(i)})$, and the image sets $\mathcal{C}^{(i)} := (\tilde{M}^{(i)} \times \tilde{N}^{(i)})([0,1]) \subset \mathbb{R}^{(K+\mathcal{K}) \times D}$. The divergence difference is a sum of line integrals:

$$\Psi_\alpha(M^{(D)}||N^{(D)}) - \Psi_\alpha(M||N) = \sum_{i=1}^{D} \Psi_\alpha(M^{(i)}||N^{(i)}) - \Psi_\alpha(M^{(i-1)}||N^{(i-1)})$$

$$= \sum_{i=1}^{D} \int_{\mathcal{C}^{(i)}} \left( \sum_{k=0}^{K} d_{\hat{\boldsymbol{n}}_i}^{(\mu_k)} \Psi_\alpha(\tilde{M}^{(i)}||\tilde{N}^{(i)}) + \sum_{\kappa=0}^{\mathcal{K}} d_{-\hat{\boldsymbol{n}}_i}^{(\nu_k)} \Psi_\alpha(\tilde{M}^{(i)}||\tilde{N}^{(i)}) \right) ds$$

$$= \sum_{i=1}^{D} \int_{t=0}^{t=1} \sum_{k=0}^{K} d_{\hat{\boldsymbol{n}}_i}^{(\mu_k)} \Psi_\alpha(\tilde{M}^{(i)}(t)||\tilde{N}^{(i)}(t)) \sum_{l=1}^{k} \left( 1 - (\boldsymbol{\sigma}_l^{(i)})^T \delta\mu_l^{(i-1)} \right)$$

$$+ \sum_{\kappa=0}^{\mathcal{K}} d_{-\hat{\boldsymbol{n}}_i}^{(\nu_k)} \Psi_\alpha(\tilde{M}^{(i)}(t)||\tilde{N}^{(i)}(t)) \sum_{\iota=1}^{\kappa} \left( 1 - (\boldsymbol{\sigma}_\iota^{(i)})^T \delta\nu_\iota^{(i-1)} \right) dt \geq 0 :$$

At the start of the proof, we made the assumption without generality loss that $\exists \boldsymbol{\sigma}^{(D)} \in \{+1, -1\}^D : \operatorname{sign}(\mu_k) = \boldsymbol{\sigma}^{(D)} = -\operatorname{sign}(\nu_\kappa) \quad \forall \mu_k \in M, \nu_\kappa \in N$. Therefore, the directional derivatives $d_{\hat{\boldsymbol{n}}_i}^{(\mu_k)}, d_{-\hat{\boldsymbol{n}}_i}^{(\nu_k)}$ along the curve directions $\pm \hat{\boldsymbol{n}}_i = \pm \left( (\boldsymbol{\sigma}^{(D)})^T \hat{\boldsymbol{e}}_i \right) \hat{\boldsymbol{e}}_i$ are nonnegative. The determinant factors $\sum_{l=1}^{k} \left( 1 - (\boldsymbol{\sigma}_l^{(i)})^T \delta\mu_l^{(i-1)} \right)$ and $\left( 1 - (\boldsymbol{\sigma}_\iota^{(i)})^T \delta\nu_\iota^{(i-1)} \right)$ are nonnegative due to the feasibility of $M^{(i)}, N^{(i)} \ \forall i \in \{0, \ldots, D\}$.

$\square$

**Lemma O.13.** *There is a dominating pair $M^*$, $N^*$ with $i \in \{1, \ldots, D\}$ and $\sigma \in \{-, +\}$ such that $\delta\mu_k = \sigma \hat{\boldsymbol{e}}_i, \delta\nu_\kappa = -\sigma \hat{\boldsymbol{e}}_i \ \forall k \in \{1, \ldots, K\} \forall \kappa \in \{1, \ldots, \mathcal{K}\}$.*

*Proof.* By Lemma O.12, we may assume without generality loss that $\exists \boldsymbol{\sigma} \in \{+1, -1\}^D : \delta\mu_k \in S_{\boldsymbol{\sigma}}$, $\delta\nu_\kappa \in S_{-\boldsymbol{\sigma}}$, with $S_{\boldsymbol{\sigma}} := \{\boldsymbol{x} \in \mathbb{R}^D \mid \boldsymbol{\sigma}^T \boldsymbol{x} = 1\}, \forall k \in \{1, \ldots, K\} \forall \kappa \in \{1, \ldots, \mathcal{K}\}$.

The following curves will be useful. For $1 \leq i \neq j \leq D, l \in \{1, \ldots, D\}$ define

$$\gamma_k^{(ijl)} : [0,1] \longrightarrow \mathbb{R}^D, \quad \gamma_k^{(ijl)}(t) = \begin{cases} \mu_k \text{ for } k < l \\ \mu_k + t(\boldsymbol{\sigma}^T \hat{\boldsymbol{e}}_i)(\hat{\boldsymbol{e}}_i^T \delta\mu_l)((\boldsymbol{\sigma}^T \hat{\boldsymbol{e}}_j)\hat{\boldsymbol{e}}_j - (\boldsymbol{\sigma}^T \hat{\boldsymbol{e}}_i)\hat{\boldsymbol{e}}_i) \text{ for } k \geq l \end{cases}$$

$$\zeta_\kappa^{(ij\iota)} : [0,1] \longrightarrow \mathbb{R}^D, \quad \zeta_\kappa^{(ij\iota)}(t) = \begin{cases} \nu_\kappa \text{ for } \kappa < \iota \\ \nu_\kappa + t(\boldsymbol{\sigma}^T \hat{\boldsymbol{e}}_i)(\hat{\boldsymbol{e}}_i^T \delta\nu_\iota)((\boldsymbol{\sigma}^T \hat{\boldsymbol{e}}_j)\hat{\boldsymbol{e}}_j - (\boldsymbol{\sigma}^T \hat{\boldsymbol{e}}_i)\hat{\boldsymbol{e}}_i) \text{ for } \kappa \geq \iota. \end{cases}$$

By construction, $\gamma_k^{(ijl)}(0) = \mu_k, \zeta_\kappa^{(ij\iota)}(0) = \nu_\kappa, \gamma_0^{(ijl)}(1) = 0, \zeta_0^{(ij\iota)}(1) = 0, \gamma_k^{(ijl)}(1) - \gamma_{k-1}^{(ijl)}(1) = \delta\mu_k$ for $1 \leq k \neq l \leq K, \zeta_\kappa^{(ij\iota)}(1) - \zeta_{\kappa-1}^{(ij\iota)}(1) = \delta\nu_\kappa$ for $1 \leq \kappa \neq \iota \leq \mathcal{K}$,

$$\left( \gamma_l^{(ijl)}(1) - \gamma_{l-1}^{(ijl)}(1) \right)^T \hat{\boldsymbol{e}}_m = \begin{cases} 0 \text{ for } m = i \\ (\hat{\boldsymbol{e}}_i^T \boldsymbol{\sigma}) \delta\mu_l^T \hat{\boldsymbol{e}}_i + (\hat{\boldsymbol{e}}_j^T \boldsymbol{\sigma}) \delta\mu_l^T \hat{\boldsymbol{e}}_j \text{ for } m = j \\ \delta\mu_l^T \hat{\boldsymbol{e}}_m \text{ else,} \end{cases}$$

and

$$\left( \zeta_\iota^{(ij\iota)}(1) - \zeta_{\iota-1}^{(ij\iota)}(1) \right)^T \hat{\boldsymbol{e}}_m = \begin{cases} 0 \text{ for } m = i \\ (-\hat{\boldsymbol{e}}_i^T \boldsymbol{\sigma}) \delta\nu_\iota^T \hat{\boldsymbol{e}}_i + (-\hat{\boldsymbol{e}}_j^T \boldsymbol{\sigma}) \delta\nu_\iota^T \hat{\boldsymbol{e}}_j \text{ for } m = j \\ \delta\nu_\iota^T \hat{\boldsymbol{e}}_m \text{ else.} \end{cases}$$

In particular, this implies that the new pairs of sets $(M^{(l)} := \{\gamma_0^{(ijl)}(1), \ldots, \gamma_K^{(ijl)}(1)\}, N)$, $(M, N^{(\iota)} := \{\zeta_0^{(ij\iota)}(1), \ldots, \zeta_{\mathcal{K}}^{(ij\iota)}(1)\})$ are feasible.

Assume that $\exists l \in \{1, \ldots, K\}, m \in \{1, \ldots, K\}, i \in \{1, \ldots, D\}, j \in \{1, \ldots, D\}, i \neq j : (\delta\mu_l^T \hat{\boldsymbol{e}}_i) \neq 0 \wedge (\delta\mu_m^T \hat{\boldsymbol{e}}_j)\} \neq 0$. Note that we include the possibility of $l = m$. We can then assume without generality loss (by Lemma O.1) that the conditions of Lemma O.6 hold for either $\hat{\boldsymbol{n}}_+ = \frac{1}{\sqrt{2}}((\boldsymbol{\sigma}^T \hat{\boldsymbol{e}}_j)e_j - (\boldsymbol{\sigma}^T \hat{\boldsymbol{e}}_i)\hat{\boldsymbol{e}}_i)$ or $\hat{\boldsymbol{n}}_- = \frac{1}{\sqrt{2}}((\boldsymbol{\sigma}^T \hat{\boldsymbol{e}}_i)\hat{\boldsymbol{e}}_i - (\boldsymbol{\sigma}^T \hat{\boldsymbol{e}}_j)\hat{\boldsymbol{e}}_j)$. In the case of $\hat{\boldsymbol{n}}_+$, consider the product curve

$$(\tilde{M}^{(l)} \times N) : [0,1] \longrightarrow \mathbb{R}^{K+\mathcal{K} \times D}, \quad t \longmapsto (\gamma_0^{(ijl)}(t), \ldots, \gamma_K^{(ijl)}(t), \nu_0, \ldots, \nu_{\mathcal{K}})$$

and the image set $\mathcal{C}^{(l)} := (\tilde{M}^{(l)} \times N)([0,1])$. The divergence difference has the form

$$\Psi_\alpha(M^{(l)}||N) - \Psi_\alpha(M||N) = \int_{\mathcal{C}^{(l)}} \left( \sum_{k=0}^{K} d_{\hat{\boldsymbol{n}}_+}^{(\mu_k)} \Psi_\alpha(\tilde{M}^{(l)}||N) \right) ds$$

$$= \int_{t=0}^{t=1} \left( \sum_{k=l}^{K} d_{\hat{\boldsymbol{n}}_+}^{(\mu_k)} \Psi_\alpha(\tilde{M}^{(l)}(t)||N) \right) \sqrt{2}(\boldsymbol{\sigma}^T \hat{\boldsymbol{e}}_i)(\hat{\boldsymbol{e}}_i^T \delta\mu_l) dt \geq 0 :$$

By construction, the directional derivatives $d_{\hat{\boldsymbol{n}}_+}^{(\mu_k)} \Psi_\alpha(\tilde{M}^{(k)}(t)||N)$ are nonnegative. Likewise, the determinant term $\sqrt{2}(\boldsymbol{\sigma}^T \hat{\boldsymbol{e}}_i)(\hat{\boldsymbol{e}}_i^T \delta\mu_l)$ is nonnegative by the simplex condition together with feasibility of $(M, N)$.

If the separability condition is fulfilled for $\hat{\boldsymbol{n}}_-$ instead, we can repeat the argument with indices $i \leftrightarrow j$ swapped and index $m$ instead of $l$.

An analogous argument holds if $\exists \iota \in \{1, \ldots, \mathcal{K}\}, \pi \in \{1, \ldots, \mathcal{K}\}, i \in \{1, \ldots D\}, j \in \{1, \ldots, D\}, i \neq j: (\delta\nu_\iota^T \hat{\boldsymbol{e}}_i) \neq 0 \wedge (\delta\nu_\pi^T \hat{\boldsymbol{e}}_j)\} \neq 0.$  $\square$

Theorem O.10 immediately follows from Lemma O.13 by induction. $\square$

## O.2 Reduction of the divergence to a univariate integral

In the previous subsections, we showed that $\Psi_\alpha(N||M)$ is maximized by collinear and equidistant sets of means. This allows to evaluate $\Psi_\alpha(N||M)$ through a one-dimensional integral via marginalization. Recall (Theorem O.9). Note furthermore that in both cases, constraining the first element of either $N$ or $M$ into the origin incurs no generality loss due to the translation invariance of $\Psi_\alpha$.

Being able to constrain the problem to a single dimension, we now show

**Lemma O.14.** *In the collinear case, we may evaluate $\Psi_\alpha$ as the one-dimensional divergence $\Psi_\alpha^{1D}$ between two mixtures $\sum_{k=0}^{K} w_k \mathcal{N}(r_k, \sigma^2), \sum_{\kappa=0}^{\mathcal{K}} \omega_\kappa \mathcal{N}(-\rho_\kappa, \sigma^2)$ of univariate Gaussians centered around the means $r_k$ and $-\rho_\kappa$,*

$$\Psi_\alpha \left( \sum_{\kappa=0}^{\mathcal{K}} \omega_\kappa \mathcal{N}(-\rho_\kappa \hat{\boldsymbol{e}}, \sigma^2 \boldsymbol{I}) || \sum_{k=0}^{K} w_k \mathcal{N}(r_k \hat{\boldsymbol{e}}, \sigma^2 \boldsymbol{I}) \right) = \Psi_\alpha^{1D} \left( \sum_{\kappa=0}^{\mathcal{K}} \omega_\kappa \mathcal{N}(-\rho_\kappa, \sigma^2) || \sum_{k=0}^{K} w_k \mathcal{N}(r_k, \sigma^2) \right).$$

*Proof.* Without loss of generality, we may assume $\hat{\boldsymbol{e}}$ to be the indicator vector of the first component due to rotation invariance of $\Psi_\alpha$. Using bracketed superscripts $(0)$ to indicate the first vector component, and $(1:)$ to indicate the rest, we can use Fubini's theorem to write

$$\Psi_\alpha \left( \sum_{\kappa=0}^{\mathcal{K}} \omega_\kappa \mathcal{N}(-\rho_\kappa \hat{\boldsymbol{e}}, \sigma^2 \boldsymbol{I}) || \sum_{k=0}^{K} w_k \mathcal{N}(r_k \hat{\boldsymbol{e}}, \sigma^2 \boldsymbol{I}) \right)$$

$$= \int_{\mathbb{R}^D} \left[ \sum_{\kappa=0}^{\mathcal{K}} \omega_\kappa \mathcal{N}(\boldsymbol{x} \mid -\rho_\kappa \hat{\boldsymbol{e}}, \sigma^2 \boldsymbol{I}) \right]^\alpha \left[ \sum_{k=0}^{K} w_k \mathcal{N}(\boldsymbol{x} \mid r_k \hat{\boldsymbol{e}}, \sigma^2 \boldsymbol{I}) \right]^{(1-\alpha)} dx^{(0)} d\boldsymbol{x}^{(1:)}$$

$$= \int_{\mathbb{R}^{D-1}} \mathcal{N}(\boldsymbol{x}^{(1:)} \mid 0, \sigma^2 \boldsymbol{I}) d\boldsymbol{x}^{(1:)} \int_{\mathbb{R}} \left[ \sum_{\kappa=0}^{\mathcal{K}} \omega_k \mathcal{N}(x^{(0)} \mid -\rho_\kappa, \sigma^2) \right]^\alpha \left[ \sum_{k=0}^{K} w_k \mathcal{N}(x^{(0)} \mid r_k, \sigma^2) \right]^{(1-\alpha)} dx^{(0)}$$

$$= 1 \cdot \Psi_\alpha^{1D} \left( \sum_{\kappa=0}^{\mathcal{K}} \omega_\kappa \mathcal{N}(-\rho_\kappa, \sigma^2) || \sum_{k=0}^{K} w_k \mathcal{N}(r_k, \sigma^2) \right).$$

$\square$

The proof is analogous for the Laplacian mechanism.

## O.3   Randomized Response Proofs

The following two results (first for the Rényi divergence, second for the hockey stick divergence) show that our (group) privacy guarantees for randomized response are given by the maximum of two terms that can be evaluated in constant time.

**Theorem O.15.**

$$\underset{\boldsymbol{\tau}\in[0,1]^{(K_++1)+(K_-+1)}}{\arg\max} \sum_{z=0}^{1}\left[\left(\sum_{i=1}^{K_++1} w_i^{(1)}(1-|z-\tau_i^{(1)}|)\right)^{\alpha}\cdot\left(\sum_{j=1}^{K_-+1} w_j^{(2)}(1-|z-\tau_j^{(2)}|)\right)^{1-\alpha}\right]$$

*subject to*

$$\tau_i^{(1)} \in \{\theta, 1-\theta\}, \forall i,$$
$$\tau_j^{(1)} \in \{\theta, 1-\theta\}, \forall j,$$
$$\tau_1^{(1)} = \tau_1^{(2)},$$

*with*

$$w_i^{(1)} = \text{Binomial}(i-1\mid K_-, r),$$
$$w_j^{(2)} = \text{Binomial}(j-1\mid K_+, r),$$

*is given by*

$$\arg\max\left\{\Psi\left(m_{\tilde{\boldsymbol{\tau}}^{(1)}}\,\|\,m_{\tilde{\boldsymbol{\tau}}^{(2)}}\right), \Psi\left(m_{\hat{\boldsymbol{\tau}}^{(1)}}\,\|\,m_{\hat{\boldsymbol{\tau}}^{(2)}}\right)\right\},$$

*where*

$$\tilde{\tau}_1^{(1)} = \tilde{\tau}_1^{(2)} =: \tau \in \{\theta, 1-\theta\}\ (symmetric),$$
$$\tilde{\tau}_{2,\ldots,(K_++1)}^{(1)} = 1-\tau,$$
$$\tilde{\tau}_{2,\ldots,(K_-+1)}^{(2)} = \tau$$

*and*

$$\hat{\tau}_1^{(1)} = \hat{\tau}_1^{(2)} =: \tau \in \{\theta, 1-\theta\}\ (symmetric),$$
$$\hat{\tau}_{2,\ldots,(K_++1)}^{(1)} = \tau,$$
$$\hat{\tau}_{2,\ldots,(K_-+1)}^{(2)} = 1-\tau$$

*Proof.*
The strategy is to show that

(a) Any mixture pair $(m_{\boldsymbol{\tau}^{(1)}}, m_{\boldsymbol{\tau}^{(2)}})$ where $m_{\boldsymbol{\tau}^{(1)}}$ has greater mass in $(1-\tau)$, i.e., $\displaystyle\sum_{i:\tau_i^{(1)}=1-\tau} w_i^{(1)} \geq \sum_{j:\tau_j^{(2)}=1-\tau} w_j^{(2)}$, is dominated by $(m_{\tilde{\boldsymbol{\tau}}^{(1)}}, m_{\tilde{\boldsymbol{\tau}}^{(2)}})$

(b) Any mixture pair $(m_{\boldsymbol{\tau}^{(1)}}, m_{\boldsymbol{\tau}^{(2)}})$ where $m_{\boldsymbol{\tau}^{(2)}}$ has greater mass in $(1-\tau)$, i.e., $\displaystyle\sum_{i:\tau_i^{(1)}=1-\tau} w_i^{(1)} \leq \sum_{j:\tau_j^{(2)}=1-\tau} w_j^{(2)}$, is dominated by $(m_{\hat{\boldsymbol{\tau}}^{(1)}}, m_{\hat{\boldsymbol{\tau}}^{(2)}})$

Notation:

$$\sum_{z=0}^{1}\left[\left(\sum_{i=1}^{K_++1} w_i^{(1)}(1-|z-\tau_i^{(1)}|)\right)^{\alpha}\cdot\left(\sum_{j=1}^{K_-+1} w_j^{(2)}(1-|z-\tau_j^{(2)}|)\right)^{1-\alpha}\right] = \sum_{z=0}^{1} f(x(z))g(y(z)),$$

where $f$ strictly convex, increasing, and $g$ strictly convex, decreasing.

Trivial case: $\theta = \frac{1}{2}$, thus assume $\theta \neq \frac{1}{2}$ for the rest of the proof.

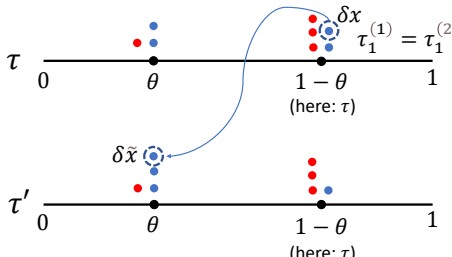

Figure 26: Visualization of $\delta x$ and $\delta x'$. Color legend: The color "blue" refers to terms with superscript $(1)$, while "red" refers to those with $(2)$.

Proof of (a):

Given: Pair of mixtures $(m_{\boldsymbol{\tau}^{(1)}}, m_{\boldsymbol{\tau}^{(2)}})$ such that $w^{(1)}_{1-\tau} := \displaystyle\sum_{i:\tau^{(1)}_i=1-\tau} w^{(1)}_i \geq \sum_{j:\tau^{(2)}_j=1-\tau} w^{(2)}_j =:$

$w^{(2)}_{1-\tau}$.

Further define $w^{(1)}_\tau = 1 - w^{(1)}_{1-\tau}, w^{(2)}_\tau = 1 - w^{(2)}_{1-\tau}$.

Define a *new* mixture via

$$\tau'^{(1)}_1 = \tau'^{(2)}_1 = \tau,$$
$$\tau'^{(1)}_{2\ldots,K_++1} = 1 - \tau,$$
$$\tau'^{(2)}_i = \tau^{(2)}_i, \forall i.$$

Assume w.l.o.g. that $\tau_0 = \max\{\theta, 1 - \theta\}$, else swap roles of $0 \leftrightarrow 1$ in the argument.

Decompose $x$: $x(z) = x_0(z) + \delta x(z)$, where

$x_0(z) = w^{(1)}_{1-\tau}(1 - |z - (1 - \tau)|) + w^{(1)}_1(1 - |z - \tau|)$, i.e., "part that stays fixed"; see Fig. 26,

$\delta x(z) = (w^{(1)}_\tau - w^{(1)}_1)(1 - |z - \tau|)$, i.e., "part that relocates"; see Fig. 26.

Similarly,

$$x'(z) = x'_0(z) + \delta x'(z),$$

and by construction,

$$x'_0(z) = x_0(z),$$
$$\delta x'(z) = \delta x(1 - z) \; (\star)$$

We can assume w.l.o.g. $\tau_0 = \max\{\theta, 1 - \theta\} \implies \delta x(1) > \delta x(0)$ $(\star\star)$ (else, swap $1 \leftrightarrow 0$)

Then, $\theta \neq \frac{1}{2}$, hence we can assume $w^{(1)}_\tau > w^{(1)}_1$ (else, $m' \equiv m$ in the first place).

$\Psi_\alpha(m_{\boldsymbol{\tau}'^{(1)}}, m_{\boldsymbol{\tau}'^{(2)}}) - \Psi_\alpha(m_{\boldsymbol{\tau}^{(1)}}, m_{\boldsymbol{\tau}^{(2)}})$

$\overset{(\star)}{=} [f(x_0(0) + \delta x(1)) - f(x_0(0) + \delta x(0))] g(y(0))) - [f(x_0(1) + \delta x(1)) - f(x_0(1) + \delta x(0))] g(y(1))$

Then, using that $f$ is strictly convex, we have

$\overset{(\star\star)}{>} f'(x_0(0) + \delta x(0))g(y(0))(\delta x(1) - \delta x(0)) - f'(x_0(1) + \delta x(0))g(y(1))(\delta x(1) - \delta x(0))$

Where $f'(x) = x^{\alpha-1}$, and thus $f'(x)g(y) = (\alpha - 1)\left(\frac{x}{y}\right)^{\alpha-1}$

$= (\alpha - 1)\left[\left(\dfrac{x_0(0) + \delta x(0)}{y(0)}\right)^{\alpha-1} - \left(\dfrac{x_0(1) + \delta x(0)}{y(1)}\right)^{\alpha-1}\right](\delta x(1) - \delta x(0))$

$= (\alpha - 1)\left[\left(\dfrac{(1 - w^{(1)}_{1-\tau})(1 - \tau) + w^{(1)}_{1-\tau}\tau}{(1 - w^{(2)}_{1-\tau})(1 - \tau) + w^{(2)}_{1-\tau}\tau}\right)^{\alpha-1} - \left(\dfrac{w^{(1)}_1\tau + (1 - w^{(1)}_1)(1 - \tau)}{(1 - w^{(2)}_{1-\tau})\tau + w^{(2)}_{1-\tau}(1 - \tau)}\right)^{\alpha-1}\right](\delta x(1) - \delta x(0))$

$> 0.$

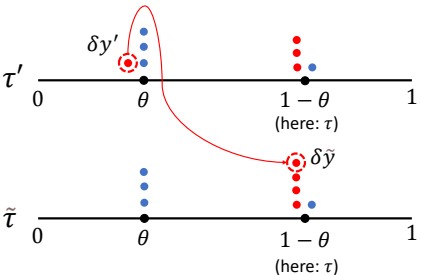

Figure 27: Visualization of $\delta y$ and $\delta y'$. The color "blue" refers to terms with superscript $(1)$, while "red" refers to those with $(2)$.

The last inequality follows from the following facts. Since we assume $w_\tau^{(1)} > w_1^{(1)}$ (else, $m \equiv m'$ in the first place), we can bound the second term by:

$$\left( \frac{w_1^{(1)}\tau + (1-w_1^{(1)})(1-\tau)}{(1-w_{1-\tau}^{(2)})\tau + w_{1-\tau}^{(2)}(1-\tau)} \right)^{\alpha-1} < \left( \frac{(1-w_{1-\tau}^{(1)})\tau + w_{1-\tau}(1-\tau)}{(1-w_{1-\tau}^{(2)})\tau + w_{1-\tau}^{(1)}(1-\tau)} \le 1 \right)^{\alpha-1} \quad (i)$$

Moreover, we have that the first term is bounded by

$$\left( \frac{(1-w_{1-\tau}^{(1)})(1-\tau) + w_{1-\tau}^{(1)}\tau}{(1-w_{1-\tau}^{(2)})(1-\tau) + w_{1-\tau}^{(2)}\tau} \right)^{\alpha-1} \ge 1 \quad (ii)$$

(strict $>$ if $w_{1-\tau}^{(1)} > w_{1-\tau}^{(2)}$). The conclusion follows from $(i)$, $(ii)$, and since WLOG $\tau > (1-\tau)$.

We have shown so far:
$$\Psi_\alpha(m_{\boldsymbol{\tau}'(1)}, m_{\boldsymbol{\tau}'(2)}) > \Psi_\alpha(m_{\boldsymbol{\tau}(1)}, m_{\boldsymbol{\tau}(2)}).$$

To complete the proof of $(a)$, we show now that $\Psi_\alpha(m_{\tilde{\boldsymbol{\tau}}(1)}, m_{\tilde{\boldsymbol{\tau}}(2)}) > \Psi_\alpha(m_{\boldsymbol{\tau}'(1)}, m_{\boldsymbol{\tau}'(2)})$.
***Idea***: Apply symmetric argument to means $\tau_{(2)}$.

Since $1 = w_\tau^{(1)} + w_{1-\tau}^{(1)} = w_\tau^{(2)} + w_{1-\tau}^{(2)}$, we have

$$w_{\tau'}^{(2)} = w_\tau^{(2)} \ge w_\tau^{(1)} > w_{\tau'}^{(1)},$$

where $w_{\tau'}^{(1)}, w_{\tau'}^{(2)}$ are defined like $w_\tau^{(1)}, w_\tau^{(2)}$ but for new means $\tau'$.

Decompose $y'$: $y'(z) = y_0'(z) + \delta y'(z)$, where

$$y_0'(z) = w_\tau^{(2)}(1 - |z - \tau|), \text{i.e., "part that stays fixed"; see Fig. 27.,}$$
$$\delta y'(z) = w_{1-\tau}^{(2)}(1 - |z - (1-\tau)|), \text{i.e., "part that relocates"; see Fig. 27.}$$

We can assume w.l.o.g. that $\tau = \max\{\theta, 1-\theta\}$, so $\delta y'(0) > \delta y'(1)$.

$\Psi_\alpha(m_{\tilde{\boldsymbol{\tau}}(1)}, m_{\tilde{\boldsymbol{\tau}}(2)}) - \Psi_\alpha(m_{\boldsymbol{\tau}'(1)}, m_{\boldsymbol{\tau}'(2)})$
$= f(x'(0)) [g(y_0'(0) + \delta y'(1)) - g(y_0'(0) + \delta y'(0))] + f(x'(1)) [g(y_0'(1) + \delta y'(0)) - g(y_0'(1) + \delta y'(1))]$

And using the strict convexity of $f$ we have

$> [f(x'(1))g'(y_0'(1) + \delta y'(1)) - f(x'(0))g'(y_0'(0) + \delta y'(0))] (\delta y'(0) - \delta y'(1))$

$= (\alpha - 1) \left[ \left( \frac{x'(0)}{y_0'(0) + \delta y'(1)} \right)^\alpha - \left( \frac{x'(1)}{y_0'(1) + \delta y'(1)} \right)^\alpha \right] (\delta y'(0) - \delta y'(1))$

$= (\alpha - 1) \left[ \left( \frac{w_\tau^{(1)}(1-\tau) + (1-w_\tau^{(1)})\tau}{1-\tau} \right)^\alpha - \left( \frac{w_\tau^{(1)}\tau + (1-w_\tau^{(1)})(1-\tau)}{w_\tau^{(2)}\tau + (1-w_\tau^{(2)})(1-\tau)} \right)^\alpha \right] \cdot (\delta y'(0) - \delta y'(1))$

$> 0.$

Proof of $(b)$ is fully analogous.

This proves the theorem statement, since condition (a) or (b) is fulfilled for any mixture pair, and hence any mixture pair is dominated by the argmax of the two mixture pairs stated in the theorem.

$\square$

**Theorem O.16.**

$$\underset{\boldsymbol{\tau}\in[0,1]^{(K_++1)+(K_-+1)}}{\arg\max}\sum_{z=0}^{1}\left[\left(\sum_{i=1}^{K_++1}w_i^{(1)}(1-|z-\tau_i^{(1)}|)\right)-e^{\varepsilon}\left(\sum_{j=1}^{K_-+1}w_j^{(2)}(1-|z-\tau_j^{(2)}|)\right)\right]$$

*subject to*

$$\tau_i^{(1)}\in\{\theta,1-\theta\},\forall i,$$
$$\tau_j^{(1)}\in\{\theta,1-\theta\},\forall j,$$
$$\tau_1^{(1)}=\tau_1^{(2)},$$

*with*

$$w_i^{(1)}=\text{Binomial}(i-1\mid K_-,r),$$
$$w_j^{(2)}=\text{Binomial}(j-1\mid K_+,r),$$

*is given by*

$$\arg\max\left\{\Psi\left(m_{\tilde{\boldsymbol{\tau}}^{(1)}}\|m_{\tilde{\boldsymbol{\tau}}^{(2)}}\right),\Psi\left(m_{\hat{\boldsymbol{\tau}}^{(1)}}\|m_{\hat{\boldsymbol{\tau}}^{(2)}}\right)\right\},$$

*where*

$$\tilde{\tau}_1^{(1)}=\tilde{\tau}_1^{(2)}=:\tau\in\{\theta,1-\theta\}\ (symmetric),$$
$$\tilde{\tau}_{2,\dots,(K_++1)}^{(1)}=1-\tau,$$
$$\tilde{\tau}_{2,\dots,(K_-+1)}^{(2)}=\tau$$

*and*

$$\hat{\tau}_1^{(1)}=\hat{\tau}_1^{(2)}=:\tau\in\{\theta,1-\theta\}\ (symmetric),$$
$$\hat{\tau}_{2,\dots,(K_++1)}^{(1)}=\tau,$$
$$\hat{\tau}_{2,\dots,(K_-+1)}^{(2)}=1-\tau$$

*Proof sketch.*
The strategy is to show that

(a) Any mixture pair $(m_{\boldsymbol{\tau}^{(1)}},m_{\boldsymbol{\tau}^{(2)}})$ where $m_{\boldsymbol{\tau}^{(1)}}$ has greater mass in $(1-\tau)$, i.e.,
$$\sum_{i:\tau_i^{(1)}=1-\tau}w_i^{(1)}\geq\sum_{j:\tau_j^{(2)}=1-\tau}w_j^{(2)},\text{ is dominated by }(m_{\tilde{\boldsymbol{\tau}}^{(1)}},m_{\tilde{\boldsymbol{\tau}}^{(2)}})$$

(b) Any mixture pair $(m_{\boldsymbol{\tau}^{(1)}},m_{\boldsymbol{\tau}^{(2)}})$ where $m_{\boldsymbol{\tau}^{(2)}}$ has greater mass in $(1-\tau)$, i.e.,
$$\sum_{i:\tau_i^{(1)}=1-\tau}w_i^{(1)}\leq\sum_{j:\tau_j^{(2)}=1-\tau}w_j^{(2)},\text{ is dominated by }(m_{\hat{\boldsymbol{\tau}}^{(1)}},m_{\hat{\boldsymbol{\tau}}^{(2)}})$$

Notation:

$$\sum_{z=0}^{1}\left[\left(\sum_{i=1}^{K_++1}w_i^{(1)}(1-|z-\tau_i^{(1)}|)\right)-e^{\varepsilon}\left(\sum_{j=1}^{K_-+1}w_j^{(2)}(1-|z-\tau_j^{(2)}|)\right)\right]=\sum_{z=0}^{1}f(x(z)-e^{\varepsilon}y(z)),$$

where $f=[\,\cdot\,]_+$ is convex and differentiable anywhere except at 0.

Trivial case: $\theta=\frac{1}{2}$, thus assume $\theta\neq\frac{1}{2}$ for the rest of the proof.

Proof of (a):

Given: Pair of mixtures $(m_{\boldsymbol{\tau}^{(1)}}, m_{\boldsymbol{\tau}^{(2)}})$ such that $w^{(1)}_{1-\tau} := \sum\limits_{i:\tau^{(1)}_i=1-\tau} w^{(1)}_i \geq \sum\limits_{j:\tau^{(2)}_j=1-\tau} w^{(2)}_j =: w^{(2)}_{1-\tau}$.

Further define $w^{(1)}_{\tau} = 1 - w^{(1)}_{1-\tau}, w^{(2)}_{\tau} = 1 - w^{(2)}_{1-\tau}$.

Define a *new* mixture via

$$\tau'^{(1)}_1 = \tau'^{(2)}_1 = \tau,$$
$$\tau'^{(1)}_{2\ldots,K_+ +1} = 1 - \tau,$$
$$\tau'^{(2)}_i = \tau^{(2)}_i, \forall i.$$

Assume w.l.o.g. that $\tau_0 = \max\{\theta, 1 - \theta\}$, else swap roles of $0 \leftrightarrow 1$ in the argument.

Decompose $x$: $x(z) = x_0(z) + \delta x(z)$, where

$x_0(z) = w^{(1)}_{1-\tau}(1 - |z - (1 - \tau)|) + w^{(1)}_1(1 - |z - \tau|)$, i.e., "part that stays fixed"; see Fig. 26,

$\delta x(z) = (w^{(1)}_\tau - w^{(1)}_1)(1 - |z - \tau|)$, i.e., "part that relocates"; see Fig. 26.

Similarly,

$$x'(z) = x'_0(z) + \delta x'(z),$$

and by construction,

$$x'_0(z) = x_0(z),$$
$$\delta x'(z) = \delta x(1 - z) \;(\star)$$

We can assume w.l.o.g. $\tau_0 = \max\{\theta, 1 - \theta\} \implies \delta x(1) > \delta x(0)$ $(\star\star)$ (else, swap $1 \leftrightarrow 0$)

Then, $\theta \neq \frac{1}{2}$, hence we can assume $w^{(1)}_\tau > w^{(1)}_1$ (else, $m' \equiv m$ in the first place).

$\Psi_\alpha(m_{\boldsymbol{\tau}'^{(1)}}, m_{\boldsymbol{\tau}'^{(2)}}) - \Psi_\alpha(m_{\boldsymbol{\tau}^{(1)}}, m_{\boldsymbol{\tau}^{(2)}})$

$\overset{(\star)}{=} [f(x_0(0) + \delta x(1) - e^\varepsilon y(0)) - f(x_0(0) + \delta x(0) - e^\varepsilon y(0))]$
$- [f(x_0(1) + \delta x(1) - e^\varepsilon y(1)) - f(x_0(1) + \delta x(0) - e^\varepsilon y(1))] g(y(1))$

Assume for now that $x_0(0) + \delta x(0) - e^\varepsilon y(0) \neq 0 \neq x_0(1) + \delta x(0) - e^\varepsilon y(1)$.

$\overset{(\star\star)}{\geq} (f'(x_0(0) + \delta x(0) - e^\varepsilon y(0)) - f'(x_0(1) + \delta x(0) - e^\varepsilon y(1)))(\delta x(1) - \delta x(0))$

$= f'\left[\left((1 - w^{(1)}_{1-\tau})(1 - \tau) + w^{(1)}_{1-\tau}\tau\right) - e^\varepsilon\left((1 - w^{(2)}_{1-\tau})(1 - \tau) + w^{(2)}_{1-\tau}\tau\right)\right](\delta x(1) - \delta x(0))$

$- f'\left[\left(w^{(1)}_1\tau + (1 - w^{(1)}_1)(1 - \tau)\right) - e^\varepsilon\left((1 - w^{(2)}_{1-\tau})\tau + w^{(2)}_{1-\tau}(1 - \tau)\right)\right](\delta x(1) - \delta x(0))$

$\geq f'\left[\left((1 - w^{(1)}_{1-\tau})(1 - \tau) + w^{(1)}_{1-\tau}\tau\right) - e^\varepsilon\left((1 - w^{(2)}_{1-\tau})(1 - \tau) + w^{(2)}_{1-\tau}\tau\right)\right](\delta x(1) - \delta x(0))$

$- f'\left[\left((1 - w^{(1)}_{1-\tau})\tau + w_{1-\tau}(1 - \tau)\right) - e^\varepsilon\left((1 - w^{(2)}_{1-\tau})\tau + w^{(2)}_{1-\tau}(1 - \tau)\right)\right](\delta x(1) - \delta x(0))$

$\geq 0.$

The second-to-last inequality follows from $w^{(1)}_\tau > w^{(1)}_1$ (else, $m \equiv m'$ in the first place). The last inequality follows from $w^{(1)}_{1-\tau} > w^{(2)}_{1-\tau}$ and the prior assumption that, WLOG, $\tau > 1 - \tau$.
If $x_0(0) + \delta x(0) - e^\varepsilon y(0) > x_0(1) + \delta x(0) - e^\varepsilon y(1)$ and $x_0(1) + \delta x(0) - e^\varepsilon y(1) = 0$, then we can arrive at the same conclusion by using any subderivative of $f$ in $[0, 1]$ at $x_0(1) + \delta x(0) - e^\varepsilon y(1)$ instead.
We have shown so far:

$$\Psi_\alpha(m_{\boldsymbol{\tau}'^{(1)}}, m_{\boldsymbol{\tau}'^{(2)}}) \geq \Psi_\alpha(m_{\boldsymbol{\tau}^{(1)}}, m_{\boldsymbol{\tau}^{(2)}}).$$

Continue in analogy to Theorem O.15.

$\square$

# P  Towards epoch-level subsampling analysis

Standard composition theorems assume that the outputs of composed mechanisms are conditionally independent, meaning each output only depends on the previous outputs and the dataset (see, e.g., the proof of Proposition 1 in [7] and the proof of Theorem 27 in [10]). For this reason, one typically considers subsampling schemes like Poisson subsampling or subsampling without replacement, which yield independent batches in each iteration.

However, there may be scenarios where batches are not independent, such as when creating batches by permuting a dataset and splitting it into equal sized chunks. Such correlated subsampling schemes are appealing because one can, for instance, limit privacy leakage by ensuring that any element appears in only one iteration per epoch when training a model. A downside of these corellated subsampling schemes is that they may brake the conditional independence assumption of standard compositions theorems.

There already exist approaches to analyzing compositions of correlated mechanisms, such as probabilistically upper-bounding a mechanism's privacy leakage with uncorellated mechanisms [75, 24]. Furthermore, permutation-based batching has already been discussed in the context of convergence guarantees for noisy SGD [38]. Nevertheless, this problem setting provides a great opportunity to demonstrate that the utility of our conditional optimal transport framework extends beyond the subsampling schemes that are typically discussed in amplification-by-subsampling literature.

## P.1  Problem setting

For this example, we consider a *non-adaptive* composition of two mechanisms, where two batches are created by permuting a dataset and splitting it in half. We assume that our dataset space is composed of size $2N$ subsets of some finite, discrete set $\mathbb{A}$, i.e., $\mathbb{X} \subseteq \{x \in \mathcal{P}(\mathbb{A}) \mid |x| = 2N\}$, and equipped with substitution relation $\simeq_\Delta$. We further define a per-iteration batch space $\mathbb{Y}$ that is composed of size $N$ subsets, i.e., $\mathbb{Y} = \{y \subseteq x \mid x \in \mathbb{X} \wedge |y| = N\}$, which is also equipped with substitution relation $\simeq_\Delta$. Finally, we assume a base mechanism $B : \mathbb{Y} \to \mathbb{Z}$ with conditional density $b_y : \mathbb{Z} \to \mathbb{R}_+$ that maps from this batch space to some output space.

To apply our framework to epoch-level subsampling distributions, we define the composed batch space $\hat{\mathbb{Y}}$, which consists of equal-sized partitions of datasets in $\mathbb{X}$, i.e.,

$$\hat{\mathbb{Y}} = \{(y_1, y_2) \in \mathbb{Y}_{\text{orig}}^2 \mid \exists x \in \mathbb{X} : y_1 \,\dot\cup\, y_2 = x\}. \tag{37}$$

On this composed batch space, we can now define our epoch-level subsampling scheme:

**Definition P.1.** The permute-and-partition subsampling scheme is the subsampling scheme $S : \mathbb{X} \to \hat{\mathbb{Y}}$ with

$$s_x((y_1, y_2)) = \begin{cases} \binom{2N}{N}^{-1} & \text{if } y_1 \cup y_2 = x \\ 0 & \text{otherwise .} \end{cases}$$

This definition follows from the fact that permuting-and-partitioning is equivalent to sampling a batch of size $N$ without replacement for the first batch and using the remaining $N$ elements for the second batch. We further consider the *non-adaptively* composed base mechanism $\hat{B} : \hat{\mathbb{Y}} \to \mathbb{Z}^2$ with $\hat{B}_{(y_1, y_2)} = (B_{y_1}, B_{y_2})$ and resultant joint density $\hat{b}_{(y_1, y_2)} = b_{y_1} \cdot b_{y_2}$.

## P.2  Optimal transport without conditioning

As a baseline, we use Theorem 3.3, i.e., optimal transport without conditioning, to provide a Rényi-DP bound for the composed, subsampled mechanism. This guarantee captures the intuition that by permuting-and-partitioning, we only leak an elements' private information once. Again, note we prove this result for *non-adaptive* composition.

**Theorem P.2.** *Assume a dataset space $\mathbb{X} \subseteq \{x \in \mathcal{P}(\mathbb{A}) \mid |x| = 2N\}$, batch space $\mathbb{Y} = \{y \subseteq x \mid x \in \mathbb{X} \wedge |y| = N\}$, and a base mechanism $B : \mathbb{Y} \to \mathbb{Z}$ that is $(\alpha, \varepsilon)$-Rényi-DP w.r.t. single-element substitution relation $\simeq_{\Delta, \mathbb{Y}}$. Let $S : \mathbb{X} \to \hat{\mathbb{Y}}$ be the permute-and-partition subsampling scheme defined in Definition P.1. Let $\hat{B}$ be the composed base mechanism with $\hat{B}_{(y_1, y_2)} = (B_{y_1}, B_{y_2})$. Then the subsampled, composed mechanism $\hat{M} = \hat{B} \circ S$ is also $(\alpha, \varepsilon)$-DP w.r.t. single-element substitution relation $\simeq_{\Delta, \mathbb{X}}$.*

*Proof.* Consider an arbitrary pair of datasets $x \simeq_{\Delta,\mathbb{X}} x'$. By definition of the substitution relation, there must be some $a \in x$ and some $a' \in x'$ such that $x' = x \setminus \{a\} \cup \{a'\}$.

We can now define the following simple coupling between $S_x$ and $S_{x'}$:

$$\gamma((y_1^{(1)}, y_2^{(1)}), (y_1^{(2)}, y_2^{(2)})) = \begin{cases} \binom{2N}{N}^{-1} & \text{if } a \in y_1^{(1)} \wedge y_2^{(2)} = y_1^{(1)} \setminus \{a\} \cup \{a'\} \wedge y_2^{(1)} = y_2^{(2)} \\ \binom{2N}{N}^{-1} & \text{if } a \in y_2^{(1)} \wedge y_2^{(2)} = y_2^{(1)} \setminus \{a\} \cup \{a'\} \wedge y_1^{(1)} = y_1^{(2)} \\ 0 & \text{otherwise.} \end{cases}$$

This coupling generates a pair of batch sequences by first applying permute-and-partition to original dataset $x$ and then replacing $a$ with $a'$ in the one batch it appears in. Because we only couple pairs that are identical in one batch and differ by a substitution in the other batch, we have per Theorem 3.3

$$\Psi(\hat{m}_x || \hat{m}_{x'}) \leq \max_{(y_1^{(1)}, y_2^{(1)}),(y_1^{(2)}, y_2^{(2)}) \in \hat{\mathbb{Y}}^2} \Psi(b_{y_1^{(1)}} \cdot b_{y_2^{(1)}} || b_{y_1^{(2)}} \cdot b_{y_2^{(2)}})$$

subject to $(y_1^{(1)} \simeq_{\Delta,\mathbb{Y}} y_1^{(2)}) \wedge (y_2^{(1)} = y_2^{(2)}) \vee (y_2^{(1)} \simeq_{\Delta,\mathbb{Y}} y_2^{(2)}) \wedge (y_1^{(1)} = y_1^{(2)})$. Either way, one of the factors is cancelled out when computing the likelihood ratio in the definition of Rényi-divergence (see Eq. (7)). We thus have

$$\Psi(\hat{m}_x || \hat{m}_{x'}) \leq \max_{y_1^{(1)}, y_1^{(2)} \in \mathbb{Y}^2} \Psi(b_{y_1^{(1)}} || b_{y_1^{(2)}})$$

subject to $y_1^{(1)} \simeq_{\Delta,\mathbb{Y}} y_1^{(2)}$. The result then follows from the definition of Rényi-DP (see Definition 2.2). $\square$

## P.3 Optimal transport with conditioning

Next, we use Theorem 3.4, i.e., optimal transport between conditional subsampling distributions, to derive a stronger guarantee. Again, note we prove this result for *non-adaptive* composition. Note that this result is similar in spirit to amplification by shuffling [76, 75], but considers central differential privacy of mechanisms operating on batches, instead of locally differentially private mechanisms that operate on individual elements. Again, note that we do not claim to be the first to consider the permute-and-partition scheme for composed mechanisms (see, e.g., [38]). It is just a nice showcase for the versatility of our framework in analyzing different subsampling schemes.

For the following result and proof we will use the following indexing convention: $y_{i,k}^{(1)}$ is a batch associated with the first mixture $\hat{m}_x$, event $A_i$, and the $k$th iteration. Similarly, $y_{j,k}^{(2)}$ is a batch associated with the second mixture $\hat{m}_{x'}$, event $E_j$, and the $k$th iteration.

**Theorem P.3.** *Assume a dataset space $\mathbb{X} \subseteq \{x \in \mathcal{P}(\mathbb{A}) \mid |x| = 2N\}$, and corresponding batch space $\mathbb{Y} = \{y \subseteq x \mid x \in \mathbb{X} \wedge |y| = N\}$, equipped with single-element substitution relation $\simeq_{\Delta,\mathbb{Y}}$. Let $S : \mathbb{X} \to \hat{\mathbb{Y}}$ be the permute-and-partition subsampling scheme defined in Definition P.1, and let $B : \mathbb{Y} \to \mathbb{Z}$ be some base mechanism. Let $\hat{B}$ be the composed base mechanism with $\hat{B}_{(y_1, y_2)} = (B_{y_1}, B_{y_2})$, and $\hat{M} = \hat{B} \circ S$ be the subsampled, coposed mechanism. Then, for all $x \simeq_{\Delta,\mathbb{X}} x'$,*

$$\Psi_\alpha(\hat{m}_x || \hat{m}_{x'})$$

$$\leq \max_{y_{1,1}^{(1)}, y_{1,2}^{(1)}, y_{1,1}^{(2)}} \Psi_\alpha \left( b_{y_{1,1}^{(1)}} \cdot b_{y_{1,2}^{(1)}} \cdot \frac{1}{2} + b_{y_{1,2}^{(1)}} \cdot b_{y_{1,1}^{(1)}} \cdot \frac{1}{2} || b_{y_{1,1}^{(2)}} \cdot b_{y_{1,2}^{(1)}} \cdot \frac{1}{2} + b_{y_{1,2}^{(1)}} \cdot b_{y_{1,1}^{(2)}} \cdot \frac{1}{2} \right)$$

*subject to $y_{1,1}^{(1)} \simeq_{\Delta,\mathbb{Y}} y_{1,1}^{(2)}$.*

*Proof.* By definition of the substitution relation, there must be some $a \in x$ and some $a' \in x'$ such that $x' = x \setminus \{a\} \cup \{a'\}$.

For the original subsampling scheme $S_x$ we let $A_1$ be the event that $a$ appears in the first batch and $A_2$ be the event that $a$ appears in the second batch. For the modified subsampling scheme $S_{x'}$ we let $E_1$ be the event that $a'$ appears in the first batch and $E_2$ be the event that $a'$ appears in the second batch. We have $P_{S_x}(A_1) = P_{S_x}(A_2) = P_{S_x'}(E_1) = P_{S_x'}(E_2) = \frac{1}{2}$.

One can sample from the distributions conditioned on $A_1$ or $E_1$ by first sampling uniformly at random $N$ elements from the $2 \cdot N - 1$ elements that are *not* $a$ or $a'$, and then using the remaining elements as the first batch. Similarly, one can sample from the distributions conditioned on $A_2$ or $E_2$ by first sampling uniformly at random $N$ elements from the $2 \cdot N - 1$ elements that are *not* $a$ or $a'$, and then using the remaining elements as the second batch.

We thus have

$$s_x(y_1^{(1)} \mid A_1) = \begin{cases} \binom{2N-1}{N}^{-1} & \text{if } y_1^{(1)} \in A_1 \\ 0 & \text{otherwise} \end{cases} \qquad s_x(y_2^{(1)} \mid A_2) = \begin{cases} \binom{2N-1}{N}^{-1} & \text{if } y_1^{(1)} \in A_2 \\ 0 & \text{otherwise} \end{cases}$$

and

$$s_{x'}(y_1^{(2)} \mid E_1) = \begin{cases} \binom{2N-1}{N}^{-1} & \text{if } y_1^{(1)} \in E_1 \\ 0 & \text{otherwise} \end{cases} \qquad s_{x'}(y_2^{(2)} \mid E_2) = \begin{cases} \binom{2N-1}{N}^{-1} & \text{if } y_1^{(1)} \in E_2 \\ 0 & \text{otherwise} \end{cases}$$

Next, we can define a coupling via

$$\gamma(y_1^{(1)}, y_2^{(1)}, y_1^{(2)}, y_2^{(2)}) = \begin{cases} s_x(y_1^{(1)} \mid A_1) & \text{if Condition P.4 is fulfilled} \\ 0 & \text{otherwise} \end{cases}$$

**Condition P.4.** Batch tuples $y_1^{(1)} \in A_1, y_2^{(1)} \in A_2, y_1^{(2)} \in E_1, y_2^{(2)} \in E_2$ fulfill Condition P.4 when $y_{1,1}^{(1)} = y_{2,2}^{(1)}$, and $y_{1,2}^{(1)} = y_{2,1}^{(1)}$, and $y_{1,2}^{(1)} = y_{1,2}^{(2)}$, and $y_{2,1}^{(1)} = y_{2,1}^{(2)}$, and $y_{1,1}^{(2)} = y_{1,1}^{(2)} \setminus \{a\} \cup \{a'\}$, and $y_{2,2}^{(2)} = y_{2,2}^{(2)} \setminus \{a\} \cup \{a'\}$.

In short: We sample uniformly at random a batch tuple $(y_{1,1}^{(1)}, y_{1,2}^{(1)})$ from $A_1$. We then generate a batch tuple from $A_2$ by permuting the $A_1$ tuple. We then generate a batch tuple from $E_2$ by replacing $a$ with $a'$ in the $A_2$ tuple. We finally generate a batch tuple from $E_1$ by permuting the $E_2$ tuple.

Because for each possible value of a batch tuple there is only one combination of the other three batch tuples such that $\gamma(\boldsymbol{y}) > 0$, and in this case $\gamma(\boldsymbol{y}) = \binom{2N-1}{N}^{-1}$, this is a valid coupling.

The result then follows from Theorem 3.4 and the constraints imposed by Condition P.4. □

Note that this result could be generalized to $K$ iterations: We can upper-bound the subsampled, composed mechanism's divergence by adversarially choosing a $K$-fold partition of $x$, constructing a mixture with $K$ (not $K!$) components, with each component corresponding to $a$ appearing in a specific batch, and finally constructing a modified mixture by replacing every occurence of $a$ with $a'$.

## P.4   Experimental Evaluation

Finally, we can compare our epoch-level amplification guarantees obtained via optimal transport with and without conditioning. As an additional baseline, we use 2-fold composition of subsampling without replacement with batch size 2. Note that, with the baseline, a modified element may appear in 0, 1 or 2 of the iterations. For the sake of this simple example, we consider output space $\mathbb{Z} = \mathbb{R}$, and let base mechanism $b$ be the Gaussian mechanism $f + \mathcal{N}(0, \sigma)$ with $f : \mathbb{Y} \to \{0, 1\}$ and univariate standard deviation $\sigma \in \{0.5, 1.0, 2.0, 5.0\}$. By this construction we do not need to reason about $f$'s sensitivity in determining the worst-case mixture components.

We make the following observation: For small $\sigma$, both epoch-level guarantees are almost identical and outperform the without replacement guarantee. With increasing $\sigma$, subsampling without replacement outperforms Theorem P.2 for some ranges of $\alpha$. Theorem P.2, which is derived via conditional optimal transport, however yields smaller $\varepsilon$.

The main takeaway of this example should be that (a) there is a benefit to using conditional optimal transport to bound privacy in terms of mixtures and (b) conditional optimal transport can be used to reason about schemes other than the typically considered Poisson subsampling, subsampling without replacement, and subsampling with replacement.

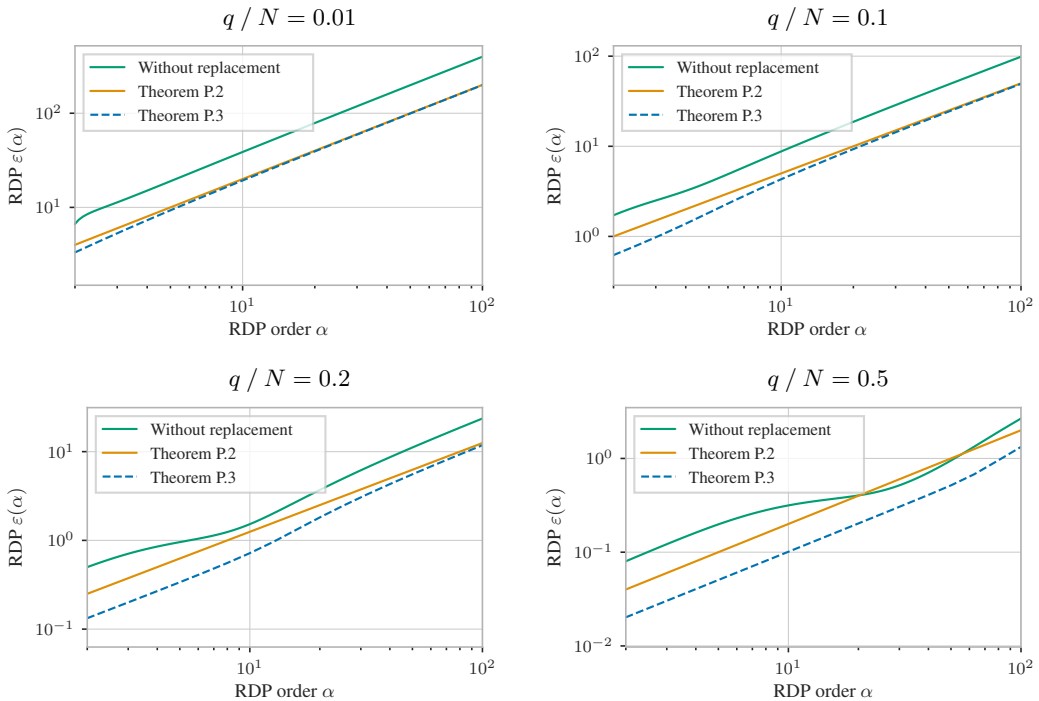

Figure 28: Comparison of our epoch-level permute-and-partition guarantees with (Theorem P.3) and without (Theorem P.2) conditioning, as well as subsampling without replacement, for 2-fold non-adaptive composition. The base mechanism is a Gaussian mechanism with $f : \mathbb{Y} \to \{0, 1\}$ and varying standard deviations $\sigma$. With increasing $\sigma$, Theorem P.3 and subsampling without replacement become more similar, while Theorem P.3 consistently yields stronger guarantees.

## Q   Broader impact

Our work contributes towards provably protecting users, and specifically groups of users, from the negative societal impact of privacy leakage. However, differential privacy may have negatively impact other aspects of trustworthy machine learning, such as fairness or robustness. Also, training for differentially private machine learning is often performed with $1 < \varepsilon \leq 10$. This is useful for relative comparisons between models, but does not impose any meaningful constraints on absolute privacy leakage (the probability of any event may increase by a factor of up to $\approx 22000$) and may thus be used to falsely advertise privacy.

