# OpenReview forum: "Unified Mechanism-Specific Amplification by Subsampling and Group Privacy Amplification"
_NeurIPS.cc/2024/Conference — NeurIPS 2024 poster_

### Official Review · Reviewer_t4hg · 2024-07-09

**Soundness:** 3
**Presentation:** 2
**Contribution:** 3
**Rating:** 5
**Confidence:** 3

**Summary:**

The study proposes a framework utilizing optimal transport theory to derive mechanism-specific and mechanism-agnostic guarantees for subsampling mechanisms in differential privacy. It focuses on leveraging conditional optimal transport to establish tight bounds on privacy parameters such as α and β, crucial for various privacy scenarios including Rényi differential privacy and moments accounting. The experiments evaluate privacy guarantees under different mechanisms including Gaussian, Laplace, and randomized response, with specific settings for noise parameters and group sizes.

**Strengths:**

- Defining partitions into events based on dataset manipulation scenarios.
        - Creating couplings between conditional distributions to ensure compatibility.
        - Bounding mixture divergences using worst-case mixture components.
        - Employing advanced joint convexity techniques to tighten divergence bounds.
        - Applying distance-compatible couplings to enhance robustness of privacy guarantees.

**Weaknesses:**

This manuscript is too long, which are 100 pages.
Lack of theoretical analysis in the main text

**Questions:**

Will the approach be adopted to LDP?

---

> ### Author Rebuttal · Authors · 2024-08-07
>
> Thank you for your review!
>
> ### Concerning appendix length
> Please let us explain how we ended up with the current page count. Our work proposes a framework for conducting subsamling analysis.
> We claim that this framework lets us prove any known subsampling guarantee (as well as novel guarantees, such as group privacy amplification) using a standard recipe.
> **We did not see any way of verifying this claim, other than going back through approximately 8 years of privacy amplification literature and re-deriving their respective results**, as well performing similar derivations for other subsamling settings.
>
> The **appendix sections that we would actually consider to be supporting  the core of our paper** are
> * D - General setting and Definitions (2 pages)
> * E - Proof of optimal transport bounds (2 pages)
> * F - Distance-compatible couplings (3 pages)
> * M.1-M.3 - Group privacy amplification & Tightness (5 pages)
> * O.1 - Worst-case mixture components for Gaussian & Laplacian mechanisms (8 pages)
>
> When focusing on these appendices, that are not merely meant to verify the above claim, this appendix length is much more in line with other machine learning conference papers on differential privacy, e.g., [1].
>
> ### Theoretical analysis in main text
> Thank you for this suggestion.
> Based on your comment, **we will move Theorem M.4 and its proof, which show the **tightness of our group privacy bounds**, to the main text.** We think that this specific theoretic result could be particularly interesting for future readers, as it helps illustrate the difference between mechanism-agnostic and mechanism-specific bounds.
>
> Please let us know if there is any other specific analysis that you would like us to move to the main text instead. Note that we cannot upload a revised manuscript during the rebuttal period, but will make these changes as soon as its possible.
>
> ### Adoption to local differential privacy
> This is an interesting question that we can answer in the positive.
> Local differential privacy is concerned with privately sharing the information of a single user. As such, there is usually not a collection of multiple records that we could subsample from.
>
> However, one can consider a scenario where this  **single user has a collection of $N$ attributes, $K$ of which are sensitive** (e.g. age, income, gender). If they randomly sample a subset of attributes (or functions thereof) to privately share via a noisy mechanism, then our results are directly applicable. They will either share $0$, $1$, or up to $K$ of the sensitive attributes with a certain probability.  We can thus use exactly the same group privacy amplification bound derived in our work.
>
> ---
>
> Again, thank you for helping us in further improving our manuscript.
> Please let us know if you have any further questions or comments during the author-reviewer discussion period!
>
> ---
>
> [1] Zhu et al. "Optimal Accounting of Differential Privacy via Characteristic Function". AISTATS'22

---

### Official Review · Reviewer_cbWU · 2024-07-10

**Soundness:** 3
**Presentation:** 2
**Contribution:** 3
**Rating:** 6
**Confidence:** 4

**Summary:**

The paper proposes a principled approach to analyzing group-privacy amplification through sub-sampling by generalizing the coupling arguments of Balle et al., 2018. This generalization extends the analysis from $1$-neighboring datasets to $K$-neighboring datasets. The core idea is to define a coupling between partitions of batches rather than the batches themselves, ensuring that the divergence of the output distribution, conditional on the partitioning, is bounded individually by the worst-case divergence. By carefully selecting the partitioning and the optimal coupling specific to the sub-sampling mechanism, the authors demonstrate that improved Rényi and Approximate DP bounds for group privacy can be achieved, surpassing the bounds obtained by applying privacy amplification by subsampling and group-privacy theorems separately. The paper shows that the combined analysis of grouping and subsampling can be tailored to a specific base mechanism, leading to mechanism-specific privacy amplification bounds that are superior to mechanism-agnostic bounds. Although the DP bounds presented are not in closed form, they can be numerically approximated to arbitrary precision using standard techniques from the literature. The paper also provides a comprehensive evaluation of how privacy is amplified for different group sizes under Gaussian, randomized response, and Laplace base mechanisms.

**Strengths:**

- The paper presents an in-depth analysis of privacy amplification due to subsampling under groupings. This topic has not been extensively studied before, making it an interesting paper to read.

- The paper demonstrates that there is a significant interplay between subsampling and grouping concerning privacy, which is not captured in standard analyses. This finding is an important contribution to the privacy community.

- The paper argues that by considering the underlying base mechanism, the amplification bounds due to subsampling can be refined beyond the tight mechanism-agnostic bounds established by Balle et al., 2018. While the results indicate this improvement for group privacy, the paper also claims it applies more generally. This could be a significant finding (although I am not entirely convinced of its validity).

**Weaknesses:**

- The authors argue that tailoring privacy amplification through subsampling to a specific mechanism can yield better bounds than the tight amplification bounds established by Balle et al., 2018. However, it appears that the paper only demonstrates privacy amplification under subsampling for Approximate-DP with groups of size 2 or larger. Does this imply that for the standard setting with no grouping (i.e., group size of 1), there is no additional subsampling amplification achievable by tailoring the analysis to the mechanism?

 - I remain unconvinced that tailoring the analysis to a specific mechanism can result in tighter amplification-due-to-subsampling bounds for a group size of 1. Could the authors provide an example of two mechanisms, $\mathcal{M}_1$ and $\mathcal{M}_2$, that both satisfy $(\epsilon, \delta)$-DP tightly, but for some subsampling mechanism $S$, mechanism $\mathcal{M}_1 \circ S$ is $(\epsilon', \delta')$-DP while $\mathcal{M}_2 \circ S$ cannot achieve $(\epsilon', \delta')$-DP?

- The figures in the paper (Figure 4 to Figure 10, Figure 13 to 15, Figure 17 to 20) do not include a baseline comparison with the separate application of tight amplification by subsampling and group privacy as described in lines 58-64. Without this comparison, it is unclear the gap between the bounds in this paper and the standard approach.

- Definition 2.2 of Rényi divergence is misleading; a concave function $f(\bullet) = \frac{1}{\alpha - 1} \log(\bullet)$ needs to be applied to your expression for $R_\alpha(m_x\Vert m_{x'})$ to obtain the Rényi divergence. The paper's expression for $R_\alpha(m_x\Vert m_{x'})$ is typically referred to as the $\alpha$th moment of the privacy loss random variable. Although the moment $R_\alpha$ may be jointly convex (a property heavily used in the paper), the Rényi divergence is not jointly convex due to the mapping $f$ being concave. Additionally, showing that $R_\alpha$ shrinks by a factor of $p$ on subsampling does not mean that the Rényi divergence shrinks by a factor of $p$ due to the concavity of $f$.

- The paper does not provide analytical bounds to help understand the asymptotics of the subsampling amplification results proved. Without the asymptotics, it's hard to verify claims of tightness.

### Minor Concerns:

- The privacy amplification results are not presented in a closed-form solution, making them difficult to operationalize.

- The subsampling bounds for Rényi DP have not been compared with other existing works, such as [1,2].

- The placement of figures does not match the respective appendices where they are discussed, making referencing the figures very challenging.

References:

[1] Steinke, Thomas. "Composition of differential privacy & privacy amplification by subsampling." arXiv preprint arXiv:2210.00597 (2022).

[2] Zhu, Yuqing, and Yu-Xiang Wang. "Poisson subsampled Rényi differential privacy." International Conference on Machine Learning. PMLR, 2019.

**Questions:**

I don't have any specific questions. I encourage authors to explain or address the problems I described in the weakness section. Based on how convincing the answers are, I'm open to adjusting my score.

**Limitations:**

Authors discuss some of the limitations of the paper. Following are some additional limitations that need addressing.

- The ansatz is unwieldy for finding general use.
- Closed-form asymptotics for the presented bounds are missing.
- It's not clear if mechanism-specific subsampling bounds can be strictly better than mechanism-agnostic subsampling bounds for group size=1.

---

> ### Author Rebuttal · Authors · 2024-08-07
>
> Thank you for your review!
>
> Please excuse our brevity, there is a character limit and you posed a lot of interesting questions. We cannot upload a revision during rebuttals, but will include your suggestions as soon as its possible.
>
> ### Specific vs agnostic for group size 1
> For group size 1, and **the special case of Poisson/WOR subsampling**, the agnostic bounds of Balle et al. [1] are tight in a mechanism-specific sense. This is a known result: Zhu et al. [2] used them to derive their tight dominating pairs (see proof of Prop. 30 in [2]).
>
> We will add the following statement to ll.256-257:
> "This bound is tight in a mechanism-agnostic sense [...]. For the **special case of group size $K=1$ and Poisson/WOR subsampling** this translates to mechanism-specific tightness (see [...])".
> Upon discussion with reviewer 1 (eK4M), we will also include group size $1$ experiments.
>
> ### Counterexample for group size 1
> There are, nevertheless, schemes whose mechanims-specific bounds are stronger even for group size 1. An example is subsampling *with* replacement under substitution. In **Fig. 2 of the pdf attached to the global rebuttal comment above**, we compare the tight agnostic bound of Balle et al. [1] to the mechanism-specific bound posited by Koskela [3], which we proved in our work.
> The bound of Koskela yields stronger privacy guarantees.
>
> Using your notation, S is **subsampling with replacement**, $M_1$ is a Gaussian mechanism, $\delta'$ is the mechanism-specific bound, and $M_2$ is the randomized membership mechanism from Section 5 of [1].
>
> ### Comparison to group privacy  + tight bounds
> Please note that **this baseline is precisely what "post-hoc" (i.e. post-hoc application of the group privacy property) refers to in our  figures**.
>
> We will add the following to l. 311:
> "In all figures, *agnostic* refers to tight mechanism-specific bounds and *specific* refers to tight mechanism-specific bounds. *Post-hoc* refers to post-hoc application of the group privacy property to tight mechanism-specific bounds for group size $1$."
>
> ### Definition of Renyi divergence
> It seems like there was a **misunderstanding due to notation, which we will correct**.
> The expression $R_{\alpha}$ is not intended to be the Renyi divergence.  We apply a concave transform to recover the Renyi divergence from $R_\alpha$ in Definition 2.2 (l. 110). Joint convexity of this term is precisely  what is used in prior work on subsampling for Renyi DP, e.g., by Zhu & Wang [4].
>
> Based on your feedback, we will
> * Replace $R$ with a different character
> * Refer to it as "$\alpha$th moment of the Renyi divergence" instead of "scaled and exponentiated Renyi divergence"
>
> ### Asymptotic bounds
> While we focused on optimally charaterizing privacy with arbitrary precision to enable differentially private training, asymptotic upper bounds could provide an interesting alternative perspective. We assume that you refer to bounds as in "Deep Learning with Differential Privacy" [5].
>
> As shown in Appendix N.4, the $\alpha$th moment of the Renyi divergence for group size $K$ is bounded by $\sum_{k=1}^K \frac{1}{w} \mathrm{Binom}(k \mid K,r)
> \sum_{l=1}^\alpha \binom{\alpha}{l}w^l (1-w)^{\alpha-l} \zeta_k(l)$ with group profile $\zeta_k(l)$ and $w=1-(1-r)^K$.
> For Gaussian mechanisms, each of the $K$ inner sums is identical to the bound from [5] with varying sensitivity. We can thus directly use their result to obtain the asymptotic bound
> $\sum_{k=1}^K \frac{1}{w} \mathrm{Binom}(k \mid K,r)
> \cdot
> \left(w^2 k^2 \alpha (\alpha+1) \mathbin{/} ((1-w) \sigma^2) +
> \mathcal{O}(w^3 k^3 \mathbin{/} \sigma^3)\right)$.
>
> Alternatively, we could apply the analysis of Steinke [6] to each summand.
>
> Finding even better asymptotic bounds could be an interesting direction for future work. We will gladly include this result once we can upload a revision, thank you.
>
> ### Judging tightness without asymptotic bounds
> Please note that we can not only judge tightness (despite not providing asymptotic bounds in our initial submission), but **in fact formally prove it**. We do so by explicitly constructing a worst-case pair of datasets and  a worst-case sensitivity-bounded function that exactly attain our bound (see Appendix M.3).
>
> ### Operationalizing the bounds
> You are right, our mechanism-specific bounds sacrifice closedness in favor of tightness. They are nevertheless **easy to operationalize, because they let us determine a dominating pair of distributions** (see Section 3.4). Given a dominating pair, the bounds can be efficiently  evaluated to arbitrary precision via binary search, see Appendix M.4.1. This procedure is implemented in accounting libraries, which can analyze arbitrary dominating pairs in a plug-and-play manner.
>
> ### Comparison to Zhu&Wang
> Please note that their bound is precisely what we use as a baseline in our experiments on RDP (e.g. Fig.11) (see "post-hoc" discussion above).
>
> ### Comparison to Steinke
> Thank you for pointing out this great resource, which we will include in the related work section!
> Note that they primarily survey known results. Tight bounds for ADP were derived by Balle et al. [1], which we compare against. Similarly, their RDP bounds are restating results from Zhu & Whang [4], which we compare against.
>
> ### Appendix figure placement
> Once we can upload a revision, we will make sure that figures match their appendix sections.
>
> ---
>
> Again, thank you.
> We look forward to the discussion period.
>
> [1] Balle et al. "Privacy Amplification by Subsampling: Tight Analyses via Couplings and Divergences". NeurIPS'18
> [2] Zhu et al. "Optimal Accounting of Differential Privacy via Characteristic Function". AISTATS'22
> [3] Koskela et al. "Computing Tight Differential Privacy Guarantees Using FFT". AISTATS'20
> [4] Zhu & Wang. "Poisson subsampled Rényi differential privacy." ICML'19
> [5] Abadi et al. "Deep Learning with Differential Privacy". CCS'16
> [6] Steinke. "Composition of Differential Privacy & Privacy Amplification by Subsampling". arXiv

---

> > ### Comment · Reviewer_cbWU · 2024-08-13
> >
> > Thank you for the detailed answers; it resolves many of my concerns. I've adjusted my score accordingly.

---

### Official Review · Reviewer_T1S1 · 2024-07-12

**Soundness:** 3
**Presentation:** 2
**Contribution:** 3
**Rating:** 6
**Confidence:** 3

**Summary:**

The authors propose a general framework for deriving mechanism-specific differential privacy guarantees for amplification by subsampling. The current methods are generally only tight in a mechanism-agnostic sense, but may possibly be significantly more private. The authors propose a new framework using conditional optimal transport, which allows stronger mechanism-specific subsampling guarantees, while still recovering the mechanism-agnostic guarantees. They also derive guarantees for privacy accountants in a unified manner. As an application, they derive tight guarantees for group privacy, where previously only a weak bound is known.

**Strengths:**

- The problem being studied is important and relevant for practical applications.
- The framework derived in this paper seems general and broadly applicable.

**Weaknesses:**

- The authors only demonstrate improvements in the privacy bounds using their framework for group privacy. It would be interesting to see if their general framework could give improvements in other practical areas. It would also be good to demonstrate utility improvement in practice when doing DP training with these improved bounds.

**Questions:**

- Is there any intuition about the types of scenarios where this new framework gives improvements over existing mechanism-agnostic bounds?

**Limitations:**

- See weaknesses.

---

> ### Author Rebuttal · Authors · 2024-08-07
>
> Thank you for you are review!
> We are glad to hear that you find the studied problem important and the proposed framework broadly applicable.
>
> ### Application to other practical areas
> We agree that, while group privacy is highly important in practice,  future work should focus on applying our framework to other practical areas where data is subsampled or transformed via random functions.
>
> We would like to point out that we do in fact demonstrate improvements for another practically relevant application besides group privacy:
> **Differentially private deep learning where batches are not created via i.i.d. subsampling, but via shuffling of the dataset**. We discuss this in Appendix P, which we reference from our "Future Work" section. In this preliminary result,  we assume the batch size to be half the dataset size. By analyzing the distribution of all batches in the epoch jointly, we can derive tight bounds that demonstrate the benefit of shuffling over i.i.d. subsampling (see Fig. 24). As such, this can be seen as a step towards "tight epoch-level subsampling analysis", as opposed to the currently prevalent "iteration-level subsampling analysis".
>
>
> ## Utility improvement in DP training
> Thank you for this suggestion!
> As the experiments in our initial submission already demonstrate (e.g. Fig. 6), our tight group privacy analysis lets us train for more epochs before exceeding a given privacy budget $\varepsilon$ at a given $\delta$ -- regardless of which specific dataset, model architecture, or hyperparameters we consider.
>
> Based on your suggestion, **we have additionally investigated the effect this longer DP-SGD training has on training a model** for MNIST image classification (for details, see Fig. 2 in the pdf attached to the global rebuttal comment above).
>
> Even for group size $2$ and a large privacy budget of $\varepsilon=8$ and $\delta=10^{-5}$, post-hoc application of the group privacy property only allows training for $283$ iterations. This causes the training to stagnate at $79.6\%$ validation accuracy. Our tight mechanism-specific analysis allows us to either train for significantly more epochs or to terminate training after a few epochs with less privacy leakage and higher accuracy.
>
> Once we can upload a revision, we plan to repeat this experiment for a wider range of parameters, and also train or fine-tune on some larger-scale image dataset.
>
> ### Intuition for improvements
> This is a great question. Our intuition is as follows: As Fig. 2 in our original submission shows, all prior work on subsampling analysis relies on a binary partitioning of the subsampling event space. This is sufficient for many scenarios in which we either have maximum or no privacy leakage, e.g., "the modified element is sampled" vs "the modified element is not sampled".
>
> **Our framework is advantageous in scenarios where there are granular levels of privacy leakage.** For instance, when considering group privacy, we may sample $0$, $1$, or up to $K$ elements simultaneously. Another example is subsampling with replacement, where a single modified element can be sampled multiple times (see Fig. 3 in the pdf attached to the global rebuttal comment).
>
> By partitioning the batch space into multiple events, we can conduct a more fine-grained analysis than the binary partitioning underlying prior work. This results in tighter bounds.
>
> ---
>
> Again, thank you for your review.
> We hope that we addressed all your comments to your satisfaction.
> Please let us know if you have any further questions or comments during the author-reviewer discussion period!
>
> ---

---

> > ### Comment · Reviewer_T1S1 · 2024-08-13
> >
> > Thanks for the clarifications, I have no further questions.

---

### Official Review · Reviewer_7Wbe · 2024-07-13

**Soundness:** 3
**Presentation:** 3
**Contribution:** 4
**Rating:** 7
**Confidence:** 2

**Summary:**

The authors propose a framework for privacy accounting of amplification by subsampling. An existing principle for this problem is to consider couplings between the output distribution of the mechanism on neighboring datasets and apply joint convexity of the privacy measure at hand. The primary contribution is to select the optimal coupling and minimize the resulting privacy bound by viewing it as an optimal transport problem.

**Strengths:**

The paper provides a novel technical framework for accounting of subsampling amplification. The authors are able to reproduce important closure properties of dominating pairs using their framework (e.g. Proposition K.3), suggesting their framework may be used to analyze mechanisms that previously evaded accurate accounting (e.g. shuffling).

**Weaknesses:**

The main technical result Theorem 3.4 did not feel clearly motivated in terms of Theorem 3.3 and the experiments in Appendix B. It was not clear to me from the justification provided on p.4 lines 171-174 why Theorem 3.4 is able to improve substantively upon the preceding result.

**Questions:**

Small editorial suggestions:
- It is difficult to follow the quantifier constraints in some places e.g. p.5 l.193
- p.4 l.178 Missing $y^{(1)}$ and $y^{(2)}$ under sum
- p.36 l.852 should this be $x'$ not $x$?

**Limitations:**

While the paper leverages their framework to provide new tighter bounds for the group privacy setting, the framework is only used in the other privacy accounting settings to reproduce known results (from e.g. Zhu et al 2022 or Balle et al 2018). A future direction for this work, which the authors do mention, is to apply the framework to address the practical gaps in privacy accounting for machine learning.

---

> ### Author Rebuttal · Authors · 2024-08-07
>
> Thank you for your review and your suggested editorial changes!
> Please let us first respond to your higher-level comments, before discussing the smaller editorial changes.
> Note that we cannot update the manuscript during the rebuttal period, but will include all your suggestions in a revision as soon as possible.
>
> ### Motivation for main theorem
> Upon re-reading the relevant section, we agree that more could have been done to even better motivate Theorem 3.4. The original manuscript only states that conditioning lets us limit the recursion depth to which joint convexity is applied (see ll. 168-171). However, it does not explain why this might be necessary / desirable.
>
> Once we can upload a revision, we intend to provide the following, more in-depth explanation:
>
> "Theorem 3.3 is the result of recursively applying the joint convexity property (Lemma 3.1) to the mixture divergence $\Psi_{\alpha}(m_x || m_{x'})$. Each application splits each mixture into two smaller mixtures and --as per joint convexity-- further upper-bounds the divergence $\Psi_{\alpha}(m_x || m_{x'})$ that is achieved **by our specific subsampled  mechanism  $m$ on our specific pair of datasets $x,x'$**. Upon fully decomposing the overall divergence into divergences between single-mixture components, this sequence of bounds is in fact larger than the divergence $\Psi_{\alpha}\left(\tilde{m}\_{\tilde{x}} || \tilde{m}\_{\tilde{x}'}\right)$ achieved **by a worst-case subsampled mechanism $\tilde{m}$ on a worst-case pair of datasets $\tilde{x},\tilde{x}'$**. We experimentally demonstrate this via comparison with the bounds of Balle et al. in Appendix B.1.4.  To overcome this limitation, we propose to limit the recursion depth in order to obtain a smaller upper bound that matches **our specific subsampled mechanism $m$ on a worst-case pair of datasets $\tilde{x},\tilde{x}'$**. Limiting the recursion depth means upper-bounding $\Psi_{\alpha}(m_x || m_{x'})$ in terms of mixture divergences that have not been fully decomposed. Specifically, we propose to do so via [...]".
>
> Please let us know if you find this explanation more helpful.
>
> ### Beyond group privacy and a unified view on known results
> We agree with you on this point. The potential applications of our framework beyond group privacy are more exciting than the typical Poisson / without replacement / with replacement subsampling setting.
>
> We would nevertheless like to point out that our unified view lets us **close gaps in existing literature that have not previously been discussed.** Specifically, we derive Renyi differential privacy bounds for various combinations of subsampling schemes, dataset relations, and neighboring relations -- effectively bringing the generality of Balle et al.'s work [1] to moments accounting (see Appendix I). While moments accounting has been supplanted by PLD accounting in the context of DP-SGD, it still finds use in various other applications of DP (e.g. [2]). We further provide the first formal derivation of dominating pairs for subsampling with replacement, which appear in prior work [3] but are not formally proven therein.
>
> ### Editorial suggestions
>
> **Simplifing quantifier constraints, e.g., l.193**
> Following your feedback, we will make the simplifying assumption that the batch neighboring relation is symmetric throughout the main text. We will further eliminate domain subscripts when clear from context.   This will allow us to rewrite the constraint in l.193 (also l.225) as $\forall t, u: d(\hat{y}^{(1)}_t, \hat{y}^{(2)}_u) \leq d(y^{(1)}_t, y^{(2)}_u)$, which should be  easier to parse.
>
> **Missing variables under sum in l. 178**
> Thank you, we will specify the summation variables in our revision.
>
> **$x'$ not $x$ in l. 852**
> Thank you for pointing out this typo, we will correct it.
>
> Again, please remember that we unfortunately cannot upload a revision during the rebuttal period.
>
> ---
>
> Again, thank you for your review!
> To conclude, we hope that we addressed all your comments to your satisfaction.
> Please let us know if you have any further questions or comments during the author-reviewer discussion period!
>
> ---
>
> [1] Balle et al. "Privacy Amplification by Subsampling: Tight Analyses via Couplings and Divergences". NeurIPS'18
> [2] Chen et al. "Improved Communication-Privacy Trade-offs in 𝐿2 Mean Estimation under Streaming Differential Privacy". ICML'24
> [3] Koskela et al. "Computing Tight Differential Privacy Guarantees Using FFT".  AISTATS'20

---

> > ### Comment · Reviewer_7Wbe · 2024-08-14
> >
> > Thank you for the clarifications! The expanded exposition for Thm 3.4 is much clearer to me. I have no further questions.

---

### Official Review · Reviewer_eK4M · 2024-07-14

**Soundness:** 3
**Presentation:** 3
**Contribution:** 3
**Rating:** 7
**Confidence:** 3

**Summary:**

This work proposed a novel analysis of mechanism-specific amplification via subsampling. The authors decompose the subsampled mechanism into two parts: batch subsampling + mechanism. Then the analysis decompose the probability density into sums of pdf of every batch. The authors then provide upper bound to the divergence and specifically applied to group privacy under poisson distribution. The empirical analysis shows that the proposed analysis is able to significantly improves upon prior mechanism-agnostic group privacy bounds.

**Strengths:**

- The paper proposed novel analysis for group privacy. The application to poisson subsampling + gaussian mechanism seems solid.
- Solid theoretical guarantees.
- Simulation results look promising.

**Weaknesses:**

- Theorem 3.4 + proposition 3.5 themselves seem very costly to compute. I appreciate the authors for providing example of poisson subsampling. Still I'm wondering whether, e.g. proposition 3.5, is easy to solve for non Poisson distribution.
- Similar to the previous point, a small weakness for Theorem 3.7 is it's unclear how hard it is to solve for mechanisms such as Laplace. If it does not have a closed form (or closed form hard to derive), it might lead to un-tight upper bound or costly numerical simulation in practice. (Edit: I see the Laplace analysis in the appendix)
- The theoretical guarantees consist of a lot of notation. Might be helpful for the authors to add small interpretation of the bounds in e.g. Eq  2 and Eq 4 for better readibility.
- Could the authors add group size = 1 in the experiment for completeness?

**Questions:**

See above.

---

> ### Author Rebuttal · Authors · 2024-08-07
>
> Thank you for the review!
> We are excited to hear that you find our submission worthy of acceptance.
>
> ### Computational cost of computing Theorem 3.4+Proposition 3.5
> Before discussing the main question of your first bullet point,
> we would like to briefly clarify the following:
> The optimal transport problem in Theorem 3.4 and the constrained optimization problem in Proposition 3.5 are not primarily intended to be solved via computational methods. Rather, they are **meant as a recipe for formally deriving amplification guarantees**, so that researchers do not need to "reinvent the wheel" whenever they want to analyze a different subsampling scheme or neighboring relation.
>
> Nevertheless, operationalizing these two results as a form of automatic theorem proving could be an exciting direction for future work. Thank you for this inspiring idea!
>
> ### Can Proposition 3.5 be solved for non-Poisson distributions?
> **Yes, in fact we do so in our paper**.
> Specifically, we solve the problem for subsampling without replacement and two mixture components to provide an alternative proof for the dominating pairs originally derived by Zhu et al. [1] (see Appendices K.2 and K.3)
>
> When considering the same neighboring relation, the optimization problem for Poisson subsampling and subsamling without replacement essentially only differ in the mixture weights / event probabilities. Our solution to the optimization problem in Theorem 3.7 is independent of the mixture weights (see Appendix O - "Worst-case mixture components").
>
> ### Can Theorem 3.7 be solved for non-Gaussian distributions?
> As you correctly point out in your edit, we do in fact also solve this problem for Laplace and Bernoulli (randomized response) distributions.
>
> In fact, our proof strategy does in principle extend to **arbitrary distributions whose isocontours are given by $\ell_p$-norm balls** (see, e.g.,  Lemma O.1). However, in order to simplify the notation of our Lemmata and to keep the scope of our paper more manageable, we have decided to focus on mechanisms commonly used by the community.
>
> If you would like to discuss this generalization further, please let us know during the discussion period!
>
> ### Interpretation of Eq. 2 for and Eq. 4 to improve readability
> Upon re-reading our manuscript, we agree that we should use the additional content page to provide more detailed explanations of these results.
>
> Unfortunately we cannot upload a revised manuscript during the rebuttal period. But **we intend to add the following explanations:**
>
> Eq. 2:
> "We now have an optimal transport problem between I+J probabability mass functions, coupled by $\gamma$. The transport cost $c$, given batch tuples $\mathbf{y}^{(1)} \in \mathbb{Y}^I$ and $\mathbf{y}^{(2)} \in \mathbb{Y}^I$, is a divergence between two mixtures. The components of the first mixture are base mechanisms densitities given batches $y_i^{(1)} \in \mathbb{Y}$ from $\mathbf{y}^{(1)}$. The weights are probabilities of events $A_i$.  The second mixture is defined analogously."
>
> Eq. 4:
> "This bound instantiates Eq. 2 with binomial mixture weights.
> Batches $y^{(1)}_u$ and $y^{(1)}_t$ have a distance bounded by $|u-t|$ because one can be obtained from the other by removing/inserting $|u-t|$ elements. The constraints for $y^{(2)}_u$ and $y^{(2)}_t$ are analogous. Batches $y^{(1)}_u$ and $y^{(2)}_t$ have a distance bounded by $(t-1) + (u-1)$ because we need to remove $t-1$ elements and insert $u-1$ elements to construct one from the other."
>
> Depending on the remaining space after implementing the other reviewers' suggestions, we intend to provide similar explanations for other results (e.g. Theorem 3.3 and Theorem 3.8). Thank you for helping us in further improving the readability of our paper!
>
> ### Could the authors add group size 1 to the experiments for completeness?
> Of course. For an example, please **see Fig. 1 in the pdf attached to the global rebuttal comment above**. We will make the same change for all figures once we can upload a revision.
>
> Note that our experiments with randomized response and Renyi differential privacy (e.g. Fig 3 in our original manuscript) already considered group size 1. There, we outperformed the best known guarantee even for group size 1.
>
> ---
> Thank you again for your efforts.
> We hope that we addressed all your comments to your satisfaction.
> Please let us know if you have any further questions during the author-reviewer discussion period!
>
> ---
>
> [1] Zhu et al. "Optimal Accounting of Differential Privacy via Characteristic Function". AISTATS'22

---

> > ### Comment · Reviewer_eK4M · 2024-08-13
> > **Response**
> >
> > Thank you for your response. I have no further questions.

---

### Author Rebuttal · Authors · 2024-08-07

We are very grateful for the helpful reviews we received.
While we have already individually responded to each of the reviewers' insightful comments, we would like to use this global rebuttal comment to
1. Provide an overview of the figures in the attached pdf file
2. For the area chair's convenience, summarize the (small) changes we intend to make make to our manuscript once we can upload a revision.

Please note that **none of the intended changes affect our main contributions or the validity of any of our results**.
Instead they provide novel perspectives or slightly more detailed explanations, in order to further improve accessibility for future readers.

## Attached pdf with figures

**Figure 1** shows that we will now also include group size 1 (i.e. traditional differential privacy) in our group privacy experiments.

**Figure 2** demonstrates that the increased number of training iterations enabled by our tight mechanism-specific group privacy analysis can in fact translate to higher model utility at a given privacy budget.

**Figure 3** showcases that mechanism-specific bounds can outperform tight mechanism-agnostic bounds even for group size 1 (here: subsampling *with* replacement).

## Intended changes

### Reviewer 1 (eK4M)
As suggested by reviewer 1, we will
* provide a short textual interpretation of Eq. 2 for accessibility (see rebuttal below for formulation),
* provide a short textual interpretation of Eq. 4 for accessibility (see rebuttal below for formulation),
* include group size 1 (i.e. traditional differential privacy) in all our group privacy experiments (see Fig.1 in attached pdf).

### Reviewer 2 (7Wbe)
As suggested to reviewer 2, we will
* slightly expand our textual motivation of Theorem 3.4 (see rebuttal below for formulation).

### Reviewer 3 (T1S1)
As suggested by reviewer 3, we will
* include an experiment on trained model utility (see Fig. 2 in attached pdf).

### Reviewer 4 (cbWU)
Following feedback by reviewer 4, we will
* clarify that mechanism-agnostic tightness *can* translate to mechanism-specific tightness for the special case of group size $1$ (see rebuttal below for formulation),
* include an experiment on subsampling with replacement to demonstrate that mechanism-agnostic tightness *does not always* translate to mechanism-specific tightness for group size 1 (see Fig. 3 in attached pdf),
* clarify that "post-hoc" in our figures refers to applying the group privacy property to tight group-size-1 bounds (see rebuttal below for formulation),
* include a novel asymptotic group privacy amplification bound at the end of Section 3.1 to provide *an additional perspective* on our tight guarantees (see rebuttal below for formula)
* cite Thomas Steinke's book chapter "Composition of Differential Privacy & Privacy Amplification by Subsampling", which surveys methods we compare against and could serve as a great introduction to future readers.

### Reviewer 5 (t4hg)
Following a suggestion by reviewer 5, we will
* move Theorem M.4 and a shortened version of its proof to the main text, to more directly showcase the theoretically proven tightness of our bounds.


### Notation / Formulations / Formatting
In addition to the above changes that improve accessibility and provide novel perspectives, we will
* Simplify constraint quantifiers in l. 193 and 225 to $\forall t, u: d(\hat{y}^{(1)}_t, \hat{y}^{(2)}_u) \leq d(y^{(1)}_t, y^{(2)}_u)$
* add missing variables under the sum operator in l.178,
* replace $x$ with $x'$ in l.852,
* correctly align figures in the appendix with their respective sections,
* replace $R_\alpha$ with a different character to avoid confusion between this "$\alpha$th moment" with the actual Renyi divergence,
* refer to it as $\alpha$th moment instead of "scaled and exponentiated Renyi divergence".

---

### Decision · Program_Chairs · 2024-09-25

**Decision:**

Accept (poster)

**Comment:**

Reviewers were rather positive on this submission. Mechanism-specific privacy accounting has been quite a trend recently, giving markedly improved privacy bounds for a number of important applications. Reviewers appreciated this novel approach which is able to incorporate subsampling into the process. There were some critiques, e.g., the bounds are a bit tough to operationalize, and the paper is rather long (100 pages) and thus hard to check every detail. Nonetheless, we all agreed that it is a very interesting framework for an important problem, and thus worth of acceptance to the conference.